# LEGALBENCH: A Collaboratively Built Benchmark for Measuring Legal Reasoning in Large Language Models

Neel Guha[*1], Julian Nyarko[*1], Daniel E. Ho[*1], Christopher Ré[*1], Adam Chilton[2], Aditya Narayana[3], Alex Chohlas-Wood[1], Austin Peters[1], Brandon Waldon[1], Daniel N. Rockmore[4], Diego Zambrano[1], Dmitry Talisman[3], Enam Hoque[5], Faiz Surani[1], Frank Fagan[6], Galit Sarfaty[7], Gregory M. Dickinson[8], Haggai Porat[9], Jason Hegland[1], Jessica Wu[1], Joe Nudell[1], Joel Niklaus[1], John Nay[10], Jonathan H. Choi[11], Kevin Tobia[12], Margaret Hagan[13], Megan Ma[10], Michael Livermore[14], Nikon Rasumov-Rahe[3], Nils Holzenberger[15], Noam Kolt[7], Peter Henderson[1], Sean Rehaag[16], Sharad Goel[17], Shang Gao[20], Spencer Williams[18], Sunny Gandhi[19], Tom Zur[9], Varun Iyer , and Zehua Li[1]

[1]Stanford University, [2]University of Chicago, [3]Maxime Tools, [4]Dartmouth College, [5]LawBeta, [6]South Texas College of Law Houston, [7]University of Toronto, [8]St. Thomas University Benjamin L. Crump College of Law, [9]Harvard Law School, [10]Stanford Center for Legal Informatics - CodeX, [11]University of Southern California, [12]Georgetown University Law Center, [13]Stanford Law School, [14]University of Virginia, [15]Télécom Paris, Institut Polytechnique de Paris, [16]Osgoode Hall Law School, York University, [17]Harvard Kennedy School, [18]Golden Gate University School of Law, [19]Luddy School of Informatics - Indiana University Bloomington, [20]Casetext

## Abstract

The advent of large language models (LLMs) and their adoption by the legal community has given rise to the question: what types of legal reasoning can LLMs perform? To enable greater study of this question, we present LEGALBENCH: a collaboratively constructed legal reasoning benchmark consisting of 162 tasks covering six different types of legal reasoning. LEGALBENCH was built through an interdisciplinary process, in which we collected tasks designed and hand-crafted by legal professionals. Because these subject matter experts took a leading role in construction, tasks either measure legal reasoning capabilities that are practically useful, or measure reasoning skills that lawyers find interesting. To enable cross-disciplinary conversations about LLMs in the law, we additionally show how popular legal frameworks for describing legal reasoning—which distinguish between its many forms—correspond to LEGALBENCH tasks, thus giving lawyers and LLM developers a common vocabulary. This paper describes LEGALBENCH, presents an empirical evaluation of 20 open-source and commercial LLMs, and illustrates the types of research explorations LEGALBENCH enables.

## 1 Introduction

Advances in large language models (LLMs) are leading American legal professionals to reexamine the practice of law [52, 61, 158, 55, 13].[2] Proponents have argued that LLMs could alter how lawyers approach tasks ranging from brief writing to corporate compliance [158]. By making legal services more accessible, they could eventually help alleviate the United States' long standing access-to-justice crisis [35, 132]. This perspective is informed by the observation that LLMs possess special properties

---

[2]In using "LLMs", we are referring to language models which evince in-context learning capabilities (also referred to as "foundation models" [13]). This behavior has traditionally been observed in models with at least a billion parameters.

37th Conference on Neural Information Processing Systems (NeurIPS 2023) Track on Datasets and Benchmarks.

which, it is argued, make them more suited for legal tasks. The models' capacity to learn new tasks from limited labeled data would reduce the manual data annotation costs that ordinarily burden the development of legal language models [13]. Their apparent proficiency at sophisticated reasoning tasks would also make them ideal for the rigor of law, which requires parsing obtuse texts with heavy jargon, and inferential processes which combine different modalities of reasoning [155].

This excitement, however, is tempered by the fact that legal applications often involve significant risk [47]. Existing work has shown that LLMs are capable of generating content that is offensive, misleading, and factually incorrect [10, 78]. Such behaviors—if replicated in legal applications [112]— could result in substantial harms [144], with much of the potential burden imposed on traditionally marginalized and under-resourced populations [125, 136]. The safety implications thus create a pressing need to develop infrastructure and processes for benchmarking LLMs in legal contexts.

However, significant challenges face practitioners seeking to assess whether LLMs can perform legal reasoning. The first challenge is the limited ecosystem of legal benchmarks [155]. The majority of existing benchmarks, for example, focus on tasks which models learn by finetuning or training on task-specific data [21]. These benchmarks do not measure the aspects of LLMs which generate excitement for law—namely, their ability to perform many different tasks using only few-shot prompts. Relatedly, benchmarking efforts have focused on professional certification exams like the Uniform Bar Exam [69], but these are not always representative of the actual use-cases for LLMs. The second challenge is the incongruity between the ways in which existing benchmarks and lawyers frame "legal reasoning." Existing benchmarks coarsely generalize all tasks involving legal data or laws as measuring "legal reasoning." In contrast, lawyers recognize that legal reasoning is a broad umbrella term encompassing many distinct types of reasoning [46]. Different legal tasks require different skills and bodies of knowledge. Because existing legal benchmarks fail to draw these distinctions, it is difficult for legal professionals to contextualize the performance of modern LLMs within their own understanding of legal competency. In short: legal benchmarks do not use the same vocabulary or conceptual frameworks as the legal profession.

In light of these limitations, we believe that rigorously evaluating the legal reasoning capabilities of LLMs will require the legal community to take a more proactive role in the process of benchmarking. To that end, we present LEGALBENCH: the first steps towards constructing an interdisciplinary collaborative legal reasoning benchmark for the English language.[3] Over the past year, the authors of this paper—drawing from their diverse legal and computer science backgrounds—came together to assemble 162 tasks (from 36 different data sources), each of which measures a specific type of legal reasoning. LEGALBENCH is thus, to the best of our knowledge, the first *open-source legal benchmarking effort*. We believe that this style of benchmark construction—where domain experts take an active and participatory role in the crafting of evaluation tasks—illustrates one approach to interdisciplinary collaboration in LLM research. Importantly, we believe it also shows that legal professionals have an essential role to play in the assessment and development of LLMs for law.

As a research project, we highlight three components of LEGALBENCH:

1. LEGALBENCH was constructed from a mix of existing legal datasets (restructured for the few-shot LLM paradigm), and hand-crafted datasets created and contributed by legal professionals (included as authors on this work). The legal professionals involved in this collaboration were asked to contribute datasets that they believed to either measure an interesting legal reasoning skill, or to capture a practically useful application for LLMs in the law. High performance on LEGALBENCH tasks thus provides useful information, allowing lawyers to validate their assessment of an LLM's legal competency, or identify an LLM that could be used in their workflow.

2. LEGALBENCH tasks are organized into an extensive typology which describes the types of legal reasoning required to perform the task. Because this typology is drawn from frameworks familiar to the legal community, it enables legal professionals to meaningfully engage in discussions of LLM performance, using a terminology and conceptual framework familiar to them [46, 122].

3. Finally, LEGALBENCH is intended as a platform to support further research. For AI researchers who lack legal expertise, LEGALBENCH comes with significant support for understanding how to prompt and evaluate different tasks. And as more of the legal community begins to engage with

---

[3]https://github.com/HazyResearch/legalbench/

the potential impact and role of LLMs, we hope to grow LEGALBENCH by continuing to solict and incorporate tasks from legal professionals.[4]

In this paper, we make several contributions. First, we present a typology for organizing and describing legal tasks in terms of the types of reasoning they require. This typology is drawn from frameworks lawyers use to describe legal reasoning [122]. Second, we provide an overview of the tasks in LEGALBENCH, describing the process by which they were constructed, important dimensions of heterogeneity, and limitations. A full description of each task is provided in the Appendix. Finally, we use LEGALBENCH to evaluate 20 LLMs from 11 different families, across a range of size points. We make observations regarding the performance of different models and present an initial study into different prompt-engineering strategies. Ultimately, these results are intended to highlight different directions of future work that LEGALBENCH may enable.

We hope that this benchmark will be interesting to a diverse set of communities. Practitioners may use these tasks to determine whether and where LLMs can be integrated into existing workflows to improve outcomes for clients. Legal academics may benefit from observing the types of annotation that LLMs are capable of [157], and different forms of empirical scholarly work they may enable. Computer scientists may benefit from studying the performance of these models in a domain like law, where distinct lexical properties and unique tasks may surface new insights.

Before we progress further, we note that the purpose of this work isn't to evaluate whether computational systems *should* replace lawyers and legal officers, or to understand the positive and negative impacts of that replacement [47, 126, 4]. Rather, our goal is to construct artifacts that enable the relevant stakeholders and affected communities to better understand, *empirically*, the capacity for LLMs to perform different types of legal tasks. Given the proliferation of computational legal tools, we believe that answering this question is vital for ensuring their safe and ethical usage.

## 2   Related work

**Benchmarking legal reasoning**   Understanding the extent to which NLP models can perform tasks or skills traditionally associated with lawyers—or be useful in legal analysis—has been the focus of significant work [70, 6, 108, 124, 82, 84, 102, 86, 22]. Prior work has identified manually arduous tasks currently performed by lawyers (e.g., document review, case summarization) and developed corresponding benchmarks [60, 140, 118, 119, 88, 67]. Researchers have focused on the technically challenging asepcts of legal tasks, like document length, jargon, or inferential reasoning [21, 109, 85, 40, 77, 21, 78, 155, 63]. Other work has focused on creating datasets for pretraining models [58, 127], non-English/multilingual tasks [93, 94, 64, 149, 51, 92, 68, 20, 23, 101, 18], legal judgement prediction [87, 37, 17, 156], legal role labeling [83], and different forms of retrieval [66].

Importantly, the majority of previous benchmarking efforts have focused on language models which learn by supervised training or finetuning (e.g., BERT variants [44]), and researchers have consequently studied questions related to the role of domain specific datasets [155, 19, 20]. More recently, researchers have begun to ask whether *large* language models (LLMs) like GPT-3/4 can perform legal reasoning [71, 151, 65, 12, 28, 30, 152], citing to evidence of these models' capacity to perform sophisticated reasoning tasks in domains like math or programming [143, 24]. Unlike BERT-based models, LLMs are evaluated on their ability to learn tasks *in-context*, primarily through prompting. While a few works have experimented with LLMs on existing benchmarks [16, 12], most evaluations focus on standardized tests or other exam equivalents [69, 91, 29]. Studies have explored the role of prompt-engineering [152, 151, 73], potential applications [91, 28, 145, 115, 114], questions regarding human-LLM interaction [30, 61], and comparisons to older finetuned-models [89].

**Connections to other LLM benchmarking efforts**   We highlight connections to two broader research efforts. First, we draw inspiration from existing efforts within NLP and machine learning to define fine-grained measures of performance, which allow researchers to discuss model capabilities with precision and specificity. Examples include the diagnostic set of the GLUE Benchmark [139], the "reasoning patterns" studied in [98], the task organization used in HELM [78], and the BigBench effort [121]. We believe this paradigm of expert-driven evaluation is essential for specialized domains like law.

---

[4]Cognizant of LEGALBENCH's current skew towards American law, we hope that additional contributions incorporate tasks from other jurisdictions.

# 3   The LEGALBENCH typology

LEGALBENCH identifies six types of legal reasoning that LLMs can be evaluated for: (1) issue-spotting, (2) rule-recall, (3) rule-application, (4) rule-conclusion, (5) interpretation, and (6) rhetorical-understanding. We first justify the selection of these types by providing background on how the legal profession frames "legal reasoning," and the connections to our typology. We then illustrate how task datasets may be used to evaluate LLMs for each type, using examples from LEGALBENCH.

Though this framework draws heavily on American legal thought, we believe it can be extended to characterize LEGALBENCH tasks that implicate non-American bodies of law. We also note that our types are non-exhaustive, and in future work hope to consider extensions.

## 3.1   Frameworks for legal reasoning

**IRAC**   American legal scholars often describe "legal reasoning" as the process of determining the legal conditions that arise from a set of events or occurrences, with reference to both prior cases and codified laws [46]. A common framework for executing this type of legal reasoning is the **I**ssue, **R**ule, **A**pplication and **C**onclusion (**IRAC**) framework [146, 122]. In this framework, legal reasoning decomposes into four sequential steps.

First, lawyers identify the legal issue in a given set of facts (**issue-spotting**). An issue is either (1) a specific unanswered legal question posed by the facts, or (2) an area of law implicated in the facts. Depending on the setting, a lawyer may be told the issue, or be required to *infer* a possible issue.

Second, lawyers identify the relevant legal rules for this issue (**rule-recall**). A rule is a statement of law which dictates the conditions that are necessary (or sufficient) for some legal outcome to be achieved. In the United States, rules come from different sources: the Constitution, federal and state statutes, regulations, and court opinions. Importantly, rules often differ between jurisdictions. Hence, the relevant rule in California might be different than the relevant rule in New York.

Third, lawyers apply these rules to the facts at hand (**rule-application**). Application, or the analysis of rule applicability, consists of identifying those facts which are most relevant to the rule, and determining how those facts influence the outcome under the rule. Application can also involve referencing prior cases involving similar rules (i.e. *precedent*), and using the similarities or differences to those cases to determine the outcome of the current dispute.

Finally, lawyers reach a conclusion with regards to their application of law to facts, and determine what the legal outcome of those facts are (**rule-conclusion**).

**Example**   Suppose that BusinessMart—a large manufacturing corporation—is being sued by Amy in federal court on diversity jurisdiction.[5] BusinessMart sells the majority of its goods in Texas, has its headquarters (where its CEO and board members sit and work) in California, and maintains a factory in Florida. A court is trying to determine—for the purposes of diversity jurisdiction—where BusinessMart's "principal place of business is."

- Issue-spotting: Here, a narrow issue is offered—where is BusinessMart's principal place of business?

- Rule-recall: A lawyer would recognize that the most relevant rule here comes from the case *Hertz Corp. v. Friend*,[6] in which the Supreme Court determined "that the phrase 'principal place of business' refers to the place where the corporation's high level officers direct, control, and coordinate the corporation's activities."

- Rule-application: Applying this rule to the facts above yields two observations. First, a corporation's CEO and board members are examples of high level officers referred to in *Hertz* that control and conduct a company. Second, the place where BusinessMart's high level officers control the company is California, as that is where the CEO and board sit and work.

- Rule-conclusion: Based on the chain of inference spelled out in the application stage, a lawyer would thus conclude that California is BusinessMart's principal place of business.

---

[5]Diversity jurisdiction gives federal courts the ability to hear cases between parties that are "citizens" of different states.

[6]Hertz Corp. v. Friend, 559 U.S. 77 (2010).

The extent to which the outcome of the application and conclusion steps follow each other is dictated by the level of ambiguity in the fact patterns. When the law on a particular question is clear and there is little ambiguity in the facts (as the case in the above example), then the application and conclusion steps point towards the same outcome. Sometimes however, the facts may be unclear or contested, and reasonable minds may differ as the conclusion step. For now, LEGALBENCH focuses entirely on the former setting (unambiguous answers), and all tasks are considered to have objectively "correct" answers.

**Other types of reasoning**  Though IRAC is the most formal framework for legal reasoning, lawyers recognize a variety of skills which are useful to practice of law [46, 75]. For instance, lawyers are often required to exercise interpretive skills, in order to identify the rights, obligations, or limitations of certain legal language (e.g., what a contractual clause may or may not enable). They must also exhibit rhetorical skills, and understand the types of arguments that are made. Though these tasks require the knowledge base and skill set of lawyers, they, arguably, do not always fit neatly within the IRAC framework. Hence, we consider these to be distinct from the examples offered in the previous section.

## 3.2  Evaluating legal reasoning in large language models

LEGALBENCH identifies six categories of legal reasoning. For each category, we describe how a LLM task may evaluate the typified legal reasoning, using examples from LEGALBENCH.

**Issue-spotting**  LEGALBENCH evaluates issue-spotting through tasks in which an LLM must determine if a set of facts raise a particular set of legal questions, implicate an area of the law, or are relevant to a specific party. Issue tasks evaluate a LLM's ability to reason over the legal implications of different activities, events, and occurrences. An example of an issue-spotting task is the `learned_hands_benefits` task, which requires an LLM to determine (Yes/No) whether a post on a public legal aid forum raises issues related to welfare law (i.e., public benefits or social services). The box below shows how a LLM might be prompted for this task.

---

**Issue-spotting example: `learned_hands_benefits`**

Does the post discuss public benefits and social services that people can get from the government, like for food, disability, old age, housing, medical help, unemployment, child care, or other social needs?

Post: "I am currently receiving support from social services, idk why, this is just how my life turned out. They have asked for all of my bank information for the past 12 months. I don't know what this means. Why would they want that?"
Answer: Yes

---

**Rule-recall**  LEGALBENCH evaluates rule-recall through tasks which require the LLM to generate the correct legal rule on an issue in a jurisdiction (e.g., the rule for hearsay in US federal court). A rule task can be an open-ended generation task—in which the LLM must generate the text of the rule for a jurisdiction—or a classification task—in which the LLM must determine whether the rule exists in that jurisdiction. Anchoring to jurisdiction is important, as legal rules differ across different jurisdictions. Rule tasks are particularly useful for measuring *hallucinations* [79]. An example of a rule-recall task is `rule_qa`, a question-answer task where questions include asking the model to state the formulations for different legal rules, identify where laws are codified, and general questions about doctrine.

---

**Rule-recall example: `rule_qa`**

Question: What are the four requirements for class certification under the Federal Rules of Civil Procedure?"
Answer: Numerosity, commonality, typicality, adequacy

---

**Rule-conclusion**  LEGALBENCH evaluates rule-conclusion through tasks which require an LLM to determine the legal outcome of a set of facts under a specified rule. LLMs are evaluated purely on whether their predicted outcome is correct. For example, the `ucc_v_common_law` task asks a LLM to determine whether a contract is governed by the Uniform Commercial Code (UCC) or the common law of contracts. The LLM is always provided with the relevant rule, via the prompt (see below).

The UCC (through Article 2) governs the sale of goods, which are defined as moveable tangible things (cars, apples, books, etc.), whereas the common law governs contracts for real estate and services. For the following contracts, determine if they are governed by the UCC or by common law.

Contract: Alice and Bob enter into a contract for Alice to sell her bike to Bob for $50. Is this contract governed by the UCC or the common law?
Governed by: UCC

**Rule-application** LEGALBENCH evaluates rule-application through the same tasks used to measure rule-conclusion. When evaluating rule-application however, we prompt the LLM to provide an explanation of how the rule applies to a set of facts, and evaluate the quality of the generated explanation along two dimensions: (1) whether the explanation is *correct*, and (2) whether it contains *analysis*. Each metric captures a different dimension upon which a particular rule-application may be good.

Correctness corresponds to the criteria that explanations should not contain errors. We focus on five types of errors: misstatements of the legal rule, misstatements of the fact pattern, incorrectly asserting the legal outcome, logic errors, and arithmetic errors. Analysis corresponds to the criteria that explanations should contain inferences from the facts that are relevant under the rule, and illustrate how a conclusion is reached. Consider, for example, an explanation which restates the rule, the fact pattern, and the predicted legal outcome. If the predicted legal outcome is correct, than the explanation in its entirety would be correct, because it contains no error. However, as prior works have noted [69, 29], examples like this are conclusory, and often unsatisfactory in the context of legal work.

To standardize evaluation and enable future work, we have released an "answer guide" for each task used for rule-application, which contains the inferences required for each sample, and describes common modes of errors. All evaluations in LEGALBENCH for rule-application have been performed with respect to this answer-guide.

Table 1 presents an examples of how three generations (corresponding to the Alice/Bob example above) would be evaluated under the above metrics. The first generation is incorrect, because it misstates the rule. The second generation is correct because it contains no falsehoods, but performs no analysis because it does not articulate inferences. The third generation is both correct and contains analysis, because it has no errors, and explicitly mentions an essential inference (e.g., that a bike is a "good").

| Incorrect | Correct/no analysis | Correct/yes analysis |
|---|---|---|
| The contract is for Alice to sell her bike to Bob. The contract is governed by the common law, because all goods are governed by the common law. | The contract is for Alice to sell her bike to Bob. The contract is governed by the UCC, because the UCC governs all goods. | The contract is for Alice to sell her bike to Bob. The contract is governed by the UCC, because a bike is a good and all goods are governed by the UCC. |

Table 1: An example of how different generations are evaluated for correctness and analysis.

**Interpretation** LEGALBENCH evaluates interpretation through tasks which require the LLM to parse and understand a legal text. Interpretive tasks provide the LLM with a text, and ask the LLM to either extract a relevant piece of information, answer a question, or categorize the text by some property. Interpretive tasks are among the most studied and practically relevant tasks in LEGALBENCH, and many have been taken from actual use-cases. An example of an interpretive task is `cuad_audit_right`, which asks the LLM to determine if a contractual clause contains an "audit right." An example is shown below:

Does the clause give a party the right to audit the books, records, or physical locations of the counterparty to ensure compliance with the contract?

Clause: "We shall have the right at all times to access the information system and to retrieve, analyze, download and use all software, data and files stored or used on the information system."
Answer: Yes

**Rhetorical-understanding** LEGALBENCH evaluates rhetorical-understanding through tasks which require an LLM to reason about legal argumentation and analysis. In these tasks, an LLM is provided with a legal argument (usually excerpted from a judicial opinion), and asked to determine whether it performs a certain

function or has a certain property. An example is the `definition_classification` task, in which an LLM must determine if a sentence from a judicial opinion provides a definition of a term.

> **Rhetorical-understanding example: `definition_classification`**
>
> Does the sentence define a term?
>
> Sentence:"To animadvert carried the broader implication of "turn[ing] the attention officially or judicially, tak[ing] legal cognizance of anything deserving of chastisement or censure; hence, to proceed by way of punishment or censure." 1 Oxford English Dictionary 474 (2d ed.1989)."
> Answer: Yes

We emphasize one aspect of LEGALBENCH: IRAC in this work is used as an organizing principle for grouping tasks. On a law exam, a student would be expected to generate an answer which structurally resembles IRAC, where each step builds on the inferences of the previous step [69, 29]. LEGALBENCH tasks, in contrast, each evaluate a single type of legal reasoning. Hence, a task like `learned_hands_benefits` can only be used to evaluate issue-spotting, and not rule-recall. In future work we hope to add tasks which evaluate multiple steps jointly.

# 4  LEGALBENCH tasks

Appendix G discusses each task in detail, providing a description of the reasoning that each task evaluates, how task data was constructed, task examples, and evaluation protocols. This section provides an overview of LEGALBENCH.

## 4.1  Construction process

**Task sources**    LEGALBENCH tasks are drawn from three sources. The first source of tasks are existing available datasets and corpora. Most of these were originally released for non-LLM evaluation settings. In creating tasks for LEGALBENCH from these sources, we often significantly reformatted data and restructured the prediction objective. For instance, the original CUAD dataset [60] contains annotations on long-documents and is intended for evaluating extraction with span-prediction models. We restructure this corpora to generate a binary classification task for each type of contractual clause. While the original corpus emphasized the long-document aspects of contracts, our restructured tasks emphasize whether LLMs can identify the distinguishing features of different types of clauses. The second source of tasks are datasets that were previously constructed by legal professionals but never released. This primarily includes datasets hand-coded by legal scholars as part of prior empirical legal projects (e.g., [27]). The last category of tasks are those that were developed specifically for LEGALBENCH, by the authors of this paper. Overall, tasks are drawn from 36 distinct corpora.

**Collaborative component**    In August 2022, we published a call for tasks, describing the goals of the project and its structure [57]. We publicized the project through mailing lists and legal computational conferences. Submitted tasks were vetted for legal correctness and task validity. Task contributors are drawn from diverse professional backgrounds within the law (e.g., academics, practitioners, computational legal researchers) and constitute the authors of this paper.

**Infrastructure**    LEGALBENCH comes with support designed to enable non-law AI researchers to use and study LEGALBENCH tasks. First, each LEGALBENCH task is accompanied by extensive documentation describing how the task is performed, its legal significance, and the construction procedure. The objective of this documentation is to provide AI researchers with a working understanding of the mechanical processes behind each task, for the purposes of better understanding LLM performance. Second, each task is accompanied by a "base" prompt, which contains task instructions and demonstrations. The base prompt is provided to promote replicability and standardization. We anticipate that future research efforts building off of LEGALBENCH will identify higher performing prompts/prompt formats. We intended to update the LEGALBENCH GitHub repository with these prompts as they are discovered.

**Limitations**    We note several limitations of the current LEGALBENCH tasks (additional limitations are noted in Appendix C). First, when this project began, most LLM context-windows were constrained to a few pages of text. As a result, the initial round of LEGALBENCH tasks does not involve longer documents. We hope to include such tasks in future work, particularly as recent technical developments have resulted in significantly longer context windows [41, 53, 42, 107]. Second, LEGALBENCH's tasks focus on legal reasoning questions with objectively correct answers. LEGALBENCH is thus not helpful for evaluating legal reasoning involving degrees of correctness or tasks where "reasonable minds may differ." Third, LEGALBENCH only considers English language tasks, is skewed towards certain jurisdictions (American law), and certain areas of the law (contracts). Thus, the current iteration of the benchmark limits inferences regarding how LLMs may generalize to legal tasks involving other jurisdictions. As we continue to solicit and incorporate contributions to LEGALBENCH, we hope

to add tasks addressing these limitations. Finally, LEGALBENCH evaluates IRAC abilities independently, while law exams and other legal work requires lawyers to generate outputs which follow IRAC in a multi-hop matter (i.e., each aspect is applied to the same fact pattern).

## 4.2 Dimensions of variation

**Task structure**    All LEGALBENCH tasks contain at least 50 samples, with an average task size of 563 samples (Appendix E.4). These tasks are comparable in size to those used in benchmarking efforts like BigBench [128], HELM [78] or RAFT [1]. LEGALBENCH tasks also span different formats: multiple-choice questions (35 tasks), open-generation (7 tasks), binary classification (112 tasks), and multi-class/multi-label classification (8 tasks).

**Reasoning types and legal domains**    LEGALBENCH provides tasks for each of the reasoning categories discussed above: rule-recall (5 tasks), issue-spotting (16 tasks), rule-application (16 tasks), rule-conclusion (16 tasks), interpretation (119 tasks), and rhetorical-understanding (10 tasks). Tasks are predominantly drawn from areas of law implicating civil matters, including contracts (58 tasks), civil procedure (8 tasks), evidence law (1 task), and corporate law (58 tasks). The skew towards interpretation tasks and tasks from contract law can be explained by the ubiquity of legal documents from these areas (e.g., contracts, terms-of-service agreements, disclosures, and etc.) and their immediate commercial implications [60, 74].

**Language variation**    Legal language is highly heterogeneous, varying in sentence structure, vocabulary, and rhetorical style across different legal areas and document types [58]. This poses a distinct challenge for LLMs, which are extremely sensitive to structure of input text and the vocabulary used [78]. LEGALBENCH tasks are drawn from a diverse set of legal language types, thus enabling researchers to study performance variation across different categories of legal text. Specifically, LEGALBENCH encompasses tasks with language drawn from plain English (32 tasks), legal opinions (11 tasks), merger agreements (34 tasks), contracts (55 tasks), statutory text (3), and other sources.

# 5  Results

We use LEGALBENCH to conduct a three-part study. In the first part (Section 5.2), we evaluate 20 LLMs from 11 different families, at four different size points. In the second part (Section 5.3), we show how LEGALBENCH can be used to conduct in-depth evaluations of models. To illustrate, we use LEGALBENCH to highlight similarities and differences in the performance of three popular commercial models: GPT-4, GPT-3.5, and Claude-1. In the final part (Section 5.4), we show how LEGALBENCH can support the development of law-specific LLM methods. We focus on prompting, and conduct experiments illustrating tradeoffs and challenges with regards to guiding LLMs towards certain tasks. Ultimately, our study here serves to illustrate the types of analyses that LEGALBENCH enables, and highlight potential directions for future work. We summarize the findings of our study here, and provide more details in the Appendix.

## 5.1  Setup

**Models**    We study 20 LLMs from 11 different families. These are: GPT-3.5 [14] (text-davinci-003) and GPT-4 from OpenAI [96]; Claude-1 (v1.3) from Anthropic [3]; Incite-Instruct-7B, Incite-Base-7B, and Incite-Instruct-3B from Together [34, 133]; OPT-2.7B, OPT-6.7B, and OPT-13B from Meta [154]; Falcon-7B-Instruct from TII [2, 103]; MPT-7B-8k-Instruct from MosaicML [129]; Vicuna-7B-16k and Vicuna-13B-16k from LMSYS [26] ; Flan-T5-XL (3B parameters) and Flan-T5-XXL (11B parameters) from Google [31]; LLaMA-2-7B, and LLaMA-2-13B from Meta [134]; WizardLM-13B [150]; and BLOOM-3b and BLOOM-7B from BigScience [116]. All inference was performed on two-GPU GCP 40GB A100s, using the Manifest library [97]. HuggingFace links for each model are provided in the Appendix.

**Prompts**    We designed a prompt for each task by manually writing instructions for the task, and selecting between zero and eight samples from the available train split to use as in-context demonstration. The number of samples selected depended on the availability of data and the sequence length of samples. For application evaluation, we augmented the prompt with an instruction for the LLM to explain its reasoning. We used the same prompt for all models with one exception (Claude-1). LLM outputs were generated using next-token generation at a temperature of 0.0. For classification/extraction tasks, we terminated at a new-line 353 token. For `rule_qa` and all application tasks except `diversity_jurisdiction_6` we generated 150 tokens. For `diversity_jurisdiction_6` we generated 300 tokens.

**Evaluation**    Classification tasks are evaluated using "exact-match" (following HELM [78]). Because some tasks contain significant label imbalances, we use balanced-accuracy as a metric. For extraction tasks, we perform normalization on generated outputs to account for differences in tense/casing/punctuation. A few tasks (e.g., `successor_liability` and `ssla_individual_defendants`) requires the LLM to produce multiple classes or extracted terms per instance. For these, we evaluate using F1. Rule-application tasks were evaluated manually by a law-trained individual, who analyzed LLM responses for both correctness and analysis. The

Appendix provides more details on our approach to manual grading. All manual evaluation was performed with reference to a grading guide, which we additionally make available.

## 5.2 Performance trends

| LLM | Issue | Rule | Conclusion | Interpretation | Rhetorical |
|---|---|---|---|---|---|
| GPT-4 | 82.9 | 59.2 | 89.9 | 75.2 | 79.4 |
| GPT-3.5 | 60.9 | 46.3 | 78.0 | 72.6 | 66.7 |
| Claude-1 | 58.1 | 57.7 | 79.5 | 67.4 | 68.9 |
| Flan-T5-XXL | 66.0 | 36.0 | 63.3 | 64.4 | 70.7 |
| LLaMA-2-13B | 50.2 | 37.7 | 59.3 | 50.9 | 54.9 |
| OPT-13B | 52.9 | 28.4 | 45.0 | 45.1 | 43.2 |
| Vicuna-13B-16k | 34.3 | 29.4 | 34.9 | 40.0 | 30.1 |
| WizardLM-13B | 24.1 | 38.0 | 62.6 | 50.9 | 59.8 |
| BLOOM-7B | 50.6 | 24.1 | 47.2 | 42.8 | 40.7 |
| Falcon-7B-Instruct | 51.3 | 25.0 | 52.9 | 46.3 | 44.2 |
| Incite-7B-Base | 50.1 | 36.2 | 47.0 | 46.6 | 40.9 |
| Incite-7B-Instruct | 54.9 | 35.6 | 52.9 | 54.5 | 45.1 |
| LLaMA-2-7B | 50.2 | 33.7 | 55.9 | 47.7 | 47.7 |
| MPT-7B-8k-Instruct | 54.3 | 25.9 | 48.9 | 42.1 | 44.3 |
| OPT-6.7B | 52.4 | 23.1 | 46.3 | 48.9 | 42.2 |
| Vicuna-7B-16k | 3.9 | 14.0 | 35.6 | 28.1 | 14.0 |
| BLOOM-3B | 47.4 | 20.6 | 45.0 | 45.0 | 36.4 |
| Flan-T5-XL | 56.8 | 31.7 | 52.1 | 51.4 | 67.4 |
| Incite-3B-Instruct | 51.1 | 26.9 | 47.4 | 49.6 | 40.2 |
| OPT-2.7B | 53.7 | 22.2 | 46.0 | 44.4 | 39.8 |

Table 2: Average performance for each LLM over the different LEGALBENCH categories. The first block of rows corresponds to large commercial models, the second block corresponds to models in the 11B-13B range, the third block corresponds to models in the 6B-7B range, and the final block corresponds to models in the 2B-3B range. The columns correspond to (in order): issue-spotting, rule-recall, rule-conclusion, interpretation, and rhetorical-understanding. For each class of models (large, 13B, 7B, and 3B), the best performing model in each category of reasoning is underlined.

Table 2 provides the average task performance for all 20 models in five reasoning categories . The first block of rows corresponds to large commercial models, the second block corresponds to models in the 11B-13B range, the third block corresponds to models in the 6B-7B range, and the final block corresponds to models in the 2B-3B range. Table 3 provides the average task performance for the three large models on rule-application. The Appendix provides full results for each model on each task.

We highlight several observations. First, within LLM families, larger models usually outperform smaller models. For instance, Flan-T5-XXL (11B parameters) outperforms 372 Flan-T5-XL (3B parameters) on average across all five reasoning categories, and LLaMA-2-13B outperforms 373 LLaMA-2-7B on average across four reasoning categories. Second, even for LLMs of the same size, we find considerable differences in performance. For instance, we observe significant gaps in performance between Flan-T5-XXL (11B parameters) and Vicuna-13B-16k (13B parameters), across all reasoning categories. This suggests, unsurprisingly, that the choice of pretraining data, regime of instruction-tuning, and architecture play an important role in determining performance, and that certain configurations may be better aligned for LEGALBENCH tasks. Interestingly, such choices may affect which types of reasoning categories LLMs appear to perform well at. For instance, WizardLM-13B performs worse than all peers on issue-spotting tasks, best on rule-recall tasks, and nearly matches the performance of the best-performing peer on rule-conclusion tasks. Third, we find evidence that open-source models are

| LLM | Correctness | Analysis |
|---|---|---|
| GPT-4 | 82.2 | 79.7 |
| GPT-3.5 | 58.5 | 44.2 |
| Claude-v1 | 61.4 | 59.0 |

Table 3: Average performance for the large LLMs on rule-application tasks.

capable of performance that matches or exceeds certain commercial models. For instance, Flan-T5-XXL outperforms GPT-3.5 and Claude-1 383 on two categories (issue-spotting and rhetorical-understanding), despite the relative gap in parameter count. Notably, the gap between closed and open-source models is largest for the rule-conclusion category. Amongst LEGALBENCH tasks, rule-conclusion tasks most like the other types of multi-step/common-sense reasoning tasks where commercial LLMs have been found to perform well.

## 5.3 Comparing commercial models

Next, we conduct a more in-depth study of performance, focusing on the three commercial models (GPT-4, GPT-3.5, and Claude-1). In particular we highlight how LEGALBENCH can provide more rigorous empirical support for anecdotal observations arising out of the legal community's use of these models, and explain performance differences between models. The Appendix provides a more in-depth analysis.

We highlight two major findings. First, evaluation on LEGALBENCH reveals that the largest performance difference between GPT-4 and GPT-3.5/Claude-1 occurs (on average) for rule-application tasks. On rule application tasks, we observe that GPT-4 outperforms both GPT-3.5 ($p < 0.01$) and Claude-1 ($p < 0.01$) on both correctness and analysis. Across LLMs, we find that variation in performance across tasks is consistent with subjective impressions of task difficulty. For instance, performance on `diversity_jurisdiction_1` (an easy task requiring a model to determine if an amount is greater than $75k and if the plaintiff and defendant are from different states) is much higher than performance on `successor_liability` (a harder task requiring a model to identify multiple successor liability exceptions in a fact pattern describing a complex transaction). Second, on the interpretation tasks, we find that on average GPT-4 outperforms GPT-3.5 ($p < 0.01$), and GPT-3.5 outperforms Claude-1 ($p < 0.01$). Here, the larger API-models are highly performant on tasks which involve binary classification over short clauses. Averaged across the 38 CUAD tasks (contract clauses), for instance, GPT-4, GPT-3.5, and Claude-1 all have a balanced-accuracy $\geq 88\%$. And on `proa` (statutory clauses), both GPT-4 and GPT-3.5 have a balanced-accuracy $\geq 90\%$. Notably, performance degrades on tasks which contain longer text sequences or involve multi-class classification. On the Supply Chain Disclosure tasks for instance—in which LLMs must classify disclosures which are 1-2 pages in length—the average balanced-accuracy of the large commercial models ranges between $74 - 75\%$. And on the MAUD tasks—which require answering multiple choice questions about merger deals—the average balanced-accuracy of GPT-4 drops to $47.8\%$ accuracy.

## 5.4 Prompt engineering strategies

Finally, we illustrate—through a series of micro-studies—how LEGALBENCH can be used to explore different aspects of prompt-engineering for LLMs in legal settings. We focus on three questions: (1) Can LLMs rely on their latent knowledge of a rule for rule-conclusion tasks? (2) Does simplifying task descriptions to plain language affect performance? (3) Are LLMs sensitive to the choice of in-context demonstrations?

We highlight our findings (full results can be found in the Appendix). With regards to (1), we find considerable variation over studied tasks—some rules appear to be sufficiently well known that explaining the rule to the LLM in the prompt has a negligible impact on performance. With regards to (2), we find that a plain-language prompt significantly outperforms the technical language prompt, by up to 21 points (balanced-accuracy) on the subset of studied tasks. And with regards to (3), we find that LLMs are highly sensitive to the choice of in-context samples, with performance differences of up to 20pts (balanced-accuracy). These results illustrate that further work on understanding how to effectively and efficiently prompt LLMs for legal tasks is needed.

# 6 Conclusion

Our work here describes LEGALBENCH: a collaboratively constructed benchmark of 162 tasks for measuring the legal reasoning capabilities of LLMs. In future work, we hope to expand this project, by continuing to solicit and collect interesting and useful tasks from the legal community

## Acknowledgments and Disclosure of Funding

We are grateful to the following individuals and groups for feedback on this project: Alex Chao, Amit Haim, Arjun Desai, Armin Thomas, Avanika Narayan, Ben Spector, Brandon Yang, Eric Nguyen, Gautam Machiraju, Javed Qadrud-Din, Jian Zhang, Jonathan Zittrain, Karan Goel, Khaled Saab, Joshua Arp, Krista Opsahl-Ong, Laurel Orr, Lisa Ouellette, Lucia Zheng, Martin Gajek, Mayee Chen, Michael Zhang, Mike Wornow, Pablo Arredondo, Percy Liang, Rishi Bommasani, Roland Vogl, Sabri Eyuboglu, Sarah Hooper, Sergio Servantez, Simran Arora, Tengyu Ma, Tony Kim, Tri Dao and Vishnu Sarukkai. We presented and recieved feedback on earlier versions of this project at various forums, including: the Center for Research on Foundation Models, the Stanford Regulation and Governance Lab, the New York LLM x Law Hackathon (June 2023), the 2023 Stanford Data Science Conference, the 2023 Stanford CodeX Conference, and the Stanford Generative AI and Foundation Models Workshop. We are grateful to the organizers and attendees of these events for engaging with our work.

We gratefully acknowledge the support of NIH under No. U54EB020405 (Mobilize), NSF under Nos. CCF1763315 (Beyond Sparsity), CCF1563078 (Volume to Velocity), 2204926 (Computational Statutory Reasoning), and 1937301 (RTML); US DEVCOM ARL under No. W911NF-21-2-0251 (Interactive Human-AI Teaming); ONR under No. N000141712266 (Unifying Weak Supervision); ONR N00014-20-1-2480: Understanding and Applying Non-Euclidean Geometry in Machine Learning; N000142012275 (NEPTUNE); NXP, Xilinx, LETI-CEA, Intel, IBM, Microsoft, NEC, Toshiba, TSMC, ARM, Hitachi, BASF, Accenture, Ericsson, Qualcomm, Analog Devices, Google Cloud, Salesforce, Total, the Center for Research on Foundation Models, the HAI-GCP Cloud Credits for Research program, the Stanford Data Science Initiative (SDSI), and members of the Stanford DAWN project: Facebook, Google, and VMWare. We thank Casetext for assistance with evaluating GPT-4. PH is supported by an Open Philanthropy AI Fellowship. The U.S. Government is authorized to reproduce and distribute reprints for Governmental purposes notwithstanding any copyright notation thereon. Any opinions, findings, and conclusions or recommendations expressed in this material are those of the authors and do not necessarily reflect the views, policies, or endorsements, either expressed or implied, of NIH, ONR, or the U.S. Government.

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

# A  Outline

For the sake of clarity, we provide a brief outline of the Appendix.

- Appendix B provides more information on the availability of LEGALBENCH.
- Appendix C discusses limitations of the project and potential negative social impacts.
- Appendix D provides a top-level datasheet.
- Appendix E provides an overview of the LEGALBENCH tasks, and contains information on licenses, which datasets have been published before, evaluation protocols, and dataset statistics.
- Appendix G provides a description of each task/task-family. For each set of tasks, we provide background on the relevant legal concepts, describe the tasks, discuss how the task was constructed, provide examples of inputs/outputs, and explain their legal significance.
- Appendix H provides the full-results for evaluated LLMs on all LEGALBENCH tasks, along with additional information about prompting.

# B  Availability

- The LEGALBENCH website is available at `https://hazyresearch.stanford.edu/legalbench/`.
- Data can be downloaded from Huggingface here: `https://huggingface.co/datasets/nguha/legalbench`.
- The Github repository with evaluation code and prompts is available here: `https://github.com/HazyResearch/legalbench/`

Gradebooks to guide manual evaluation can be found on the website.

# C  Limitations and social impact

**Limitations**   We note several limitations of our work. Legal applications—and what constitutes "legal reasoning"—is broad. Thus, LEGALBENCH will necessarily be an incomplete effort, and important tasks/document types/reasoning types are not included. To enumerate a few examples:

- LEGALBENCH does not include tasks over *long* documents. Long documents are significant for legal practice, as writings like contracts, corporate filings, statutory codes, and judicial opinions can be hundreds of pages long [77].
- The legal reasoning dimensions identified in LEGALBENCH constitute a subset of the possible legal reasoning abilities for which we wish to evaluate LLMs. An example of a reasoning ability which is not currently evaluated in LEGALBENCH would be analogical reasoning grounded in case law.
- LEGALBENCH tasks are skewed towards certain legal domains (e.g., contracts and civil procedure) and others are unrepresented.
- LEGALBENCH tasks skew towards US Federal law, and thus may not be representative for studies of other jurisdictions, or tasks involving international law.
- LEGALBENCH does not enable evaluation for multilingual, or non-English, legal tasks.
- LEGALBENCH does not evaluate more subjective legal tasks, or tasks which contain more ambiguity. These tasks are common to the legal field.

We hope to work on these limitations as part of future work. In particular, we would like to expand LEGALBENCH to include other jurisdictions and a broader cross-section of legal domains.

Nothing in LEGALBENCH should be construed as legal advice.

**Social impact**   A potential negative social impact of our work would be if others either (1) construed our work as unequivocally endorsing automation in the legal industry, or (2) used performance on LEGALBENCH as the sole justification for AI deployments. We therefore take efforts to mitigate these impacts, noting the following.

As we state in Section 1, the purpose of our work is not to determine whether large language models are capable of replacing legal professionals, the types of legal work that should/can be automated, or the broader implications of new technology on the practice of law. Rather, our focus is on developing technical artifacts which better enable stakeholders and affected parties to answer these questions themselves. Rigorous evaluation is essential to the safe and ethical usage of AI. LEGALBENCH, as a benchmark, is intended to *improve* the ability for

stakeholders to conduct evaluations. We additionally note that LEGALBENCH, as a tool for research, is not a substitute for more in-depth and context-specific evaluation efforts. The deployment of any AI application in the law must be accompanied by evaluation on in-domain data, and assessments for ethical and legal compliance.

We finally note that potential negative impact will depend significantly on the task studied and the broader social context. The consequences of mistakes in using LLMs to annotate datasets, for instance, has significantly different consequences from the cost of mistakes when LLMs are used to answer legal aid questions.

# D  Datasheet

Following recent work, we provide a datasheet [54] below. The datasheet below provides general answers to each of the questions, while Appendix G provides more in-depth details for each individual task. In addition, a number of LEGALBENCH tasks have been adapted from previously released datasets, and the datasheets accompanying their publication provide further details.

## D.1  Motivation

**For what purpose was the data set created? Was there a specific task in mind? If so, please specify the result type (e.g. unit) to be expected.**

LEGALBENCH was created to evaluate LLMs on legal tasks and better understand their legal reasoning capabilities. Recent advances in language modeling techniques have led to the emergence of "large" language models, and spurred interest within the legal community. This has led to two questions:

- What technical adaptations are necessary to enable LLMs to perform legal tasks? Legal tasks often involve longer text sequences, jargon, and multi-step reasoning, making them more difficult than traditional NLP tasks.
- For which legal tasks can current LLMs be trusted to perform safely and reliably?

LEGALBENCH encompasses many different tasks. The specification for each task and the expected output can be found in the full task descriptions (Section G).

**Who created the dataset (e.g., which team, research group) and on behalf of which entity (e.g. company, institution, organization)?**

LEGALBENCH consists of novel datasets (which were created by the authors of this paper), and transformed/adapted datasets (which were originally released as part of prior research). In Section G we discuss the origins of each dataset.

**Who funded the creation of the dataset? If there is an associated grant, please provide the name of the grantor and the grant name and number.**

LEGALBENCH and its contributors have been generously funded by a range of entities that include the institutional affiliations provided for each author, governmental grants, and other sources.

**Any other comments?**

None.

## D.2  Composition

**What do the instances that comprise the dataset represent (e.g., documents, photos, people, countries)? Are there multiple types of instances (e.g., movies, users, and ratings; people and interactions between them; nodes and edges)? Please provide a description.**

All LEGALBENCH tasks consist of instances which are text. These include: sentences, paragraphs, and documents. Some instances are drawn from real world sources of text (e.g., actual contracts, corporate disclosures, judicial opinions, or complaints). Other instances were synthetically crafted. Section G provides details for each task.

**How many instances are there in total (of each type, if appropriate)?**

Section E provides details for each task.

**Does the dataset contain all possible instances or is it a sample (not necessarily random) of instances from a larger set? If the dataset is a sample, then what is the larger set? Is the sample representative of the larger set (e.g., geographic coverage)? If so, please describe how this representativeness was validated/verified. If it is not representative of the larger set, please describe why not (e.g., to cover a more diverse range of instances, because instances were withheld or unavailable).**

Nearly every LEGALBENCH task corresponds to a sample of a population, or entirely synthetic data. Section G contains a more detailed description for each dataset. We highlight several broader explanations for the difficulty in acquiring complete or representative data which generalizes across tasks:

- As prior work on legal benchmarks has noted [58, 118], not all legal documents are published or reported. Hence, many are only accessible through special request, or only available in paper. The lack of easily available representative data is a noted challenge in many justice systems [58, 100].

- Acquiring legal annotations is exceedingly expensive. The CUAD project, for instance, estimated that a modestly sized dataset of 500 contracts (relative to the standards of NLP) had an estimated cost of $2 million US dollars [60]. As a result, it is often possible to only annotate a small sample of data, even when a larger population is available.

**What data does each instance consist of? "Raw" data (e.g., unprocessed text or images) or features? In either case, please provide a description.**

Instances in LEGALBENCH largely correspond to unprocessed text. Section G contains a more detailed description for each dataset.

**Is there a label or target associated with each instance? If so, please provide a description.**

Yes. Labels correspond to: classes, extracted entities, and open-ended generation. Section G contains a more detailed description of the labels/targets for each dataset.

**Is any information missing from individual instances? If so, please provide a description, explaining why this information is missing (e.g., because it was unavailable). This does not include intentionally removed information, but might include, e.g., redacted text.**

For reused/adapted datasets, we refer readers to the original data sheets which document redactions/missing data. Newly contributed tasks should not be missing information.

**Are relationships between individual instances made explicit (e.g., users' movie ratings, social network links)? If so, please describe how these relationships are made explicit.**

Not applicable.

**Are there recommended data splits (e.g.,training, development/validation,testing)? If so, please provide a description of these splits, explaining the rationale behind them.**

Yes. Tasks are split into train and test splits. Train splits consist of a small random sample of the original dataset (i.e., between 2-8 instances). We select small training splits in order to capture the true few-shot setting [105], in which a practitioner only has access to a handful of labeled instances. This design choice is also reflected in the structure of the RAFT benchmark [1].

**Are there any errors, sources of noise, or redundancies in the dataset? If so, please provide a description.**

A significant amount of legal data is the product of scanning and OCR. Hence, this data often contains artifacts of these processes, which appear as errant or missing characters.

**Is the dataset self-contained, or does it link to or otherwise rely on external resources (e.g., websites, tweets, other datasets)? If it links to or relies on external resources, a) are there guarantees that they will exist, and remain constant, over time; b) are there official archival versions of the complete dataset (i.e., including the external resources as they existed at the time the dataset was created); c) are there any restrictions (e.g., licenses, fees) associated with any of the external resources that might apply to a future user? Please provide descriptions of all external resources and any restrictions associated with them, as well as links or other access points, as appropriate.**

LEGALBENCH is self-contained.

**Does the dataset contain data that might be considered confidential (e.g., data that is protected by legal privilege or by doctor–patient confidentiality, data that includes the content of individuals' non-public communications)? If so, please provide a description.**

No. All LEGALBENCH data is derived from public sources or was generated by authors. There is no confidential information in our dataset.

**Does the dataset contain data that, if viewed directly, might be offensive, insulting, threatening, or might otherwise cause anxiety? If so, please describe why.**

No.

**Does the dataset relate to people? If not, you may skip the remaining questions in this section.**

LEGALBENCH data relates to people to the extent that LEGALBENCH contains tasks which contain language drawn from judicial cases involving individuals, or posts by individuals to legal forums (i.e., the Learned Hands Tasks).

**Does the dataset identify any subpopulations (e.g., by age, gender)? If so, please describe how these subpopulations are identified and provide a description of their respective distributions within the dataset.**

No.

**Is it possible to identify individuals (i.e., one or more natural persons), either directly or indirectly (i.e., in combination with other data) from the dataset? If so, please describe how.**

As LEGALBENCH is drawn entirely from public datasets—which themselves may contain additional information—it is possible to identify the original documents that LEGALBENCH data was drawn from.

**Does the dataset contain data that might be considered sensitive in any way (e.g., data that reveals racial or ethnic origins, sexual orientations, religious beliefs, political opinions or union memberships, or locations; financial or health data; biometric or genetic data; forms of government identification, such as social security numbers; criminal history)? If so, please provide a description.**

The Learned Hands tasks correspond to posts on public forums. In these posts individuals discuss legal questions, and sometimes disclose information that would meet the above definition of "sensitive."

**Any other comments?**

We note that the data distributions from which some LEGALBENCH tasks were drawn—like judicial cases or legal forums—have been used by prior work published in the NeurIPS Datasets and Benchmarks Track [58, 118]. These works offer additional information.

## D.3 Collection process

**How was the data associated with each instance acquired? Was the data directly observable (e.g., raw text, movie ratings), reported by subjects (e.g., survey responses), or indirectly inferred/derived from other data (e.g., part-of-speech tags, model-based guesses for age or language)? If data was reported by subjects or indirectly inferred/derived from other data, was the data validated/verified? If so, please describe how.**

Data underlying LEGALBENCH tasks were collected using different processes, and Section G contains a detailed discussion for each task.

**What mechanisms or procedures were used to collect the data (e.g., hardware apparatus or sensor, manual human curation, software program, software API)? How were these mechanisms or procedures validated?**

Please refer to Section G for background on each task.

**If the dataset is a sample from a larger set, what was the sampling strategy (e.g., deterministic, probabilistic with specific sampling probabilities)?**

Please see the discussion in the Composition section above.

**Who was involved in the data collection process (e.g., students, crowdworkers, contractors) and how were they compensated (e.g., how much were crowdworkers paid)?**

Section G contains a detailed discussion for each task.

**Over what timeframe was the data collected? Does this timeframe match the creation timeframe of the data associated with the instances (e.g., recent crawl of old news articles)? If not, please describe the timeframe in which the data associated with the instances was created.**

Section G contains a detailed discussion for each task.

**Were any ethical review processes conducted (e.g., by an institutional review board)? If so, please provide a description of these review processes, including the outcomes, as well as a link or other access point to any supporting documentation.**

Where applicable, Section G provides information relevant to each task.

**Does the dataset relate to people? If not, you may skip the remaining questions in this section.**

The dataset relates to people insofar as it draws text from documents which relate to people, or people created.

**Did you collect the data from the individuals in question directly, or obtain it via third parties or other sources (e.g., websites)?**

Section G contains a detailed discussion for each task.

**Were the individuals in question notified about the data collection? If so, please describe (or show with screenshots or other information) how notice was provided, and provide a link or other access point to, or otherwise reproduce, the exact language of the notification itself.**

No. Following other works which incorporate data from public judicial sources [58, 118], we note that judicial filings are public, and the individuals implicated in those proceedings are aware of the public nature.

**Did the individuals in question consent to the collection and use of their data? If so, please describe (or show with screenshots or other information) how consent was requested and provided, and provide a link or other access point to, or otherwise reproduce, the exact language to which the individuals consented.**

Individuals whose names and circumstances appear in the original datasets did not separately consent to be a part of LEGALBENCH. Again, we note that these documents are generally public, and already accessible to a wide range of parties, through many different judicial data services.

**If consent was obtained, were the consenting individuals provided with a mechanism to revoke their consent in the future or for certain uses? If so, please provide a description, as well as a link or other access point to the mechanism (if appropriate).**

N/A.

**Has an analysis of the potential impact of the dataset and its use on data subjects (e.g., a data protection impact analysis) been conducted? If so, please provide a description of this analysis, including the outcomes, as well as a link or other access point to any supporting documentation.**

No.

**Any other comments?**

None.

## D.4    Preprocessing, cleaning, labeling

**Was any preprocessing/cleaning/labeling of the data done (e.g., discretization or bucketing, tokenization, part-of-speech tagging, SIFT feature extraction, removal of instances, processing of missing values)? If so, please provide a description. If not, you may skip the remainder of the questions in this section.**

No.

## D.5    Use

**Has the dataset been used for any tasks already? If so, please provide a description.**

We have used the constructed datasets to evaluate several LLMs.

**Is there a repository that links to any or all papers or systems that use the dataset? If so, please provide a link or other access point.**

LEGALBENCH is available at `https://github.com/HazyResearch/legalbench/`.

**What (other) tasks could the dataset be used for?**

We envision this dataset could be used for the following:

- Evaluation of LLMs.
- Finetuning LLMs, either on task data directly, or self-instruct style generations derived from task data.

**Is there anything about the composition of the dataset or the way it was collected and preprocessed/cleaned/labeled that might impact future uses? For example, is there anything that a future user might need to know to avoid uses that could result in unfair treatment of individuals or groups (e.g., stereotyping, quality of service issues) or other undesirable harms (e.g., financial harms, legal risks) If so, please provide a description. Is there anything a future user could do to mitigate these undesirable harms?**

We emphasize that LEGALBENCH—like all generalized benchmarks—can offer only a preliminary understanding of LLM performance. LEGALBENCH tasks do not generalize to all legal reasoning tasks or all types of legal documents. We thus emphasize that practitioners seeking to deploy LLMs within their own applications should perform their own data collection and validation specific to their use case.

**Are there tasks for which the dataset should not be used? If so, please provide a description.**

These datasets should not be used to predict the legality of real world events, the outcome of lawsuits, or as legal advice.

## D.6 Distribution

**Will the dataset be distributed under a copyright or other intellectual property (IP) license, and/or under applicable terms of use (ToU)? If so, please describe this license and/or ToU, and provide a link or other access point to, or otherwise reproduce, any relevant licensing terms or ToU, as well as any fees associated with these restrictions.**

Table 4 provides the license that applies to each individual LEGALBENCH task.

**Have any third parties imposed IP-based or other restrictions on the data associated with the instances? If so, please describe these restrictions, and provide a link or other access point to, or otherwise reproduce, any relevant licensing terms, as well as any fees associated with these restrictions.**

Yes. Tasks which consist of adapted/transformed data are released under the same license as the original dataset. Table 4 provides these licenses, and Section G provides a reference to the original dataset for transformed tasks.

**Do any export controls or other regulatory restrictions apply to the dataset or to individual instances? If so, please describe these restrictions, and provide a link or other access point to, or otherwise reproduce, any supporting documentation.**

No.

## D.7 Maintenance

**Who will be supporting/hosting/maintaining the dataset?**

Neel Guha will be supporting this dataset.

**How can the owner/curator/manager of the dataset be contacted (e.g., email address)?**

Neel Guha can be reached at nguha@cs.stanford.edu. He will be available to answer any questions.

**Is there an erratum? If so, please provide a link or other access point.**

We have currently not found any, but will make them available on the website.

**Willthe dataset be updated (e.g.,to correct labeling errors, add new instances, delete instances)? If so, please describe how often, by whom, and how updates will be communicated to users (e.g., mailing list, GitHub)?**

Yes. There will be two types of updates to LEGALBENCH:

- First, we will update LEGALBENCH to reflect new contributions from the legal community.
- Second, we will update LEGALBENCH to reflect identified errors in the data.

We will strive to make and publicize updates as soon as errors are identified and new tasks are contributed. Neel Guha will be in charge of managing these updates.

**If the dataset relates to people, are there applicable limits on the retention of the data associated with the instances (e.g., were individuals in question told thattheir data would be retained for a fixed period of time and then deleted)? If so, please describe these limits and explain how they will be enforced.**

N/A.

**Will older versions ofthe dataset continue to be supported/hosted/maintained? If so, please describe how. If not, please describe how its obsolescence will be communicated to users.**

Yes. We will make older versions available on request by email.

**If others want to extend/augment/build on/contribute to the dataset, is there a mechanism for them to do so? If so, please provide a description. Will these contributions be validated/verified? If so, please describe how. If not, why not? Is there a process for communicating/distributing these contributions to other users?If so, please provide a description.**

Yes. We encourage members of the legal community to contribute new tasks. We are in the process of formalizing procedures for reviewing, validating, and incorporating submissions.

We additionally note that many of the LEGALBENCH tasks are available under permissive licenses, and other researchers may thus modify them.

# E Task overview

## E.1 Licenses

LEGALBENCH tasks are subject to different licenses, due to the choices of dataset contributors, or the license under which the original data was released. Table 4 summarizes the licenses. The authors bear all responsibility in case of violation of rights, and confirm the dataset licenses.

| License | Tasks |
|---|---|
| Creative Commons Attribution 4.0 | Abercrombie, CUAD Tasks, Citation Prediction Tasks, Contract NLI Tasks, Contract QA, Corporate Lobbying, Diversity Tasks, Function of Decision Section, Hearsay, Insurance Policy Interpretation, International Citizenship Questions, J.Crew Blocker, Legal Reasoning Causality, MAUD Tasks, Oral Argument Question Purpose, Overruling, Personal Jurisdiction, Private Right of Action, Rule QA, SCALR, Securities Complaint Extraction Tasks, Successor Liability, Supply Chain Disclosure Tasks, Telemarketing Sales Rule, UCC v. Common Law, Unfair Terms of Service |
| Attribution-NonCommercial 4.0 International | Canada Tax Court Outcomes, Consumer Contracts QA, Textualism Tools |
| Attribution-ShareAlike 4.0 International | Definition Tasks |
| Attribution-NonCommercial-ShareAlike 4.0 International | Learned Hands Tasks |
| MIT | New York State Judicial Ethics, Privacy Policy QA, SARA Tasks |
| Creative Commons Attribution-NonCommercial License | OPP-115 Tasks |
| Attribution-NonCommercial 3.0 Unported | Privacy Policy Entailment |

Table 4: Licenses

## E.2 Public availability status

Given that many commercially available LLMs are trained on the "entirety of the web"—and little is known as to how they are trained—there are concerns that many benchmarks have inadvertently become part of the training data for these models. Therefore, this section identifies and organizes LEGALBENCH tasks into three categories:

- Previously published tasks, which were derived from datasets that were initially published as part of other works and available on the web for download.
- Original but available tasks, which are original creations of the LEGALBENCH project but previously made available online.
- Original and unavailable tasks, which are original creations of the LEGALBENCH project but have not been released online.

Table 5 summarizes the availability status of each of the tasks.

## E.3 Reasoning type

Table 6 organizes tasks by the LEGALBENCH reasoning type they can be used to assess. Table 7 similarly organizes tasks according to reasoning-types recognized in the NLP literature. For each reasoning type, we provide examples of general-domain NLP benchmarks which are similar. For more information on the types of reasoning required for each task, please see the individual task descriptions provided in Appendix G.

| Publication status | Tasks |
|---|---|
| Previously published tasks | CUAD Tasks, Contract NLI Tasks, MAUD Tasks, OPP-115 Tasks, Overruling, Privacy Policy Entailment, Privacy Policy QA, SARA Tasks, Unfair Terms of Service |
| Original but available tasks | Abercrombie, Diversity Tasks, Hearsay, International Citizenship Questions, Learned Hands Tasks, New York State Judicial Ethics, Personal Jurisdiction, Private Right of Action, Rule QA |
| Original and unavailable tasks | Canada Tax Court Outcomes, Citation Prediction Tasks, Consumer Contracts QA, Contract QA, Corporate Lobbying, Definition Tasks, Function of Decision Section, Insurance Policy Interpretation, J.Crew Blocker, Legal Reasoning Causality, Oral Argument Question Purpose, SCALR, Securities Complaint Extraction Tasks, Successor Liability, Supply Chain Disclosure Tasks, Telemarketing Sales Rule, Textualism Tools, UCC v. Common Law |

Table 5: Task publication status

| LEGALBENCH reasoning type | Tasks |
|---|---|
| Issue-spotting | Corporate Lobbying, Learned Hands Tasks |
| Rule-recall | Citation Prediction Tasks, International Citizenship Questions, New York State Judicial Ethics, Rule QA |
| Rule-application | Abercrombie, Diversity Tasks, Hearsay, Personal Jurisdiction, Successor Liability, Telemarketing Sales Rule, UCC v. Common Law |
| Rule-conclusion | Abercrombie, Diversity Tasks, Hearsay, Personal Jurisdiction, Successor Liability, Telemarketing Sales Rule, UCC v. Common Law |
| Interpretation | CUAD Tasks, Consumer Contracts QA, Contract NLI Tasks, Contract QA, Insurance Policy Interpretation, J.Crew Blocker, MAUD Tasks, OPP-115 Tasks, Privacy Policy Entailment, Privacy Policy QA, Private Right of Action, SARA Tasks, Securities Complaint Extraction Tasks, Supply Chain Disclosure Tasks, Unfair Terms of Service |
| Rhetorical-understanding | Canada Tax Court Outcomes, Definition Tasks, Function of Decision Section, Legal Reasoning Causality, Oral Argument Question Purpose, Overruling, SCALR, Textualism Tools |

Table 6: Tasks by LEGALBENCH reasoning type.

## E.4  Task statistics

Table 9 provides statistics for the LEGALBENCH tasks. For each task, we list the number of samples and the average length (in words) of each input. LEGALBENCH encompasses tasks ranging from short (a single sentence)

| NLP reasoning type | Tasks |
|---|---|
| Knowledge (e.g., MMLU [59], WikiFact in HELM [78, 106]) | Citation Prediction Tasks, International Citizenship Questions, New York State Judicial Ethics, Rule QA |
| Linguistic inference (e.g., CoLA [142]) | Canada Tax Court Outcomes, Definition Tasks, Function of Decision Section, Legal Reasoning Causality, Oral Argument Question Purpose, Overruling, Textualism Tools |
| Topic classification (e.g., RAFT [1]) | Corporate Lobbying, Learned Hands Tasks, CUAD Tasks, J.Crew Blocker, OPP-115 Tasks, Private Right of Action, Unfair Terms of Service |
| Entailment (e.g., RTE [138]) | Contract NLI Tasks, Privacy Policy Entailment, Privacy Policy QA, Insurance Policy Interpretation, SARA Tasks |
| Arithmetic (e.g., GSM8K [33]) | Diversity Tasks, SARA Tasks |
| Multi-step reasoning (e.g., STREET [111]) | Abercrombie, Hearsay, Personal Jurisdiction, Successor Liability, Telemarketing Sales Rule, UCC v. Common Law |
| Document-based QA (e.g., BoolQ [32]) | Consumer Contracts QA, Contract QA, MAUD Tasks, Supply Chain Disclosure Tasks |
| Named entity recognition (e.g., CoNLL-2003 [113]) | Securities Complaint Extraction Tasks |
| Casual reasoning (e.g., CoPA [138]) | SCALR |

Table 7: Tasks by NLP reasoning type. For each reasoning type, we provide examples of general-domain NLP benchmarks which are similar. For more information on the types of reasoning required for each task, please see the individual task descriptions provided in Appendix G.

to longer inputs (two pages of text) (Figure 1). The average LEGALBENCH task contains between 500-600 instances. All tasks consist of at least 50 instances. A more detailed breakdown is available in Table 8.

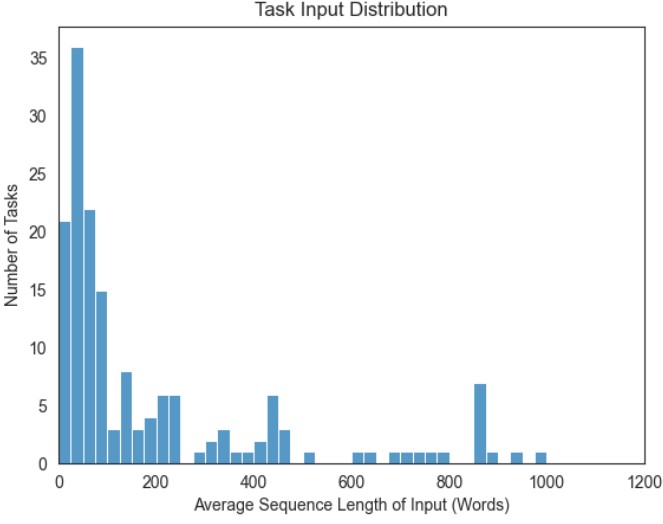

Figure 1: LEGALBENCH task sizes and input text lengths.

| Size range (samples) | Number of tasks |
| --- | --- |
| 50-100 | 28 |
| 100-500 | 97 |
| 500-2000 | 29 |
| 2000+ | 8 |

Table 8: Number of LEGALBENCH tasks at different dataset sizes.

Table 9: Task Statistics

| Task | Number of Samples | Mean Sample Length (Words) |
| --- | --- | --- |
| abercrombie | 100 | 7.1 |
| canada_tax_court_outcomes | 250 | 99.2 |
| citation_prediction_classification | 110 | 35.9 |
| citation_prediction_open | 55 | 32.4 |
| consumer_contracts_qa | 400 | 486.8 |
| contract_nli_confidentiality_of_agreement | 90 | 73.9 |
| contract_nli_explicit_identification | 117 | 75.7 |
| contract_nli_inclusion_of_verbally_conveyed_ information | 147 | 78.1 |
| contract_nli_limited_use | 216 | 63.8 |
| contract_nli_no_licensing | 170 | 65.6 |
| contract_nli_notice_on_compelled_disclosure | 150 | 77.9 |
| contract_nli_permissible_acquirement_of_ similar_information | 186 | 66.6 |
| contract_nli_permissible_copy | 95 | 60.2 |
| contract_nli_permissible_development_of_ similar_information | 144 | 61.0 |
| contract_nli_permissible_post-agreement_ possession | 119 | 81.7 |
| contract_nli_return_of_confidential_ information | 74 | 74.2 |
| contract_nli_sharing_with_employees | 178 | 83.9 |
| contract_nli_sharing_with_third-parties | 188 | 78.9 |
| contract_nli_survival_of_obligations | 165 | 64.6 |
| contract_qa | 88 | 48.3 |
| corporate_lobbying | 500 | 878.1 |
| cuad_affiliate_license-licensee | 204 | 75.9 |
| cuad_affiliate_license-licensor | 94 | 99.7 |
| cuad_anti-assignment | 1178 | 54.6 |
| cuad_audit_rights | 1222 | 53.6 |
| cuad_cap_on_liability | 1252 | 59.8 |
| cuad_change_of_control | 422 | 62.8 |
| cuad_competitive_restriction_exception | 226 | 67.1 |

Table 9 – continued from previous page

| Task | Number of Samples | Mean Sample Length (Words) |
| --- | --- | --- |
| cuad_covenant_not_to_sue | 314 | 64.6 |
| cuad_effective_date | 242 | 44.5 |
| cuad_exclusivity | 768 | 58.0 |
| cuad_expiration_date | 882 | 49.9 |
| cuad_governing_law | 882 | 46.9 |
| cuad_insurance | 1036 | 56.8 |
| cuad_ip_ownership_assignment | 582 | 65.3 |
| cuad_irrevocable_or_perpetual_license | 286 | 72.1 |
| cuad_joint_ip_ownership | 198 | 59.1 |
| cuad_license_grant | 1402 | 63.5 |
| cuad_liquidated_damages | 226 | 57.2 |
| cuad_minimum_commitment | 778 | 57.9 |
| cuad_most_favored_nation | 70 | 68.0 |
| cuad_no-solicit_of_customers | 90 | 61.2 |
| cuad_no-solicit_of_employees | 148 | 66.6 |
| cuad_non-compete | 448 | 60.6 |
| cuad_non-disparagement | 106 | 63.8 |
| cuad_non-transferable_license | 548 | 61.5 |
| cuad_notice_period_to_terminate_renewal | 228 | 57.0 |
| cuad_post-termination_services | 814 | 67.3 |
| cuad_price_restrictions | 52 | 53.6 |
| cuad_renewal_term | 392 | 55.2 |
| cuad_revenue-profit_sharing | 780 | 59.8 |
| cuad_rofr-rofo-rofn | 696 | 64.0 |
| cuad_source_code_escrow | 124 | 64.4 |
| cuad_termination_for_convenience | 436 | 52.0 |
| cuad_third_party_beneficiary | 74 | 42.4 |
| cuad_uncapped_liability | 300 | 70.1 |
| cuad_unlimited-all-you-can-eat-license | 54 | 56.8 |
| cuad_volume_restriction | 328 | 49.0 |
| cuad_warranty_duration | 326 | 55.9 |
| definition_classification | 1345 | 41.0 |
| definition_extraction | 695 | 53.9 |
| diversity_1 | 306 | 16.2 |
| diversity_2 | 306 | 23.4 |
| diversity_3 | 306 | 21.1 |
| diversity_4 | 306 | 23.5 |
| diversity_5 | 306 | 28.3 |
| diversity_6 | 306 | 47.9 |
| function_of_decision_section | 374 | 87.4 |
| hearsay | 100 | 25.3 |

| Task | Number of Samples | Mean Sample Length (Words) |
|---|---|---|
| insurance_policy_interpretation | 138 | 87.2 |
| international_citizenship_questions | 9310 | 33.2 |
| intra_rule_distinguishing | 60 | 33.8 |
| jcrew_blocker | 60 | 167.2 |
| learned_hands_benefits | 72 | 253.2 |
| learned_hands_business | 180 | 225.3 |
| learned_hands_consumer | 620 | 246.3 |
| learned_hands_courts | 198 | 225.6 |
| learned_hands_crime | 694 | 233.7 |
| learned_hands_divorce | 156 | 240.2 |
| learned_hands_domestic_violence | 180 | 262.3 |
| learned_hands_education | 62 | 265.7 |
| learned_hands_employment | 716 | 242.0 |
| learned_hands_estates | 184 | 230.9 |
| learned_hands_family | 2271 | 258.4 |
| learned_hands_health | 232 | 286.8 |
| learned_hands_housing | 4500 | 254.8 |
| learned_hands_immigration | 140 | 232.8 |
| learned_hands_torts | 438 | 272.2 |
| learned_hands_traffic | 562 | 229.6 |
| legal_reasoning_causality | 59 | 245.7 |
| maud_"ability_to_consummate"_concept_is_ subject_to_mae_carveouts | 70 | 688.6 |
| maud_"financial_point_of_view"_is_the_sole_ consideration | 113 | 307.5 |
| maud_accuracy_of_fundamental_target_ r&ws:_bringdown_standard | 176 | 143.7 |
| maud_accuracy_of_target_"general"_r&w:_ bringdown_timing_answer | 182 | 142.5 |
| maud_accuracy_of_target_capitalization_ r&w_(outstanding_shares):_bringdown_ standard_answer | 182 | 142.0 |
| maud_additional_matching_rights_period_ for_modifications_(cor) | 159 | 314.3 |
| maud_application_of_buyer_consent_ requirement_(negative_interim_covenant) | 181 | 85.6 |
| maud_buyer_consent_requirement_(ordinary_ course) | 182 | 121.3 |
| maud_change_in_law:__subject_to_ "disproportionate_impact"_modifier | 100 | 702.6 |
| maud_changes_in_gaap_or_other_ accounting_principles:__subject_to_ "disproportionate_impact"_modifier | 99 | 703.1 |
| maud_cor_permitted_in_response_to_ intervening_event | 101 | 305.6 |

| Task | Number of Samples | Mean Sample Length (Words) |
|---|---|---|
| maud_cor_permitted_with_board_fiduciary_determination_only | 101 | 303.1 |
| maud_cor_standard_(intervening_event) | 85 | 326.1 |
| maud_cor_standard_(superior_offer) | 101 | 308.0 |
| maud_definition_contains_knowledge_requirement_-_answer | 148 | 243.4 |
| maud_definition_includes_asset_deals | 147 | 314.7 |
| maud_definition_includes_stock_deals | 149 | 313.6 |
| maud_fiduciary_exception:__board_determination_standard | 180 | 246.8 |
| maud_fiduciary_exception:_board_determination_trigger_(no_shop) | 180 | 245.1 |
| maud_fls_(mae)_standard | 78 | 705.1 |
| maud_general_economic_and_financial_conditions:_subject_to_"disproportionate_impact"_modifier | 99 | 704.6 |
| maud_includes_"consistent_with_past_practice" | 182 | 122.7 |
| maud_initial_matching_rights_period_(cor) | 159 | 313.4 |
| maud_initial_matching_rights_period_(ftr) | 133 | 336.4 |
| maud_intervening_event_-_required_to_occur_after_signing_-_answer | 148 | 242.2 |
| maud_knowledge_definition | 168 | 334.8 |
| maud_liability_standard_for_no-shop_breach_by_target_non-d&o_representatives | 157 | 38.1 |
| maud_ordinary_course_efforts_standard | 182 | 122.7 |
| maud_pandemic_or_other_public_health_event:__subject_to_"disproportionate_impact"_modifier | 99 | 707.5 |
| maud_pandemic_or_other_public_health_event:_specific_reference_to_pandemic-related_governmental_responses_or_measures | 99 | 707.5 |
| maud_relational_language_(mae)_applies_to | 91 | 705.5 |
| maud_specific_performance | 179 | 96.8 |
| maud_tail_period_length | 180 | 95.1 |
| maud_type_of_consideration | 173 | 128.2 |
| nys_judicial_ethics | 300 | 25.7 |
| opp115_data_retention | 96 | 31.5 |
| opp115_data_security | 1342 | 38.6 |
| opp115_do_not_track | 118 | 37.1 |
| opp115_first_party_collection_use | 2094 | 32.6 |
| opp115_international_and_specific_audiences | 988 | 52.1 |
| opp115_policy_change | 439 | 33.2 |
| opp115_third_party_sharing_collection | 1598 | 35.3 |
| opp115_user_access,_edit_and_deletion | 470 | 35.1 |

| Task | Number of Samples | Mean Sample Length (Words) |
|---|---|---|
| opp115_user_choice_control | 1554 | 33.5 |
| oral_argument_question_purpose | 319 | 50.2 |
| overruling | 2400 | 27.5 |
| personal_jurisdiction | 54 | 67.8 |
| privacy_policy_entailment | 4343 | 111.9 |
| privacy_policy_qa | 10931 | 41.1 |
| proa | 100 | 42.6 |
| rule_qa | 50 | 11.7 |
| scalr | 571 | 275.4 |
| ssla_company_defendants | 1231 | 310.0 |
| ssla_individual_defendants | 1015 | 313.7 |
| ssla_plaintiff | 1036 | 308.4 |
| sara_entailment | 276 | 148.0 |
| sara_numeric | 100 | 12222.1 |
| successor_liability | 50 | 71.5 |
| supply_chain_disclosure_best_practice_ accountability | 387 | 510.0 |
| supply_chain_disclosure_best_practice_audits | 387 | 508.3 |
| supply_chain_disclosure_best_practice_ certification | 386 | 508.4 |
| supply_chain_disclosure_best_practice_ training | 387 | 508.2 |
| supply_chain_disclosure_best_practice_ verification | 387 | 507.0 |
| supply_chain_disclosure_disclosed_ accountability | 386 | 510.4 |
| supply_chain_disclosure_disclosed_audits | 387 | 508.0 |
| supply_chain_disclosure_disclosed_ certification | 386 | 509.9 |
| supply_chain_disclosure_disclosed_training | 387 | 506.9 |
| supply_chain_disclosure_disclosed_ verification | 387 | 507.6 |
| telemarketing_sales_rule | 51 | 58.4 |
| textualism_tool_dictionaries | 111 | 151.3 |
| textualism_tool_plain | 169 | 160.9 |
| ucc_v_common_law | 100 | 20.9 |
| unfair_tos | 3822 | 34.3 |

# F   Evaluation

This section describes metrics and evaluation protocols.

**Rule-application**   To evaluate an LLM's performance on a rule-application task, a law-trained expert manually validated each generation. We computed two metrics. The first metric—*correctness*—corresponds to the proportion of generations which do not misstate the fact pattern, the legal outcome, the rule, or contain a logical error. Logical errors include arithmetic mistakes, or assertions which are plainly wrong (e.g., that an apple is not a tangible object).

- The LLM would incorrectly assert the legal outcome. For instance, an LLM would assert diversity jurisdiction existed, when it actually did not.

- The LLM would incorrectly assert an intermediate conclusion. For instance, an LLM would assert that a rental agreement for a boat was a contract for a service, rather than a moveable and tangible good. In this case, the LLM would fail to realize that a boat is a moveable or tangible good.

- The LLM would hallucinate a piece of information not explicitly stated in the fact pattern.

- The LLM would misstate the content of a rule. For instance, the LLM would assert that subprovision of a statute barred one type of conduct, when in fact, that conduct was barred by a different subprovision.

The second metric—*anlaysis*—corresponds to the proportion of generations which contain the necessary inferences to reach the correct legal conclusion from the provided fact-pattern. Thus, it is insufficient for a LLM explanation to merely state that a piece of evidence is hearsay is insufficient: an explanation must reference the qualities of the evidence which make it hearsay. We introduced this measurement after discovering that for some tasks, LLMs often generate explanations which—though correct—largely restate the rule being applied, without any reference to the underlying facts. Explanations which are incorrect are automatically deemed to be insufficient on analysis. We compute an overall analysis score for a LLM on a task by measuring the proportion of samples for which the explanation contains sufficient analysis.

To standardize evaluation and enable future work, we have released an "answer guide" for each task used for rule-application, which contains the inferences required for each sample, and describes common modes of errors. All evaluations in LEGALBENCH for rule-application have been performed with respect to this answer-guide.

**Generation tasks**   LEGALBENCH contains the following generation-tasks, which are evaluated as follows:

- `rule_qa` is a question-answer task in which a LLM must generate a response to an open-ended question. A law-trained individual evaluated the generations against an answer-key for the task, which is available for download from the website.

- The Securities Complaint Extraction tasks require an LLM to extract the names of different parties from excerpts of securities class-action complaints. Because some samples require the extraction of multiple entities, we evaluate using F1 score.

- `definition_extraction` requires the LLM to identify the term that is being defined in a sentence from a Supreme Court opinion. For a small number of sentences, any one of multiple terms may constitute the correct answer. We therefore evaluate performance on this task using accuracy, and by counting the fraction of sentences for which the LLM identified a permissible term. To account for edge-cases involving word tense, we compare stemmed versions of the answers to stemmed versions of the generation.

- `sara_numeric` requires an LLM to generate an estimate of the amount of tax that is owed. We compute performance here using an accuracy metric, which treats a prediction as accurate if it is within 10% of the true amount.

- `citation_open` requires an LLM to predict the name of the case that should be cited for a particular sentence of judicial text. We evaluate by checking if the LLM generation contains the correct case name.

- `successor_liability` requires an LLM to identify the multiple possible successor liability exceptions to a fact pattern. We evaluate using F1.

**Classification tasks**   We evaluate all classification tasks in LEGALBENCH using exact-match on class-balanced-accuracy. We do this because a number of LEGALBENCH tasks are class-imbalanced.

# G Task descriptions

This section provides a detailed description for each family of tasks.

## G.1 Abercrombie

In LEGALBENCH, the Abercrombie task is denoted as `abercrombie`.

**Background** A particular mark (e.g., a name for a product or service) is only eligible for trademark protection if it is considered to be distinctive. In assessing whether a mark is distinctive, lawyers and judges follow the framework set out in the case *Abercrombie & Fitch Co. v. Hunting World, Inc.*,[7] which enumerates five categories of distinctiveness. These categories characterize the relationship between the dictionary definition of the term used in the mark, and the service or product it is being attached to. They are:

- Generic: A name is generic with respect to a product or service if it connotes the basic nature of the product/service, rather than more individualized characteristics of the product. For example, the mark "Salt" for packaged sodium chloride would be generic under Abercrombie because "salt" is the common name for sodium chloride. It is also common to think of generic marks as merely referring to the class of goods for which a particular product is a species.

- Descriptive: A name is descriptive if it identifies a characteristic or quality of an article or service, such as color, odor, function, dimensions, or ingredients. For example, the name "Sharp" for a television would be descriptive, because it describes a plausible characteristic of television (i.e., their sharp image quality).

- Suggestive: A name is suggestive if it suggests, rather than describes, some particular characteristic of the goods or services to which it applies. An essential aspect of suggestive names is that it requires the consumer to exercise the imagination in order to draw a conclusion as to the nature of the goods and services. For example, the name "Greyhound" would be suggestive for a bus service, because greyhounds are considered to be fast, and "fast" is an adjective that could be used to describe a bus service.

- Arbitrary: A name is arbitrary if it is a "real" word but seemingly "arbitrary" with respect to the product or service. For example, the mark "Apple" for a software company is arbitrary, because apples are unrelated to software.

- Fanciful: A name is fanciful if it is entirely made up, and not found in the English dictionary. For example, "Lanmbe" is a fanciful mark, because it is a made-up word.

The Abercrombie spectrum is commonly taught as part of Intellectual Property courses in law school, and students are expected to understand how to determine the Abercrombie classification for a particular product/mark combination.

Performing the Abercrombie task requires reasoning about the literal meaning of a word and the degree of its connection to a particular product/service. It requires having some understanding of the types of words that could plausibly be used to describe a particular good/service, and the extent to which those words relate to a particular mark. It also requires reasoning as to whether a particular word is a real English word.

**Task** The Abercrombie task requires an LLM to determine–given a candidate mark and a description of a product/service–which of the five Abercrombie categories above apply.

| Facts | Abercrombie Classification |
|---|---|
| The mark "Whirlpool" for an oven. | arbitrary |
| The mark "Compact" for wallets. | descriptive |
| The mark "Imprion" for a line of sports drinks. | fanciful |
| The mark "Car" for a line of automobiles. | generic |
| The mark "Quick Green" for grass seed. | suggestive |

Table 10: Task examples.

---

[7]*Abercrombie & Fitch Co. v. Hunting World*, 537 F.2d 4 (2nd Cir. 1976).

**Construction process** We manually create a dataset to evaluate a model's ability to classify a mark's distinctiveness (into one of the above 5 categories) with respect to a product. In writing samples, we draw inspiration from similar exercises available in legal textbooks and practice study guides. Hence, the samples provided have a definite answer, and are not subject to ambiguity. There is an expectation that a law student learning intellectual property would be able to answer these questions to a high degree of accuracy.

We create approximately 20 samples for each category of distinctiveness, and randomly select a single sample from each category to constitute the train set. The remaining 19 samples (for each category) are assigned to the test set (for a total of 95 samples).

| Class | Number of samples |
|-------------|-------------------|
| generic | 19 |
| descriptive | 19 |
| suggestive | 20 |
| arbitrary | 18 |
| fanciful | 19 |

Table 11: Test set class distribution.

**Significance and value** Given how easy this task is for lawyers with a basic training in intellectual property law, it is unlikely that LLMs will be called on to perform this task in the actual practice of law, or that the ability for LLMs to perform this task would alter the way in which lawyers approach IP practice. Instead, the Abercrombie task is significant as a measurement of reasoning ability. Because it is "simplistic" by the standards of human lawyers, it provides a useful objective measure of reasoning progress for LLMs.

## G.2 Canada Tax Court Outcomes

In LEGALBENCH, the Canada Tax Court Outcomes task is also denoted as `canada_tax_court_outcomes`.

**Background**   The Tax Court of Canada hears appeals of government decisions related to taxation.[8] The Court's decisions, which are written in natural language, are published on the Court's website, in both French and English.[9] Decisions typically include a section at the beginning summarizing the outcome of the appeal, followed by sections describing the factual background and various procedural steps, a section identifying the issues under consideration, sections with legal analysis, and a concluding section. While this is the standard format, judges are free to use other formats if they prefer. Decision length varies depending on the complexity of the litigation, with some decisions being only a few hundred words, and others being many thousands of words.

Appeals in Tax Court of Canada cases are brought by individuals or organizations who ask the Court to overturn a government taxation decision. Outcomes of appeals are generally binary: appeals are either granted, in which case the government taxation decision is overturned in whole or in part, or appeals are denied in which case the government taxation decision is upheld. Occasionally published decisions will not involve the outcome of an appeal, including where the decision is about a procedural step (e.g. the admissibility of particular evidence).

The `canada_tax_court_outcomes` task involves identifying whether an excerpt from a Tax Court of Canada decision includes the outcome of the appeal and, if so, what the outcome is. While the task is straightforward, one challenge is that the model must distinguish between outcomes of the appeal as a whole and outcomes of particular aspects of the appeal. Another challenge is that where the excerpt does not include the outcome, the model must avoid predicting the outcome – even if the model might plausibly correctly infer the likely outcome from the excerpt provided.

**Task**   The Canada Tax Court Outcomes task requires an LLM to classify whether an excerpt from a given decision includes the outcome of the appeal, and if so whether the appeal was allowed or dismissed. Some excerpts do not include an outcome of the appeal, in which case the model should return 'other'. Where the excerpt includes the outcome and the appeal is allowed in whole or in part, the model should return 'allowed'. Where the excerpt includes an outcome, and the appeal is dismissed the model should return 'dismissed'. The model should disregard outcomes that are not about the ultimate outcome of the appeal, such as costs awards (i.e. orders requiring a party to pay the other party's legal costs).

**Construction process**   We obtained the full text of English-language versions of decisions from 2001 to 2022 by scraping the Tax Court of Canada website.[10] We then cleaned and parsed the text to extract excerpts that are most likely to contain the outcome of the appeal. For example, many decisions contain a brief introductory section describing the outcome of the appeal using a specific header, and if the decision contained a section with such a header, we excerpted only that section. Where our parsing code could not identify such a section, we excerpted the first and last 2,500 characters, because outcomes are generally described at either the beginning or end of decisions. After initially attempting outcome classification on these excerpts using OpenAI's ChatGPT, we selected a quasi-random sample of 250 excerpts (quasi random because we selected these manually, we over-sampled excerpts where the outcome is 'other', and we chose some excerpts that were challenging due to factors such as length or unusual format). We manually reviewed outcomes for these excerpts, correcting some that had been miscategorized.

Two random cases from each class are selected for the training split, while the remainder are used as the test set.

**Significance and value**   Legal scholars frequently gather data about outcomes in large numbers of legal decisions in order to examine patterns in judicial decision-making. For example, a legal scholar may be interested in comparing outcomes in similar processes across jurisdictions or they might examine whether a legislative change resulted in different outcomes over time. Lawyers and legal information technology companies may also be interested in gathering data on outcomes for the purposes of judicial analytics or to predict future outcomes.

Gathering such data is typically straightforward. It is, for example, a common task assigned to first year law student research assistants who can frequently achieve close to 100% accuracy on such tasks with only minimal training. However, because the data is often useful only when gathered on large numbers of decisions, this type of data gathering using human research assistants can be cost prohibitive. If LLMs can obtain high accuracy on these tasks, substantial savings could be achieved – which would increase the ability of researchers to pursue new projects.

---

[8]*Tax Court of Canada Act*, RSC, 1985, c T-2, online: `https://laws-lois.justice.gc.ca/eng/acts/t-2/index.html`, s 12.

[9]Tax Court of Canada, "Find a Decision", online: `https://decision.tcc-cci.gc.ca/tcc-cci/en/nav.do`.

[10]Ibid. As per the terms of service of the website, we are required to note that the text of the scraped decisions are not the official versions (official versions can be obtained from the website), and that the reproduction of these cases has not been produced in affiliation with or with the endorsement of the Government of Canada.

| Excerpt | Outcome |
|---|---|
| The appeal is allowed in part and the assessment is referred back to the Minister of National Revenue for reconsideration and reassessment to reflect a 25% reduction of the tax owed by the appellant and adjustments to the interest and penalties, as agreed to by the respondent. Costs are to be determined after hearing both parties. In all other respects, the assessment is confirmed. Signed this 23rd day of February 2012. "Franois Angers" Angers J. | allowed |
| IN ACCORDANCE with the Reasons for Judgment attached, the appeal from the decision of the Respondent in relation to the income of the Appellant for the purposes of determining his entitlement to the Guaranteed Income Supplement under the Old Age Security Act for the payment period from July 1, 2014 to June 30, 2015 is dismissed, without costs. Signed at Ottawa, Canada, this 24th day of October 2017. R.S. Bocock Bocock J. | dismissed |
| (These Reasons for Judgment are issued in substitution for the Reasons for Judgment signed on January 22, 2002) Lamarre, J. [1] These are appeals under the informal procedure against assessments made by the Minister of National Revenue ("Minister") under the Income Tax Act ("Act") for the 1995, 1996, 1997, 1998 and 1999 taxation years. [2] In filing her 1995 income tax return, the appellant claimed a business investment loss of $268,897 with respect to investments in eight mortgages held "in trust" for the appellant and her father Henry Sokolowski by Kiminco Acceptance Co. Ltd. ("Kiminco"), a member of the Glen Coulter group of companies. The eight mortgage investments were made in 1987 and 1988 and are identified as follows in paragraph 13 of the Reply to the Notice of Appeal: Account/Mortgage Number Ultimate Borrower ... For the Appellant: Name: Firm: For the Respondent: Morris Rosenberg Deputy Attorney General of Canada Ottawa, Canada | other |

Table 12: Task examples.

| Outcome | Number of samples |
|---|---|
| allowed | 101 |
| dismissed | 131 |
| other | 12 |

Table 13: Test set class distribution.

### G.3 Citation Prediction Tasks

In the LEGALBENCH, the Citation Prediction tasks are also denoted as `citation_prediction_*`.

**Background**  The importance of locating relevant legal materials, or "law search" has long been recognized as an essential aspect of legal practice. This process involves uncovering case law, statutes, and other materials pertinent to legal questions or arguments. As a fundamental aspect of legal reasoning, law search plays a crucial role in bridging the gap between the initial translation of behaviors into legal questions and the subsequent interpretation and application of the relevant law.

Legal professionals are often valued for their ability to find and apply the appropriate law to their clients' situations. Given the intricate nature of the contemporary legal domain, the process of law search has evolved into a complex and nuanced task that demands a comprehensive understanding of the law.

A core component of law search is legal relevance. From a sociological perspective, the relevance of legal documents to a specific legal question is a social fact. This fact is determined by the judgments made by members of the legal community, who must determine which legal materials are applicable to a given question. Relevance relates legal questions to sources of legal authority.

In functional legal communities, law search leads to some degree of convergence over legal materials. Convergence occurs when competent members of a legal community, faced with the same legal question, identify the same sources of relevant legal authority. This process is essential to ensuring that the legal system operates consistently, predictably, and coherently.

As a critical process that connects the translation of behaviors into legal questions and the subsequent interpretation and application of the relevant law, law search is indispensable to legal reasoning.

The Citation Prediction task requires reasoning concerning the relationship between the text of judicial opinions and legal propositions. Successful prediction would entail encoding a notion of legal relevance and would allow a system to determine whether a legal proposition was or was not supported by the extant body of law.

**Task**  The citation task is based on a version of the evaluation approaches used in [38]. There are two Citation Prediction tasks. The first (`citation_prediction_classification`) requires an LLM to predict whether a given sentence (i.e. legal proposition) is or is not supported by a given case. The second (`citation_prediction_open`) requires an LLM to predict a case (by name) that supports a provided sentence.

**Construction process**  We collected a sample of circuit court opinions published after January 1, 2023. To the best of our knowledge, most existing LLMs haven't been trained on any data generated in 2023. For each opinion, we manually collected sentences which were supported by a citation to a judicial opinion, where (1) the sentence contained some quotation from the original case, and (2) the sentence was supported by a single cite. We chose sentences which included quotation fragments and were only supported by a single cite to avoid sentences which could be supported by a broad set of cases. When a sentence is supported by a much larger universe of cases, verifying that an LLM answer is incorrect is difficult. We also recorded the circuit for each opinion that we pulled language from. As a result, we can include the circuit information in the prompt, since circuits prefer citing their previous decisions. We collected 55 sentences using this process. For the citation generation task (`citation_prediction_open`), we ask the LLM to predict the citation given the sentence.

The `citation_prediction_classification` task is then constructed as follows. We use each sentence-citation pair to create two task samples. The first sample corresponds to the sentence and the correct citation (positive label). The second sample corresponds to the sentence and a randomly selected citation from the remainder of the data (negative label). This generates a dataset of 110 sentence-citation pairs, two of which are assigned to the training split.

**Significance and value**  Law search is a core function of legal thinking. In addition, the difficulty of identifying relevant law is a core barrier in the public's ability usefully access the law. The ability of an LLM to accurately engage in citation prediction would have important practical value in providing access to law, and would also allow the LLM to more reliably support legal statements with relevant authority.

| Input | Citation | Supported? |
|---|---|---|
| Exclusions are always strictly construed against the insurer and in favor of the insured. | Nationwide Mut. Ins. Co. v. Cosenza | Yes |
| The Supreme Court and this court have repeatedly "held that environmental plaintiffs adequately allege injury in fact when they aver that they use the affected area and are persons for whom the aesthetic and recreational values of the area will be lessened by the challenged activity." | United States v. Pearce | No |

Table 14: Examples for `citation_prediction_classification`.

| Input | Citation |
|---|---|
| In other words, the DJA "creates a means by which rights and obligations may be adjudicated in cases involving an actual controversy that has not reached the stage at which either party may seek a coercive remedy." | United States v. Doherty |
| To be "equivalent to a demotion," the action need not "result in a decrease in pay, title, or grade; it can be a demotion if the new position proves objectively worse—such as being less prestigious or less interesting or providing less room for advancement." | Alvarado v. Tex. Rangers |

Table 15: Examples for `citation_prediction_open`.

## G.4 Clause Classification Tasks

LEGALBENCH includes a number of tasks in which the LLM must determine the "type" or "category" of a provision/clause in a legal document. Specifically:

- The Contract QA task (Section G.4.4), in which the LLM is provided with the name of a common type of contractual clause and a clause, and must determine if the clause is an example of the example type.

- 38 tasks derived from the CUAD dataset (Section G.4.1), where each task is a binary-classification task requiring the LLM to identify whether a clause (from an EDGAR contract) belongs to a certain category (e.g., audit rights clauses) [60].

- The J.Crew blocker task (Section G.4.2), in which the LLM must classify whether a clause (from a loan agreement) is a J-Crew blocker provision.

- The Unfair Terms of Service task (Section G.4.3), in which the LLM must classify a clause (from a terms of service agreement) to one of eight types, where seven of the types denote clauses that would potentially be considered "unfair" under European law [80].

**Significance and value** Lawyers spend significant time and energy reviewing legal documents (e.g., contracts, leases, etc.). Manual review serves an important purpose, allowing parties to identify potentially problematic terms [117]. Parties will sometimes review agreements that have already been signed, in response to changing world events. For instance, the COVID-19 pandemic led many firms to inspect agreements for the existence of *force majeure* clauses, which ordinarily specify how contractual expectations should be handled in the event of major world crises [48]. Because legal documents are long and require legal training to understand, the process of reviewing is often extremely expensive [60]. This, in turn, presents significant access-to-justice concerns. Because most individuals do not have the financial capacity to consult lawyers prior to entering legal agreements, they are oblivious to when those agreements contain predatory, oppressive, or unconscionable terms. A rich legal scholarship has noted, for instance, the frequency at which legal agreements contain terms that would be invalidated by a court [25].

The clause classification tasks in LEGALBENCH are thus amongst the most *practically useful* tasks in LEGAL-BENCH, as they capture an actual current-day use case for LLMs. As the complexity of clause classification depends both on the clause category and document type, LEGALBENCH tasks span a range of clause categories and source documents.

### G.4.1 CUAD Tasks

We adapt the CUAD dataset for LEGALBENCH [60]. The original dataset consists of 500 contracts, each annotated with up to 41 different clause types. These contracts varied significantly in length, ranging from a few pages to over one-hundred pages. In the original word, [60] studied the ability for BERT-base language models to identify the text spans corresponding to different types of clauses. The principal difficulties were (1) the length of the contract, and (2) the lack of significant training data.

We adapt the CUAD dataset as follows. We select 38 of the 41 clause categories. For each selected category, we construct a dataset consisting of (1) clauses in the CUAD contracts which are assigned to that category, and (2) an equal number of clauses randomly sampled from other categories. This produces a balanced binary classification task for clause category, where the purpose is to identify which clauses belong to the respective category. A table with the selected categories, and their descriptions is found below.

Table 16 lists each task, a "description" of the category corresponding to the task, and an example of a clause which meets the category criteria. In accordance with [60], the description is presented as the question posed to the annotators during data labeling. If a clause yields an affirmative answer with regards to the question, then the label is "Yes". Otherwise the label is "No".

In LEGALBENCH, the CUAD tasks are denoted as cuad_*.

### Table 16: CUAD Tasks

| Task |
| --- |
| **Task name**: cuad_affiliate_license-licensee
**Description**: Does the clause describe a license grant to a licensee (incl. sublicensor) and the affiliates of such licensee/sublicensor?
**Example**: [\*\*\*], Valeant hereby grants to Dova a fully paid-up, royalty free, non-transferable, non- exclusive license (with a limited right to sub-license to its Affiliates) to any Valeant Property that appears on, embodied on or contained in the Product materials or Product Labeling solely for use in connection with Dova's promotion or other commercialization of the Product in the Territory. |

**Task**

**Task name**: cuad_affiliate_license-licensor
**Description**: Does the clause describe a license grant by affiliates of the licensor or that includes intellectual property of affiliates of the licensor?
**Example**: "Company Licensed Know-How" means all Know-How owned by any Company Entity as of the Effective Date and used or held for use in the Arizona Field as of the Effective Date.<omitted>Subject to the terms and conditions of this Agreement, the Company hereby grants to Seller a perpetual, non-exclusive, royalty-free license in, to and under the Company Licensed Know-How for use in the Arizona Field throughout the world.

**Task name**: cuad_anti-assignment
**Description**: Does the clause require consent or notice of a party if the contract is assigned to a third party?
**Example**: Except as otherwise set forth herein, neither party shall transfer, assign or cede any rights or delegate any obligations hereunder, in whole or in part, whether voluntarily or by operation of law, without the prior written consent of the other party, which consent may be withheld at the other party's reasonable business discretion; provided, however, that either party may transfer this Agreement without prior written consent of the other to an Affiliate of such party, or to the surviving party in a merger or consolidation, or to a purchaser of all or substantially all of its assets.

**Task name**: cuad_audit_rights
**Description**: Does the clause give a party the right to audit the books, records, or physical locations of the counterparty to ensure compliance with the contract?
**Example**: For avoidance of doubt, all audits under this Section shall be conducted solely by an independent public accountant as described in the foregoing sentence.

**Task name**: cuad_cap_on_liability
**Description**: Does the clause specify a cap on liability upon the breach of a party's obligation? This includes time limitation for the counterparty to bring claims or maximum amount for recovery.
**Example**: EXCEPT FOR LIABILITIES UNDER SECTION 7.2 [Indemnity], NEITHER PARTY'S AGGREGATE LIABILITY ARISING OUT OF OR IN CONNECTION WITH THIS AGREEMENT OR THE TRANSACTIONS CONTEMPLATED HEREBY, WHETHER IN CONTRACT, TORT (INCLUDING NEGLIGENCE), WARRANTY OR OTHERWISE, SHALL EXCEED [***].

**Task name**: cuad_change_of_control
**Description**: Does the clause give one party the right to terminate or is consent or notice required of the counterparty if such party undergoes a change of control, such as a merger, stock sale, transfer of all or substantially all of its assets or business, or assignment by operation of law?
**Example**: Notwithstanding the foregoing, if any Party to this Agreement (or any of its successors or permitted assigns) (a) shall enter into a consolidation or merger transaction in which such Party is not the surviving entity and the surviving entity acquires or assumes all or substantially all of such Party's assets, (b) shall transfer all or substantially all of such Party's assets to any Person or (c) shall assign this Agreement to such Party's Affiliates, then, in each such case, the assigning Party (or its successors or permitted assigns, as applicable) shall ensure that the assignee or successor- in-interest expressly assumes in writing all of the obligations of the assigning Party under this Agreement, and the assigning Party shall not be required to seek consent, but shall provide written notice and evidence of such assignment, assumption or succession to the non-assigning Party.

**Task name**: cuad_competitive_restriction_exception
**Description**: Does the clause mention exceptions or carveouts to Non-Compete, Exclusivity and No-Solicit of Customers?
**Example**: Notwithstanding the foregoing, Excite may make available opportunities on the Excite Site to purchase Music Products from parties other than Sponsor if such Music Products are not available from Sponsor so long as, prior to entering into arrangements to make available opportunities to purchase Music Products from parties other than Sponsor, Excite notifies Sponsor of its interest in the Music Products and gives Sponsor thirty (30) days to make the desired Music Products available through the Sponsor Site.

**Task name**: cuad_covenant_not_to_sue
**Description**: Is a party restricted from contesting the validity of the counterparty's ownership of intellectual property or otherwise bringing a claim against the counterparty for matters unrelated to the contract?
**Example**: In connection with any reference to the Trademarks, Distributor shall not in any manner represent that it has an ownership interest in the Trademarks or registration(s) thereof, and Distributor acknowledges that no action by it or on its behalf shall create in Distributor's favor any right, title, or interest in or to the Trademarks.

**Task name**: cuad_exclusivity

**Task**

**Description**: Does the clause specify exclusive dealing commitment with the counterparty? This includes a commitment to procure all "requirements" from one party of certain technology, goods, or services or a prohibition on licensing or selling technology, goods or services to third parties, or a prohibition on collaborating or working with other parties), whether during the contract or after the contract ends (or both).
**Example**: Bosch hereby grants to Client the exclusive rights to sell and distribute the Product, subject to the Territory as set forth below, to certain select companies in the Automotive Industry, each of which shall be approved by Bosch in writing as requested by the Client on a case by case basis.

**Task name**: cuad_insurance
**Description**: Is there a requirement for insurance that must be maintained by one party for the benefit of the counterparty?
**Example**: Throughout the entire Term, you must maintain such types of insurance, in such amounts, as we may require.

**Task name**: cuad_ip_ownership_assignment
**Description**: Does intellectual property created by one party become the property of the counterparty, either per the terms of the contract or upon the occurrence of certain events?
**Example**: Upon written request of ArTara, University will assign the IND to ArTara.

**Task name**: cuad_irrevocable_or_perpetual_license
**Description**: Does the clause specify a license grant that is irrevocable or perpetual?
**Example**: Subject to the terms and conditions of this Agreement, as of the Distribution Date, SpinCo hereby grants to Nuance and the members of the Nuance Group a worldwide, non-exclusive, fully paid-up, perpetual and irrevocable, transferable (subject to ARTICLE VIII), sublicensable (subject to Section 4.01(g)) license to install, access, use, reproduce, perform, display, modify (including the right to create improvements and derivative works), further develop, sell, manufacture, distribute and market products and services based on, using or incorporating the SpinCo Shared Technology Assets within the Nuance Field of Use, together with natural extensions and evolutions thereof.

**Task name**: cuad_joint_ip_ownership
**Description**: Does the clause provide for joint or shared ownership of intellectual property between the parties to the contract?
**Example**: If the Domain Name is deemed a combination mark, neither party shall use the Domain Name for any purpose except as expressly provided herein or attempt to register the Domain Name, and the parties will jointly cooperate on any enforcement action of infringement of the Domain Name.

**Task name**: cuad_license_grant
**Description**: Does the clause contain a license granted by one party to its counterparty?
**Example**: Neoforma hereby grants VerticalNet a non-exclusive, non-transferable, royalty-free, right and license to link to the Neoforma Sites through a Neoforma Link.

**Task name**: cuad_liquidated_damages
**Description**: Does the clause award either party liquidated damages for breach or a fee upon the termination of a contract (termination fee)?
**Example**: You and each of your principals agree that the liquidated damages provision does not give us an adequate remedy at law for any default under, or for the enforcement of, any provision of this Agreement other than the Royalty Fee sections.

**Task name**: cuad_minimum_commitment
**Description**: Does the clause specify a minimum order size or minimum amount or units pertime period that one party must buy from the counterparty?
**Example**: If the Quarterly Average Sales Force Size is less than [***] Sales Representatives for an applicable Calendar Quarter, then in calculating the promotion fee due under Section 6.1.1, the Applicable Percentage for such Calendar Quarter shall be reduced to a new percentage equal to [***].

**Task name**: cuad_most_favored_nation
**Description**: Does the clause state that if a third party gets better terms on the licensing or sale of technology/goods/services described in the contract, the buyer of such technology/goods/services under the contract shall be entitled to those better terms?

**Task**

**Example**: Eutectix agrees that in the event any Licensed Products shall be sold (1) to any Affiliate (as defined herein), or (2) to a corporation, firm, or association with which, or individual with whom Eutectix or its stockholders or Affiliates shall have any agreement, understanding, or arrangement (such as, among other things, an option to purchase stock, or an arrangement involving a division of profits or special rebates or allowances) without which agreement, understanding, or arrangement, prices paid by such a corporation, firm, association or individual for the Licensed Products would be higher than the Net Sales Price reported by Eutectix, or if such agreement, understanding, or arrangement results in extending to such corporation, firm, association, or individual lower prices for Licensed Products than those charged to outside concerns buying similar products in similar amounts and under similar conditions, then, and in any such events, the royalties to be paid hereunder in respect of such Licensed Products shall be computed based on an assumed or deemed Net Sales Price equal to those charged to such outside concerns.

**Task name**: cuad_no-solicit_of_customers
**Description**: Does the clause restrict a party from contracting or soliciting customers or partners of the counterparty, whether during the contract or after the contract ends (or both)?
**Example**: During the Term of this Agreement, and for a period of one year thereafter, except as expressly provided in this Agreement, PlanetCAD shall not market any services to Customers without the prior written approval of Dassault Systemes.

**Task name**: cuad_no-solicit_of_employees
**Description**: Does the clause restrict a party's soliciting or hiring employees and/or contractors from the counterparty, whether during the contract or after the contract ends (or both)?
**Example**: You covenant that during the term of this Agreement, except as otherwise approved in writing by us, you will not, either directly or indirectly, for yourself, or through, on behalf of, or in conjunction with any person, persons, partnership, corporation or company:<omitted>2. Employ or seek to employ any person who is at that time employed by us, our affiliates, or by any other franchisee of ours, or otherwise directly or indirectly induce or seek to induce such person to leave his or her employment threat.

**Task name**: cuad_non-compete
**Description**: Does the clause restrict the ability of a party to compete with the counterparty or operate in a certain geography or business or technology sector?
**Example**: Agent may not offer or promote competitive products without the consent of Kallo.

**Task name**: cuad_non-disparagement
**Description**: Does the clause require a party not to disparage the counterparty?
**Example**: The Company shall not tarnish or bring into disrepute the reputation of or goodwill associated with the Seller Licensed Trademarks or Arizona.

**Task name**: cuad_non-transferable_license
**Description**: Does the clause limit the ability of a party to transfer the license being granted to a third party?
**Example**: Subject to the terms and conditions of this Agreement, Licensor hereby grants to Licensee, and Licensee hereby accepts from Licensor, a personal, non-exclusive, royalty-free right and license to use the Licensed Mark solely and exclusively as a component of Licensee's own corporate name and in connection with marketing the investment management, investment consultation and investment advisory services that Investment Advisor may provide to Licensee.

**Task name**: cuad_post-termination_services
**Description**: Does the clause subject a party to obligations after the termination or expiration of a contract, including any post-termination transition, payment, transfer of IP, wind-down, last-buy, or similar commitments?
**Example**: Cisco agrees to repurchase all Product in Distributor's inventory within [*****] days following the effective date of termination or expiration.

**Task name**: cuad_price_restrictions
**Description**: Does the clause place a restriction on the ability of a party to raise or reduce prices of technology, goods, or services provided?
**Example**: The prices set forth in Section 2.4(a) shall be subject to adjustment annually on the first day of each Product Year beginning in the calendar year 2000 and on the first day of each succeeding Product Year for the remainder of the Term and all renewals of this Agreement in proportion to the increase or decrease in the Consumer Price Index (CPI) as compared to the CPI as it existed on the first day of the Term of this Agreement.

**Task name**: cuad_revenue-profit_sharing
**Description**: Does the clause require a party to share revenue or profit with the counterparty for any technology, goods, or services?

| Task |
| --- |

**Example**: In consideration for the licenses granted to Corio pursuant to Section 2 (except Section 2.5) of this Agreement, Corio shall pay the revenue sharing fees specified in EXHIBIT B hereto.

**Task name**: cuad_rofr-rofo-rofn
**Description**: Does the clause grant one party a right of first refusal, right of first offer or right of first negotiation to purchase, license, market, or distribute equity interest, technology, assets, products or services?
**Example**: If Licensee shall have exercised such right, the closing shall be held at the corporate offices of Licensee on the closing date specified in the Offering Notice or the date that is ninety (90) days after the date of Licensee's notice of its exercise of such right, whichever is later.

**Task name**: cuad_source_code_escrow
**Description**: Does the clause require one party to deposit its source code into escrow with a third party, which can be released to the counterparty upon the occurrence of certain events (bankruptcy, insolvency, etc.)?
**Example**: With each delivery of Software to Bank of America hereunder, Supplier shall deliver to Bank of America the Source Code for all Software and for all Updates, Upgrades and new releases of the Software.

**Task name**: cuad_termination_for_convenience
**Description**: Does the clause specify that one party can terminate this contract without cause (solely by giving a notice and allowing a waiting period to expire)?
**Example**: Customer may terminate this Agreement during the Term upon at least one (1) years' written notice to M&I, provided that Customer pays M&I an early termination fee ("Termination for Convenience Fee") in an amount equal to REDACTED of the Estimated Remaining Value.

**Task name**: cuad_third_party_beneficiary
**Description**: Does the clause specify that that there a non-contracting party who is a beneficiary to some or all of the clauses in the contract and therefore can enforce its rights against a contracting party?
**Example**: Such covenants must be on a form that we provide, which form will, among other things, designate us as a third party beneficiary of such covenants with the independent right to enforce them.

**Task name**: cuad_uncapped_liability
**Description**: Does the clause specify that a party's liability is uncapped upon the breach of its obligation in the contract? This also includes uncap liability for a particular type of breach such as IP infringement or breach of confidentiality obligation
**Example**: Subject to the foregoing as wen as Mobimagic's obligations under this Agreement, Mobimagic shall not in any manner be held or be responsible or liable for any unforeseen contingency, claims, liabilities, demands. losses, damages or expenses arising due to absence of storage or retention of any PC Financial data which shall be the sole responsibility of PC Financial .

**Task name**: cuad_unlimited-all-you-can-eat-license
**Description**: Does the clause grant one party an "enterprise," "all you can eat" or unlimited usage license?
**Example**: Subject to the terms and conditions of this Agreement, Commerce One hereby grants to Corio a fee-bearing, perpetual and irrevocable, nonexclusive, nontransferable (except in accordance with Section 14.1 of this Agreement), right and license in the Territory to<omitted>(iv) sublicense an unlimited number of Customers to access and use the Software and MarketSite.net Service only through the installation on Corio servers;

**Task name**: cuad_volume_restriction
**Description**: Does the clause specify a fee increase or consent requirement, etc. if one party's use of the product/services exceeds certain threshold?
**Example**: Make himself available for four (4) sessions for production of photographs, or radio, television, video or other multi-media programming for use in Bizzingo's advertising or promotional materials, with each such session not exceeding eight (8) hours.

**Task name**: cuad_effective_date
**Description**: Does the clause specify the date upon which the agreement becomes effective?
**Example**: This JV Agreement shall become effective on the signing date and shall have a duration of * years, extendable for a further * years, unless notice of non- renewal is sent one year before the natural expiry date.<omitted>2 April 2020

**Task name**: cuad_expiration_date
**Description**: Does the clause specify the date upon which the initial term expires?
**Example**: This Agreement shall be effective as of the Effective Date and shall continue in effect through December 31, 2021 and any Renewal Term (the "Term"), unless terminated earlier as set forth herein.

**Task name**: cuad_governing_law
**Description**: Does the clause specify which state/country's law governs the contract?

| Task |
| --- |

**Example**: This Agreement shall be governed by and construed under the laws of the State of California, excluding conflict of laws provisions and excluding the 1980 United Nations Convention on Contracts for the International Sale of Goods.

**Task name**: cuad_notice_period_to_terminate_renewal
**Description**: Does the clause specify a notice period required to terminate renewal?
**Example**: Unless either party gives written notice to terminate this Agreement at least six (6) months prior to the end of said Initial Term, this Agreement shall continue on a year to year basis ("Extended Term(s)") until terminated by either party by giving written notice of termination thereof to the other party at least six (6) months prior to the end of the then current Extended Term.

**Task name**: cuad_renewal_term
**Description**: Does the clause specify a renewal term?
**Example**: This Agreement shall commence on the Effective Date and, unless sooner terminated in accordance with its terms, including by Ginkgo pursuant to Section 7.3 (Buy-Down Election) or extended by the mutual written agreement of the Parties, shall continue until the Intended End of Term (such time period, as may be extended pursuant to this Section 13.3.1 (Term - General), the "Term"); provided that, if,<omitted>at the expiration of the Intended End of Term, Ginkgo has paid the Minimum Cumulative Purchase Commitment, but will not have paid to BLI the Full Purchase Target, then the Term of this Agreement shall automatically extend for an additional [***] ([***]) year period from the date of the expiration of the then-Intended End of Term so that, among other things, BLI may potentially receive the benefit of the Full Purchase Target and Ginkgo may receive the continuing benefit of royalty-free licenses.

**Task name**: cuad_warranty_duration
**Description**: Does the clause specify a duration of any warranty against defects or errors in technology, products, or services provided under the contract?
**Example**: Airspan warrants that, following repair or replacement, the repaired or replaced Equipment or Software by Airspan shall be free from defects in materials and faulty workmanship and that the Software will conform in all material respects to Airspan's published specifications therefor for ninety (90) days from date of shipment from Airspan to Distributor or until the end of the Initial Warranty Period, whichever is longer.

## G.4.2  J.Crew Blocker

In LEGALBENCH, the J.Crew Blocker task is denoted as `jcrew-blocker`.

**Background**    Loan agreements often contain restrictive covenants that place limits on a borrower's activities to protect the lender's interests. One such restrictive covenant that has become popular in recent years is the "J.Crew blocker" provision. This provision was created in response to actions taken by the retailer J.Crew in 2016. J.Crew transferred valuable intellectual property assets out of the collateral pool for its existing loans by moving them into a new unrestricted subsidiary. This subsidiary was then able to use the IP assets as collateral to obtain new financing.

The J.Crew blocker provision aims to prevent this type of activity by prohibiting borrowers from transferring IP assets out of the reach of existing lenders. There are two key components to a J.Crew blocker:

1. A prohibition on transferring IP assets to unrestricted subsidiaries. This prevents the borrower from moving assets outside the scope of lender restrictions.

2. A requirement to obtain lender consent for any IP transfers to subsidiaries. This gives lenders oversight and control over how IP assets are distributed within the corporate group.

The presence of a robust J.Crew blocker in a loan agreement is designed to keep material assets within the collateral pool, and thereby protect lenders from borrowers' attempts to secure additional debt through unexpected transfers of IP. For this reason, J.Crew blocker provisions have been widely adopted in leveraged loan agreements.

**Task**    The J.Crew blocker task requires determining whether a given provision in a loan agreement qualifies as a J.Crew blocker. To make this determination, the provision must be analyzed to assess whether it contains:

1. A prohibition on transferring IP assets to unrestricted subsidiaries

AND/OR

2. A requirement to obtain lender consent for IP transfers to any subsidiary.

If the provision includes one or both of these components, it can be classified as a J.Crew blocker. If not, the provision does not meet the criteria.

**Construction process**    The dataset for this task was constructed by legal experts extracting real examples of provisions from public loan agreements. Each example was labeled as either meeting the criteria for a J.Crew blocker or not. The dataset contains 60 total examples, organized into two columns: "Text" (containing the clause in question) and "Label" (indicating whether the clause is a J.Crew Blocker provision). Each clause was analyzed and classified as a J.Crew Blocker provision ("Yes") or not ("No"). The construction process involved manually reviewing and annotating these samples, ensuring that each clause was accurately categorized. This process, carried out by legal experts, provides definitive answers to each sample, eliminating ambiguity.

**Significance and value**    The ability to identify J.Crew blocker provisions is important for both lenders and borrowers in leveraged finance. For lenders, it helps ensure key protections are included in loan agreements. For borrowers, it provides insight into restrictions being placed on their activities. Given the widespread adoption of J.Crew blockers, this is a task that requires proficiency to actively participate in the leveraged loan market. The task serves as an important measure of an LLM's ability to understand and apply legal concepts, particularly those related to secured lending and intellectual property law. It also tests the LLM's capacity to analyze and interpret legal provisions. Given the increasing complexity and sophistication of financial transactions, the ability to accurately identify and understand such provisions is a valuable skill for any LLM. This task, therefore, provides a useful measure of progress for LLMs in their understanding and interpretation of complex legal clauses.

| Clause | J.Crew Blocker Provision? |
|---|---|
| provided that (i) immediately before and after such designation, no Event of Default shall have occurred and be continuing, (ii) in the case of the designation of any Subsidiary as an Unrestricted Subsidiary, such designation shall constitute an Investment in such Unrestricted Subsidiary (calculated as an amount equal to the sum of (x) the fair market value of the Equity Interests of the designated Subsidiary and any of its Subsidiaries that are owned by Holdings or any Restricted Subsidiary, immediately prior to such designation (such fair market value to be calculated without regard to any Obligations of such designated Subsidiary or any of its Subsidiaries under the Guaranty Agreement) and (y) the aggregate principal amount of any Indebtedness owed by such Subsidiary and any of its Subsidiaries to Holdings or any of the Restricted Subsidiaries immediately prior to such designation, all calculated, except as set forth in the parenthetical to clause (x) above, on a consolidated basis in accordance with U.S. GAAP), and such Investment shall be permitted under Section 10.05, (iii) no Subsidiary may be designated as an Unrestricted Subsidiary if it or any of its Subsidiaries is a Restricted Subsidiary for the purpose of any Refinancing Notes Indenture, any Permitted Pari Passu Notes Document, any Permitted Pari Passu Loan Documents, any Permitted Junior Notes Document or other debt instrument, with a principal amount in excess of the Threshold Amount, (iv) following the designation of an Unrestricted Subsidiary as a Restricted Subsidiary, Holdings shall comply with the provisions of Section 9.12 with respect to such designated Restricted Subsidiary, (v) no Restricted Subsidiary may be a Subsidiary of an Unrestricted Subsidiary (and any Subsidiary of an Unrestricted Subsidiary that is acquired or formed after the date of designation shall automatically be designated as an Unrestricted Subsidiary) and (vi) in the case of the designation of any Subsidiary as an Unrestricted Subsidiary, each of (x) the Subsidiary to be so designated and (y) its Subsidiaries has not, at the time of designation, and does not thereafter, create, incur, issue, assume, guarantee or otherwise become directly or indirectly liable with respect to any Indebtedness pursuant to which the lender has recourse to any of the assets of Holdings or any Restricted Subsidiary (other than Equity Interests in an Unrestricted Subsidiary). | No |
| provided, that (i) immediately before and after such designation, no Event of Default exists (including after giving effect to the reclassification of Investments in, Indebtedness of and Liens on the assets of, the applicable Restricted Subsidiary or Unrestricted Subsidiary), (ii) as of the date of the designation thereof, no Unrestricted Subsidiary shall own any Capital Stock in any Restricted Subsidiary of the Borrower or hold any Indebtedness of or any Lien on any property of the Borrower or its Restricted Subsidiaries and (iii) no subsidiary may be designated as an Unrestricted Subsidiary if it owns intellectual property that is material to the business of the Borrower and its Restricted Subsidiaries, taken as a whole (such intellectual property, Material Intellectual Property), at the time of designation, other than in connection with transactions that have a bona fide business purpose so long as such transactions are not undertaken to facilitate a financing or a Restricted Payment or undertaken in connection with a liability management transaction. Notwithstanding anything contained in this Section 6.05 to the contrary, in no event shall (a) the Borrower or any Restricted Subsidiary be permitted to make or own any Investment in the Holdings direct or indirect equityholders constituting Material Intellectual Property (other than pursuant to a bona fide transition service or similar arrangement or in the same manner as other customers, suppliers or commercial partners of the relevant transferee generally) or (b) any Restricted Subsidiary transfer ownership of, or license on an exclusive basis, any Material Intellectual Property to any Unrestricted Subsidiary, other than in connection with transactions that have a bona fide business purpose and so long as such transactions are not undertaken to facilitate a financing or a Restricted Payment or undertaken in connection with a liability management transaction. Notwithstanding anything contained in this Section 6.06 to the contrary, in no event shall (a) the Borrower or any Restricted Subsidiary be permitted to make any Disposition of Material Intellectual Property to Holdings direct or indirect equityholders (other than pursuant to a bona fide transition service or similar arrangement or in the same manner as other customers, suppliers or commercial partners of the relevant transferee generally) or (b) any Restricted Subsidiary make any Disposition, constituting either a transfer of ownership or an exclusive license, of any Material Intellectual Property to any Unrestricted Subsidiary, other than in connection with transactions that have a bona fide business purpose and so long as such transactions are not undertaken to facilitate a financing or a Restricted Payment or undertaken in connection with a liability management transaction. | Yes |

Table 17: Examples from `jcrew_blocker`.

### G.4.3 Unfair Terms of Service

In LEGALBENCH, the Unfair Terms of Service task is denoted as `unfair_tos`.

**Background**    An array of recent work has found that consumers rarely read terms of service agreements [95, 62]. As a result, consumers regularly sign agreements or contracts containing provisions that (1) they lack awareness of, and/or (2) would consider as "unfair" or "predatory." Reasons for this phenomenon include the sheer amount of time it would take to read every terms of service agreement, the obtuse language of these agreements, and the lack of actual recourse on an individual basis.

With reference to European consumer law, [80] identify eight categories of clauses in terms-of-service agreements which could be considered "potentially unfair":

- Arbirtration: clauses which mandated that all disputes between the parties would be resolved through arbitration.

- Unilateral change: clauses which allow the provider to modify the terms of service and/or the service itself.

- Content removal: clauses which give the provider a right to modify/delete a user's content

- Jurisdiction: clauses which specify a jurisdiction in which claims must be brought, regardless of where the user lives.

- Choice of law: clauses which specify the country's law which governs disputes arising under the contract, regardless of where the user lives.

- Limitation of liability: clauses which limit the liability of the service provider.

- Unilateral termination: clauses which empower the service provider to terminate/suspend the service at their discretion.

- Contract by using: clauses which stipulate that a consumer is bound by the terms of service simply by using the service.

A more detailed description of these categories can be found in [80].

**Task**    The Unfair Terms of Service task requires an LLM to determine—given a clause from a terms of service agreeement—whether it belongs to one of the above eight categories, and if so, which one.

**Construction process**    We use the version of data available in [21], which takes a subset from [80]. Unlike [21]—which frames the task as distinguishing "fair" from "unfair" clauses—we cast the task as 8-way multiclassification task across the original categories identified in [80].

**Significance and value**    Unlike the CUAD and J.Crew Blocker task, the Unfair TOS task evaluates a LLM's ability to perform multiclass clause classification across a highly imbalanced dataset.

| Clause | Clause type |
|---|---|
| you also acknowledge that a variety of evernote actions may impair or prevent you from accessing your content or using the service at certain times and/or in the same way , for limited periods or permanently , and agree that evernote has no responsibility or liability as a result of any such actions or results , including , without limitation , for the deletion of , or failure to make available to you , any content . | Arbitration |
| if you do not terminate your agreement before the date the revised terms become effective , your continued access to or use of the airbnb platform will constitute acceptance of the revised terms . | Choice of law |
| we may at any time and from time to time , in our sole discretion , change the fees and charges , or add new fees and charges , in relation to any of the products . | Content removal |
| you and academia.edu agree that any dispute , claim or controversy arising out of or relating to these terms or the breach , termination , enforcement , interpretation or validity thereof or the use of the site or services ( collectively , " disputes " ) will be settled by binding arbitration , except that each party retains the right : ( i ) to bring an individual action in small claims court and ( ii ) to seek injunctive or other equitable relief in a court of competent jurisdiction to prevent the actual or threatened infringement , misappropriation or violation of a party 's copyrights , trademarks , trade secrets , patents or other intellectual property rights ( the action described in the foregoing clause ( ii ) , an " ip protection action " ) . | Contract by using |
| if we find that any shared content in your account violates our terms of service ( including by violating another person 's intellectual property or privacy rights ) , we reserve the right to un-share or take down such content . | Jurisdiction |
| if you live in the european union : you agree that the laws of ireland , excluding conflict of laws rules , shall exclusively govern any dispute relating to this contract and/or the services . | Limitation of liability |
| oculus does not endorse or guarantee the opinions , views , advice or recommendations posted or sent by users . | Other |
| if you object to the changes , nintendo reserves the right to terminate this agreement or any portion of it upon reasonable notice and you will have to register again if you wish to continue using the nintendo account service under the new terms and conditions . | Unilateral change |
| unless you and we agree otherwise , in the event that the agreement to arbitrate above is found not to apply to you or to a particular claim or dispute , either as a result of your decision to opt out of the agreement to arbitrate or as a result of a decision by the arbitrator or a court order , you agree that any claim or dispute that has arisen or may arise between you and ebay must be resolved exclusively by a state or federal court located in salt lake county , utah . | Unilateral termination |

Table 18: Examples from `unfair_tos`.

### G.4.4 Contract QA

In LEGALBENCH, the Contract QA task is denoted as `contract_qa`.

**Background** Each of the above tasks evaluates the capacity for LLMs to learn to recognize a single type of clause, given a description of that clause and/or examples of it. The Contract QA task generalizes this across

| Class | Number of samples |
|---|---|
| Other | 3454 |
| Contract by using | 15 |
| Choice of law | 38 |
| Content removal | 53 |
| Unilateral change | 70 |
| Arbitration | 98 |
| Limitation of liability | 28 |
| Unilateral termination | 32 |
| Jurisdiction | 25 |

Table 19: Test set class distribution.

multiple clause types, evaluating an LLM's ability to recognize legal provisions that are *not* described in the prompt.

**Task**  Each sample in the dataset consists of (1) a contract clause, and (2) a question asking if the clause is an example of a provision type (e.g., "Is this a severability clause?"). Across the dataset, the questions correspond to 22 different legal provisions. Questions and provisions are paired such that for each provision type, the LLM is presented with two clauses that are an example of the type, and two clauses which are not.

**Construction Process**  The data was manually extracted from a set of sample agreements contributed by a LegalTech vendor and from public sources. It represents a variety of contracts, such as:

- Vendor or Partner Data Protection Agreements (DPA)
- Master Services Agreements (MSA)
- Licensing Terms
- BIPA consents

| Clause | Question | Answer |
|---|---|---|
| This Agreement shall be governed by and construed in accordance with the laws of the State of New York, without giving effect to any choice of law or conflict of law provisions. | Does the clause discuss BIPA consent? | No |
| If a dispute arises between the parties under this Agreement that cannot be resolved through good faith negotiations within a reasonable period of time, such dispute shall be escalated to an executive officer of each party for resolution. If such executive officers are unable to resolve such dispute within a reasonable period of time after escalation, either party may pursue any available legal remedies. | Does the clause discuss how disputes may be escalated? | Yes |

Table 20: Examples for `contract_qa`.

## G.5  Consumer Contracts QA

In LEGALBENCH, the Consumer Contracts QA task is denoted as `consumer_contracts_qa`.

**Background**  Consumer contracts govern many economic and social activities, ranging from retail purchases and online search to social media and entertainment. These contracts can affect consumers' access to services, control terms of payment, and determine the remedies available when consumers' rights are violated. Despite the importance of these legal agreements, consumers typically lack the time, expertise, and incentive to properly examine how consumer contracts impact their rights and interests. This issue is known as the "no-reading" problem [8, 7]. LLMs may offer a solution. By reading consumer contracts and explaining their legal ramifications, LLMs could enable consumers to better understand and exercise their legal rights in many everyday contexts.

**Task**  The Consumer Contracts QA task, first introduced in [71], aims to examine the degree to which an LLM can understand certain consumer contracts. Specifically, the task is comprised of 200 yes/no legal questions relating to the terms of service of popular websites. Examples of questions are provided in the table below.

In addition to the original 200 questions, the task includes an alternatively worded version of all 200 questions. While each question's content is substantially the same across both versions of the question, the alternatively worded questions are, by design, less readable, that is, more difficult for a human to read. Comparing performance across the original questions and the alternatively worded questions can help assess an LLM's brittleness in performing the task at hand. An example is provided in the table below:

**Construction process**  The task was introduced in [71]. To construct the dataset, an attorney drafted 200 yes/no questions relating to the terms of service of the 20 most-visited U.S. websites (10 questions per document), as well as an alternatively worded version of all 200 questions. The questions relate to a wide range of legal issues arising in the terms of service, including eligibility to access services, payment for services, limitations of liability, intellectual property rights, and dispute resolution procedures. Answers to all questions can be obtained from the applicable terms of service.

**Significance and value**  Given the ubiquity of consumer contracts, LLMs capable of reading these documents and communicating their contents to consumers might offer significant benefits. These benefits, however, are contingent on a model's accuracy and reliability. LLMs that misinterpret the provisions of consumer contracts may hinder consumers' ability to understand and exercise their contractual rights. The Consumer Contracts QA task is a preliminary attempt at evaluating the ability of LLMs to read certain consumer contracts.

**Contract**: Content Removal and Disabling or Terminating Your Account

We can remove any content or information you share on the Service if we believe that it violates these Terms of Use, our policies (including our Instagram Community Guidelines), or we are permitted or required to do so by law. We can refuse to provide or stop providing all or part of the Service to you (including terminating or disabling your your access to the Facebook Products and Facebook Company Products) immediately to protect our community or services, or if you create risk or legal exposure for us, violate these Terms of Use or our policies (including our Instagram Community Guidelines), if you repeatedly infringe other people's intellectual property rights, or where we are permitted or required to do so by law. We can also terminate or change the Service, remove or block content or information shared on our Service, or stop providing all or part of the Service if we determine that doing so is reasonably necessary to avoid or mitigate adverse legal or regulatory impacts on us. If you believe your account has been terminated in error, or you want to disable or permanently delete your account, consult our Help Center.When you request to delete content or your account, the deletion process will automatically begin no more than 30 days after your request. It may take up to 90 days to delete content after the deletion process begins. While the deletion process for such content is being undertaken, the content is no longer visible to other users, but remains subject to these Terms of Use and our Data Policy. After the content is deleted, it may take us up to another 90 days to remove it from backups and disaster recovery systems.

Content will not be deleted within 90 days of the account deletion or content deletion process beginning in the following situations:

where your content has been used by others in accordance with this license and they have not deleted it (in which case this license will continue to apply until that content is deleted); or

where deletion within 90 days is not possible due to technical limitations of our systems, in which case, we will complete the deletion as soon as technically feasible; or

where deletion would restrict our ability to: investigate or identify illegal activity or violations of our terms and policies (for example, to identify or investigate misuse of our products or systems); protect the safety and security of our products, systems, and users; comply with a legal obligation, such as the preservation of evidence; or comply with a request of a judicial or administrative authority, law enforcement or a government agency; in which case, the content will be retained for no longer than is necessary for the purposes for which it has been retained (the exact duration will vary on a case-by-case basis).

If you delete or we disable your account, these Terms shall terminate as an agreement between you and us, but this section and the section below called "Our Agreement and What Happens if We Disagree" will still apply even after your account is terminated, disabled, or deleted.

**Question**: According to the terms, 30 days after Ive asked to delete content, can other users see that content?

**Answer**: No

---

**Contract**: 16. Termination You may terminate these Terms at any time and for any reason by deleting your Account and discontinuing use of all Services. If you stop using the Services without deactivating your Account, your Account may be deactivated due to prolonged inactivity.

We may suspend or terminate your Account, moderator status, or ability to access or use the Services at any time for any or no reason, including for violating these Terms or our Content Policy.

The following sections will survive any termination of these Terms or of your Account: 4 (Your Content), 6 (Things You Cannot Do), 10 (Indemnity), 11 (Disclaimers), 12 (Limitation of Liability), 13 (Governing Law and Venue), 16 (Termination), and 17 (Miscellaneous).

17. Miscellaneous These Terms constitute the entire agreement between you and us regarding your access to and use of the Services. Our failure to exercise or enforce any right or provision of these Terms will not operate as a waiver of such right or provision. If any provision of these Terms is, for any reason, held to be illegal, invalid, or unenforceable, the rest of the Terms will remain in effect. You may not assign or transfer any of your rights or obligations under these Terms without our consent. We may freely assign any of our rights and obligations under these Terms.

**Question**: Will certain terms remain in force notwithstanding a users termination of the service?

**Answer**: Yes

Table 21: Task examples.

| Original wording | Alternative wording |
| --- | --- |
| Am I allowed to be paid for writing a Wikipedia article, assuming I disclose who's paying me? | Are Wikipedia contributors permitted to receive payment in respect of their contributions, provided they disclose the identity of the person or institution providing such payment? |

Table 22: Example of reworded question.

## G.6 Contract NLI Tasks

In LEGALBENCH, the Contract NLI tasks are denoted as `contract_nli_*`.

**Task**  The Contract NLI tasks require a LLM—given an excerpt of a contract and an assertion about the legal effect of that excerpt—to determine whether the assertion is supported or unsupported by the excerpt.

**Construction process**  These tasks are constructed by transforming data released by [72]. The original dataset consists of 607 contracts and 17 assertions (e.g., "Receiving Party shall not disclose the fact that Agreement was agreed or negotiated"). Each contract is labeled for each assertion as supporting, negating, or not mentioning the assertion. Please refer to the original paper for details on annotation.

We restructure this dataset for a short-context LLM setting. Specifically, we treat each assertion as a separate task, where the objective is to determine whether a contract excerpt is supportive (or not) of the assertion. For each instance where a contract is supportive of an assertion, [72] has annotated the excerpt of the contract that is supportive. When creating a task, we use the supportive excerpts for the assertion from the test set as positive instances. To generate negative instances, we combine excerpts where the assertion is contradicted with a random sample of excerpts associated with other assertions. We treat both groups of excerpts as instances which are "unsupportive" of the assertion. We transform the assertion into a Yes/No question, where the LLM is asked to determine if a clause satisfies the assertion.

Table 23 lists each task, the assertion associated with the task, and an example of an excerpt which supports the assertion.

### Table 23: ContractNLI Tasks

| Task |
| --- |
| **Task name**: contract_nli_return_of_confidential_information
**Question**: Identify if the clause provides that the Receiving Party shall destroy or return some Confidential Information upon the termination of Agreement.
**Example**: Upon receipt by the Recipient of a written demand from the Disclosers: 8.1.1 the Recipient must return or procure the return to the Disclosers or, as the Disclosers may require, destroy or procure the destruction of any and all materials containing the Confidential Information together with all copies; 8.1.2 if the Disclosers requires, the Recipient must provide the Disclosers with a certificate or such other evidence as the Disclosers may reasonably require duly signed or executed by an officer of the Recipient confirming that the Recipient has complied with all of its obligations under this Agreement including about return, destruction and deletion of Confidential Information and media; 8.1.3 the Recipient must delete or procure the deletion of all electronic copies of Confidential Information; and 8.1.4 the Recipient must make, and procure that the Authorised Persons shall make, no further Use of the Confidential Information. |
| **Task name**: contract_nli_no_licensing
**Question**: Identify if the clause provides that the Agreement shall not grant Receiving Party any right to Confidential Information.
**Example**: No license to the receiving party under any trade secrets or patents or otherwise with respect to any of the Proprietary Information is granted or implied by conveying proprietary Information or other information to such party, and none of the information transmitted or exchanged shall constitute any representation, warranty, assurance, guaranty or inducement with respect to the infringement of patents or other rights of others. |
| **Task name**: contract_nli_confidentiality_of_agreement
**Question**: Identify if the clause provides that the Receiving Party shall not disclose the fact that Agreement was agreed or negotiated.
**Example**: In addition, except as permitted herein, Recipient shall not disclose the fact that the parties are exchanging Confidential Information and having discussions. In connection therewith, it is agreed that no public release or disclosure of any contemplated transaction shall be made except by a mutually agreed disclosure except that each party may make such disclosure if advised by its outside securities counsel in writing that such disclosure is required; PROVIDED, HOWEVER, that in such event such party will notify the other party that it intends, as a preliminary matter, to take such action and the outside securities counsel of such party shall first discuss the mater with the outside securities counsel of the other party before any definitive decision is made on the disclosure. |
| **Task name**: contract_nli_explicit_identification
**Question**: Identify if the clause provides that all Confidential Information shall be expressly identified by the Disclosing Party. |

**Task**

**Example**: 1. As used herein, the term "Proprietary Information" refers to any and all Information of a confidential, proprietary, or secret nature which is applicable to or related In any way to (i) the business, present or future, of the Disclosing Party, (ii) the research and development or investigations of the Disclosing Party or (iii) the business of any customer of the Disclosing Party; provided, in each case, that such information is delivered to the Receiving Party by the Disclosing Party and (a) is marked or identified in writing as "Confidential", (b) if verbal or visual disclosure, is identified as "Confidential" in a writing within ten (10) business days of such disclosure, or

**Task name**: contract_nli_survival_of_obligations
**Question**: Identify if the clause provides that ome obligations of Agreement may survive termination of Agreement.
**Example**: b. This Agreement shall be valid when signed by duly authorised representatives of the Parties and shall be binding on each Party for 10 (ten) years as from the date of signature of the last signatory, even if at the end of the negotiations a data sharing agreement is not signed between the Parties, or until such time as the Information enters into the public domain.

**Task name**: contract_nli_permissible_development_of_similar_information
**Question**: Identify if the clause provides that the Receiving Party may independently develop information similar to Confidential Information.
**Example**: "Confidential Information" of a disclosing party ("Discloser") means the following, regardless of its form and including copies made by the receiving party ("Recipient"), whether the Recipient becomes aware of it before or after the date of this Agreement: except where that information is: Independently developed by the Recipient without use, directly or indirectly of Confidential Information received from the Discloser.

**Task name**: contract_nli_permissible_post-agreement_possession
**Question**: Identify if the clause provides that the Receiving Party may retain some Confidential Information even after the return or destruction of Confidential Information.
**Example**: 9. Upon the Disclosing Party's written request, the Receiving Party shall (at the Receiving Party's election) promptly return or destroy (provided that any such destruction shall be certified by a duly authorized Representative of the Receiving Party) all Confidential Information of the Disclosing Party and all copies, reproductions, summaries, analyses or extracts thereof or based thereon (whether in hard-copy form or an intangible media, such as electronic mail or computer files) in the Receiving Party's possession or in the possession of any Representative of the Receiving Party; provided, however: (i) that if a legal proceeding has been instituted to seek disclosure of the Confidential Information, such material shall not be destroyed until the proceeding is settled or a final judgment with respect thereto has been rendered; (ii) that the Receiving Party shall not, in connection with the foregoing obligations, be required to identify or delete Confidential Information held electronically in archive or back-up systems in accordance with general systems archiving or backup policies; and (iii) that the Receiving Party shall not be obligated to return or destroy Confidential Information of the Disclosing Party to the extent the Receiving Party is required to retain a copy pursuant to applicable law, and further provided that the Receiving Party will not, and the Receiving Party will use reasonable measures to cause its employees not to, access such Confidential Information so archived or backed-up.

**Task name**: contract_nli_inclusion_of_verbally_conveyed_information
**Question**: Identify if the clause provides that Confidential Information may include verbally conveyed information.
**Example**: I acknowledge that The Business Partnership has provided, and/or has agreed to provide in the future, to me information of a confidential or proprietary nature (the Confidential Information) Confidential Information shall mean any information or data relating to any clients of The Business Partnership business or affairs disclosed whether in writing, orally or by any other means.

**Task name**: contract_nli_sharing_with_third-parties
**Question**: Identify if the clause provides that the Receiving Party may share some Confidential Information with some third-parties (including consultants, agents and professional advisors).
**Example**: Receiving Party shall carefully restrict access to Sensitive Information to employees, contractors and third parties as is reasonably required and shall require those persons to sign nondisclosure restrictions at least as protective as those in this Agreement.

**Task name**: contract_nli_permissible_copy
**Question**: Identify if the clause provides that the Receiving Party may create a copy of some Confidential Information in some circumstances.

| Task |
| --- |

**Example**: If any party makes copies of the Confidential Information of the other party, such copies shall also constitute Confidential Information and any and all confidential markings on such documents shall be maintained.

**Task name**: contract_nli_notice_on_compelled_disclosure
**Question**: Identify if the clause provides that the Receiving Party shall notify Disclosing Party in case Receiving Party is required by law, regulation or judicial process to disclose any Confidential Information.
**Example**: If the Receiving Party or its Representatives are requested or required in any judicial, arbitral or administrative proceeding or by any governmental or regulatory authority to disclose any Evaluation Material (whether by deposition, interrogatory, request for documents, subpoena, civil investigative demand, or otherwise), or the Receiving Party is so requested or required to disclose any of the facts disclosure of which is prohibited under paragraph (3)(e) of this Agreement, the Receiving Party shall give the Furnishing Party prompt notice of such request so that the Furnishing Party may seek an appropriate protective order or other appropriate remedy and/or waive compliance with the provisions of this Agreement, and, upon the Furnishing Party's request and at the Furnishing Party's expense, shall reasonably cooperate with the Furnishing Party in seeking such an order. (d) Notice If either Party proposes to make any disclosure in reliance on clause (i) above, the disclosing Party shall, to the extent practicable, provide the other Party with the text of the proposed disclosure as far in advance of its disclosure as is practicable and shall in good faith consult with and consider the suggestions of the other Party concerning the nature and scope of the information it proposes to disclose. Notwithstanding the foregoing, a Party may make such public announcement or public statement if in the opinion of such Party's outside counsel or General Counsel, such public announcement or public statement is necessary to avoid committing a violation of law or of any rule or regulation of any securities association, stock exchange or national securities quotation system on which such Party's securities are listed or trade. In such event, the disclosing Party shall use its reasonable best efforts to give advance notice to the other Party and to consult with the other Party on the timing and content of any such public announcement or public statement.

**Task name**: contract_nli_permissible_acquirement_of_similar_information
**Question**: Identify if the clause provides that the Receiving Party may acquire information similar to Confidential Information from a third party.
**Example**: For the purposes of this Agreement, the term "Confidential Information" shall mean all trade secrets and confidential or proprietary information (and any tangible representation thereof) owned, possessed or used in connection with The Company Business or by the Buyer Parties and its Affiliates; provided, however, that "Confidential Information" does not include information which is or becomes generally available to the public other than as a result of a disclosure by a Seller Party..

**Task name**: contract_nli_sharing_with_employees
**Question**: Identify if the clause provides that the Receiving Party may share some Confidential Information with some of Receiving Party's employees.
**Example**: We and our representatives will keep the Evaluation Materials completely confidential; provided, however, that (i) any of such information may be disclosed to those of our directors, officers, employees, agents, representatives (including attorneys, accountants and financial advisors), lenders and other sources of financing (collectively, "our representatives") who we reasonably determine need to know such information for the purpose of evaluating a Possible Transaction between us and the Company (it being understood that our representatives shall be informed by us of the confidential nature of such information and shall be directed by us, and shall each agree to treat such information confidentially) and

**Task name**: contract_nli_limited_use
**Question**: Identify if the clause provides that the Receiving Party shall not use any Confidential Information for any purpose other than the purposes stated in Agreement.
**Example**: 2.1. A Receiving Party agrees: 2.1.2. to use the Confidential Information of the other solely in, and to the extent necessary for the Purpose and not to copy or use any Confidential Information of the other save to the extent necessary for the Purpose;

**Significance and value**    The Contract NLI tasks evaluate an LLM's capacity to reason over the rights and obligations created by a contract. The ability to perform this skill is essential to many types of legal work.

### G.7 Corporate Lobbying

In LEGALBENCH, the Corporate Lobbying task is denoted as `corporate_lobbying`.

**Background** A significant amount of effort is devoted to identifying developing sources of law which implicate client or issue interests. Examples of such sources include: legislative bills, proposed regulations, or in-progress litigation. Identifying these sources serves multiple purposes. From a scholarly standpoint, researchers often aggregate sources into issue-focused databases, enabling them to identify emerging trends or patterns across different sources [36]. From an advocacy standpoint, identifying sources allows affected groups to better understand how their rights or obligations may be affected, and how to focus efforts on interacting with courts, legislatures, and other governmental bodies [15, 81, 43].

**Task** The Corporate Lobbying task requires an LLM to determine whether a proposed Congressional bill may be relevant to a company based on a company's self-description in its SEC 10K filing. The following information about a bill and a company are available:

- The title of the bill.
- A summary of the bill.
- The name of the company.
- A description of the company.

We expect higher accuracy of LLM predictions if we were to provide the model with more data about a bill, and especially if we provide it with more data about a company. Proprietary applications of this approach could leverage significant internal company data. More expensive deployments could leverage the full text of the bill

**Construction process** This data was manually labeled. This work was an extension of the research described in [90].

**Significance and value** Determining whether a particular bill is relevant for a company requires (1) identifying the legal consequences of the bill, and (2) whether those consequences are relevant to a company's business model, structure, or activities. As discuss above, this type of prognostication is a common legal practice. For instance, law firms regularly publish "client alerts" which seek to keep clients updated on new legal developments [120].

| Class | Number of samples |
|-------|-------------------|
| No    | 345               |
| Yes   | 145               |

Table 24: Test set class distribution

| Field | Text |
|---|---|
| Bill Title | A bill to provide standards relating to airline travel by Federal employees for official business. |
| Bill Summary | Fly Smart Act
This bill establishes standards for airline travel by federal employees for official business, including a general requirement to use coach-class accommodations and a ban on military aircraft for domestic official travel. It allows use of first-class and business class for federal employees under certain circumstances, such as to accommodate a disability or special need or because of exceptional security circumstances |
| Company Name | Alaska Air Group, Inc. |
| Company Description | Virgin America has been a member of Air Group since it was acquired in 2016. In 2018, Virgin America and Alaska combined operating certificates to become a single airline, and legally merged into a single entity. The Company also includes McGee Air Services, an aviation services provider that was established as a wholly-owned subsidiary of Alaska in 2016. Together with our regional partner airlines, we fly to 115 destinations with over 1,200 daily departures through an expansive network across the United States, Mexico, Canada, and Costa Rica. With global airline partners, we provide our guests with a network of more than 900 destinations worldwide. Our adjusted net income was $554 million, which excludes merger-related costs, special items and mark-to-market fuel hedge adjustments. Refer to "Results of Operations" in Management's Discussion and Analysis for our reconciliation of Non-GAAP measures to the most directly comparable GAAP measure. Mainline - includes scheduled air transportation on Alaska's Boeing or Airbus jet aircraft for passengers and cargo throughout the U.S., and in parts of Canada, Mexico, and Costa Rica. other third-party carriers' scheduled air transportation for passengers across a shorter distance network within the U.S. under capacity purchase agreements (CPA). Horizon - includes the capacity sold to Alaska under CPA. Expenses include those typically borne by regional airlines such as crew costs, ownership costs and maintenance costs. We believe our success depends on our ability to provide safe air transportation, develop relationships with guests by providing exceptional customer service and low fares, and maintain a low cost structure to compete effectively. In 2018 , we focused much of our energy on the integration of Virgin America, completing over 95% of our integration milestones. In January 2018, Alaska and Virgin America received a Single Operating Certificate (SOC) from the Federal Aviation Administration (FAA), which recognizes Alaska and Virgin America as one airline. In April 2018, we transitioned to a single Passenger Service System (PSS), which allows us to provide one reservation system, one website and one inventory of flights to our guests. This transition to a single PSS enables us to unlock many of the revenue synergies expected from the acquisition, and to provide consistent branding to our guests at all airport gates, ticketing, and check-in areas. The two most important milestones we have yet to complete include combining the maintenance operations of Boeing and Airbus, and reconfiguring our Airbus fleet. In 2018 , we painted 33 Airbus aircraft with the Alaska livery and we are in process of reconfiguring all Airbus aircraft to achieve a cabin experience for our guests that is consistent with our Boeing fleet. In early 2019, we will also complete the integration of our crew management systems and aim to reach a collective bargaining agreement with our aircraft technicians, the last remaining labor group that has not yet reached a joint collective bargaining agreement. With the integration largely behind us, we remain committed to our vision to become the favorite airline for people on the West Coast. The acquisition of Virgin America positioned us as the fifth largest airline in the U.S., with an unparalleled ability to serve West Coast travelers. ' evolving needs by offering a relevant network and schedule, upgrading our onboard offerings, and retaining our unique West Coast vibe. Some of the more notable product enhancements underway include adding high-speed satellite connectivity to our entire Boeing and Airbus fleets, updating and expanding our airport lounges, and working with the Port of Seattle to open a state-of-the-art 20-gate North Satellite Concourse 4 at Sea-Tac Airport, including a 15,000 square-foot flagship lounge. We have also introduced new food and beverage menus, which include more fresh, local, and healthy offerings including salads, protein plates, and fresh snacks, as well as new beverage offerings, including craft beers, juices and an updated wine selection. We are also active in the communities we serve and strive to be an industry leader in environmental and community stewardship. |

Table 25: An example of a relevant bill for `corporate_lobbying`.

## G.8 Definition Tasks

In LEGALBENCH, the Definition Tasks are denoted as `definition_classification` and `definition_extraction`.

**Background** Judicial opinions regularly involve *definition*, assigning a particular meaning to words or phrases (Let us define words and phrases as "terms"). Definition of terms can occur when judges introduce or discuss legal concepts (e.g. *parol evidence*), and it frequently occurs when judges interpret terms in legal texts. This can include language from past judicial opinions and language appearing in legal texts like contracts, statutes, and the Constitution. Historically, interpreters have often evaluated the definition(s) of individual words. For example, in interpreting the meaning of "keep and bear arms" in the Second Amendment, courts consider the definition(s) of individual words (like "bear"). This approach—focusing on terms' definitions—has only increased in recent decades with the rise of textualist approaches to constitutional and statutory interpretation.

Judicial opinions define a wide range of terms, including ordinary terms, legal terms, and scientific terms. They also appeal to a wide range of defining sources, including ordinary dictionaries, legal dictionaries, and legal texts. For an example of the last, consider statutory definitions: 1 U.S.C. 1 offers generally applicable definitions of many frequent statutory terms.

It is useful for lawyers to identify *when definition occurs* (definition classification), as well as *which terms* have been defined (definition extraction). These tasks might seem simple at first. There are some intuitively plausible indicators of definition classification and extraction. For example, defined terms often (but not always) appear in quotation marks or near a citation to a dictionary.

However, these tasks are not entirely straightforward. Indicators like quotation will not lead to perfect definition classification and extraction. Consider for example, this sentence from the dataset related to the definition of "confidential": *The term "confidential" meant then, as it does now, "private" or "secret." Webster's Seventh New Collegiate Dictionary 174 (1963).*[11] As another example from the dataset, consider this definition of "brought": *But a natural reading of § 27's text does not extend so far. "Brought" in this context means "commenced," Black's Law Dictionary 254 (3d ed. 1933).*[12] Other examples exclusively quote the definition, rather than defined terms: *Stare decisis ("to stand by things decided") is the legal term for fidelity to precedent. Black's Law Dictionary 1696 (11th ed. 2019).*[13] In all of these examples, the presence of a dictionary would not indicate which term is extracted. In other examples, there is no dictionary cited; there is not a perfect correlation between dictionary citation and classification of a sentence as a defining one.[14]

**Tasks** The Definition Classification task requires an LLM to determine–given an excerpt from a Supreme Court opinion–whether the excerpt is defining any term (Yes/No). The Definition Extraction task requires an LLM to determine–given an excerpt from a Supreme Court opinion–which term the excerpt is defining (Open-ended response).

**Construction process** An original hand-coded dataset was constructed to study how the Supreme Court relies on dictionaries over time. Any case citing a dictionary was included in the dataset, and human coders identified relevant excerpts that defined terms and *which* terms were defined.

That dataset has been repurposed for the task here. For the definition extraction task, the original dataset includes the relevant information (excerpts, with the defined term coded separately).

For the definition classification task, the original dataset includes examples of language defining terms. To create a set of non-defining language, Neel Guha randomly selected similarly long excerpts of text from the same Supreme Court opinions. Kevin Tobia analyzed those randomly selected excerpts, identifying any that include definitions (for removal). The resulting dataset has 691 sentences which define sentences, and 646 sentences which do not.

**Significance and value** This is not a particularly difficult task for human lawyers, and it is unlikely that LLMs would replace lawyers as experts in this process. However, it is possible that LLMs successful in these tasks could provide beneficial legal research roles (e.g. quickly identifying all prior definitions of a specific term in a particular jurisdiction).

Moreover, the definition extraction task serves as a useful test of LLMs abilities, given the task's open-ended nature. The task is not limited to a small set of possible answers (e.g. Yes, No). Rather, it requires identifying which term of all terms in an excerpt is defined. Most of these choices will admit of over ten possible answers (i.e. excerpts of over ten words). Moreover, there is great variety in the language used across the examples. There are hundreds of possible answers, across all items.

---

[11] Food Mktg. Inst. v. Argus Leader Media, 139 S. Ct. 2356, 2363 (2019).

[12] Merrill Lynch, Pierce, Fenner & Smith Inc. v. Manning, 136 S. Ct. 1562, 1568 (2016).

[13] June Medical Services L.L.C. v. Russo, 140 S. Ct. 2103, 2134 (2020).

[14] E.g. "And "remuneration" means "a quid pro quo," "recompense" or "reward" for such services. Id., at 1528." BNSF Ry. Co. v. Loos, 139 S. Ct. 893, 905 (2019).

| Sentence | Definition sentence? |
|---|---|
| The risk of that consequence ought to tell us that something is very wrong with the Court's analysis. | No |
| This term has long referred to a class of expenses commonly recovered in litigation to which attorney's fees did not traditionally belong. See Black's Law Dictionary 461 (1891) (defining "expensae litis" to mean "generally allowed" costs); 1 J. Bouvier, Law Dictionary 392 (1839) (defining the term to mean the "costs which are generally allowed to the successful party"); id., at 244 (excluding from the definition of "costs" the "extraordinary fees [a party] may have paid counsel"). | Yes |

Table 26: Examples for `definition_classification`.

| Sentence | Defined term |
|---|---|
| The term "plaintiff" is among the most commonly understood of legal terms of art: It means a "party who brings a civil suit in a court of law." Black's Law Dictionary 1267 (9th ed. 2009) see also Webster's Third New International Dictionary 1729 (1961)" | plaintiff |
| The ordinary understanding of law enforcement includes not just the investigation and prosecution of offenses that have already been committed, but also proactive steps designed to prevent criminal activity and to maintain security. | law enforcement |

Table 27: Examples for `definition_extraction`.

## G.9 Diversity Jurisdiction

In LEGALBENCH, the Diversity Jurisdiction tasks are denoted as `diversity_*`.

**Background**  Diversity jurisdiction is one of two ways in which a federal court may have jurisdiction over a lawsuit pertaining to state law. Diversity jurisdiction exists when there is (1) complete diversity between plaintiffs and defendants, and (2) the amount-in-controversy (AiC) is greater than $75,000.

"Complete diversity" requires that there is no pair of plaintiff and defendant that are citizens of the same state. However, it is acceptable for multiple plaintiffs to be from the same state, or for multiple defendants to be from the same state.

The AiC requirement allows for certain forms of aggregation. Specifically, if plaintiff **A** asserts two independent claims against defendant **B**, the value of the claims may be added together when considering if the AiC requirement is met. However, a plaintiff may not aggregate the value of claims against two separate defendants, and two plaintiffs may not aggregate claims against the same defendant.

**Tasks**  We define six different tasks, each of which tests the diversity jurisdiction rule under a different pattern of facts. The diversity jurisdiction tasks are:

- `diversity_1`: The fact patterns consists of one plaintiff, one defendant, and one claim per plaintiff-defendant pair.
- `diversity_2`: The fact patterns consists of one plaintiff, two defendants, and one claim per plaintiff-defendant pair.
- `diversity_3`: The fact patterns consists of one plaintiff, one defendant, and two claims per plaintiff-defendant pair.
- `diversity_4`: The fact patterns consists of two plaintiffs, one defendant, and one claim per plaintiff-defendant pair.
- `diversity_5`: The fact patterns consists of two plaintiffs, one defendant, and two claims per plaintiff-defendant pair.
- `diversity_6`: The fact patterns consists of two plaintiffs, two defendants, and two claims per plaintiff-defendant pair.

**Construction process**  We programmatically construct a dataset to test the diversity jurisdiction. We generate randomness over the names of the parties, the claims, and the amounts.

**Significance and value**  It is extremely unlikely LLMs would ever be used to evaluate diversity jurisdiction in practical settings. However, because the task is considered extremely simplistic—and one that first year law students are expected to perform perfectly—it offers a useful evaluation benchmark for LLMs. The structure of the task is potentially non-trivial for LLMs, as it requires identifying the relationships between parties (i.e., who are plaintiffs and defendants), understanding which claims may be aggregated, and computing whether the aggregated amounts meet the AiC requirement.

| Task | Facts | Diversity Jurisdiction? |
|------|-------|-------------------------|
| diversity_1 | Oliver is from Oregon. William is from Oregon. Oliver sues William for defamation for $3,000. | No |
| diversity_1 | James is from South Dakota. Sophia is from Virginia. James sues Sophia for negligence for $9,010,000. | Yes |
| diversity_2 | Benjamin is from South Carolina. Amelia is from Indiana. Mia is from South Carolina. Benjamin sues Amelia and Mia each for wrongful eviction for $22,000. | No |
| diversity_2 | James is from Colorado. Elijah is from West Virginia. Theodore is from Washington. James sues Elijah and Theodore each for negligence for $2,864,000. | Yes |
| diversity_3 | Ava is from Rhode Island. Theodore is from Rhode Island. Ava sues Theodore for securities fraud for $70,000 and trespass for $6,000. | No |
| diversity_3 | Charlotte is from Colorado. Harper is from Oklahoma. Charlotte sues Harper for breach of contract for $74,000 and securities fraud for $88,000. | Yes |
| diversity_4 | Harper is from New Jersey. Benjamin is from Colorado. Isabella is from Colorado. Harper and Benjamin both sue Isabella for breach of contract for $6,165,000. | No |
| diversity_4 | Noah is from Indiana. Sophia is from West Virginia. Benjamin is from Montana. Noah and Sophia both sue Benjamin for defamation for $3,996,000. | Yes |
| diversity_5 | Noah is from Idaho. Elijah is from Connecticut. Theodore is from Wyoming. Noah and Elijah both sue Theodore for medical malpractice for $57,000 and legal malpractice for $16,000. | No |
| diversity_5 | Charlotte is from Oregon. Mia is from Virginia. Elijah is from Tennessee. Charlotte and Mia both sue Elijah for trademark infringement for $57,000 and medical malpractice for $20,000. | Yes |
| diversity_6 | Lucas is from South Dakota. Amelia is from New Hampshire. Benjamin is from South Dakota. Benjamin is from South Dakota. Lucas and Amelia both sue Benjamin for negligence for $16,000 and wrongful eviction for $76,000. Lucas and Amelia both sue Olivia for medical malpractice for $3,000 and breach of contract for $76,000. | No |
| diversity_6 | Emma is from Kansas. Noah is from Delaware. Elijah is from South Dakota. Elijah is from New Jersey. Emma and Noah both sue Elijah for trademark infringement for $4,000 and trespass for $85,000. Emma and Noah both sue Liam for negligence for $10,000 and defamation for $67,000. | Yes |

Table 28: Examples for the Diversity Tasks.

## G.10 Function of Decision Section

In LEGALBENCH, the Function of Decision Section task is denoted as `function_of_decision_section`.

**Background**   In common-law legal systems, written judicial decisions serve two functions. First, they resolve the dispute that litigants brought before the court and explain the reason for the court's decision. Second, they become new law, binding on future parties and future courts should another case arise that presents sufficiently similar facts.

Because judicial decisions not only describe the law, but are themselves the law, lawyers in common-law legal systems must be able to read and digest case law to extract key legal principles and apply those principles to their own cases. This skill takes time and practice to develop.

Importantly, not every word in a judicial decision is binding, only the facts and reasoning that were required for the court to reach its decision. Thus, lawyers must distinguish important from trivial facts across numerous past decisions before they can conclude what the law on a particular issue is. One of the most foundational case-reading skills is the ability to review a legal decision and identify the function that each section of the decision serves. In the American legal education system, this skill is taught beginning in the first year of law school, often by encouraging students to identify the function of each section of a decision. A typical classification scheme is as follows:

- Facts: A section of the decision that recounts the historical events and interactions between the parties that gave rise to the dispute.
- Procedural History: A section of the decision that describes the parties' prior legal filings and prior court decisions that led up to the issue to be resolved by the decision.
- Issue: A section of the decision that describes a legal or factual issue to be considered by the court.
- Rule: A section of the decision that states a legal rule relevant to resolution of the case.
- Analysis: A section of the decision that evaluates an issue before the court by applying governing legal principles to the facts of the case
- Conclusion: A section of the decision that articulates the court's conclusion regarding a question presented to it.
- Decree: A section of the decision that announces and effectuates the court's resolution of the parties' dispute, for example, granting or denying a party's motion or affirming, vacating, reversing, or remanding a lower court's decision.

Identifying the function of sections within judicial decisions is a fundamental skill for lawyers in common-law legal systems. Without it, precedent-based legal reasoning would be impossible.

**Task**   The Function of Decision Sections task requires an LLM to determine–given a one-paragraph excerpt of a legal decision–which of the seven functions above that paragraph serves in the context of the entire decision.

**Construction process**   We created a dataset of paragraphs from legal decisions, classified into one of the seven functions above. Paragraphs were taken from decisions in West Publishing's fourth Federal Reporter series, which publishes the decisions of the United States Courts of Appeals. To avoid selection bias and achieve a degree of randomness, paragraphs were selected from sequential decisions, in the order they appeared, spanning all areas of civil and criminal law that fall within the jurisdiction of the federal courts.

**Significance and value**   Beginning law students may initially have trouble identifying the function of a particular section within a judicial opinion, but it quickly becomes a simple task. LLMs would not be called on to perform this task in the actual practice of law, but because it is a foundational legal reasoning skill, it provides a useful measure of reasoning progress for LLMs.

| Class | Number of samples |
|---|---|
| Facts | 49 |
| Procedural History | 58 |
| Issue | 51 |
| Rule | 56 |
| Analysis | 56 |
| Conclusion | 50 |
| Decree | 47 |

Table 29: Test set class distribution.

| Excerpt | Function |
|---|---|
| The Commission's notice and orders, however, are to the contrary. From the very outset, the Commission has made clear that the Governance Order was no more than a call for a proposal that would then be subject to further notice, comment, and revision. | Analysis |
| Donna and Hurley contend that the Supreme Court's decision in Honeycutt v. United States, —— U.S. ——, 137 S. Ct. 1626, 198 L.Ed.2d 73 (2017), should be applied retroactively to invalidate the forfeiture judgments against them. | Conclusion |
| For the reasons stated, we affirm the district court's judgment. | Decree |
| "The Game of Life" is a classic family board game, introduced in 1960 by the Milton Bradley Company to great success. This case involves a long-running dispute between Rueben Klamer, a toy developer who came up with the initial concept of the game, and Bill Markham, a game designer whom Klamer approached to design and create the actual game prototype. Eventually, their dispute (which now involves various assignees, heirs, and successors-in-interest) reduced to one primary issue: whether the game qualified as a "work for hire" under the Copyright Act of 1909. If it did, Markham's successors-in-interest would not possess the termination rights that would allow them to reassert control over the copyright in the game. After considering the evidence produced at a bench trial, the district court concluded that the game was, indeed, such a work. Plaintiff-appellants, who all trace their interest in the game to Markham, challenge that determination. We affirm. | Facts |
| Officers of the Puerto Rico Police Department watched Julio Casiano-Santana ("Casiano") engage in a drug deal. They arrested him, recovering a loaded pistol and three bags of crack cocaine from the scene. Casiano was charged with possession of a firearm in furtherance of a drug trafficking crime, 18 U.S.C. § 924(c)(1)(A)(i), two counts of possession with intent to distribute controlled substances, 21 U.S.C. § 841(a)(1) and (b)(1)(C), and possession of a firearm by a convicted felon, 18 U.S.C. § 922(g)(1). | Issue |
| On remand, the district court held a new sentencing hearing, in which Lawrence allocuted. Resentencing Transcript at 11–12, United States v. Lawrence, No. 03-cr-00092-CKK (D.D.C. Oct. 5, 2009), ECF No. 103. Lawrence told the court that, while incarcerated, he had "been trying to do the right things as far as * * * becoming a man so I can provide for my son, he's 11 and very big." Id. Lawrence's mother was "getting old" and does "the best that she can[,]" but his son had "health issues as far as * * * weight gain and a lot of other things." Id. at 12. Lawrence explained that he "just want[ed] a chance to be a father" to his son, and that he "was just hoping that it's possible that * * * I can get out in his life before * * * the streets * * * or anything that maybe I have done affect him[.]" Id. He said he wanted to "be a productive citizen[,]" and noted that he "read the Bible" and "attended church, school, [and] college." Id. He admitted that he had "gotten into some altercations," but "not because I wanted to, but it's prison, and you know, there's all types of people in prison." Id. While "making no excuses" for his actions, he said he "was just hoping the Court would have leniency" in his "particular case." Id. | Procedural History |
| The border between interpretation and bare consultation can be hazy and, therefore, "difficult to plot." Lawless, 894 F.3d at 18 (citing Livadas, 512 U.S. at 124 n.18, 114 S.Ct. 2068). This case, however, does not closely approach the border: on their face, Rose's state-law claims require more than bare consultation of the CBA. They substantially depend on construing the terms of the agreement (the CBA) that RTN and the Union negotiated. We explain briefly. | Rule |

Table 30: Examples for function_of_decision_section

### G.11  Hearsay

In LEGALBENCH, the hearsay task is denoted as `hearsay`.

**Background**   The Federal Rules of Evidence dictate that "hearsay" evidence is inadmissible at trial. Hearsay is defined as an "out-of-court statement introduced to prove the truth of the matter asserted." In determining whether a piece of evidence meets the definition of hearsay, lawyers ask three questions:

1. Was there a statement? The definition of statement is broad, and includes oral assertions, written assertions, and non-verbal conduct intended to communicate (i.e. *assert*) a message. Thus, for the purposes of the hearsay rule, letters, verbal statements, and pointing all count as statements.

2. Was it made outside of court? Statements not made during the trial or hearing in question count as being out-of-court.

3. Is it being introduced to prove the truth of the matter asserted? A statement is introduced to prove the truth of the matter asserted if its truthfulness is essential to the purpose of its introduction. Suppose that at trial, the parties were litigating whether Alex was a soccer fan. Evidence that Alex told his brother "I like soccer," would be objectionable on hearsay grounds, as (1) the statement itself asserts that Alex likes soccer, and (2) the purpose of introducing this statement is to prove/disprove that Alex likes soccer. In short, the truthfulness of the statement's assertion is central to the issue being litigated. However, consider if one of the parties wished to introduce evidence that Alex told his brother, "Real Madrid is the greatest soccer team in the world." This statement would **not** be hearsay. It's assertion—that Real Madrid is the greatest soccer team in the world—is unrelated to the issue being litigated. Here, one party is introducing the statement not to prove what the statement says, but to instead show that a particular party (i.e. Alex) was the speaker of the statement.

**Task**   Given a legal issue and a piece of prospective evidence, the LLM must determine whether the evidence constitutes hearsay under the above test.

We note that in practice, many pieces of evidence which are hearsay are nonetheless still admissible under one of the many hearsay exception rules. We ignore these exceptions for our purposes, and leave the construction of benchmarks corresponding to these exceptions for future work.

**Construction process**   We create the hearsay dataset by hand, drawing inspiration from similar exercises available in legal casebooks and online resources. The dataset consists of 5 slices, where each slice tests a different aspect of the hearsay rule. We randomly select 1 sample from each slice to be in the train set. The remainder of the slice constitutes the test set (for a total of 95 samples). The slices (with test set counts) are:

- Statement made in court ($n = 14$): Fact patterns where the statement in question is made during the course of testimony at trial. Thus, the statement is not hearsay.

- Non-assertive conduct ($n = 19$): Fact patterns where the evidence does not correspond to a statement. Hence, the hearsay rule is inapplicable.

- Standard hearsay ($n = 29$): Fact patterns where there is an oral statement, it is said out of court, and it is introduced to prove the truth of the matter asserted. Thus, these fact patterns correspond to hearsay.

- Non-verbal hearsay ($n = 12$): Fact patterns where the statement is hearsay, but made in writing or through assertive conduct (e.g. pointing).

- Not introduced to prove truth ($n = 20$): Fact patterns where an out-of-court statement is introduced to prove something other than what it asserts.

**Significance and value**   The hearsay rule is commonly taught in law school as part of Evidence. Law students are expected to understand the rule, and how to apply it. The hearsay task is interesting for LLM evaluation because it emphasizes multi-step reasoning—the test for hearsay encompasses several different steps, where each step differs in difficulty. These include:

- Event detection: The LLM must determine whether the fact pattern mentions a statement being made.

- Spatial reasoning: The LLM must determine whether the statement was made inside a court room.

- Argument extraction: The LLM must determine what the statement is asserting.

- Argument relevance: The LLM must finely determine whether the assertion is relevant to the issue being litigated.

| Facts | Hearsay? |
|---|---|
| On the issue of whether Will knew that the company intended to announce its drug trials had been cancelled, the fact that he told the jury that "he didn't know the first thing about how medicines worked." | No |
| On the issue of whether Gerald was alive immediately after being attacked by Kathryn, Gerald's statement, "I was attacked by Kathryn." | No |
| On the issue of whether Susan was familiar with Shakespeare, the fact that she had once played the role of Macbeth and recieved a standing ovation after her monologue. | No |
| To prove that the insured under a life policy is dead, his wife offers a death certificate. | Yes |
| On the issue of whether Albert bought a knife, Angela testified that he shook his head when she asked him. | Yes |
| On the issue of whether the brakes were faulty, Amy testifies that she heard Arthur claim that he thought something was wrong with the car. | Yes |

Table 31: Examples for hearsay

## G.12 Insurance Policy Interpretation

In LEGALBENCH, the Insurance Policy Interpretation task is denoted as `insurance_policy_interpretation`.

**Background**  Insurance disputes often arise when parties disagree on whether a claim is covered under a certain insurance policy. To study such disagreements in interpretation, researchers at Stanford recruited crowdsource workers to review a pair of an insurance policy and a claim and respond whether they believe the claim is covered. A policy-claim pair whose applicability the workers disagree with each other on suggests ambiguity in the policy text.

**Task**  The Insurance Interpretation task requires an LLM to review a pair of an insurance policy and a claim and determine whether the policy clearly covers the claim, clearly does not cover it, or if it is unclear whether it covers it or not.

**Construction process**  The clause-claim pairs are manually constructed before being reviewed by crowdsource workers [137]. To convert the numbers of Covered/Not_Covered/Can't_Decide responses to discrete labels, we first calculate the 95% multinomial confidence interval of the proportion of each response. We then choose the label for which the confidence interval lower bound is greater than or equal to .5. If no label has a lower bound $\geq$ .5, we classify the policy-claim pair as "It's ambiguous." This conversion process ensures that individual crowdsource workers do not arbitrarily sway the labels. Examples for each label can be found in Table 32.

---

**Policy**: Harper's insurance covers damage from "House Removal," which includes "damage to belongings that occurs while being stored by professional removal contractors."
**Claim**: Harper is moving to a new home on the other side of town. Because her old home has already sold and her new home is not yet ready for her to move in, she checks into a hotel and asks a professional moving company to store some of her belongings at the company warehouse. A couple days before she is set to move in, the warehouse floods, which ruins the items that the movers were storing for Harper. Harper files a claim with her insurance company for the damage to her belongings.
**Label**: Covered

---

**Policy**: Denise's insurance covers damage from "House Removal," defined as "damage to belongings caused while being removed by professional removal contractors from the home."
**Claim**: Denise is moving to a new home on the other side of town. She asks her uncle, a retired professional mover, to help move her belongings out of her current home. During the move, her uncle's truck is involved in a minor accident that damages several pieces of her furniture and other belongings. Denise files a claim with her insurance company for the damage to her belongings.
**Label**: Not Covered

---

**Policy**: Jason has insurance coverage against loss and damage from "Identity Theft," which excludes "identity theft connected with the policyholder's business."
**Claim**: Jason is a successful car salesman. One day, while Jason is at home, hackers manage to infiltrate Jason's home WiFi network. The hackers steal Jason's social security number and open a number of fraudulent lines of credit in his name. To resolve the fraud, Jason must spend thousands of dollars in legal fees. Jason files a claim with his insurance company for his losses.
**Label**: It's ambiguous.

---

Table 32: Examples for `insurance_interpretation`.

**Significance and value**  The ability to determine whether an insurance claim is covered under a given policy can significantly reduce claim processing time. It can also shine light on potential ambiguity in existing policies. Additionally, this task represents one of the rare benchmarks where an LLM is required to predict laypeople's legal interpretations, as we retrieve the ground truth labels based on crowdsourced responses.

### G.13 International Citizenship Questions

In LEGALBENCH, the International Citizenship Questions task is denoted as `international_citizenship_questions`.

**Background**  The GLOBALCIT Citizenship Law Dataset is a valuable resource that comprehensively categorizes citizenship acquisition and loss methods in 190 countries. It enables cross-country comparisons and offers insights into global trends in citizenship laws and examines 28 different ways in which citizenship can be acquired, as well as 15 ways laws allow citizenship to be lost. The original dataset is formulated as a tabular survey dataset. We change this survey format into Yes/No questions about specific countries and their laws as of 2020 resulting in 9300 question- answer pairs.

**Task**  The model must answer yes/no questions about global citizenship law.

| Question | Answer |
|---|---|
| Consider the country of Central African Republic. Does the country provide for acquisition of citizenship by a person who is in the public service (including military service) and, if so, under which conditions? | No |
| Consider the country of Bolivia. Does the country provide for involuntary loss of citizenship by a person who is adopted by or in legal guardianship of a citizen of another country and, if so, under which conditions? | No |
| Consider the country of Denmark. Which residence conditions does the country provide for residence-based acquisition? | Yes |
| Consider the country of Germany. Does the country require the demonstration of civic knowledge or cultural integration for residence-based acquisition? | Yes |

Table 33: Examples for international_citizenship_questions

**Contruction process**  We download the GLOBALCIT Citizenship Law Dataset [135] and craft a script that converts the tabular survey data into yes/no questions, appending information about the country and the time at which the survey was created.

**Significance and value**  Understanding knowledge about the law globally is important to evaluate. To successfully answer legal reasoning questions globally language models must be able to retrieve a rule and then reason about it.

### G.14 Learned Hand Tasks

In LEGALBENCH, the Learned Hand tasks are denoted as `learned_hand_*`.

**Background** A person may experience problems in many areas of their lives – their family, work, finances, housing, education, driving, and more – which legal professionals would recognize as being 'legal issues'. The person may not know that a problem with a credit card company, a landlord, a spouse, or an employer is a 'legal issue', or what terminology or categorization a lawyer would use to make sense of it.

The designation of a legal issue means that a person may benefit from getting specialized guidance from a legal professional to resolve this problem because they can guide them on their rights, liabilities, possible options, procedures, and specialized legal tasks. Not all people may want to pursue legal services to resolve a legal issue. But legal issue-spotting can help them both make sense of the problem they are experiencing, and what services and laws might be available if they wish to make use of them.

Legal professionals typically carry out issue-spotting during an intake process. They receive a person's verbal or written description of the situation they are in. Then the professional identifies the main legal issues that are apparent in this situation, starting at the main top-level categories and then sometimes proceeding to identify more specific sub-categories of legal issues. For example, the professional may identify that a person's situation involves a legal issue with the top-level category of 'housing' and specific sub-categories of 'possible eviction for non-payment of rent' and 'poor living conditions of their rental'.

The professional may identify multiple overlapping legal issue categories in one situation. For example, the professional may identify that a person has a housing law issue and a family law issue if their landlord is threatening to evict them because of the police being called to the rental home because of a domestic violence incident.

The main categories of legal issues that professionals would identify in people's situations are:

- **Benefits**: A situation would have a benefits issue if it involves the person attempting to resolve a problem with applying for, receiving, or discontinuing public benefits and social services from the government. This could include benefits that support them regarding food, disability, old age, housing, health, unemployment, child care, or other social needs.

- **Business**: A situation would have a business issue if the person is running a small business or nonprofit, and encounters a problem around incorporation, licenses, taxes, regulations, bankruptcies, or natural disasters. This category is not meant to apply to larger corporate legal issues, but rather the kinds of business problems that an individual might bring to a legal professional for help.

- **Consumer**: A situation would have a consumer legal issue if the person was dealing with problems around debt and money, insurance, consumer goods and contracts, taxes, or small claims about the quality of service.

- **Courts**: A situation would be categorized as a courts issue if the person is dealing with a problem around how to interact with the court system or with lawyers more broadly. This may involve the person attempting to follow legal procedures, court rules, or filing requirements, or it may involve them attempting to hire, manage, or address lawyers.

- **Crime**: A situation would have a crimes issue if the person is dealing with the criminal justice system as a defendant, victim, or family member. They may be experiencing problems around being investigated, searched, or charged with a crime, or going to a criminal trial and prison, or being a victim of a crime.

- **Divorce**: A situation would be categorized as a divorce issue if a person is dealing with a separation, divorce, or annulment while splitting with a spouse or partner. The problem may involve separation, spousal support, splitting money and property, child support, visitation, or following the related court process.

- **Domestic Violence**: A situation would have a domestic violence issue if the person is dealing with abuse with a partner, family member, or other intimate acquaintance. The situation may involve understanding rights and laws related to domestic violence, getting protective orders, enforcing them, reporting abuse, and dealing with collateral consequences to housing, finances, employment, immigration, and education.

- **Education**: A situation has an education issue if the person is dealing with a problem around school for themselves or a family member. The situation may involve accommodations for special needs, discrimination, student debt, discipline, or other issues in education.

- **Employment**: A situation would be identified as having an employment issue if the person has a problem with a job, including during the application process, during the job, or after ending employment. Problems may include discrimination, harassment, payment, unionizing, pensions, termination, drug testing, background checks, worker's compensation, classification as a contractor, or more.

- **Estates**: A situation would have an estates issue if a person is dealing with an estate, wills, or guardianship. This may include issues around end-of-life planning, health and financial directives, trusts, guardianships, conservatorships, and other estate issues that people and family deal with.

- **Family**: A situation would have a family law issue if a person is dealing with an issue involving a family member. This may include issues around divorce, child custody, domestic violence, adoption, paternity, name change, and other family issues.

- **Health**: A situation would be categorized as a health law issue if the person is dealing with problems around accessing health services or protecting their rights in medical settings. This may involve problems with accessing health care, paying for care, getting public benefits for care, privacy of medical records, problems with quality of care, or other issues.

- **Housing**: A situation would have a housing law issue if a person is dealing with problems around the housing where they live, or that they own. These include problems with renting a home, eviction, living conditions, discrimination, foreclosure, post-disaster housing, housing assistance, and more.

- **Immigration**: A situation would have an immigration issue if a person is not a full citizen in the US and is dealing with problems related to their status. This may include understanding visa options, working as an immigrant, political asylum, border searches, deportation, human trafficking, refugees, immigration court hearings, and more.

- **Torts**: A situation would be categorized as having a torts issue if the person is dealing with an accident or conflict with another person that involves some perceived harm. These problems may include a car accident, conflicts with neighbors, dog bites, bullying, harassment, data privacy breaches, being sued, or suing someone else.

- **Traffic**: A situation would have a traffic law issue if the person is experiencing a problem with traffic, parking, or car ownership. This might include problems with getting ticketed, getting or reinstating a driver's license, car accidents, purchasing a car, repossession, and more.

This set of categories is commonly used by legal professionals as they triage potential clients. The LIST Taxonomy from Stanford Legal Design Lab has formalized these categories into a machine-readable taxonomy, available at `https://taxonomy.legal/`. The LIST taxonomy builds on the taxonomies built by legal aid groups, like the Legal Services Corporation categories list that most legal aid groups use to encode their matters, signifying what issues they helped clients with.[15]. LIST also builds off of the legal aid community's National Subject Matter Index, which was a more extensive list of categories to further assist legal aid groups in tracking the issues they helped people with.[16].

Performing the Legal Issue-Spotting task requires parsing through informal wording and structures that a person may use to convey the situation they are struggling with. Typically the person is writing this narrative down in an informal, quick manner (like in an online intake form on a legal services website) or speaking it aloud (like on an intake hotline, or during an in-person interview). The narratives are not typically structured into a concise order. They use informal terminology rather than legal terms.

**Task**   The Legal Issue-spotting task requires an LLM to consider a person's narrative about their situation. The LLM must use this narrative to determine which legal issue category (or categories) apply to the person's situation.

**Construction process**   There is a crowdsourced dataset, created via the online labeling game Learned Hands, that has established when and how these legal issue categories apply to people's narratives. The narratives are drawn from the subreddit r/legaladvice, in which people share several lines or paragraphs about the situation they are dealing with, which they think might involve legal issues. The moderators of the subreddit are active in managing activity, so the posts do not contain personal identifying information or off-topic postings.

The Stanford Legal Design Lab and Suffolk LIT Lab built the Learned Hands game so that law students and lawyers could read narratives one by one, and then answer a series of yes-no-skip questions about what legal issue seems to be present. Once there are sufficient consistent votes for a certain label to apply, or to not apply, to a narrative, then the label is finalized. A narrative may have more than one label, as mentioned above.

**Significance and value**   The legal issue categorization helps the professional triage the person to the right services, resources, and procedures that can assist them in resolving the legal issue. If an LLM is able to identify legal issues in people's informal narratives, this demonstrates an ability to perform a key task in people's justice journeys. Issue-spotting by an LLM may be able to help a person who is just starting to explore whether or how they should engage with legal services, courts, or exercising their rights.

The issue-spotting task may be provided online, when people are visiting legal help websites and trying to find what guide, form, or service would best help them with their problem. Or it may be integrated into the intake

---

[15]See the LSC's list at `https://www.lsc.gov/i-am-grantee/grantee-guidance/lsc-reporting-requirements/case-service-reporting/csr-handbook-2017`

[16]See the NSMI database at `https://nsmi.lsntap.org/`

process that paralegals or justice workers carry out over hotlines or in-person, to speed up the often lengthy intake process.

## G.15 Legal Reasoning Causality

In LEGALBENCH, the Legal Reasoning Causality task is denoted as `legal_reasoning_causality`.

**Background**    In many legal domains, systematic evidential barriers hinder the substantiation of causal claims through direct evidence. To address these shortcomings, courts have recognized the power of statistical evidence in establishing causation in various contexts, such as product liability,[17] medical malpractice,[18] discrimination,[19] and more. For instance, when pursuing a labor discrimination claim, the plaintiff must establish that her protected trait was the underlying reason for the alleged discriminatory decision (e.g., firing or not hiring). However, direct evidence of discriminatory intent rarely exists, so it is often nearly impossible to refute the possibility that other (legitimate) differences between two employees or candidates were the cause for favoring one over the other. In such cases, litigants can and often do try to substantiate a causal link between the plaintiff's group affiliation and the defendant's behavior through statistical analysis. For instance, plaintiffs might send fictitious resumes that differ only by the suspected demographic characteristic,[20] akin to a field experiment. Likewise, statistical analysis of observational data that controls for the major factors affecting the employment practice can be used to demonstrate whether a specific social group suffers from inferior outcomes (relative to some control group) vis-à-vis a particular employer, landlord, or lender that engages with a sufficiently large number of employees or customers.[21]

**Task**    The "causal reasoning" task requires an LLM to determine whether the court's reasoning regarding the finding of whether a causal link exists between the plaintiff's protected trait and the allegedly discriminatory decision relied on either statistical or direct-probative evidence. It requires understanding the types of words that are used to describe statistical evidence in any given context (regression, correlation, variables, control, and more), and the extent to which those words relate to substantiating a finding of causality (as opposed to other legal components).

**Construction process**    We manually created a dataset of fifty-nine excerpts from court decisions in lawsuits alleging labor market discrimination filed in US Federal District Courts. First, fifty-nine court decisions involving claims of labor discrimination were identified using the LexisNexis database. Second, the passages in which the finding of causality appeared were identified and extracted. Third, we coded the passages as either relying on statistical evidence (e.g., regression analysis, findings of correlation, etc.) or on direct evidence (e.g., witnesses, documents, etc.).

We selected two random samples from each class to use as part of the train split.

**Significance and value**    The potential of LLMs to identify different types of legal reasoning in general, and the finding of causality in particular, has implications both for the legal profession and for the academic study of law and judicial decision-making. First, algorithmic tools are gradually being utilized by lawyers to assist them in preparing for litigation. Specifically, given the heterogeneity among judges, a key element of a successful litigation strategy is a lawyer's ability to construct their arguments based on the specific inclinations of the judge assigned to the case. Gaining an accurate understanding of judges' unique mode of reasoning (including, e.g., the types of evidence they tend to rely on), based on their prior decisions, is crucial for winning any lawsuit. Second, databases consisting of court decisions are the most common source for studying the law and judicial decision-making in legal academia. However, these databases are typically limited to rather technical information, such as the names of the parties and the judge(s), the legal area of the case, and the like. The essential part of any judicial opinion – the legal reasoning – is typically treated as a black box. An LLM that could classify the various types of legal reasoning – e.g., what evidence is used to establish causation – can facilitate studying judicial decision- making in ways currently not feasible at large scales.

---

[17]See, e.g, Neurontin v. Pfizer, 712 1st Cir. 52 (2013).

[18]O'Neal v. St. John Hosp. & Med. Ctr, 487 Mich SC, 485 (2010).

[19]See, e.g., International Broth. of Teamsters v. U.S., 431 U.S. 324 ("[i]n many cases the only available avenue of proof is the use of racial statistics to uncover clandestine and covert discrimination by the employer or union involved"); Bazemore v. Friday, 478 U.S 385 (1986); Marcus Jones v. Lee Way Motor Freigh, 431 10 th cir 245 (1970) ("In racial discrimination cases, statistics often demonstrate more than the testimony of many witnesses").

[20]Havens Realty Corp. v. Coleman, 455 U.S. 363, 374-75, 71 L. Ed. 2d 214, 102 S. Ct. 1114 (1982).

[21]See Bazemore v. Friday, 478 U.S 385 (1986) ("although it need not include every conceivable factor. Given the frequency of employment discrimination litigation in the contemporary United").

| Excerpt | Relies on statistical evidence? |
|---|---|
| However, a review of the "over base level" numbers of the four comparators and Escalera in core endourology reflects significant differences in the severity of losses between the comparators and Escalera during the January through June 2015 time period. Escalera's utilization of different time periods for each comparator within 2015 is not appropriate when examining the team managers' performances given Bard Medical's Solo/Skylite production products. Using the same time frame for each comparator, the record reflects that between January through June of 2015, Kunzinger was $55,626.89 below base, Santoro was $160,651.77 above base, Peters was $20,070.56 above base, and Martin was $79,932.38 above base. (Ottley Dep. Exs. 3, 12, 14, 16.) These numbers demonstrate that the "losses" experienced by the comparators during the same time period as Escalera are not substantially identical. Escalera's loss of base was $68,799.06 more than [**25] the closest comparator he identified.

Additionally, comparing the "over base level" numbers of the comparators and Escalera between January through October 2015 reflects that at the time Escalera was terminated he had suffered significantly more loss over base than his identified comparators: Escalera was [*805] $174,792.44 below base, Kunzinger was $101,132.60 below base,3 Santoro was $110,078.73 above base, Peters was $31,876.80 below base, and Martin was $1,611.79 below base. Because of these significant differences in losses, no reasonable jury could find that these four comparators and Escalera are similarly situated in all relevant respects. | No |
| Equally without evidentiary significance is the statistical analysis of the list of 17; indeed, the analysis was not even admissible under HN4 the standard of Daubert v. Merrell Dow Pharmaceuticals, Inc., 509 U.S. 579, 125 L. Ed. 2d 469, 113 S. Ct. 2786 (1993), governing the admissibility of expert testimony, which requires the district judge to satisfy himself that the expert is being as careful as he would be in his regular professional work outside his paid litigation consulting. E.g., Braun v. Lorillard Inc., 84 F.3d 230, 234-35 (7th Cir. 1996); Rosen v. Ciba-Geigy Corp., 78 F.3d 316, 318 (7th Cir. 1996); Daubert v. Merrell Dow Pharmaceuticals, Inc., 43 F.3d 1311, 1316-19 (9th Cir. 1995); cf. Mid-State Fertilizer Co. v. [**7] Exchange National Bank, 877 F.2d 1333, 1339 (7th Cir. 1989). Although the expert used standard statistical methods for determining whether there was a significant correlation between age and retention for the 17 persons on the list, see Michael O. Finkelstein & Bruce Levin, Statistics for Lawyers 157 (1990) (Fisher's exact test), the omission of Sebring and Shulman from the sample tested was arbitrary. The expert should at least have indicated the sensitivity of his analysis to these omissions. More important is the expert's failure to correct for any potential explanatory variables other than age. Completely ignored was the more than remote possibility that age was correlated with a legitimate job-related qualification, such as familiarity with computers. Everyone knows that younger people are on average more comfortable with computers than older people are, just as older people are on average more comfortable with manual-shift cars than younger people are. Three weeks of training might go some distance toward closing the computer-literacy gap, yet it would be more surprising than otherwise if so short a period of training could close the gap completely. The expert could easily [**8] have inquired about the feasibility of ascertaining through discovery the history of the use of computers by each of the employees on the list of 17. | Yes |

Table 34: Examples for `legal_reasoning_causality`.

| Class | Number of samples |
|-------|-------------------|
| Yes   | 31                |
| No    | 24                |

Table 35: Test set class distribution.

## G.16 MAUD Tasks

In LEGALBENCH, the MAUD tasks are denoted as `maud_*`.

**Background**  We adapt the Merger Agreement Understanding Dataset (MAUD) for LEGALBENCH. MAUD consists of over 37,000 expert-annotated examples for a set of legal reading comprehension tasks based on the American Bar Association's 2021 Public Target Deal Point Study. In the Study, lawyers review merger agreements and identify key legal clauses ("deal points") within those contracts. The lawyers then specify the nature of the clauses by answering a predetermined set of questions that cover a wide range of topics, including conditions to closing, the definition of material adverse effect, and remedies to breach of contract. MAUD's multiple-choice format, according to [140], assesses an LLM's ability to interpret the meaning of specialized legal language.

**Task**  The tasks take advantage of MAUD's reading comprehension component. They require an LLM—given a key legal clause and a set of descriptions for the clause—to choose the option that best describes the clause.

**Construction process**  These tasks are constructed by transforming the abridged dataset released by [140]. The abridged dataset contains 14,928 examples with deal points extracted from 94 merger agreements covering 92 multiple-choice questions. We narrow down to 57 questions by filtering out the ones with fewer than 50 examples. Each example consists of the text of a deal point, the question, options, and the answer key.

We create translations that map the questions into human-readable multiple-choice prompts. For instance, the prompt for the question "Accuracy of Target 'General' R&W: Bringdown Timing" is "When are representations and warranties required to be made according to the bring down provision?" It is then followed by an enumeration of the options for the LLM to choose among.

We focus on MAUD's abridged examples because we are interested in assessing an LLM's legal reading comprehension capability rather than its ability to extract relevant text segment given a complete deal point. Additionally, inputs of examples from the main dataset, which contains complete deal point texts, are oftentimes far longer than what an average open-source LLM could ingest at once, rendering them unsuitable for benchmarking purposes.

The table below lists the question and options for each MAUD-based LEGALBENCH task along with an example input-answer pair.

Table 36: MAUD Tasks

| Task |
| --- |
| **Task name**: `maud_type_of_consideration`
**Question**: What type of consideration is specified in this agreement?
**Options**: A: All Cash; B: All Stock; C: Mixed Cash/Stock; D: Mixed Cash/Stock: Election
**Example**: each Share <omitted> shall be converted into the right to receive the Offer Price in cash, without interest (the "Merger Consideration"), minus any withholding of Taxes required by applicable Laws in accordance with Section 3.6(d) (Page 20)
**Answer**: A |
| **Task name**: `maud_accuracy_of_target_general_rw_bringdown_timing_answer`
**Question**: When are representations and warranties required to be made according to the bring down provision?
**Options**: A: At Closing Only; B: At Signing & At Closing
**Example**: Section 7.2 Conditions to Obligations of Parent and Acquisition Sub to Effect the Merger. The obligations of Parent and Acquisition Sub to effect the Merger are, in addition to the conditions set forth in Section 7.1, further subject to the satisfaction or (to the extent not prohibited by Law) waiver by Parent at or prior to the Effective Time of the following conditions: (a) each of the representations and warranties of the Company contained in this Agreement, without giving effect to any materiality or "Company Material Adverse Effect" or similar qualifications therein, shall be true and correct as of the Closing Date, except for such failures to be true and correct as would not, individually or in the aggregate, have a Company Material Adverse Effect (except to the extent such representations and warranties are expressly made as of a specific date, in which case such representations and warranties shall be so true and correct as of such specific date only); (Page 67)
**Answer**: A |
| **Task name**: `maud_accuracy_of_target_capitalization_rw_(outstanding_shares)_bringdown_standard_answer`
**Question**: How accurate must the capitalization representations and warranties be according to the bring down provision?
**Options**: A: Accurate in all material respects; B: Accurate in all respects; C: Accurate in all respects with below-threshold carveout; D: Accurate in all respects with de minimis exception |

**Task**

**Example**: Conditions to the Offer
Notwithstanding any other term of the Offer or this Agreement to the contrary, Merger Sub will not be required to accept for payment or, subject to any applicable rules and regulations of the SEC, including Rule 14e-l(c) under the Exchange Act (relating to Merger Sub's obligation to pay for or return tendered Shares promptly after the termination or withdrawal of the Offer), to pay for any Shares tendered pursuant to the Offer, and may delay the acceptance for payment of or, subject to any applicable rules and regulations of the SEC, the payment for, any tendered Shares, and (subject to the provisions of this Agreement) may terminate the Offer and not accept for payment any tendered Shares, at any scheduled Expiration Date (as it may have been extended pursuant to Section 2.1 of this Agreement) if <omitted> (ii) any of the additional conditions set forth below are not satisfied or waived in writing by Parent at the Expiration Time:
<omitted> (d) Representations and Warranties. Each of the representations and warranties set forth in: <omitted> (iv) this Agreement (other than those set forth in the foregoing clauses (i), (ii) and (iii) of this clause (d) of Annex I), without giving effect to any "materiality" or "Material Adverse Effect" qualifiers or qualifiers of similar import set forth therein, shall be true and correct as of the consummation of the Offer as though made as of the consummation of the Offer (Page 107)
**Answer**: D

**Task name**: maud_accuracy_of_fundamental_target_rws_bringdown_standard
**Question**: How accurate must the fundamental representations and warranties be according to the bring down provision?
**Options**: A: Accurate at another materiality standard (e.g., hybrid standard); B: Accurate in all material respects; C: Accurate in all respects
**Example**: (b) Additional Conditions to Obligation of Parent and Merger Sub. <omitted> the representations and warranties of the Company set forth in Article 3 shall be true and correct <omitted> at and as of the Closing as if made at and as of such time (Page 11)
**Answer**: A

**Task name**: maud_ability_to_consummate_concept_is_subject_to_mae_carveouts
**Question**: Is the "ability to consummate" concept subject to Material Adverse Effect (MAE) carveouts?
**Options**: A: No; B: Yes
**Example**: "Material Adverse Effect" means, with respect to Huntington, TCF or the Surviving Corporation, as the case may be, any effect, change, event, circumstance, condition, occurrence or development that, either individually or in the aggregate, has had or would reasonably be likely to have a material adverse effect on (i) the business, properties, assets, liabilities, results of operations or financial condition of such party and its Subsidiaries, taken as a whole (provided, however, that, with respect to this clause (i), Material Adverse Effect shall not be deemed to include the impact of (A) changes, after the date hereof, in U.S. generally accepted accounting principles ("GAAP") or applicable regulatory accounting requirements, (B) changes, after the date hereof, in laws, rules or regulations (including the Pandemic Measures) of general applicability to companies in the industries in which such party and its Subsidiaries operate, or interpretations thereof by courts or Governmental Entities, (C) changes, after the date hereof, in global, national or regional political conditions (including the outbreak of war or acts of terrorism) or in economic or market (including equity, credit and debt markets, as well as changes in interest rates) conditions affecting the financial services industry generally and not specifically relating to such party or its Subsidiaries (including any such changes arising out of the Pandemic or any Pandemic Measures), (D) changes, after the date hereof, resulting from hurricanes, earthquakes, tornados, floods or other natural disasters or from any outbreak of any disease or other public health event (including the Pandemic), (E) public disclosure of the execution of this Agreement, public disclosure or consummation of the transactions contemplated hereby (including any effect on a party's relationships with its customers or employees) (it being understood that the foregoing shall not apply for purposes of the representations and warranties in Sections 3.3(b), 3.4, 4.3(b) or 4.4) or actions expressly required by this Agreement or that are taken with the prior written consent of the other party in contemplation of the transactions contemplated hereby, or (F) a decline in the trading price of a party's common stock or the failure, in and of itself, to meet earnings projections or internal financial forecasts, but not, in either case, including any underlying causes thereof; except, with respect to subclauses (A), (B), (C) or (D), to the extent that the effects of such change are materially disproportionately adverse to the business, properties, assets, liabilities, results of operations or financial condition of such party and its Subsidiaries, taken as a whole, as compared to other companies in the industry in which such party and its Subsidiaries operate) or (ii) the ability of such party to timely consummate the transactions contemplated hereby. (Page 18)
**Answer**: A

**Task name**: maud_fls_(mae)_standard
**Question**: What is the Forward Looking Standard (FLS) with respect to Material Adverse Effect (MAE)?
**Options**: A: "Could" (reasonably) be expected to; B: "Would"; C: "Would" (reasonably) be expected to; D: No; E: Other forward-looking standard

**Task**

**Example**: "Material Adverse Effect" means, with respect to BancorpSouth, Cadence or the Surviving Entity, as the case may be, any effect, change, event, circumstance, condition, occurrence or development that, either individually or in the aggregate, has had or would reasonably be expected to have a material adverse effect on (i) the business, properties, assets, liabilities, results of operations or financial condition of such party and its Subsidiaries taken as a whole (provided, however, that, with respect to this clause (i), Material Adverse Effect shall not be deemed to include the impact of (A) changes, after the date hereof, in U.S. generally accepted accounting principles ("GAAP") or applicable regulatory accounting requirements, (B) changes, after the date hereof, in laws, rules or regulations (including the Pandemic Measures) of general applicability to companies in the industries in which such party and its Subsidiaries operate, or interpretations thereof by courts or Governmental Entities (as defined below), (C) changes, after the date hereof, in global, national or regional political conditions (including the outbreak of war or acts of terrorism) or in economic or market (including equity, credit and debt markets, as well as changes in interest rates) conditions affecting the financial services industry generally and not specifically relating to such party or its Subsidiaries (including any such changes arising out of a Pandemic or any Pandemic Measures), (D) changes, after the date hereof, resulting from hurricanes, earthquakes, tornados, floods or other natural disasters or from any outbreak of any disease or other public health event (including a Pandemic), (E) public disclosure of the execution of this Agreement, public disclosure or consummation of the transactions contemplated hereby (including any effect on a party's relationships with its customers or employees) or actions expressly required by this Agreement or that are taken with the prior written consent of the other party in contemplation of the transactions contemplated hereby, or (F) a decline in the trading price of a party's common stock or the failure, in and of itself, to meet earnings projections or internal financial forecasts (it being understood that the underlying causes of such decline or failure may be taken into account in determining whether a Material Adverse Effect has occurred), except to the extent otherwise excepted by this proviso); except, with respect to subclauses (A), (B), (C), or (D) to the extent that the effects of such change are materially disproportionately adverse to the business, properties, assets, liabilities, results of operations or financial condition of such party and its Subsidiaries, taken as a whole, as compared to other companies in the industry in which such party and its Subsidiaries operate), or (ii) the ability of such party to timely consummate the transactions contemplated hereby. (Page 19)

**Answer**: C

**Task name**: `maud_general_economic_and_financial_conditions_subject_to_disproportionate_impact_modifier`

**Question**: Do changes caused by general economic and financial conditions that have disproportionate impact qualify for Material Adverse Effect (MAE)?

**Options**: A: No; B: Yes

**Example**: "Material Adverse Effect" means, with respect to Huntington, TCF or the Surviving Corporation, as the case may be, any effect, change, event, circumstance, condition, occurrence or development that, either individually or in the aggregate, has had or would reasonably be likely to have a material adverse effect on (i) the business, properties, assets, liabilities, results of operations or financial condition of such party and its Subsidiaries, taken as a whole (provided, however, that, with respect to this clause (i), Material Adverse Effect shall not be deemed to include the impact of (A) changes, after the date hereof, in U.S. generally accepted accounting principles ("GAAP") or applicable regulatory accounting requirements, (B) changes, after the date hereof, in laws, rules or regulations (including the Pandemic Measures) of general applicability to companies in the industries in which such party and its Subsidiaries operate, or interpretations thereof by courts or Governmental Entities, (C) changes, after the date hereof, in global, national or regional political conditions (including the outbreak of war or acts of terrorism) or in economic or market (including equity, credit and debt markets, as well as changes in interest rates) conditions affecting the financial services industry generally and not specifically relating to such party or its Subsidiaries (including any such changes arising out of the Pandemic or any Pandemic Measures), (D) changes, after the date hereof, resulting from hurricanes, earthquakes, tornados, floods or other natural disasters or from any outbreak of any disease or other public health event (including the Pandemic), (E) public disclosure of the execution of this Agreement, public disclosure or consummation of the transactions contemplated hereby (including any effect on a party's relationships with its customers or employees) (it being understood that the foregoing shall not apply for purposes of the representations and warranties in Sections 3.3(b), 3.4, 4.3(b) or 4.4) or actions expressly required by this Agreement or that are taken with the prior written consent of the other party in contemplation of the transactions contemplated hereby, or (F) a decline in the trading price of a party's common stock or the failure, in and of itself, to meet earnings projections or internal financial forecasts, but not, in either case, including any underlying causes thereof; except, with respect to subclauses (A), (B), (C) or (D), to the extent that the effects of such change are materially disproportionately adverse to the business, properties, assets, liabilities, results of operations or financial condition of such party and its Subsidiaries, taken as a whole, as compared to other companies in the industry in which such party and its Subsidiaries operate) or (ii) the ability of such party to timely consummate the transactions contemplated hereby. (Page 18)

| Task |
| --- |

**Answer**: A

**Task name**: `maud_change_in_law__subject_to_disproportionate_impact_modifier`
**Question**: Do changes in law that have disproportionate impact qualify for Material Adverse Effect (MAE)?
**Options**: A: No; B: Yes
**Example**: "Material Adverse Effect" means, with respect to SVB Financial, Boston Private or the Surviving Corporation, as the case may be, any effect, change, event, circumstance, condition, occurrence or development that, either individually or in the aggregate, has had or would reasonably be expected to have a material adverse effect on (i) the business, properties, assets, liabilities, results of operations or financial condition of such party and its Subsidiaries taken as a whole (provided, however, that, with respect to this clause (i), Material Adverse Effect shall not be deemed to include the impact of (A) changes, after the date hereof, in U.S. generally accepted accounting principles ("GAAP") or applicable regulatory accounting requirements, (B) changes, after the date hereof, in laws, rules or regulations of general applicability to companies in the industries in which such party and its Subsidiaries operate, or interpretations thereof by courts or Governmental Entities, (C) changes, after the date hereof, in global, national or regional political conditions (including the outbreak of war or acts of terrorism) or in economic or market (including equity, credit and debt markets, as well as changes in interest rates) conditions affecting the financial services industry generally and not specifically relating to such party or its Subsidiaries, (D) changes, after the date hereof, resulting from hurricanes, earthquakes, tornados, floods or other natural disasters or from any outbreak of any disease or other public health event (including the COVID-19 pandemic and the implementation of the Pandemic Measures), (E) public disclosure or consummation of the transactions contemplated hereby or actions expressly required by this Agreement or that are taken with the prior written consent of the other party in contemplation of the transactions contemplated hereby (it being understood and agreed that this clause (E) shall not apply with respect to any representation or warranty that is intended to address the consequences of the execution, announcement or performance of this Agreement or consummation of the Merger) or (F) the failure, in and of itself, to meet earnings projections or financial forecasts, but not including the underlying causes thereof; except, with respect to subclause (A), (B), (C) or (D), to the extent that the effects of such change are disproportionately adverse to the business, properties, assets, liabilities, results of operations or financial condition of such party and its Subsidiaries, taken as a whole, as compared to similar companies in the industry in which such party and its Subsidiaries operate); or (ii) the ability of such party to timely consummate the transactions contemplated hereby. (Page 20)
**Answer**: B

**Task name**: `maud_changes_in_gaap_or_other_accounting_principles__subject_to_disproportionate_impact_modifier`
**Question**: Do changes in GAAP or other accounting principles that have disproportionate impact qualify for Material Adverse Effect (MAE)?
**Options**: A: No; B: Yes

**Task**

**Example**: "Company Material Adverse Effect" shall mean any state of facts, circumstance, condition, event, change, development, occurrence, result, effect, action or omission (each, an "Effect") that, individually or in the aggregate with any one or more other Effects, (i) results in a material adverse effect on the business, financial condition or results of operations of the Company and its Subsidiaries, taken as a whole or (ii) prevents, materially impairs, materially impedes or materially delays the consummation of the Merger and the other transactions contemplated hereby on or before the End Date; provided, however, that with respect to clause (i) only, no Effect to the extent resulting or arising from any of the following, shall, to such extent, be deemed to constitute, or be taken into account in determining the occurrence of, a Company Material Adverse Effect: (A) general economic, political, business, financial or market conditions; (B) any outbreak, continuation or escalation of any military conflict, declared or undeclared war, armed hostilities, or acts of foreign or domestic terrorism; (C) any pandemic (including the SARS-CoV-2 virus and COVID-19 disease), epidemic, plague, or other outbreak of illness or public health event, hurricane, flood, tornado, earthquake or other natural disaster or act of God; (D) any failure by the Company or any of its Subsidiaries to meet any internal or external projections or forecasts or any decline in the price or trading volume of Company Common Stock (but excluding, in each case, the underlying causes of such failure or decline, as applicable, which may themselves constitute or be taken into account in determining whether there has been, or would be, a Company Material Adverse Effect); (E) the public announcement or pendency of the Merger and the other transactions contemplated hereby; (F) changes in applicable Legal Requirements; (G) changes in GAAP or any other applicable accounting standards; or (H) any action expressly required to be taken by the Company pursuant to the terms of the Agreement or at the express written direction or consent of Parent; provided, further, that any Effect relating to or arising out of or resulting from any change or event referred to in clause (A), (B), (C), (F) or (G) above may constitute, and be taken into account in determining the occurrence of, a Company Material Adverse Effect to the extent that such change or event has a disproportionate impact (but solely to the extent of such disproportionate impact) on the Company and its Subsidiaries as compared to other participants that operate in the industry in which the Company and its Subsidiaries operate. (Pages 87-88)

**Answer**: B

**Task name**: `maud_pandemic_or_other_public_health_event_specific_reference_to_pandemic-related_governmental_responses_or_measures`
**Question**: Is there specific reference to pandemic-related governmental responses or measures in the clause that qualifies pandemics or other public health events for Material Adverse Effect (MAE)?
**Options**: A: No; B: Yes
**Example**: "Company Material Adverse Effect" shall mean any state of facts, circumstance, condition, event, change, development, occurrence, result, effect, action or omission (each, an "Effect") that, individually or in the aggregate with any one or more other Effects, (i) results in a material adverse effect on the business, financial condition or results of operations of the Company and its Subsidiaries, taken as a whole or (ii) prevents, materially impairs, materially impedes or materially delays the consummation of the Merger and the other transactions contemplated hereby on or before the End Date; provided, however, that with respect to clause (i) only, no Effect to the extent resulting or arising from any of the following, shall, to such extent, be deemed to constitute, or be taken into account in determining the occurrence of, a Company Material Adverse Effect: (A) general economic, political, business, financial or market conditions; (B) any outbreak, continuation or escalation of any military conflict, declared or undeclared war, armed hostilities, or acts of foreign or domestic terrorism; (C) any pandemic (including the SARS-CoV-2 virus and COVID-19 disease), epidemic, plague, or other outbreak of illness or public health event, hurricane, flood, tornado, earthquake or other natural disaster or act of God; (D) any failure by the Company or any of its Subsidiaries to meet any internal or external projections or forecasts or any decline in the price or trading volume of Company Common Stock (but excluding, in each case, the underlying causes of such failure or decline, as applicable, which may themselves constitute or be taken into account in determining whether there has been, or would be, a Company Material Adverse Effect); (E) the public announcement or pendency of the Merger and the other transactions contemplated hereby; (F) changes in applicable Legal Requirements; (G) changes in GAAP or any other applicable accounting standards; or (H) any action expressly required to be taken by the Company pursuant to the terms of the Agreement or at the express written direction or consent of
Parent; provided, further, that any Effect relating to or arising out of or resulting from any change or event referred to in clause (A), (B), (C), (F) or (G) above may constitute, and be taken into account in determining the occurrence of, a Company Material Adverse Effect to the extent that such change or event has a disproportionate impact (but solely to the extent of such disproportionate impact) on the Company and its Subsidiaries as compared to other participants that operate in the industry in which the Company and its Subsidiaries operate. (Pages 87-88)

**Answer**: A

**Task name**: `maud_pandemic_or_other_public_health_event__subject_to_disproportionate_impact_modifier`

**Task**

**Question**: Do pandemics or other public health events have to have disproportionate impact to qualify for Material Adverse Effect (MAE)?

**Options**: A: No; B: Yes

**Example**: "Material Adverse Effect" means, with respect to BancorpSouth, Cadence or the Surviving Entity, as the case may be, any effect, change, event, circumstance, condition, occurrence or development that, either individually or in the aggregate, has had or would reasonably be expected to have a material adverse effect on (i) the business, properties, assets, liabilities, results of operations or financial condition of such party and its Subsidiaries taken as a whole (provided, however, that, with respect to this clause (i), Material Adverse Effect shall not be deemed to include the impact of (A) changes, after the date hereof, in U.S. generally accepted accounting principles ("GAAP") or applicable regulatory accounting requirements, (B) changes, after the date hereof, in laws, rules or regulations (including the Pandemic Measures) of general applicability to companies in the industries in which such party and its Subsidiaries operate, or interpretations thereof by courts or Governmental Entities (as defined below), (C) changes, after the date hereof, in global, national or regional political conditions (including the outbreak of war or acts of terrorism) or in economic or market (including equity, credit and debt markets, as well as changes in interest rates) conditions affecting the financial services industry generally and not specifically relating to such party or its Subsidiaries (including any such changes arising out of a Pandemic or any Pandemic Measures), (D) changes, after the date hereof, resulting from hurricanes, earthquakes, tornados, floods or other natural disasters or from any outbreak of any disease or other public health event (including a Pandemic), (E) public disclosure of the execution of this Agreement, public disclosure or consummation of the transactions contemplated hereby (including any effect on a party's relationships with its customers or employees) or actions expressly required by this Agreement or that are taken with the prior written consent of the other party in contemplation of the transactions contemplated hereby, or (F) a decline in the trading price of a party's common stock or the failure, in and of itself, to meet earnings projections or internal financial forecasts (it being understood that the underlying causes of such decline or failure may be taken into account in determining whether a Material Adverse Effect has occurred), except to the extent otherwise excepted by this proviso); except, with respect to subclauses (A), (B), (C), or (D) to the extent that the effects of such change are materially disproportionately adverse to the business, properties, assets, liabilities, results of operations or financial condition of such party and its Subsidiaries, taken as a whole, as compared to other companies in the industry in which such party and its Subsidiaries operate), or (ii) the ability of such party to timely consummate the transactions contemplated hereby. (Page 19)

**Answer**: B

**Task name**: maud_relational_language_(mae)_applies_to

**Question**: What carveouts pertaining to Material Adverse Effect (MAE) does the relational language apply to?

**Options**: A: All MAE carveouts; B: No; C: Some MAE carveouts

| Task |
| --- |
| **Example**: "Material Adverse Effect" means, with respect to Huntington, TCF or the Surviving Corporation, as the case may be, any effect, change, event, circumstance, condition, occurrence or development that, either individually or in the aggregate, has had or would reasonably be likely to have a material adverse effect on (i) the business, properties, assets, liabilities, results of operations or financial condition of such party and its Subsidiaries, taken as a whole (provided, however, that, with respect to this clause (i), Material Adverse Effect shall not be deemed to include the impact of (A) changes, after the date hereof, in U.S. generally accepted accounting principles ("GAAP") or applicable regulatory accounting requirements, (B) changes, after the date hereof, in laws, rules or regulations (including the Pandemic Measures) of general applicability to companies in the industries in which such party and its Subsidiaries operate, or interpretations thereof by courts or Governmental Entities, (C) changes, after the date hereof, in global, national or regional political conditions (including the outbreak of war or acts of terrorism) or in economic or market (including equity, credit and debt markets, as well as changes in interest rates) conditions affecting the financial services industry generally and not specifically relating to such party or its Subsidiaries (including any such changes arising out of the Pandemic or any Pandemic Measures), (D) changes, after the date hereof, resulting from hurricanes, earthquakes, tornados, floods or other natural disasters or from any outbreak of any disease or other public health event (including the Pandemic), (E) public disclosure of the execution of this Agreement, public disclosure or consummation of the transactions contemplated hereby (including any effect on a party's relationships with its customers or employees) (it being understood that the foregoing shall not apply for purposes of the representations and warranties in Sections 3.3(b), 3.4, 4.3(b) or 4.4) or actions expressly required by this Agreement or that are taken with the prior written consent of the other party in contemplation of the transactions contemplated hereby, or (F) a decline in the trading price of a party's common stock or the failure, in and of itself, to meet earnings projections or internal financial forecasts, but not, in either case, including any underlying causes thereof; except, with respect to subclauses (A), (B), (C) or (D), to the extent that the effects of such change are materially disproportionately adverse to the business, properties, assets, liabilities, results of operations or financial condition of such party and its Subsidiaries, taken as a whole, as compared to other companies in the industry in which such party and its Subsidiaries operate) or (ii) the ability of such party to timely consummate the transactions contemplated hereby. (Page 18)
**Answer**: C |
| **Task name**: `maud_knowledge_definition`
**Question**: What counts as Knowledge?
**Options**: A: Actual knowledge; B: Constructive knowledge
**Example**: provided, however, that with respect to clause (i) only, no Effect to the extent resulting or arising from any of the following, shall <omitted> be deemed to constitute <omitted> a Company Material Adverse Effect (Pages 87-88)
**Answer**: B |
| **Task name**: `maud_buyer_consent_requirement_(ordinary_course)`
**Question**: In case the Buyer's consent for the acquired company's ordinary business operations is required, are there any limitations on the Buyer's right to condition, withhold, or delay their consent?
**Options**: A: Yes. Consent may not be unreasonably withheld, conditioned or delayed.; B: No. |

| Task |
| --- |

**Example**: Section 5.1 Interim Operations of the Company and Parent.
(a) From the date of this Agreement and until the Effective Time or the earlier termination of this Agreement in accordance with its terms, except as (v) otherwise expressly contemplated by this Agreement, (w) set forth in the applicable subsection of Section 5.1 of the Company Disclosure Letter (it being agreed that disclosure of any item in any subsection of Section 5.1 of the Company Disclosure Letter shall be deemed disclosure with respect to any other subsection of Section 5.1 of the Company Disclosure Letter only to the extent that the relevance of such item to such subsection is reasonably apparent on its face), (x) required by applicable Law, (y)(A) required to comply with COVID-19 Measures or otherwise taken (or not taken) by the Company or any of its Subsidiaries reasonably and in good faith to respond to COVID-19 or COVID-19 Measures or (B) taken (or not taken) by the Company or any of its Subsidiaries reasonably and in good faith to respond to any other extraordinary event that was not reasonably foreseeable as of the date of this Agreement and occurring after the date of this Agreement that is outside of the control of the Company or its Affiliates and is outside of the ordinary course of business of the Company and its Subsidiaries and Joint Ventures (and is not related to a Company Takeover Proposal); provided that prior to taking any actions in reliance on this clause (y), which would otherwise be prohibited by any provision of this Agreement, the Company will use commercially reasonable efforts to provide advance notice to and consult with Parent (if reasonably practicable) with respect thereto or (z) consented to in writing by Parent (which consent shall not be unreasonably withheld, conditioned or delayed), the Company shall, and shall cause each of its Subsidiaries to, use its commercially reasonable efforts to conduct its business in all material respects in the ordinary course of business consistent with past practice and in compliance in all material respects with all material applicable Laws, and shall, and shall cause each of its Subsidiaries to, use its commercially reasonable efforts to preserve intact its present business organization, keep available the services of its directors, officers and employees and maintain existing relations and goodwill with customers, distributors, lenders, partners (including Joint Venture partners and others with similar relationships), suppliers and others having material business associations with it or its Subsidiaries; (Pages 40-41)
**Answer**: A

**Task name**: `maud_includes_consistent_with_past_practice`
**Question**: Does the wording of the Efforts Covenant clause include "consistent with past practice"?
**Options**: A: No; B: Yes
**Example**: 5.2 Operation of the Acquired Corporations' Business. (a) During the Pre-Closing Period, except (w) as required or otherwise contemplated under this Agreement or as prohibited or required by applicable Legal Requirements, (x) with the written consent of Parent (which consent shall not be unreasonably withheld, delayed or conditioned, and provided that no consent shall be required if the Company reasonably believes after consulting with outside legal counsel that seeking such consent would violate Antitrust Law), (y) for any action required to be or reasonably taken, or omitted to be taken, pursuant to any COVID-19 Measures or which is otherwise required or reasonably taken, or omitted to be taken, in response to COVID-19 or any other pandemic, epidemic or disease outbreak, as determined by the Company in its reasonable discretion, or (z) as set forth in Section 5.2 of the Company Disclosure Schedule, the Company shall, and shall cause each Acquired Corporation to, use commercially reasonable efforts to conduct its business and operations in the ordinary course in all material respects (Page 41)
**Answer**: A

**Task name**: `maud_ordinary_course_efforts_standard`
**Question**: What is the efforts standard?
**Options**: A: Commercially reasonable efforts; B: Flat covenant (no efforts standard); C: Reasonable best efforts
**Example**: "Ordinary Course of Business" means, with respect to any Person, the conduct of such Person's business that is consistent with the past practices of such Person prior to the date of this Agreement and taken in the ordinary course of normal, day-to-day operations of such Person, but excluding any conduct that would reasonably be expected to violate applicable Law in any material respect. <omitted> 7.1. Interim Operations. (a) The Company shall, and shall cause each of its Subsidiaries to, from and after the date of this Agreement until the earlier of the Effective Time and the termination of this Agreement pursuant to Article IX (unless Parent shall otherwise approve in writing (such approval not to be unreasonably withheld, conditioned or delayed), and except as otherwise expressly required by this Agreement or as required by a Governmental Entity or applicable Law and any Material Contract in effect prior to the date of this Agreement), conduct its business in the Ordinary Course of Business (Page 66)
**Answer**: B

**Task name**: `maud_application_of_buyer_consent_requirement_(negative_interim_covenant)`
**Question**: What negative covenants does the requirement of Buyer consent apply to?
**Options**: A: Applies only to specified negative covenants; B: Applies to all negative covenants

| Task |
| --- |

**Example**: Except (w) with respect to the Specified Exceptions (other than as applied to Section 5.1(a), Section 5.1(b), or Section 5.1(k)), (x) 25

as otherwise expressly contemplated or permitted by this Agreement, (y) as set forth in Section 5.1 of the Company Disclosure Schedule, or (z) with the Parent's consent (which shall not be unreasonably withheld, conditioned or delayed), during the Pre-Closing Period the Company shall not, and shall not permit any of its Subsidiaries to, directly or indirectly, do any of the following: (Pages 29-30)

**Answer**: B

---

**Task name**: `maud_fiduciary_exception__board_determination_standard`

**Question**: Under what circumstances could the Board take actions on a different acquisition proposal notwithstanding the no-shop provision?

**Options**: A: If failure to take actions would lead to "breach" of fiduciary duties; B: If failure to take actions would be "inconsistent" with fiduciary duties; C: If failure to take actions would lead to "reasonably likely/expected breach" of fiduciary duties; D: If failure to take actions would lead to "reasonably likely/expected to be inconsistent" with fiduciary duties; E: If failure to take actions would lead to "reasonably likely/expected violation" of fiduciary duties; F: If taking such actions is "required to comply" with fiduciary duties; G: If failure to take actions would lead to "violation" of fiduciary duties; H: Under no circumstances could the Board do so.; I: Other circumstances

**Example**: Section 5.4 No Company Solicitation. <omitted> (b) Notwithstanding anything in Section 5.4(a) to the contrary, until the Company Stockholder Approval is obtained, if the Company receives a bona fide written Alternative Acquisition Proposal made after the date hereof that does not result from a material breach of this Section 5.4, and the Company Board determines in good faith (after consultation with outside legal counsel and a nationally recognized financial advisor) that such Alternative Acquisition Proposal is, or could reasonably be expected to lead to, a Superior Acquisition Proposal, (i) the Company may negotiate and enter into an Acceptable Confidentiality Agreement with the Person making such Alternative Acquisition Proposal; provided, that the Company shall promptly (and in no event later than twenty-four (24) hours after execution thereof) deliver a copy of such Acceptable Confidentiality Agreement to Parent, (ii) following entry into such Acceptable Confidentiality Agreement by the Company, the Company and its Representatives may provide information (including nonpublic information) subject to such executed Acceptable Confidentiality Agreement; provided, that any nonpublic information provided to such Person, including if posted to an electronic data room, shall be provided to Parent prior to or substantially concurrently with the time it is provided to such Person, and (iii) the Company and its Representatives may engage in discussion or negotiations for such Alternative Acquisition Proposal with such Person and its Representatives. (Page 59)

**Answer**: H

---

**Task name**: `maud_fiduciary_exception_board_determination_trigger_(no_shop)`

**Question**: What type of offer could the Board take actions on notwithstanding the no-shop provision?

**Options**: A: Acquisition Proposal only; B: Superior Offer, or Acquisition Proposal reasonably likely/expected to result in a Superior Offer

**Example**: SECTION 5.02. Acquisition Proposals. <omitted> (c) Information Exchange; Discussions or Negotiation. Notwithstanding anything to the contrary contained in Section 5.02(a), prior to obtaining the Company Requisite Vote, in the event that the Company, any of its Subsidiaries or its or their Representatives receive from any Person, after the date of this Agreement, an unsolicited, bona fide written Acquisition Proposal that did not result from a breach of this Section 5.02, and that the Company Board determines in good faith, after consultation with its financial advisors and outside legal counsel, is, or is reasonably likely to lead to, a Superior Proposal, the Company may (i) furnish or provide information to the Person making such Acquisition Proposal and its Representatives pursuant to an Acceptable Confidentiality Agreement; provided, however, that the Company shall as promptly as is reasonably practicable (and in any event within one (1) Business Day) make available to Parent and Merger Sub any written material non-public information concerning the Company or its Subsidiaries that is provided to any Person pursuant to this Section 5.02(c)(i), to the extent such information was not previously made available to Parent, Merger Sub or their Representatives, and (ii) engage in discussions and negotiations with such Person and its Representatives with respect to such Acquisition Proposal. (Page 35)

**Answer**: B

---

**Task name**: `maud_cor_permitted_with_board_fiduciary_determination_only`

**Question**: Is Change of Recommendation permitted as long as the board determines that such change is required to fulfill its fiduciary obligations?

**Options**: A: No; B: Yes

**Task**

**Example**: SECTION 5.3 No Solicitation by the Company; Company Recommendation. <omitted> (d) <omitted> Notwithstanding the foregoing or any other provision of this Agreement to the contrary, prior to the time the Company Stockholder Approval is obtained (but not thereafter), the Company Board or the Company Special 41

Committee may make a Company Adverse Recommendation Change if either (x) in the case of a Company Adverse Recommendation Change made in response to a Company Acquisition Proposal, the Company Board or the Company Special Committee has determined in good faith, after consultation with its outside financial advisors and outside legal counsel, that such Company Acquisition Proposal constitutes a Company Superior Proposal and that failure to take such action would reasonably be expected to be inconsistent with the directors' fiduciary duties under applicable Law or (y) in the case of a Company Adverse Recommendation Change made in response to a Company Intervening Event, the Company Board or the Company Special Committee has determined in good faith, after consultation with its outside financial advisors and outside legal counsel, that, as a result of a Company Intervening Event, the failure to take such action would reasonably be expected to be inconsistent with its fiduciary duties under applicable Law; (Pages 46-47)

**Answer**: A

**Task name**: `maud_cor_standard_(superior_offer)`
**Question**: What standard should the board follow when determining whether to change its recommendation in connection with a superior offer?
**Options**: A: "Breach" of fiduciary duties; B: "Inconsistent" with fiduciary duties; C: "Reasonably likely/expected breach" of fiduciary duties; D: "Reasonably likely/expected to be inconsistent" with fiduciary duties; E: "Reasonably likely/expected violation" of fiduciary duties; F: "Required to comply" with fiduciary duties; G: "Violation" of fiduciary duties; H: More likely than not violate fiduciary duties; I: None; J: Other specified standard

**Example**: Section 5.2. No Solicitation. <omitted>

(c) Notwithstanding anything to the contrary contained in this Agreement, at any time prior to obtaining the Company Stockholder Approval, the Company Board may make a Change in Recommendation in response to an unsolicited bona fide written Acquisition Proposal or cause the Company to enter into an Alternative Acquisition Agreement concerning an Acquisition Proposal, in each case only if: (i) such Acquisition Proposal or Superior Proposal did not result from a breach of Section 5.2(a); (ii)the Company Board (or a committee thereof) determines in good faith (A) after consultation with the Company's outside legal counsel and Independent Financial Advisor, that such Acquisition Proposal constitutes a Superior Proposal and (B) after consultation with the Company's outside legal counsel, that in light of such Acquisition Proposal, a failure to make a Change in Recommendation or to cause the Company to enter into such Alternative Acquisition Agreement would be inconsistent with the Company Board's fiduciary obligations to the Company's stockholders under the DGCL; (Page 27)

**Answer**: B

**Task name**: `maud_cor_permitted_in_response_to_intervening_event`
**Question**: Is Change of Recommendation permitted in response to an intervening event?
**Options**: A: No; B: Yes

| Task |
| --- |

**Example**: 6.1 No Solicitation. <omitted>
Notwithstanding the foregoing or anything to the contrary set forth in this Agreement (including the provisions of this Section 6.1), at any time prior to receipt of the Company Stockholder Approval, the Company Board may effect a Company Board Recommendation Change in response to a Superior Proposal or an Intervening Event if: (i) the Company Board shall have determined in good faith (after consultation with outside counsel and outside financial advisor) that the failure to effect a Company Board Recommendation Change would be reasonably likely to be inconsistent with its fiduciary obligations under applicable law; (ii) so long as the Company and its Subsidiaries are not in material breach of their obligations pursuant to this Section 6.1 with respect to an Acquisition Proposal underlying such Company Board Recommendation Change; (iii) the Company has notified the Parent in writing that it intends to effect a Company Board Recommendation Change, describing in reasonable detail the reasons for such Company Board Recommendation Change (a "Recommendation Change Notice") (it being understood that the Recommendation Change Notice shall not constitute a Company Board Recommendation Change or a Trigger Event for purposes of this Agreement); (iv) if requested by the Parent, the Company shall have made its Representatives available to negotiate (to the extent that Parent desires to so negotiate) with the Parent's Representatives any proposed modifications to the terms and conditions of this Agreement during the three (3) Business Day period following delivery by the Company to the Parent of such Recommendation Change Notice; and (v) if the Parent shall have delivered to the Company a written, binding and irrevocable offer to alter the terms or conditions of this Agreement during such three (3) Business Day period, the Company Board shall have determined in good faith (after consultation with outside counsel), after considering the terms of such offer by the Parent, that the failure to effect a Company Board Recommendation Change would still be reasonably likely to be inconsistent with its fiduciary obligations under applicable law; provided, however, that in the event of any material revisions to an Acquisition Proposal underlying a potential Company Board Recommendation Change, the Company will be required to notify Parent of such revisions and the applicable three (3) Business Day period described above shall be extended until two (2) Business Days after the time Parent receives notification from the Company of such revisions. (Page 34)

**Answer**: B

**Task name**: `maud_cor_standard_(intervening_event)`
**Question**: What standard should the board follow when determining whether to change its recommendation in response to an intervening event?
**Options**: A: "Breach" of fiduciary duties; B: "Inconsistent" with fiduciary duties; C: "Reasonably likely/expected breach" of fiduciary duties; D: "Reasonably likely/expected to be inconsistent" with fiduciary duties; E: "Reasonably likely/expected violation" of fiduciary duties; F: "Required to comply" with fiduciary duties; G: "Violation" of fiduciary duties; H: More likely than not violate fiduciary duties; I: Other specified standard
**Example**: 6.3 Shareholders' Approval and Stockholder Approval. <omitted> (c) <omitted> if the Board of Directors of <omitted> the Company, after receiving the advice of its outside counsel and, with respect to financial matters, its outside financial advisors, determines in good faith that it would more likely than not result in a violation of its fiduciary duties under applicable law to make or continue to make the Parent Board Recommendation or the Company Board Recommendation, as applicable, such Board of Directors may <omitted> submit this Agreement to its shareholders or stockholders, respectively, without recommendation (which, for the avoidance of doubt, shall constitute a Recommendation Change) (Page 57)
**Answer**: I

**Task name**: `maud_initial_matching_rights_period_(cor)`
**Question**: How long is the initial matching rights period in case the board changes its recommendation?
**Options**: A: 2 business days or less; B: 3 business days; C: 3 calendar days; D: 4 business days; E: 4 calendar days; F: 5 business days; G: Greater than 5 business days
**Example**: 6.3 No Solicitation by Golden. <omitted> in response to a <omitted> Golden Competing Proposal <omitted> the Golden Board may effect a Golden Change of Recommendation; provided, however, that such a Golden Change of Recommendation may not be made unless and until: <omitted>; provided that in the event of any material amendment or material modification to any Golden Superior Proposal <omitted> , Golden shall be required to deliver a new written notice to Labrador and to comply with the requirements of this Section 6.3(e)(iv) with respect to such new written notice, except that the advance written notice obligation set forth in this Section 6.3(e)(iv) shall be reduced to two Business Days (Pages 34-35)
**Answer**: D

**Task name**: `maud_additional_matching_rights_period_for_modifications_(cor)`
**Question**: How long is the additional matching rights period for modifications in case the board changes its recommendation?
**Options**: A: 2 business days or less; B: 3 business days; C: 3 days; D: 4 business days; E: 5 business days; F: > 5 business days; G: None

**Task**

**Example**: Section 5.4 Non-Solicitation. <omitted>
(b) <omitted> Notwithstanding the foregoing, at any time prior to obtaining the East Stockholder Approval, and subject to East's compliance in all material respects at all times with the provisions of this Section 5.4 and Section 5.3, in response to a Superior Proposal with respect to East that was not initiated, solicited, knowingly encouraged or knowingly facilitated by East or any of the East Subsidiaries or any of their respective Representatives, the East Board may make an East Adverse Recommendation Change; provided, however, that East shall not be entitled to exercise its right to make an East Adverse Recommendation Change in response to a Superior Proposal with respect to East (x) until three (3) Business Days after East provides written notice to Central (an "East Notice") advising Central that the East Board or a committee thereof has received a Superior Proposal, specifying the material terms and conditions of such Superior Proposal, and identifying the Person or group making such Superior Proposal, (y) if during such three (3) Business Day period, Central proposes any alternative transaction (including any modifications to the terms of this Agreement), unless the East Board determines in good faith (after consultation with East's financial advisors and outside legal counsel, and taking into account all financial, legal, and regulatory terms and conditions of such alternative transaction proposal, including any conditions to and expected timing of consummation, and any risks of non-consummation of such alternative transaction proposal) that such alternative transaction proposal is not at least as favorable to East and its stockholders as the Superior Proposal (it being understood that any change in the financial or other material terms of a Superior Proposal shall require a new East Notice and a new two (2) Business Day period under this Section 5.4(b)) and (z) unless the East Board, after consultation with outside legal counsel, determines that the failure to make an East Adverse Recommendation Change would be inconsistent with its fiduciary duties. (Page 76)
**Answer**: A

**Task name**: `maud_definition_includes_stock_deals`
**Question**: What qualifies as a superior offer in terms of stock deals?
**Options**: A: "All or substantially all"; B: 50%; C: Greater than 50% but not "all or substantially all"; D: Less than 50%
**Example**: 5.4 No Solicitation by the Company; Other Offers. <omitted> the Company shall not be entitled to: (i) make a Change in Company Board Recommendation <omitted> unless: <omitted> the Company shall have first provided prior <omitted> notice to Parent that it is prepared to <omitted> make a Change in Company Board Recommendation (a "Recommendation Change Notice") <omitted> Any material changes with respect to the Intervening Event <omitted> or material changes to the financial terms of such Superior Proposal <omitted> shall require the Company to provide to Parent a new Recommendation Change Notice <omitted> and a new three (3) Business Day period. (Pages 45-46)
**Answer**: C

**Task name**: `maud_definition_includes_asset_deals`
**Question**: What qualifies as a superior offer in terms of asset deals?
**Options**: A: "All or substantially all"; B: 50%; C: Greater than 50% but not "all or substantially all"; D: Less than 50%
**Example**: Section 5.4 Acquisition Proposals. <omitted> (d) <omitted> following receipt of a <omitted> Acquisition Proposal <omitted> that the Company Board determines <omitted> constitutes a Superior Proposal, the Company Board may <omitted> make an Adverse Recommendation Change <omitted> if <omitted> (i) (A) the Company shall have provided to Parent <omitted> notice, <omitted> (it being understood and agreed that any amendment to the financial terms or any other material term or condition of such Superior Proposal shall require a new notice and an additional three Business Day period) (Pages 44-45)
**Answer**: B

**Task name**: `maud_financial_point_of_view_is_the_sole_consideration`
**Question**: Is "financial point of view" the sole consideration when determining whether an offer is superior?
**Options**: A: No; B: Yes
**Example**: 5.4 No Solicitation by the Company; Other Offers. <omitted> the Company shall not be entitled to: (i) make a Change in Company Board Recommendation <omitted> unless: <omitted> the Company shall have first provided prior <omitted> notice to Parent that it is prepared to <omitted> make a Change in Company Board Recommendation (a "Recommendation Change Notice") <omitted> Any material changes with respect to the Intervening Event <omitted> or material changes to the financial terms of such Superior Proposal <omitted> shall require the Company to provide to Parent a new Recommendation Change Notice <omitted> and a new three (3) Business Day period. (Pages 45-46)
**Answer**: A

**Task name**: `maud_definition_contains_knowledge_requirement_-_answer`
**Question**: What is the knowledge requirement in the definition of "Intervening Event"?

**Task**

**Options**: A: Known, but consequences unknown or not reasonably foreseeable, at signing; B: Known, but consequences unknown, at signing; C: Not known and not reasonably foreseeable at signing; D: Not known at signing

**Example**: "Acquisition Proposal" means any inquiry, proposal or offer from any Person or group of Persons other than Parent or one of its Subsidiaries made after the date of this Agreement relating to (A) a merger, reorganization, consolidation, share purchase, share exchange, business combination, recapitalization, liquidation, dissolution, joint venture, partnership, spin-off, extraordinary dividend or similar transaction involving the Company or any of its Subsidiaries, which is structured to permit such Person or group of Persons to, directly or indirectly, acquire beneficial ownership of 20% or more of the outstanding equity securities of the Company, or 20% or more of the consolidated net revenues, net income or total assets of the Company and its Subsidiaries, taken as a whole or (B) the acquisition in any manner, directly or indirectly, of over 20% of the equity securities or consolidated total assets of the Company and its Subsidiaries, in each case other than the Merger and the other transactions contemplated by this Agreement. <omitted> "Superior Proposal" means any bona fide written Acquisition Proposal (A) on terms which the Company Board determines in good faith, after consultation with its outside legal counsel and financial advisors, to be more favorable from a financial point of view to the holders of Shares than the Merger and the other transactions contemplated by this Agreement, taking into account all the terms and conditions of such proposal and this Agreement and (B) that the Company Board determines in good faith is capable of being completed, taking into account all financial, regulatory, legal and other aspects of such proposal; provided, that for purposes of the definition of "Superior Proposal," the references to "20%" in the definition of Acquisition Proposal shall be deemed to be references to "50%." (Page 47)

**Answer**: A

**Task name**: `maud_intervening_event_-_required_to_occur_after_signing_-_answer`
**Question**: Is an "Intervening Event" required to occur after signing?
**Options**: A: No. It may occur or arise prior to signing.; B: Yes. It must occur or arise after signing.
**Example**: "Superior Proposal" shall mean, with respect to a party hereto, any <omitted> Acquisition Proposal with respect to such party made by a third party to acquire, directly or indirectly, pursuant to a tender offer, exchange offer, merger, share exchange, consolidation or other business combination, (A) all or substantially all of the assets of such party and its Subsidiaries, taken as a whole, (Page 120)
**Answer**: A

**Task name**: `maud_initial_matching_rights_period_(ftr)`
**Question**: How long is the initial matching rights period in connection with the Fiduciary Termination Right (FTR)?
**Options**: A: 2 business days or less; B: 3 business days; C: 3 calendar days; D: 4 business days; E: 4 calendar days; F: 5 business days; G: 5 calendar days; H: Greater than 5 business days
**Example**: SECTION 5.3 No Solicitation by the Company; Company Recommendation. <omitted> (d) <omitted> provided, however, that the Company Board and the Company Special Committee shall not, and shall cause the Company not to, make a Company Adverse Recommendation Change in connection with a Company Superior Proposal unless (I) the Company has given Parent at least four (4) Business Days' prior written notice of its intention to take such action (which notice shall reasonably describe the material terms of the Company Superior Proposal or attach the agreement and all material related documentation providing for such Company Superior Proposal), (II) the Company has negotiated, and has caused its Representatives to negotiate, in good faith with Parent during such notice period, to the extent Parent wishes to negotiate, to enable Parent to propose in writing a binding offer to effect revisions to the terms of this Agreement such that it would cause such Company Superior Proposal to no longer constitute a Company Superior Proposal, (III) following the end of such notice period, the Company Board or the Company Special Committee shall have considered in good faith any such binding offer from Parent, and shall have determined that the Company Superior Proposal would continue to constitute a Company Superior Proposal if the revisions proposed in such binding offer were to be given effect and (IV) in the event of any material change to the material terms of such Company Superior Proposal, the Company shall, in each case, have delivered to Parent an additional notice consistent with that described in clause (I) above and the notice period shall have recommenced, except that the notice period shall be at least two (2) Business Days (rather than the four (4) Business Days otherwise contemplated by clause (I) above); (Page 47)
**Answer**: D

**Task name**: `maud_tail_period_length`
**Question**: How long is the Tail Period?
**Options**: A: 12 months or longer; B: Other; C: within 12 months; D: within 6 months; E: within 9 months
**Example**: Section 7.3 Termination Fees. <omitted> (b) <omitted> if <omitted> Parent or the Company terminates this Agreement <omitted> (iii) <omitted> the Company shall have consummated an Alternative Acquisition Proposal or entered into an Alternative Acquisition Agreement for any Alternative Acquisition Proposal <omitted> which Alternative Acquisition Proposal is ultimately consummated (Page 80)

| Task |
| --- |
| **Answer**: C |
| **Task name**: `maud_specific_performance`
**Question**: What is the wording of the Specific Performance clause regarding the parties' entitlement in the event of a contractual breach?
**Options**: A: "entitled to seek" specific performance; B: "entitled to" specific performance
**Example**: Section 9.10 Specific Performance. The parties hereto hereby agree that irreparable damage would occur in the event that any provision of this Agreement were not performed in accordance with its specific terms or were otherwise breached, and that money damages or other legal remedies would not be an adequate remedy for any such damages. Accordingly, the parties acknowledge and agree that each party shall be entitled to, in accordance with the provisions of this Agreement, an injunction or injunctions, specific performance or other equitable relief to prevent breaches of this Agreement and/or to enforce specifically the terms and provisions hereof in any court, in addition to any other remedy to which they are entitled at law or in equity (Page 73)
**Answer**: B |

**Significance and value**   Reading comprehension is a particularly challenging part of contract review, both to human and to machine. The MAUD tasks evaluate an LLM's ability to understand and categorize a wide spectrum of legal clauses in the context of merger agreements.

## G.17 New York State Judicial Ethics

In LEGALBENCH, the New York State Judicial Ethics task is denoted as `nys_judicial_ethics`.

**Background** The New York State Advisory Committee on Judicial Ethics posts rulings on real ethical scenarios. The Committee, established in 1987, offers guidance to roughly New York State judges and justices, as well as other judicial personnel and candidates in the state. By interpreting the Rules Governing Judicial Conduct and the Code of Judicial Conduct, the Committee assists these individuals in maintaining high ethical standards. Actions taken by judges in accordance with the Committee's formal opinions are deemed proper, which helps protect them during any future investigations by the New York State Commission on Judicial Conduct.

**Task** 300 real-world scenarios and fact patterns have been reformulated into yes or no questions to understand whether models understand ethical rules and how they might apply to different judicial situations.

For example in a 2022 decision the Committee noted that: "A judge who previously served as the District Attorney may not preside over a parole recognizance hearing concerning a parolee or releasee who had originally been convicted and sentenced during the judge's former tenure as the District Attorney."

This is converted to a Yes/No question:

> Question: Can a judge who previously served as the District Attorney preside over a parole recognizance hearing concerning a parolee or releasee who had originally been convicted and sentenced during the judge's former tenure as the District Attorney?
> Answer: No

| Question | Answer |
| --- | --- |
| Can a judge's law clerk assist the judge with any election-related matters during a general election when the judge is on-call? | No |
| Can a part-time town justice be employed part-time as a community school liaison with the county sheriff's office simultaneously? | No |
| Can an appellate judge who successfully sought to vacate a vexatious lien filed by a disgruntled litigant against their real property preside over appeals from other decisions or orders rendered by the lower court judge who granted the petition to vacate? | Yes |
| Can a part-time town justice serve as a part-time assistant conflict defender in the same county as their court? | Yes |

Table 37: Examples for `nys_judicial_ethics`.

**Construction process** We collect digest statements from the New York State Unified Court System Advisory Committee on Judicial Ethics.[22] We collect samples from 2010, 2021, 2022, and 2023 and then use ChatGPT to reformulate the statements into yes or no questions. To ensure that data is not used for training OpenAI models, we opt out of data use for accounts used for task creation. We leave 2010 and 2021 data for understanding scope of data leakage from opinions being online. 2022 and 2023 data should not have been seen by most models that were trained prior to these years.

**Significance and value** An important part of legal practice is abiding by ethics rules. As agents become more involved in the legal process it will be important to understand not only whether they can understand and reason about rules for the public, but also whether they can reason about ethical principles and rules governing judges and lawyers.

---

[22]https://www.nycourts.gov/legacyhtm/ip/judicialethics/opinions/

### G.18 OPP-115 Tasks

In LEGALBENCH, the OPP-115 tasks are denoted as opp115_*.

**Background**   The OPP-115 Corpus, consisting of 115 online privacy policies, provides a comprehensive collection of privacy statements expressed in natural language [147]. Each of these policies has been meticulously read and annotated by a team of three law graduate students. The annotations present in the text specifically outline various data practices.

These privacy policies are classified into ten distinct categories:

1. First Party Collection/Use: This describes how and why a service provider collects user information.
2. Third Party Sharing/Collection: This explains how user information may be shared with or collected by third parties.
3. User Choice/Control: This delineates the choices and control options available to users.
4. User Access, Edit, & Deletion: This describes if and how users may access, edit, or delete their information.
5. Data Retention: This states how long user information is stored.
6. Data Security: This communicates how user information is protected.
7. Policy Change: This explains if and how users will be informed about changes to the privacy policy.
8. Do Not Track: This discusses if and how Do Not Track signals for online tracking and advertising are honored.
9. International & Specific Audiences: This focuses on practices that pertain only to a specific group of users (e.g., children)
10. Other: This encompasses additional sub-labels for introductory or general text, contact information, and practices not covered by the other categories.

**Task and construction process**   A separate binary classification task has been created for each category, with negative samples drawn from the rest of the text. To ensure consistency, any text with less than 10 words has been eliminated. The 'other' category was not included in the categorization because it was deemed too broad and wouldn't provide much value in terms of specific classification.

**Significance and value**   The classification task associated with the OPP-115 Corpus serves as a significant measure of an LLM's logical reasoning ability. By assigning privacy policy segments to the right categories, LLMs demonstrate their understanding and interpretation of the language and nuances within these privacy policies. Although the task may be seen as "simple" from a human legal practitioner's viewpoint, it provides an invaluable and objective gauge of an LLM's progress in logical reasoning and language comprehension.

| Task/category | Example of clause |
|---|---|
| opp115_data_retention | The name of the domain from which you access the Internet (for example, gmail.com, if you are connecting from a Google account); |
| opp115_data_security | However, no system can be 100% secure and human errors occur, so there is the possibility that there could be unauthorized access to your information. By using our services, you assume this risk. |
| opp115_do_not_track | Do Not Track Signals Our websites do not treat browsers that send a do not track signal differently from browsers that do not send one. |
| opp115_first_party_collection_use | Send-a-friend: In the case of send-a-friend email or card, we only collect |
| opp115_international_and_specific_audiences | CalOPPA is the first state law in the nation to require commercial websites and online services to post a privacy policy. The law's reach stretches well beyond California to require a person or company in the United States (and conceivably the world) that operates websites collecting personally identifiable information from California consumers to post a conspicuous privacy policy on its website stating exactly the information being collected and those individuals with whom it is being shared, and to comply with this policy. - See more at: http://consumercal.org/california-online-privacy-protection-act-caloppa/sthash.0FdRbT51.dpuf |
| opp115_policy_change | If we make a significant or material change in the way we use your personal information, t |
| opp115_third_party_sharing_collection | We use third-party payment service providers such as Amazon.com ( Privacy Policy), Stripe.com ( Privacy Policy), and PayPal ( Privacy Policy) |
| opp115_user_access,_edit_and_deletion | you can access your personal information by contacting ABITA.COM as described at the bottom of this statement, or through alternative means of access described by the service. |
| opp115_user_choice_control | do not wish to receive any additional marketing material, you can indicate your preference on our store partners order form. |

Table 38: Examples for OPP-115 tasks.

## G.19 Purpose of Oral Argument Questions

In LEGALBENCH, the Purpose of Oral Argument Questions task is denoted as `oral_argument_question_purpose`.

**Background**  Before a court decides a case, it typically calls before it the attorneys for the parties to the lawsuit to orally present their arguments for why the case should be resolved in favor of their clients and to answer any questions that the judge or judges have of them. In modern times, however, oral argument is not lawyers' primary avenue for explaining their positions. Instead, parties submit their arguments in written form ("briefs") and use their oral argument time primarily to supplement those submissions by reiterating their key positions, clarifying areas of ambiguity, and seeking to persuade judges who are uncertain how the case should be resolved.

Although there is no universally accepted listing, judges questions at oral argument tend to fall into a few categories:[23]

- Background: A question seeking factual or procedural information that is missing or not clear in the briefing
- Clarification: A question seeking to get an advocate to clarify her position or the scope of the rule being advocated.
- Implications: A question about the limits of a rule or its implications for future cases.
- Support: A question offering support for the advocate's position.
- Criticism: A question criticizing an advocate's position.
- Communicate: A question designed primarily to communicate with one or more other judges on the court.
- Humor: A question designed to interject humor into the argument and relieve tension.

A lawyer presenting her case before a court at oral argument must be able to quickly and accurately determine why the judge has asked a particular question so as to answer it on behalf of her client in the most persuasive way possible. It is also a difficult task. Under the pressure of persistent and difficult questioning it is easy for a lawyer to misread a question and offer an unresponsive or misguided answer. Skillful lawyers learn to quickly understand not only judges' questions but the reasons they are asked.

**Task**  The Purpose of Oral Argument Questions task requires an LLM to determine–given a question from an oral argument transcript–for which of the seven purposes above the judge asked the question.

**Construction process**  We created a dataset of questions from U.S. Supreme Court oral argument transcripts, classified into one of the seven functions above. Questions were taken from cases argued in the 2022-23 Supreme Court term, in reverse chronological order. A question was defined as the totality of a judge's words prior to an advocate's response, regardless whether the words constituted a true interrogatory sentence. To include a sufficient number of questions of each type in the dataset, questions were not drawn at random. Instead rarer question types (e.g. humor and communication) were targeted for inclusion, questions of more common types (e.g. clarification) were frequently omitted.

**Significance and value**  Young attorneys, and even many experienced ones, struggle with oral advocacy. It requires comfort in the courtroom, quick thinking, and careful demeanor to assess from the tone and content of a question why the judge is asking it and how most effectively to respond. In one regard, LLMs will be superior. They do not suffer from human nervousness. However, given only a text prompt rather than an audible question from which to infer tone, and given only a judge's question and not the full case context, this task would be a challenge even for a seasoned lawyer. Whether LLMs can succeed will be an extremely interesting measure of progress in legal analysis.

---

[23]This categorization scheme is from [148], employed and discussed along with other possible classification schemes at [45].

| Question | Purpose |
|---|---|
| May I ask you a question about standing? So it's the case, isn't it, that if any party in either of these two cases has standing, then it would be permissible for us to reach the merits of the issue? | Background |
| I guess I don't understand that answer. In other words, is it simply adding for religious reasons to the label that would change whether it could be regulated or not? | Clarification |
| And we have amicus brief from different stakeholders, some saying it may not apply in parody, but it could apply in movie titles, it might apply in something else and not this, in novels, et cetera. Why should we rule broadly? And if we rule narrowly, on what basis? You heard earlier at least three alliterations, one, the – Justice Kagan's, one Justice Jackson, one me, limit this just to parodies, because parodies really do rely on is this a joke that people are going to get. | Communicate |
| Mr. Martinez, I think one of the problems that you have, as evidenced by a lot of the questions that you've been getting, is with the derivative works protection, you know, which, in, you know, 106(2), actually talks about transforming any other form in which a work may be recast, transformed, or adapted. And it seems to me like your test, this meaning or message test, risks stretching the concept of transformation so broadly that it kind of eviscerates Factor 1 and puts all of the emphasis on Factor 4. I mean, when you've been asked about book to movie and – and – and, you know, songs, you keep flipping to Factor 4. So, if a work is derivative, like Lord of the Rings, you know, book to movie, is your answer just like, well, sure, that's a new meaning or message, it's transformative, so all that matters is 4? | Criticism |
| What are your two ones that you're like killers? | Humor |
| So what's the limiting line of yours – of yours? Justice Kagan asked you about another website designer. But how about people who don't believe in interracial marriage or about people who don't believe that What's – where's the line? I choose to serve whom I want. If I disagree with their personal characteristics, like race or disability, I can choose not to sell to those peopledisabled people should get married? | Implications |
| There were several questions earlier about the justification for granting preference for foster or adoptive parents who are members of an entirely different tribe. Could you speak to that? | Support |

Table 39: Examples for `oral_argument_question_purpose`

### G.20 Overruling

In LEGALBENCH, the Overruling task is denoted as `overruling`.

**Background**  A hallmark of the common-law legal system is that courts will *overrule* previous judicial decisions. The act of overruling is significant, and it indicates that the overruled decision was either in accurate in its articulation/application of a particular law, or that overruling court wishes to announce a substantive change in law.

**Task**  In this task, an LLM is required—given an excerpt of judicial text—to determine if the text overrules a previous decision.

**Construction process**  This task is taken from [155], which previously studied the capacity for finetuned BERT models to perform this task. Please refer to [155] for more information on the construction process.

**Significance and value**  Identifying when judicial text overrules another case is a basic but essential lawyering skill. From a practical standpoint, the capacity for LLMs to correctly classify overruling sentences could have practical applications for the design and construction of legal opinion databases. When using or citing a case in legal arguments, lawyers must ensure that the case hasn't been an overruled, and is still "good law." Tools which automatically parse legal databases and extract cases which have been overruled would thus be helpful for constructing legal arguments.

| Sentence | Overruling sentence? |
| --- | --- |
| brockett v. spokane arcades, inc., 472 u.s. 491, 501 (1985) (citations omitted). | No |
| we overrule so much of kerwin as holds that a criminal defendant is not entitled to inspect and make an analysis of the seized controlled substance. | Yes |

Table 40: Examples for `overruling`.

### G.21 Personal Jurisdiction

In LEGALBENCH, the Personal Jurisdiction task is denoted as `personal_jurisdiction`.

**Background**  Personal jurisdiction refers to the ability of a particular court (e.g. a court in the Northern District of California) to preside over a dispute between a specific plaintiff and defendant. A court (sitting in a particular forum) has personal jurisdiction over a defendant only when that defendant has a relationship with the forum. We focus on a simplified version of the rule for federal personal jurisdiction, using the rule:

> There is personal jurisdiction over a defendant in the state where the defendant is domiciled, or when (1) the defendant has sufficient contacts with the state, such that they have availed themselves of the privileges of the state and (2) the claim arises out of the nexus of the defendant's contacts with the state.

Under this rule, there are two paths for a court have jurisdiction over a defendant: through domicile or through contacts.

- Domicile: A defendant is domiciled in a state if they are a citizen of the state (i.e. they live in the state). Changing residency affects a change in citizenship.

- Contacts: Alternatively, a court may exercise jurisdiction over a defendant when that defendant has *sufficient contacts* with the court's forum, and the legal claims asserted arise from the *nexus* of the defendant's contacts with the state. In evaluating whether a set of contacts are sufficient, lawyers look at the extent to which the defendant interacted with the forum, and availed themselves of the benefits and privileges of the state's laws. Behavior which usually indicates sufficient contacts include: marketing in the forum or selling/shipping products into the forum. In assessing nexus, lawyers ask if the claims brought against the defendant arise from their contacts with the forum. In short: is the conduct being litigated involve the forum or its citizens in some capacity?

**Task**  The personal jurisdiction task requires an LLM to determine—given a fact pattern describing the events leading up to a legal claim—whether a particular court has personal jurisdiction over the defendant.

**Construction process**  We manually construct a dataset to test application of the personal jurisdiction rule, drawing inspiration from exercises found online and in legal casebooks. Each sample in our dataset describes a "fact pattern," and asks if a court located in particular state (**A**) can exercise personal jurisdiction over an individual (**B**) named in the fact pattern. In designing the dataset, we use 5 base fact patterns, and create 4 slices, where each slice evaluates a different aspect of the personal jurisdiction rule:

- Domicile: Fact patterns where **B** is domiciled in **A**. Hence, personal jurisdiction exists.

- No contacts: Fact patterns where **B** has insufficient contacts with **A**. Hence there is no personal jurisdiction.

- Yes contacts, no nexus: Fact patterns where **B** has sufficient contacts with **A**, but the claims against **B** do not arise from those contacts. Hence, personal jurisdiction does not exist.

- Yes contacts, yes nexus: Fact patterns where **B** has sufficient contacts with **A**, and the claims against **B** arise from those contacts. Hence, there is personal jurisdiction.

Caveat. Personal jurisdiction is a rich and complex doctrine. Our dataset focuses on a narrow class of fact patterns, related to jurisdiction over individuals. We don't consider, for instance, more complex questions related to adjudicating citizenship (e.g. the *Hertz* test) or the classic stream-of-commerce problems. We leave this to future work.

**Significance and value**  Identifying when personal jurisdiction exists is a skill that law students learn in their first-year civil procedure course. The personal jurisdiction task is interesting because applying even the simplified version of the rule requires reasoning over the degree of connection between a defendant and the forum state.

| Facts | Personal jurisdiction? |
|---|---|
| Dustin is a repairman who lives in Arizona and repairs computers in California, Oregon, and Washington. Dustin is an avid skier, so his favorite place to go on vacation is Colorado. While travelling to repair a computer in Washington, Dustin is involved in a car crash in Oregon with Laura, a citizen of Oregon. After the accident, Dustin returns to Arizona. Laura sues him in Colorado. | No |
| David is a citizen of California. He flies to New York for a vacation, where he meets Maggie, who is also visiting from Rhode Island. While they chat, Dave fraudulently tricks Maggie into giving him her savings. David then continues his vacation and visits Texas, Oregon, Florida, and New Mexico. After he returns home, Maggie sues David for fraud in Oregon. | No |
| Ana is a lawyer who resides in Texas. While visiting Louisiana, she meets David, who runs a bike shop. David's bike shop is famous, and he frequently advertises his bikes in Texas newspapers. Ana buys a bike from David and rides it back home. Right after she crosses the border, the bike seat explodes, injuring Ana. Ana sues David in Texas. | Yes |
| Tony (from Texas) is a regional manager for a cookbook company, Tasty Eats Books (incorporated and principal place of business in Delaware). Tony's job requires him to travel from city to city to show new cookbooks to chefs. In January 2022, he was scheduled to visit restaurants in Illinois, Indiana, and Michigan. While in Michigan, Tony goes to Lake Erie to blow off some steam. He ends up getting into a fight with Arthur, a lawyer from Detroit, Michigan. Tony and Arthur each blame the other for starting the fight. Arthur sues Tony in Texas. | Yes |

Table 41: Examples for `personal_jurisdiction`.

### G.22 Privacy Policy Entailment

In LEGALBENCH, the Privacy Policy Entailment task is denoted as `privacy_policy_entailment`.

**Background**   The Privacy Policy Entailment task is created from the APP-350 corpus [159], which consists of 350 Android app privacy policies annotated with different privacy practices. In this corpus, individual clauses are annotated based on whether they do or do not perform a certain practice (e.g., "We access your location information").

**Task**   Given a clause from a privacy policy and a description of the practice, the LLM must determine if the clause describes the performance of that practice. This is analogous to an entailment task, where the premise is the clause, and the hypothesis is the practice description.

**Construction process**   For each practice coded in the APP-350 corpus, we derive a natural language description of that practice, which serves as our "hypothesis." Each instance of this task corresponds to a triple containing a clause, a practice description, and a binary classification (Yes/No) based on whether the clause performs the practice. Across the dataset there are 57 unique policy descriptions.

| Clause | Description | Performed? |
|---|---|---|
| We may collect and record information through the SN Service in accordance with the policies and terms of that SN Service. The information we collect when you connect your user account to an SN Service may include: (1) your name, (2) your SN Service user identification number and/or user name, (3) locale, city, state and country, (4) sex, (5) birth date, (6) email address, (7) profile picture or its URL, and (8) the SN Service user identification numbers for your friends that are also connected to Supercell's game(s). | The policy describes receiving data from an unspecified single sign on service | Yes |
| Your e-mail address will not be stored. | The policy describes collection of the user's e-mail by a party to the contract. | No |

Table 42: Example clause-description-label pairs for Privacy Policy Entailment task.

**Significance and value**   The privacy policy entailment task is similar to ContractNLI, in that it evaluate a LLM's capacity to perform entailment-style reasoning over formal legal language. From a lawyerly perspective, understanding whether a policy performs certain functions or empowers one of the parties to pursue practices is an essential element of legal comprehension. From a practical perspective, the ability for LLMs to perform this task could empower researchers to conduct broader studies of privacy agreements. As [159] observes, annotation cost limitations often restrict the scope of empirical studies of privacy agreements.

### G.23 Privacy Policy QA

In name, the Privacy Policy QA task is denoted as `privacy_policy_qa`.

**Background**    The Privacy Policy QA task is derived from [110], which annotated clauses in mobile application privacy policies based on whether they contain the answer to a question.

**Task**    Given an excerpt from a privacy policy and a question, the LLM must determine whether the excerpt is relevant to answering the question or not.

**Construction process**    We used the snippet annotations available in [110] to construct this task, removing all snippets with fewer than 10 words. Examples of excerpt/question/relevant tuples are shown in the table below. 5449 instances correspond to a relevant question-clause pair, and 5474 instances correspond to an irrelevant question-clause pair.

| Excerpt | Question | Class |
|---|---|---|
| We also use cookies, tags, web beacons, local shared objects, files, tools and programs to keep records, store your preferences, improve our advertising, and collectNon-Identifying Information, includingDevice Dataand information about your interaction with the Site and ourBusiness Partners'web sites. | is my search and purchase history shared with advertisers? | Relevant |
| We collect information about the value added services you are using over Viber and/or apps (such as games) you have downloaded through Viber. | does viber sell my information to advertisers and marketers? | Irrelevant |

Table 43: Examples for Privacy Policy QA

**Significance and value**    Determining when a particular legal excerpt is relevant to answering a question is essential to interpreting legal documents. This task allows us to evaluate LLMs for this capability. From a more practical standpoint, the Privacy Policy QA task is a helpful evaluation task when developing LLM systems which involve decompositions over long documents. A common approach—in order to account for the fact that many long documents exceed ordinary context windows—is to chunk documents into smaller segments, and apply a LLM independently to filter out irrelevant segments. For QA tasks involving long policies, this task allows practitioners to measure performance for the filtering step.

## G.24 Private Right of Action (PROA)

In LEGALBENCH, the Private Right of Action task is denoted as `proa`.

**Background**   A private right of action (PROA) exists when a statute empowers an ordinary individual (i.e. a private person) to legally enforce their rights by bringing an action in court. In short, a PROA creates the ability for an individual to sue someone in order to recover damages or halt some offending conduct. PROAs are ubiquitous in antitrust law (in which individuals harmed by anti-competitive behavior can sue offending firms for compensation) and environmental law (in which individuals can sue entities which release hazardous substances for damages) [50].

**Task**   In the PROA task, an LLM must determine if a statutory clause contains a private right of action.

**Construction process**   We construct a dataset of PROAs by hand, drawing inspiration from clauses found in different state codes. We construct 50 clauses which do contain a PROA, and 50 clauses which do not. Clauses which do not contain a private right may either create no cause of action, or empower a non-private individual (e.g., an attorney general) to bring a claim. 5 randomly sampled clauses constitute the training set, and the remaining 95 form the test set.

| Input | Answer |
|---|---|
| The attorney general or an attorney representing the state may bring an action for an injunction to prohibit a person from violating this chapter or a rule adopted under this chapter. | No |
| The administrator may bring an action in a court of competent jurisdiction to enforce this chapter. | No |
| The sheriff or the sheriff's designee shall maintain a permanent personnel file on each department employee. | No |
| If any laborer, without good cause, shall abandon his or her employer before the expiration of his or her contract, he or she shall be liable to his or her employer for the full amount of any account he or she may owe his or her employer. | Yes |
| No employer may discharge any employee by reason of the fact that earnings have been subjected to garnishment or execution. If an employer discharges an employee in violation of this section, the employee may within ninety days of discharge bring a civil action for recovery of twice the wages lost as a result of the violation and for an order requiring reinstatement. | Yes |
| In addition to all other penalties, rights, or remedies provided by law, an individual or entity that uses or attempts to use its official authority or influence for the purpose of interfering with the right of a legislative employee to make a protected disclosure is liable in a civil action for damages brought by a legislative employee. | Yes |

Table 44: Examples for proa

**Significance and value**   The PROA task evaluates an LLM's ability to perform a two-step reasoning test: (1) does the statute allow a party to bring a claim in court, and (2) is that party private? Law students and legal professionals should be capable of performing this task at near-perfect accuracy.

The PROA task derives additional significance from a recent movement towards studying state statutory language [153]. Legal scholars have long been unable to conduct large scale empirical studies of state statutory language, given the sheer volume of state statutes. The ability for LLMs to accurately classify or annotate statutes could thus empower new empirical studies of state statues.

## G.25 Rule QA

In LEGALBENCH, the Rule QA task is denoted as `rule_qa`.

**Background**   Lawyers are regularly required to recall specific legal *rules* that are drawn from cases, statutes, or other sources. Rules can take many shapes and forms. For instance, the rule pertaining to the federal requirements for a class (in a class action lawsuit) are codified in Rule 23(a) of the Federal Rules of Civil Procedure and are simply known as the need for "numerosity, commonality, typicality, and adequacy."

**Task**   The Rule QA task evaluates a LLM's ability to answer questions on different legal rules. The rules are drawn from subjects typically studied in the first year of law school (e.g., civil procedure, constitutional law, etc.). This is an open-generation task.

**Construction process**   We manually wrote 50 question-answer pairs, focusing on the types of rules which are regularly tested in law school courses on civil procedure, evidence, and intellectual property. The questions ask the LLM to either (1) restate a rule, (2) identify where a rule is codified, or (3) list the factors employed in a particular rule. Several questions explicitly narrow their scope to a jurisdiction (e.g., California state evidence law), in order to avoid bias towards merely federal law.

| Question | Answer | Area of law |
|---|---|---|
| What are the four categories of patentable subject matter? | "process, machine, manufacture, or composition of matter." | IP |
| What are the requirements for diversity jurisdiction? | Diversity jurisdiction exists when the amount in controversy exceeds $75,000 and the plaintiffs and defendants are completely diverse (i.e. no plaintiff shares a state of citizenship with any defendant) | Civil Procedure |
| Under which statute are patentable subject matter requirements codified? | 35 USC 101 | IP |
| What are the factors of the Mathews balancing test? | A three-part test that determines whether an individual has received due process under the Constitution. The test balances (1) the importance of the interest at stake; (2) the risk of an erroneous deprivation of the interest because of the procedures used, and the probable value of additional procedural safeguards; and (3) the government's interest. | Constitutional law |

Table 45: Examples for Rule QA

**Significance and value**   The Rule QA task is an initial effort to evaluate the propensity for legal hallucination in LLMs. The questions asked are exceedingly basic, and law students taking the relevant course would be expected to answer them nearly perfectly.

### G.26 SARA Tasks

In LEGALBENCH, the SARA tasks are denoted as `sara_*`.

**Background**   An important skill for lawyers is to determine, given the facts of a case, whether a given law applies and what it prescribes. For example, *does the payment received by the defendant on August 21st , 2017 qualify as wages under §3306(b) of the US Tax Code?* This task has been introduced by [76] as statutory reasoning. [63] further introduce the Statutory Reasoning Assessment dataset (SARA). SARA contains (1) a set of 9 sections, taken from US federal tax law statutes, pruned and simplified; and (2) hand-crafted cases that test the understanding of those 9 sections. In this context, a case is a paragraph of text stating facts in plain language. Each case comes either with an entailment prompt — a statement about the statutes and the case that may be true or false — or a question — asking how much tax one of the case's protagonists owes. The SARA dataset is a simplified version of real-world cases, that retains many of the features of statutory reasoning for tax law. It poses, however, a significant challenge to NLP models [12].

**Tasks**   There are two SARA tasks. The first, `sara_entailment`, corresponds to the entailment cases. The entailment cases state that a given law applies to a given case, and require the LLM to produce a binary answer — akin to Recognising Textual Entailment [39]. This is an approximation of real-world statutory reasoning, where the answer is usually not strictly binary.

The second task, `sara_numeric`, consists of the numeric cases. Here, the goal is to compute the amount of tax owed. We frame this as a floating point number. To measure numerical accuracy, we use the metric introduced by [63], which includes a tolerance for inaccurate predictions.

**Construction process**   We framed the SARA dataset for the paradigm of language modeling. Due to dependencies across sections in the statutes, the entirety of the statutes are generally relevant to determine the answer to any of the cases. However, all 9 sections do not fit into the LLM's context window, and must be pruned. In entailment cases, the entailment prompt specifies which law from the statutes to apply. We automatically extract the text of that law, and use it as the language model prompt. For numerical cases, the entirety of the statutes are relevant, and we use that as the language model prompt. Pruning is left to the LLM's pre-processing.

**Significance and value**   Statutory reasoning is an important skill for lawyers, that is used within many other legal tasks. It is a fundamental task for legal AI, probing whether a computational model can understand and reason with legal rules expressed in natural language. The types of reasoning involved in SARA are diverse — defeasible, temporal, numerical reasoning, inter alia — and relevant beyond the legal domain. Statutory reasoning combines natural language understanding and logical reasoning, a major goal for AI.

If statutory reasoning were solved, it could serve as a basis for more complex legal tasks. For example, it could be used to automate the computation of taxes and benefits, without the need for coding the expert systems in use in many parts of the world [99]. A system that can do statutory reasoning would also be a step towards machine reading models that can analyze legislation and anticipate its effects, coming up with possible application scenarios [11]. As a final example, a statutory reasoning agent could be used for basic legal advice, increasing access to justice.

**Task name**: `sara_entailment`
**Statute**: (2) an individual legally separated from his spouse under a decree of divorce or of separate maintenance shall not be considered as married.
**Description**: Alice and Bob got married on April 5th, 2012. Alice and Bob were legally separated under a decree of divorce on September 16th, 2017.
**Statement**: Section 7703(a)(2) applies to Alice for the year 2018.
**Answer**: Entailment

**Task name**: `sara_numeric`
**Statute**: §3301. Rate of tax 
 
 There is hereby imposed on every employer (as defined in section 3306(a)) for each calendar year an excise tax, with respect to having individuals....**[Ommitted from clarity]**...This section shall not apply to any taxable year beginning after December 31, 2017, and before January 1, 2026. 
 
**Description**: Bob is Charlie and Dorothy's son, born on April 15th, 2015. Alice married Charlie on August 8th, 2018. Alice's and Charlie's gross incomes in 2018 were $324311 and $414231 respectively. Alice, Bob and Charlie have the same principal place of abode in 2018. Alice and Charlie file jointly in 2018, and take the standard deduction.
**Question**: How much tax does Alice have to pay in 2018?
**Answer**: $259487

Table 46: Example of each SARA task. For `sara_numeric`, we ommit part of the statute for brevity.

## G.27 SCALR

In LEGALBENCH, the SCALR task is denoted as `scalr`.

**Background**  Each case decided by the Supreme Court addresses a specific *question presented for review*. Both the questions and the Court's opinions are published on the Supreme Court's website.

Many of the Court's opinions are briefly described by other judges who recount the holdings of the Court in their own writing. For example, consider the following passage from State of South Carolina v. Key, 27971 (S.C. 2020; emphasis added):

> The United States Supreme Court has addressed the constitutionality of warrantless blood draws in several DUI cases. See Schmerber, 384 U.S. at 770-71 (**holding** the warrantless blood draw of a DUI suspect was valid because the law enforcement officer, dealing with a car accident, could "reasonably have believed that he was confronted with an emergency, in which the delay necessary to obtain a warrant, under the circumstances, threatened 'the destruction of evidence'")...

We refer to these brief descriptions as 'holding statements' or 'holding parentheticals,' since they are often enclosed by parentheses. Identifying the holding parenthetical that corresponds to a question presented for review requires a notion of 'responsiveness' or relevance between questions and answers as well as an understanding of the kinds of legal issues that could be implicated by a specific question presented for review.

**Task**  The SCALR benchmark is a collection of 571 multiple choice questions designed to assess the legal reasoning and reading comprehension ability of large language models. Each multiple-choice task gives the question presented for review in a particular Supreme Court case. The solver must determine which holding parenthetical describes the Court's ruling in response to the question presented. Here is an example from *AT&T Mobility LLC v. Concepcion*, 563 U.S. 333 (2011) with the correct response emphasized:

> Question: Whether the Federal Arbitration Act preempts States from conditioning the enforcement of an arbitration agreement on the availability of particular procedures–here, class -wide arbitration–when those procedures are not necessary to ensure that the parties to the arbitration agreement are able to vindicate their claims.
>
>> A: holding that when the parties in court proceedings include claims that are subject to an arbitration agreement, the FAA requires that agreement to be enforced even if a state statute or common-law rule would otherwise exclude that claim from arbitration
>> B: holding that the Arbitration Act "leaves no place for the exercise of discretion by a district court, but instead mandates that district courts shall direct the parties to proceed to arbitration on issues as to which an arbitration agreement has been signed"
>> C: holding that class arbitration "changes the nature of arbitration to such a degree that it cannot be presumed the parties consented to it by simply agreeing to submit their disputes to an arbitrator"
>> **D: holding that a California law requiring classwide arbitration was preempted by the FAA because it "stands as an obstacle to the accomplishment and execution of the full purposes and objectives of Congress," which is to promote arbitration and enforce arbitration agreements according to their terms**
>> E: holding that under the Federal Arbitration Act, a challenge to an arbitration provision is for the courts to decide, while a challenge to an entire contract which includes an arbitration provision is an issue for the arbitrator

**Construction**  The data used to create this task comes from two sources:

1. Questions presented were gathered from the Supreme Court of the United States' website, which hosts PDFs of questions granted for review in each case dating back to the 2001 Term.

2. Holding statements that comprise the "choices" for each question were compiled from (a) CourtListener's collection of parenthetical descriptions and (b) extraction of parenthetical descriptions from Courtlistener's and the Caselaw Access Project's collections of court decisions using Eyecite.

Because questions presented for review in Supreme Court cases are not easily available prior to 2001, the benchmark is limited to questions from cases decided in the 2001 Term and later. To ensure that "holding" statements would address the particular question presented, we limited the set of cases to those in which exactly one question was granted for review. We also perform some manual curation to exclude questions which are not

answerable without specific knowledge of a case. For example, we eliminated a case that presented this question: "Whether this Court's decision in Harris v. United States, 536 U.S. 545 (2002), should be overruled."

To create choices for each question presented, we first filter our set of parenthetical descriptions as follows:

1. We limited our parenthetical descriptions to only those that begin with "holding that...", as these are most likely to describe the core holding of the case, rather than some peripheral issue.

2. We use only parentheticals that describe Supreme Court cases. This avoids the creation of impossible questions that ask the solver to distinguish between "holding" statements dealing with exactly the same issue at different stages of appellate review.

3. We then select for each case the longest parenthetical meeting the above criteria. We use the longest parenthetical because it is most likely to be descriptive enough to make the question answerable.

We then create a task for each case which has both a question presented and a "holding" statement meeting the above requirements. (While question-correct holding pairs are only for cases decided after 2001, we allow the use of parentheticals describing any Supreme Court case as alternative answer choices.) We then need to select the four alternative answer choices for each question in a manner that makes the task challenging. To select choices that are at least facially plausible, we find the four "holding" statements from the remaining pool that are most TF-IDF similar to the question presented. The inclusion of difficult alternative choices requires the solver to draw nuanced distinctions between legal issues that share overlap in terminology.

**Significance and value**   This task is significant because it tracks the useful and challenging skill of identifying a passage as relevant or responsive to a given query. LLMs that are able to perform well at this task have the potential to be more useful for complex legal question-answering and retrieval. The poor performance of simpler models on this task demonstrates that it is a challenging one that requires a level of understanding beyond the word/synonym level.

## G.28  Securities Complaint Extraction

In LEGALBENCH, the Securities Complaint Extraction tasks are denoted as `ssla_*`.

**Background**  Securities Class Actions (SCAs) are lawsuits filed by, and on behalf of, investors alleging economic injury as a result of material misstatements or omissions in public disclosures made by corporate directors and officers. These actions allege violations of the Securities Act of 1933 and Securities Exchange Act of 1934 and are predominately filed in federal court, though in 2018 the United States Supreme Court determined actions brought under the '33 Act were permitted in state court.

"Plaintiff(s)" is the legal term to describe the individual, company, or organization bringing forth a lawsuit. Under the class action system, one or more plaintiffs are appointed "Lead Plaintiff" by the court to represent the interests of a larger group of "similarly situated" parties. In securities class actions, investors that suffered the greatest financial loss, often public pensions or unions, are appointed lead plaintiff.

"Defendant(s)" is the legal term to describe the individual, company, or organization that must defend themself against the alleged violations or misconduct outlined in the lawsuit. There is always at least one defendant. The majority of securities class actions name the company, its CEO and its CFO. Many name additional C-suite level officers, members of the Board of Directors and additional third-parties such as the company's independent auditor and the underwriters of public offerings.

Each designated lead plaintiff, and all named defendants, are explicitly identified under the "Parties" section of the class action complaint.

**Tasks**  There are three extraction tasks.

- The plaintiff task requires an LLM to extract the named plaintiffs within a text.

- The individual defendants tasks require an LLM to extract named defendants who are individuals from within a text.

- The company defendants tasks require an LLM to extract named defendants who are corporations/companies from within a text.

For certain samples, the complaint excerpt may not exactly contain the answer. For example, the correct answer may be "Strongbridge Biopharma PLC," while the complaint only mentions "Strongbridge." In order to maintain fidelity to the workflow used by SSLA, we evaluate an LLM's ability to generate the official name of the entity, as represented in the answer. This requires the LLM to possess some background knowledge regarding official corporation names. We find that larger models are generally able to account for this.

Sometimes, the provided text will not explicitly name the plaintiff, an individual defendant, or a corporate defendant. In these cases, the LLM is expected to return "Not named".

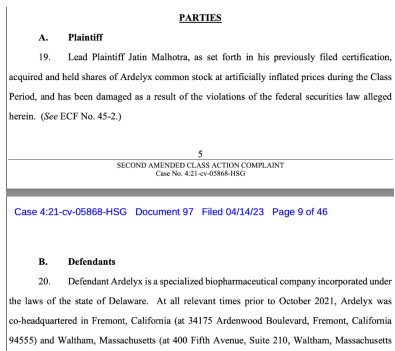

Figure 2: Example of the typical SCA structure (Case 4:21-cv-05868-HSG)

**Construction process**  Stanford Securities Litigation Analytics (SSLA) identifies, tracks, and aggregates data on the several hundred private shareholder lawsuits and public SEC/DOJ enforcements filed each year. SSLA fellows manually extract and analyze information including plaintiffs, defendants, judges, mediators, plaintiff and defense firms, key litigation events, real-time case statuses, settlement timing, settlement dollar amounts, attorneys' fees and expenses, and imposed SEC / DOJ penalties. There is no ambiguity regarding the answers for this task given its nature.

This dataset is an extract from the corpus of texts of securities class action complaints in the SSLA database. Given the typical structure and headings for these types of cases, this dataset represents text extracted from the complaint, between the sections titled "Parties" and "Substantive Allegations". For cases where the second

heading was not found, texts fragments were limited to 2,000 characters. Cases with both headings were then filtered to include only those with texts up to 2,000 characters, which excluded cases with longer "Parties" sections. Thus, all observations in this dataset are 2,000 characters or less.

Text was scraped from complaints using python's PyPDF2 library and left unformatted and uncleaned. This training set includes several observations where the text does not include all or some of the named entities due to the method of text collection and variation in case structure. In several of these cases, plaintiff names are not present in the selected text because the plaintiff had been named earlier in the complaint.

Table 47: Examples from Securities Complaint Tasks

**Task**

**Task name**: `ssla_company_defendants`
**Excerpt**: 6. Plaintiff Don L. Gross, as set forth in the accompanying certification and incorporated by reference herein, purchased the common stock of KCS during the Cla ss Period and has been damaged thereby. 7. Defendant KCS, headquartered in Kansas C ity, Missouri, operates railroads in the Midwest and Mexico that run north to south, unlike most other U.S. ra ilroads that run east to west. The Company's stock traded on the NYSE, an ef ficient market, during th e Class Period under the ticker symbol "KSU." As of October 11, 2013, there were more than 110 million shares issued and outstanding. 8. Defendant David L. Starling ("Starling"), at all relevant times, served as KCS's President and Chief Executive Officer ("CEO"). Case 4:14-cv-00345-BCW Document 1 Filed 04/15/14 Page 2 of 293 9. Defendant David R. Ebbrecht ("Ebbrecht"), at all relevant times, served as KCS's Executive Vice President and Chief Operating Officer ("COO"). 10. Defendant Patrick J. Ottensmeyer ("Ottensmeye r"), at all relevant times, served as KCS's Executive Vice President Sales & Marketing. 11. Defendant Michael W. Upchurch ("Upchurch "), at all relevant times, served as KCS's Executive Vice President and Chief Financial Officer ("CFO"). 12. Defendants Starling, Ebbrecht, Ottensmeyer an d Upchurch are collec tively referred to herein as the "Individual Defendants." 13. During the Class Period, the Individual Defe ndants, as senior executive officers and/or directors of KCS, were privy to confidential and proprie tary information concerning KCS, its operations, finances, financial cond ition and present and future business prospects. The Individual Defendants also had access to material adverse non-public information concerning KCS, as discussed in detail below. Because of their positions with KCS, the Individual Defendants had access to non-public information about its busine ss, finances, products, markets and present and future business prospects via interna
**Answer**: Kansas City Southern

**Task name**: `ssla_individual_defendants`
**Excerpt**: 6. Plaintiff Don L. Gross, as set forth in the accompanying certification and incorporated by reference herein, purchased the common stock of KCS during the Cla ss Period and has been damaged thereby. 7. Defendant KCS, headquartered in Kansas C ity, Missouri, operates railroads in the Midwest and Mexico that run north to south, unlike most other U.S. ra ilroads that run east to west. The Company's stock traded on the NYSE, an ef ficient market, during th e Class Period under the ticker symbol "KSU." As of October 11, 2013, there were more than 110 million shares issued and outstanding. 8. Defendant David L. Starling ("Starling"), at all relevant times, served as KCS's President and Chief Executive Officer ("CEO"). Case 4:14-cv-00345-BCW Document 1 Filed 04/15/14 Page 2 of 293 9. Defendant David R. Ebbrecht ("Ebbrecht"), at all relevant times, served as KCS's Executive Vice President and Chief Operating Officer ("COO"). 10. Defendant Patrick J. Ottensmeyer ("Ottensmeye r"), at all relevant times, served as KCS's Executive Vice President Sales & Marketing. 11. Defendant Michael W. Upchurch ("Upchurch "), at all relevant times, served as KCS's Executive Vice President and Chief Financial Officer ("CFO"). 12. Defendants Starling, Ebbrecht, Ottensmeyer an d Upchurch are collec tively referred to herein as the "Individual Defendants." 13. During the Class Period, the Individual Defe ndants, as senior executive officers and/or directors of KCS, were privy to confidential and proprie tary information concerning KCS, its operations, finances, financial cond ition and present and future business prospects. The Individual Defendants also had access to material adverse non-public information concerning KCS, as discussed in detail below. Because of their positions with KCS, the Individual Defendants had access to non-public information about its busine ss, finances, products, markets and present and future business prospects via interna
**Answer**: David L Starling, David R Ebbrecht, Patrick J. Ottensmeyer, Michael W. Upchurch

**Task name**: ssla_plaintiff

**Task**

**Excerpt**: 11. Plaintiff, as set forth in th e attached certification, purchas ed Catalyst securities at artificially inflated prices dur ing the Class Period and has b een damaged upon the revelation of the alleged corrective disclosures. 12. Defendant Catalyst is a Coral Gates, Florida headquartered company located at 355 Alhambra Circle Suite 1500 Coral Gates, FL 33134. The common stock is traded on the NASDAQ Stock Market ("NASDAQ") unde r the ticker symbol "CPRX." 13. Defendant Patrick J. McEnany ("McEna ny") is the Company's co-founder, CEO and President. 14. Defendant Dr. Hubert E. Huckel M.D. ("Huckel") is the Company's co-founder and one of its directors. 15. Defendant Steven R. Miller Ph. D. ("M iller") is the company's COO and CSO. 16. The defendants referenced above in ¶¶ 13- 15 are sometimes referred to herein as the "Individual Defendants." DEFENDANTS' WRONGDOING

Background Case 1:13-cv-23878-UU Document 1 Entered on FLSD Docket 10/25/2013 Page 4 of 20 5 17. Catalyst is a specialty pharmaceutical company which develops and commercializes drugs treating orphan (rare) neuromuscular an d neurological diseases. 18. Lambert-Eaton Myasthenic Syndrome ("LEM S") is an extremely serious disase which is also extremely rare, afflicting about 3.4 persons per million, and about one to two thousand patients in the United States. 19. FDA rules permit so-called "compassionate use" – use of a drug that has not been approved by the FDA outside of clinical trials. A patient may be given drugs under a compassionate use program if the patient may benefit from the treatment, the therapy can be given safely outside the clinical trial setting, no other alternative therapy is available, and the drug developer agrees to provide access to the drug. 20. Jacobus is a tiny privat e pharmaceutical company in New Jersey, with only dozens of employees, and only 35 as of 2009. Jacobus has b een manufacturing 3,4 DAP and providing it to patients through a

**Answer**: Not named

**Significance and value**  Extracting data from legal documents is an extraordinarily resource- and time-intensive effort prone to human error. As a result, there are no known databases of non-securities class action litigation, despite the obvious public policy implications of the class action system. Automation of identification tasks coupled with human approval would improve efficiency and reduce collection costs and data errors. This task may be useful to other legal researchers and industry practitioners extracting structured data from complex texts. Identification is a very simple task that can be done by those with an understanding and familiarity with the underlying legal documents and legal system, but an LLM's ability to accurately and precisely identify entities is a useful metric to assess.

### G.29 Successor Liability

In LEGALBENCH, the Successor Liability task is denoted as `successor_liability`.

**Background** When one company sells its assets to another company, the purchaser is generally not liable for the seller's debts and liabilities. Successor liability is a common law exception to this general rule. In order to spot a successor liability issue, lawyers must understand how courts apply the doctrine.

The doctrine holds purchasers of all, or substantially all, of a seller's assets liable for the debts and liabilities of the seller if:

1. the purchaser expressly agrees to be held liable;
2. the assets are fraudulently conveyed to the purchaser in order to avoid liability;
3. there is a de facto merger between the purchaser and seller; or
4. the purchaser is a mere continuation of the seller.

Express agreement is governed by standard contract law rules. In practice, if a purchase agreement contains a provision to assume liabilities, litigation will rarely arise. Courts, however, sometimes interpret an implied agreement in the absence of a written provision.

Assets are fraudulently conveyed when the seller intends to escape liability through a sale or knows that liability will be avoided through a sale.

De facto merger is a multifactor test that consists of (1) continuity of ownership; (2) cessation of ordinary business and dissolution of the acquired corporation as soon as possible; (3) assumption by the purchaser of the liabilities ordinarily necessary for the uninterrupted continuation of the business of the acquired corporation; and (4) continuity of management, personnel, physical location, assets, and general business operation. Some jurisdictions require a showing of all four elements. Others do not, and simply emphasize that the substance of the asset sale is one of a merger, regardless of its form.

Mere continuation typically requires a showing that after the asset sale, only one corporation remains and there is an overlap of stock, stockholders, and directors between the two corporations. There are two variations of the mere continuation exception. The first variation is the "continuity of enterprise" exception. In order to find continuity of enterprise, and thus liability for the purchaser of assets, courts engage in a multifactor analysis. Factors include: (1) retention of the same employees; (2) retention of the same supervisory personnel; (3) retention of the same production facilities in the same physical location; (4) production of the same product; (5) retention of the same name; (6) continuity of assets; (7) continuity of general business operations; and (8) whether the successor holds itself out as the continuation of the previous enterprise. The second variation is the product line exception. This exception imposes liability on asset purchasers who continue manufacturing products of a seller's product line. This exception generally requires that defendants show that the purchaser of assets is able to assume the risk spreading role of the original manufacturer, and that imposing liability is fair because the purchaser enjoys the continued goodwill of the original manufacturer.

Scholars have noted that fraud, de facto merger, and mere continuation (and its variants) overlap. They share the common thread of inadequate consideration, that is, the consideration given in exchange for the assets is unable to fund the liabilities that underwrite those assets. Because of the overlap, different courts might apply different doctrines to identical sets of facts, but arrive at the same policy [49].

Successor liability doctrine is commonly taught in a course on corporate law or business associations in law school. Sometimes it is reserved for upper level courses in corporate finance or mergers and acquisitions. Students are expected to spot successor liability issues and understand how to determine if a successor will be held liable.

**Task** The Successor Liability task requires an LLM to spot a successor liability issue and identify its relevant exception to no liability. If more than one exception is relevant, the LLM is required to state the additional exception(s). The task does not include identification of the two variations to the mere continuation exception (continuity of enterprise and product line).

| Facts | Issue | Relevant exception |
|---|---|---|
| Large Incarceration Services purchased a substantial amount of Small Prison's assets last year. The asset purchase agreement expressly disclaimed Small Prison's potential liability for an employment discrimination claim arising out of its prison service activities. Small conveyed its assets to Large because Small's owners were concerned that the liability arising from the lawsuit would lead to bankruptcy. Several months following the asset purchase, Small Prison lost the discrimination lawsuit. The plaintiffs now seek relief from Large Incarceration Services. | successor liability | fraudulent conveyance, mere continuation |
| Large Incarceration Services purchased a substantial amount of Small Prison's assets last year. The asset purchase agreement expressly assumed any liability arising out of its prison service activities. Several months following the asset purchase, Small Prison lost a number of discrimination lawsuits. The plaintiffs now seek relief from Large Incarceration Services. | successor liability | express agreement |
| Big Pharma purchases substantially all of DW I's assets. The purchase agreement expressly provides for assumption of only those liabilities necessary for continuing operations of DW I. DW I had developed a successful drug that regulated oxygen levels in the blood. After the purchase of DW I's assets, DW I dissolves. DW I's shareholders maintain ownership in Big Pharma equivalent to their ownership in DW I. In addition, there is some overlap between the two companies' management teams. Moreover, Big Pharma continues to employ seventy percent of DW I's workforce. Past users of DW I's drug bring a mass tort claim against Big Pharma alleging that DW I's drug incorrectly measured oxygen levels in the blood leading to harm. | successor liability | de facto merger, mere continuation |
| Big Pharma purchases substantially all of DW I's assets. The purchase agreement expressly provides for assumption of only those liabilities necessary for continuing operations of DW I. DW I had developed a successful drug that regulated oxygen levels in the blood. After the purchase of DW I's assets, DW I dissolves. DW I's shareholders do not own any stock in Big Pharma. However, there is some overlap between the two companies' management teams. In addition, Big Pharma continues to employ seventy percent of DW I's workforce. Past users of DW I's drug bring a mass tort claim against Big Pharma alleging that DW I's drug incorrectly measured oxygen levels in the blood leading to harm. | successor liability | mere continuation |

Table 48: Examples for successor liability.

### G.30 Supply Chain Disclosure Tasks

In LEGALBENCH, the Supply Chain Disclosure Tasks are denoted as `supply_chain_disclosure_*`.

**Background**   Corporations are frequently legally required to disclose information that may be relevant to investors, regulators, or members of the public. One example of this kind of disclosure requirement is laws that require corporations doing business in particular jurisdictions to provide detailed information on their supply chains, which is intended to ensure that the company's business practices are not supporting things like human trafficking or human rights violations. One example of these kind of disclosure requirements is the California Transparency in Supply Chains Act (CTSCA). The CTSCA applies to corporations that are a: "[1] retail seller and manufacturer [2] doing business in this state [of California] and [3] having annual worldwide gross receipts that exceed one hundred million dollars ($100,000,000)."[24] If a corporation meets these criteria, they are required to post information on five topics:

- **Verification**: "[A]t a minimum, disclose to what extent, if any, that the retail seller or manufacturer . . . [e]ngages in verification of product supply chains to evaluate and address risks of human trafficking and slavery. The disclosure shall specify if the verification was not conducted by a third party."[25]

- **Audits**: "[A]t a minimum, disclose to what extent, if any, that the retail seller or manufacturer . . . [c]onducts audits of suppliers to evaluate supplier compliance with company standards for trafficking and slavery in supply chains. The disclosure shall specify if the verification was not an independent, unannounced audit."[26]

- **Certification**: "[A]t a minimum, disclose to what extent, if any, that the retail seller or manufacturer . . . [r]equires direct suppliers to certify that materials incorporated into the product comply with the laws regarding slavery and human trafficking of the country or countries in which they are doing business."[27]

- **Accountability**: "[A]t a minimum, disclose to what extent, if any, that the retail seller or manufacturer . . . [m]aintains internal accountability standards and procedures for employees or contractors failing to meet company standards regarding slavery and trafficking."[28]

- **Training**: "[A]t a minimum, disclose to what extent, if any, that the retail seller or manufacturer . . . [p]rovides company employees and management, who have direct responsibility for supply chain management, training on human trafficking and slavery, particularly with respect to mitigating risks within the supply chains of products."[29]

In addition to requiring corporations that meet the specified criteria to post disclosures that provide this information, the California Attorney General's office has also posted a guide informing firms of "Best Practices" for what specific information to provide on each of these five topics.[30]

However, prior research has suggested that companies do not always post disclosures that cover each of these topics; and, even when they do, the disclosures are not always consistent with the recommended best practices [27].

**Construction process**   We constructed this task based on an existing dataset of supply chain disclosures. In the summer of 2015, we, with the help of research assistants, searched the websites of corporations that had previously been identified by an organization called "KnowTheChain" as being required to post supply chain disclosures to be compliant with the California Supply Chain Transparency Act. Through this process, we found disclosures for roughly 400 firms out of roughly 500 firms for which KnowTheChain suggested were required to post disclosures.

For each of these roughly 400 firms, we saved copies of their supply chain disclosures. We then had research assistants read the disclosures and code whether they included each of the five required topics for disclosure and, if so, whether the disclosures on those five topics were consistent with the best practices outlined by the California Attorney General's office.

We convert each of these 10 coded variables into a distinct binary classificationt task, producing 10 tasks. Table 49 lists each task, along with the precise question used to code the disclosure.

---

[24]*See* California Transparency in Supply Chains Act, CAL. CIV. CODE § 1714.43(a)(1) (West 2012).

[25]CAL. CIV. CODE § 1714.43(c)(1).

[26]CAL. CIV. CODE § 1714.43(c)(2).

[27]CAL. CIV. CODE § 1714.43(c)(3).

[28]CAL. CIV. CODE § 1714.43(c)(4).

[29]CAL. CIV. CODE § 1714.43(c)(5).

[30]CAL. DEP'T OF JUSTICE, THE CALIFORNIA TRANSPARENCY IN SUPPLY CHAINS ACT: A RESOURCE GUIDE (2015), https://oag.ca.gov/sites/all/files/agweb/pdfs/sb657/resource-guide.pdf.

**Significance and value** Corporate disclosure requirements are a commonly used regulatory tool, but evidence suggests that firms do not always fully comply with these disclosure requirements. The Supply Chain Disclosure task evaluates whether LLMs may be able to determine whether corporations are complying with those disclosure requirements. Because these disclosures are often formatted very differently, written in complex language, and may be designed to obfuscate relevant information, this task provides a useful measure of whether LLMs can parse the content covered in legal documents.

| Task | Question |
|---|---|
| disclosed_ verification | Does the statement disclose to what extent, if any, that the retail seller or manufacturer engages in verification of product supply chains to evaluate and address risks of human trafficking and slavery? If the company conducts verification], the disclosure shall specify if the verification was not conducted by a third party. |
| disclosed_audits | Does the statement disclose to what extent, if any, that the retail seller or manufacturer conducts audits of suppliers to evaluate supplier compliance with company standards for trafficking and slavery in supply chains? The disclosure shall specify if the verification was not an independent, unannounced audit. |
| disclosed_ certification | Does the statement disclose to what extent, if any, that the retail seller or manufacturer requires direct suppliers to certify that materials incorporated into the product comply with the laws regarding slavery and human trafficking of the country or countries in which they are doing business? |
| disclosed_ accountability | Does the statement disclose to what extent, if any, that the retail seller or manufacturer maintains internal accountability standards and procedures for employees or contractors failing to meet company standards regarding slavery and trafficking? |
| disclosed_ training | Does the statement disclose to what extent, if any, that the retail seller or manufacturer provides company employees and management, who have direct responsibility for supply chain management, training on human trafficking and slavery, particularly with respect to mitigating risks within the supply chains of products? |
| best_practice_ verification | Does the statement disclose whether the retail seller or manufacturer engages in verification and auditing as one practice, expresses that it may conduct an audit, or expressess that it is assessing supplier risks through a review of the US Dept. of Labor's List? |
| best_practice_ audits | Does the statement disclose whether the retail seller or manufacturer performs any type of audit, or reserves the right to audit? |
| best_practice_ certification | Does the statement disclose whether the retail seller or manufacturer requires direct suppliers to certify that they comply with labor and anti-trafficking laws? |
| best_practice_ accountability | Does the statement disclose whether the retail seller or manufacturer maintains internal compliance procedures on company standards regarding human trafficking and slavery? This includes any type of internal accountability mechanism. Requiring independently of the supply to comply with laws does not qualify or asking for documentary evidence of compliance does not count either. |
| best_practice_ training | Does the statement disclose whether the retail seller or manufacturer provides training to employees on human trafficking and slavery? Broad policies such as ongoing dialogue on mitigating risks of human trafficking and slavery or increasing managers and purchasers knowledge about health, safety and labor practices qualify as training. Providing training to contractors who failed to comply with human trafficking laws counts as training. |

Table 49: Supply Chain Disclosure Tasks

## G.31 Telemarketing Sales Rule

In LEGALBENCH, the Telemarketing Sales Rule task is denoted as `telemarketing_sales_rule`.

**Background**   The Telemarketing Sales Rule (16 C.F.R. Part 310) is a set of regulations promulgated by the Federal Trade Commission to implement the Telemarketing and Consumer Fraud and Abuse Prevention Act. Its purpose is to protect consumers from specified deceptive and abusive telemarketing practices. This task focuses on 16 C.F.R. § 310.3(a)(1) and 16 C.F.R. § 310.3(a)(2), which outline a series of specific telemarketing practices prohibited as "deceptive." 16 C.F.R. § 310.3(a)(1) lists information that must be disclosed to a consumer before a sale is made, and 16 C.F.R. § 310.3(a)(2) lists categories of information that a telemarketer is prohibited from misrepresenting. 16 C.F.R. § 310.2 provides definitions relevant to both of these subsections.

The Telemarketing Sales Rule (TSR) is not commonly taught in law school as its own topic, but may be used as examples in courses on consumer protection law, administrative law, telecommunications law, and the like. Because of its simplicity, it has also been used in beginner-level legal research exercises tasking students with finding the TSR in the Code of Federal Regulations and applying it to a set of facts.

Applying the TSR would require an LLM to classify a set of facts as either falling within or outside of the specific prohibitions outlined in the rule. For example, the TSR requires that telemarketers disclose certain material information before a sale is made, such as the total cost of the goods or services, the quantities of goods or services being purchased, and exchange and return restrictions. It also forbids telemarketers from making material misrepresentations as to cost, quantity, quality, endorsement or sponsorship, and the like. In many real-life situations, it would be ambiguous whether certain telemarketer behavior would violate the TSR; for example, it could be contentious whether a given misrepresentation fits the definition of "material." However, this task is limited to clear, unambiguous violations or non-violations, such as if a telemarketer told a consumer that they were selling four apples for four dollars, when in fact they were selling four apples for six dollars.

The following subsections 16 C.F.R. § 310.3(a)(1) and 16 C.F.R. § 310.3(a)(2) were ignored in the task, given their complexity or their reference to other statutes and regulations:

- 16 C.F.R. § 310.3(a)(2)(vi)
- 16 C.F.R. § 310.3(a)(2)(viii)

**Task**   The TSR task is meant to test whether an LLM can classify simple sets of facts as describing a violation of the TSR, or not describing a violation of the TSR.

**Construction process**   We manually created 50 samples, such that examples of at least one violation and at least one non-violation of each relevant subsection of 16 C.F.R. § 310.3(a)(1) and 16 C.F.R. § 310.3(a)(2) were present.

**Significance and value**   Determining whether a simple and unambiguous set of facts falls within the ambit of 16 C.F.R. § 310.3(a)(1) or 16 C.F.R. § 310.3(a)(2) would be an easy task for law students and lawyers, as well as many non-lawyers. However, an LLM that was trained to recognize clear violations of consumer protection laws could help administrative agencies like the Federal Trade Commission inform normal citizens of their rights.

| Input | Answer |
|---|---|
| Acme Toys is a telemarketer subject to the Telemarketing Sales Rule. Acme Toys sold a customer a frisbee. It disclosed the brand of the frisbee, but did not tell the customer the frisbee was manufactured in Portugal. Is this a violation of the Telemarketing Sales Rule? | No |
| Acme Toys is a telemarketer subject to the Telemarketing Sales Rule. Acme Toys told a customer that it would sell them a handful of frisbees at a very reasonable price, and that shipping would be $5. Then, the customer agreed to the sale. Is this a violation of the Telemarketing Sales Rule? | Yes |

Table 50: Examples for `telemarketing_sales_rule`

## G.32 Textualism Tasks

In LEGALBENCH, the Textualism tasks are denoted as `textualism_tool_*`.

**Background**   Courts regularly interpret statutes to determine the precise meaning of words contained in the statute. For instance, suppose a statute specifies that "It shall be illegal to park a vehicle inside public parks for longer than thirty minutes." A court may be asked to determine whether the statute prohibits persons from parking bicycles inside public parks. This requires defining the term "vehicle" and determining if a bicycle is a type of vehicle.

To guide the interpretation of ambiguous statutory terms, American jurisprudence has developed numerous *principles* of statutory construction or interpretation. These principles—also referred to as tools or canons—are rules which dictate how terms in statutes should be interpreted. For instance, the principle of *ejusdem generis* states that where general words or phrases follow a number of specific words or phrases, the general words are specifically construed as limited and apply only to persons or things of the same kind or class as those expressly mentioned [130].

One approach to statutory interpretation—known as *textualism*—states that only the text of the statute should be considered [131]. In contrast, other approaches to interpreting an ambiguous term might call for a court to analyze the purpose of the statute, the history of the statute, or the intent of the legislature.

**Task**   The Textualism tasks ask a LLM to determine if an excerpt of judicial text is applying a specific textual tool when performing statutory interpretation. There are two tasks: dictionaries (textualism_tool_dictionaries) and .

- The first task is plain-meaning (textualism_tool_plain), and it requires an LLM to determine if a court is applying the "plain meaning" rule. The plain meaning rule says that statutory text should be interpreted according to its plain or ordinary meaning.

- The second task is dictionaries (textualism_tool_dictionaries), and it requires an LLM to determine if a court is using dictionaries to define the statutory text.

**Construction process**   For each task we extracted paragraphs from Court of Appeals opinions and manually annotated whether the paragraphs showed the court as "using" the respective tool.

- In order to count as using plain meaning, the paragraph must reference the plain or ordinary meaning of the text. This includes directly saying "plain meaning" or referencing the general logic of the plain meaning rule. There must also be evidence that the court used the tool in its decision. This latter condition is notable because legal scholars often care about whether the court actually used the tool when defending its decision. Common examples of using include stating it as a general rule of decision ("[O]ur obligation is to look to the plain language of the statute to effectuate the intent of congress") or applying it to the facts ("The statute's plain language indicates the 150% fee cap applies if (1) the plaintiff was "a prisoner" at the time he brought the action and (2) he was awarded attorney's fees pursuant to § 1988."). "Using" does not, for example, include paragraphs that criticize the use of the plain meaning rule.

- In order to count as using dictionaries, the paragraph must reference a dictionary. There must also be evidence that the court used a dictionary as part of its rationale. This latter condition is notable because legal scholars often care about whether the court actually used the tool when defending its decision. Common examples of using include stating it as a general rule of decision ("[We use a dictionary to help determine the plain meaning of the statutory text") or applying it to the facts ("According to the Websters dictionary, a vehicle is any *means in or by which someone travels, or something is carried or conveyed*"). "Using" does not, for example, include paragraphs that criticize the use of dictionaries.

**Significance and value**   Recognizing when a court is applying a particular canon of interpretation is a classical skill law students are expected to learn. LLM performance on this task thus offers a heuristic for comparing LLM comprehension of judicial text to that of a law student's. More practically, the capacity for LLMs to detect when certain canons are being applied could make them a valuable tool for empirical legal scholars.

| Input | Answer |
|---|---|
| overcome our prior interpretation of a statute depends, in turn, on whether we regarded the statute as unambiguously compelling our interpretation. | No |
| the statutory waiver is express, and its range is defined in unmistakable language. to say that a private person, but not the united states, is liable under title vii for interest as an element of an attorneys fee would rob the unambiguous statutory language of its plain meaning. it would defeat the statutory imposition upon the united states of a liability for costs, and the statutory inclusion of a reasonable attorneys fee as part of the costs, identical to that of a private party in similar circumstances. the scope-setting statutory words the same as a private person mark out the united states liability for attorneys fees as well as costs in the traditional sense. our responsibility as judges is to enforce this provision according to its terms. | Yes |

Table 51: Examples for `textualism_tool_plain`.

| Input | Answer |
|---|---|
| we pause to note that even if congress sought, through the csra, to regulate the nonuse of interstate channels, it would still be within its constitutional command to do so. the supreme court has often held, in several contexts, that the defendants nonuse of interstate channels alone does not shield him from federal purview under the commerce clause. in heart of atlanta motel, inc. v. united states, 379 u.s. 241, 250, 85 s.ct. 348, 353, 13 l.ed.2d 258 (1964), the court upheld commerce clause jurisdiction over a local motel that failed to engage in interstate commerce when it refused to rent rooms to black guests. the court held that by failing to rent the rooms, the hotel inhibited black travelers from crossing state lines and thus obstructed interstate commerce that otherwise would have occurred. id. at 253, 85 s.ct. at 356. in standard oil co. v. united states, 221 u.s. 1, 68, 31 s.ct. 502, 518, 55 l.ed. 619 (1911), the court upheld the sherman act, 15 u.s.c. 1, 2, as permissible congressional action under the commerce clause. the sherman act prohibits restraints of trade and obstructions of interstate commerce in order to facilitate commerce that otherwise would occur absent the defendants monopolistic behavior. finally, in united states v. green, 350 u.s. 415, 420, 76 s.ct. 522, 525, 100 l.ed. 494 (1956), the court found constitutional the hobbs act, 18 u.s.c. 1951, which punishes interference with interstate commerce by extortion, robbery or physical violence [by] ... outlaw[ing] such interference in any way or degree. to accept baileys nonuse argument would mean, as emphasized by the second circuit, that congress would have no power to prohibit a monopoly so complete as to thwart all other interstate commerce in a line of trade[;] or to punish under the hobbs act someone who successfully prevented interstate trade by extortion and murder. sage, 92 f.3d at 105. | No |
| our primary area of concern with the district courts determination is its confident assertion that the language of 326(a) is unambiguous. see lan assocs., 237 b.r. at 56-57. in this day and age when we exchange by a keystroke or series of keystrokes what we used to handle only in cash, we do not think that the term moneys is so clear as the district court indicated. in fact, one of the definitions cited by the district court refers to money as a measure of value, see id. at 55-56 (citing websters third new intl dictionary 1458 (1986)), which surely is a concept that evolves along with and is dependent upon changing cultural, social, and economic practices and institutions. for example, in todays society the term money could easily encompass the concept of credit, which increasing numbers of people use as a method of payment. the term money might also encompass property, especially when property is used as a method of payment or a measure of wealth. see websters ii new college dictionary 707 (defining money as [a] medium that can be exchanged for goods and services and is used as a measure of their values on the market and as [p]roperty and assets considered in terms of monetary value); supra note 5 (describing the nabts argument that an exchange of property involves an exchange of value). but see in re brigantine beach hotel corp., 197 f.2d 296, 299 (3d cir.1952) (referring to precode statute governing receiver compensation and stating that [i]t is clear that the word moneys in the clause ... upon all moneys disbursed or turned over ... is not the equivalent of property.). these reasonable interpretations of the term moneys render it ambiguous for purposes of our interpretation of 326(a). see taylor v. continental group change in control severance pay plan, 933 f.2d 1227, 1232 (3d cir.1991) (a term is ambiguous if it is subject to reasonable alternative interpretations.); accord united states v. gibbens, 25 f.3d 28, 34 (1st cir.1994) (a statute is ambiguous if it reasonably can be read in more than one way.). | Yes |

Table 52: Examples for `textualism_tool_dictionaries`

## G.33 UCC vs Common Law

In LEGALBENCH, the UCC vs Common Law task is denoted as `ucc_v_common_law`.

**Background**   In the United States, contracts are typically governed by one of two different bodies of law depending on the subject matter of the contract. Contracts for the sale of goods (physical, moveable things) are governed by the Uniform Commercial Code (UCC), a uniform set of laws created by the Uniform Law Commission and adopted in all US jurisdictions. Contracts for services and real estate, on the other hand, are governed by state common law. For example, a contract for Alice to sell Bob her bike would be governed by the UCC (sale of a good), but a contract for Bob to repair Alice's bike would be governed by the common law (service).

This distinction is significant because the UCC and the common law diverge on numerous important legal issues such as:

- Offer and acceptance: The common law requires an offeree's acceptance to exactly match the terms of the offeror's offer in order for a contract to be formed (the "mirror image" rule). The UCC, on the other hand, allows for some variation in the terms under UCC Section 2-207.

- Definiteness: For a common law contract to be enforceable, it must be reasonably definite with respect to all material terms. For example, a service contract would not be enforceable without a price term or an adequate description of the service to be provided. The UCC only requires that a goods contract include the good being sold and the quantity. If any other term is missing from the contract (such as price or delivery), it will be filled in by UCC default rules.

- Options: To create an option contract (by which the offeror provides the offeree with a defined period of irrevocability), the common law requires that the offeree give the offeror separate consideration for the option. The UCC allows merchants to make "firm offers" (effectively option contracts) without the offeree providing separate consideration.

- Modification: To modify an existing contract, the common law requires both parties to provide new consideration (the "preexisting duty rule") whereas the UCC only requires that modifications be made in good faith.

**Task**   The UCC vs. Common Law task requires an LLM to determine whether a contract is governed by the UCC or by the common law.

**Construction process**   The dataset was manually created to test an LLM's ability to determine whether a contract is governed by the UCC or by the common law. The dataset is composed of 100 descriptions of simple contracts such as "Alice and Bob enter into a contract for Alice to sell her bike to Bob for $50" (UCC) and "Aria pays Owen $100 to mount a television on the wall of her living room" (common law). Each description is followed the question, "Is this contract governed by the UCC or the common law?"

The dataset does not include "mixed purpose" contracts which incorporate both the sale of a good and a service. For example, a contract in which Alice sells Bob her bike for $100 and agrees to inflate the tires each week for the first month would be a mixed purpose contract. To determine whether a mixed purpose contract is governed by the UCC or the common law, most jurisdictions apply the "predominant purpose" test under which the predominant purpose of the contract (good or service) determines which body of law applies.

**Significance and value**   The UCC vs. Common Law task is significant for a number of reasons. First, it provides a measure of an LLM's legal reasoning ability relative to a human lawyer (who would almost certainly score a 100% on the task). Second, it demonstrates an LLM's ability to determine the subject matter of a legal text, which has implications for the use of LLMs for legal tasks far beyond contract classification. Third, this task could prove useful in the context of contract lifecycle management (CLM) in which a CLM software product could automatically sort contracts by subject matter for review purposes. Fourth, while the sample contracts in the dataset were simple and easily identifiable as either UCC or common law contracts, real-world mixed purpose contracts can be difficult to classify and sometimes generate costly litigation. This task could be used as a starting point for developing a more fine-tuned task that can classify mixed purpose contracts.

# H   Extended empirical study

We now present a fuller discussion of LEGALBENCH experiments. Specifically, we use LEGALBENCH to conduct a three-part study.

- In the first part (Section H.2), we conduct a sweeping evaluation of 20 LLMs from 11 different families, at four different size points. We use this study to make initial observations on performance differences across families, the role of model size, and the gap between open-source and commercial LLMs.

- In the second part (Section H.3), we show how LEGALBENCH can be used to conduct in-depth evaluations of models. To illustrate, we use LEGALBENCH to highlight similarities and differences in the performance of three popular commercial models: GPT-4, GPT-3.5, and Claude-1.

- In the final part (Section H.4), we show how LEGALBENCH can support the development of law-specific LLM methods. We focus on prompting, and conduct a series of experiments that begin to surface tradeoffs and challenges with regards to guiding LLMs towards certain tasks.

Ultimately, our study here serves to illustrate the types of analyses that LEGALBENCH enables, and highlight potential directions for future work.

## H.1   Setup

### H.1.1   Models

**Commercial models**   We study three commercial API-access models. From the OpenAI GPT family, we study GPT-3.5 [14] (text-davinci-003) and GPT-4 [96]. Results from these models were retrieved between May and August of 2023. From the Anthropic family, we study Claude-1 (v1.3) [3]. Results from this model were retrieved in July of 2023. These models are believed to be large (hundreds of billions of parameters), though exact details on their architecture and training process are unknown. It is thus possible that some LEGALBENCH tasks leaked into pretraining data.

**Open-source models**   We study 17 open-source models at three different size points: 3B parameters, 7B parameters, and 13B parameters. All inference was performed on two-GPU GCP 40GB A100s, using the Manifest library [97].

- From Together, we study three models: Incite-Instruct-7B, Incite-Base-7B, and Incite-Instruct-3B [34, 133].
- From Meta's OPT family, we study three models: OPT-2.7B, OPT-6.7B, and OPT-13B [154].
- From TII's Falcon family, we study Falcon-7B-Instruct [2, 103].
- From MosaicML's MPT family, we study MPT-7B-8k-Instruct [129].
- From LMSYS' Vicuna family, we study Vicuna-7B-16k and Vicuna-13B-16k [26].
- From Google's FLAN-T5 family, we study Flan-T5-XL (3B parameters) and Flan-T5-XXL (11B parameters) [31].
- From Meta's LLama-2 family, we study LLaMA-2-7B, and LLaMA-2-13B [134].
- From the Wizard family, we study WizardLM-13B [150].
- From the BigScience BLOOM family, we study BLOOM-3b and BLOOM-7B [116].

HuggingFace links for the studied open-source models can be found below.

**Future work**   Our selected LLMs represent only a sample of the models available. For instance, we do not evaluate LLMs larger than 13B parameters, which have been observed to perform well [9]. Studied LLMs are also "general domain," in that we don't find evidence that any were specifically customized to perform well on legal text.[31] In future work we hope to expand our evaluation to a broader set of LLMs.

### H.1.2   Prompts

We designed a prompt for each task by manually writing instructions for the task, and selecting between zero and eight samples from the available train split to use as in-context demonstration. The number of samples selected depended on the availability of data and the sequence length of samples. For instance, the inputs to the Supply Chain Disclosure tasks are disclosure statements between 1-2 pages long, making the inclusion of

---

[31]We note that as of July 2023, we were unable to identify public law-specific English large language models to evaluate.

| LLM | HuggingFace URL |
|---|---|
| Incite-Instruct-3B | togethercomputer/RedPajama-INCITE-Instruct-3B-v1 |
| Incite-Base-7B | togethercomputer/RedPajama-INCITE-Base-7B-v0.1 |
| Incite-Instruct-7B | togethercomputer/RedPajama-INCITE-Instruct-7B-v0.1 |
| BLOOM-3B | bigscience/bloom-3b |
| BLOOM-7B | bigscience/bloom-7b1 |
| OPT-2.7B | facebook/opt-2.7b |
| OPT-6.7B | facebook/opt-6.7b |
| OPT-13B | facebook/opt-13b |
| Falcon-7B-Instruct | tiiuae/falcon-7b-instruct |
| MPT-7B-8k-Instruct | mosaicml/mpt-7b-instruct |
| Vicuna-7B-16k | lmsys/vicuna-7b-v1.5-16k |
| Vicuna-13B-16k | lmsys/vicuna-13b-v1.5-16k |
| Flan-T5-XL | google/flan-t5-xl |
| Flan-T5-XXL | google/flan-t5-xxl |
| LLaMA-2-7B | meta-llama/LLaMA-2-7b-hf |
| LLaMA-2-13B | meta-llama/LLaMA-2-13b-hf |
| WizardLM-13B | WizardLM/WizardLM-13B-V1.2 |

Table 53: HuggingFace links for open-source models.

multiple demonstrations infeasible. For application evaluation, we augmented the prompt with an instruction for the LLM to explain its reasoning.

We used the same prompts across all LLMs with one exception. In contrast to the OpenAI and open-source LLMs, Anthropic recommends specific prompting formats when using Claude.[32] This includes surrounding in-context samples with `<example>`/`<example>` tags, and adding instructions specifying the output space. We observed that failing to adhere to these guidelines led Claude to generate text which made extracting a prediction challenging. Therefore, when prompting Claude, we added example-tags to the in-context demonstrations and instructions specifying the prediction space (e.g., "Reply with either: generic, descriptive, suggestive, arbitrary, fanciful").

LLM outputs were generated using next-token generation at a temperature of 0.0. For classification/extraction tasks, we terminated at a new-line token. For `rule_qa` and all application tasks except `diversity_jurisdiction_6` we generated 150 tokens. For `diversity_jurisdiction_6` we generated 300 tokens.

We believe there is significant scope for improving and refining prompts on LEGALBENCH. Hence, our results here provide a lower-bound on performance, as better prompts may elicit higher scores. Our prompts correspond to what we believe would be reasonable, based on experience with prompt engineering in other settings, and the guidance provided by model developers. We make all prompts available as a starting point for future work on LEGALBENCH.

Prompts for all LEGALBENCH experiments are available on the Github repository. Table 54 provides the number of in-context demonstrations used.

### H.1.3 Evaluation

Classification tasks are evaluated using "exact-match" (following HELM [78]). Because some tasks contain significant label imbalances, we use balanced-accuracy as a metric. For extraction tasks, we perform normalization on generated outputs to account for differences in tense/casing/punctuation. A few tasks (e.g., `successor_liability` and `ssla_individual_defendants`) requires the LLM to produce multiple classes or extracted terms per instance. For these, we evaluate using F1.

---

[32]https://docs.anthropic.com/claude/docs/introduction-to-prompt-design

| Number of in-context demonstrations | Tasks |
|---|---|
| 0 | Canada Tax Court Outcomes, Consumer Contracts QA, Corporate Lobbying, International Citizenship Questions, Rule QA, Supply Chain Disclosure Tasks, SARA (Numeric) |
| 1 | MAUD Tasks, SCALR |
| 2 | Citation Prediction Tasks |
| 3 | Securities Complaint Extraction Tasks |
| 4 | Legal Reasoning Causality, Personal Jurisdiction, Successor Liability, SARA (Entailment) Telemarketing Sales Rule, Textualism Tools |
| 5 | Abercrombie, Hearsay, Insurance Policy Interpretation, Private Right of Action |
| 6 | CUAD Tasks, Diversity Tasks, J.Crew Blocker, Learned Hands Tasks, Overruling, UCC v. Common Law |
| 7 | Function of Decision Section, Oral Argument Question Purpose |
| 8 | Contract NLI Tasks, Contract QA, Definition Tasks, New York State Judicial Ethics, OPP-115 Tasks, Privacy Policy Entailment, Privacy Policy QA |
| 9 | Unfair Terms of Service |

Table 54: Number of in-context demonstrations used for each type.

Rule-application tasks were evaluated manually by a law-trained individual, who analyzed LLM responses for both correctness and analysis.[33] This type of manual evaluation is consistent with previous works evaluating LLM generations in the legal domain [29, 69]. As rule-application requires LLMs to generate "explanations" detailing legal reasoning—a capability primarily exhibited by larger models—we only evaluated GPT-4, GPT-3.5, and Claude-1. `rule_qa` was also manually evaluated by a law-trained individual. All manual evaluation was performed with reference to a grading guide, which we additionally make available.

### H.2    Performance trends

Table 55 provides the average task performance for all 20 models in five reasoning categories (issue-spotting, rule-recall, rule-conclusion, interpretation, and rhetorical-understanding). The first block of rows corresponds to large commercial models, the second block corresponds to models in the 11B-13B range, the third block corresponds to models in the 6B-7B range, and the final block corresponds to models in the 2B-3B range. Table 56 provides the average task performance for the three large models on rule-application.

Overall, we find significant variation in performance across tasks, suggesting that LEGALBENCH captures a diverse spectrum of difficulty. These results emphasize that assessments of LLM capabilities for legal applications must be made on a task-by-task basis, and informed by the nuances of specific tasks. While certain types of tasks appear beyond the scope of current-day LLMs, others seem more within reach. In this section, we offer preliminary observations on performance trends across model size, family, and reasoning categories.

**Parameter count**    Within LLM families, we observe that larger models usually outperform smaller models. For instance, Flan-T5-XXL (11B parameters) outperforms Flan-T5-XL (3B parameters) on average across all five reasoning categories, and LLaMA-2-13B outperforms LLaMA-2-7B on average across four reasoning categories. Notably, the margin of the gap varies across LLM families and reasoning categories. For instance, on rule-recall, the 7B Incite-Instruct model outperforms the 3B Incite-Instruct model by almost 10pts, while the 6.7B OPT model outperforms the 2.7B OPT model by less than 1pt. We additionally note that the largest LLM (GPT-4) outperforms virtually all other models.

---

[33]For the six diversity jurisdiction tasks, we sampled 30 instances from each task. For all other rule-application tasks, we manually evaluated the entirety of the dataset.

| LLM | Issue | Rule | Conclusion | Interpretation | Rhetorical |
|---|---|---|---|---|---|
| GPT-4 | 82.9 | 59.2 | 89.9 | 75.2 | 79.4 |
| GPT-3.5 | 60.9 | 46.3 | 78.0 | 72.6 | 66.7 |
| Claude-1 | 58.1 | 57.7 | 79.5 | 67.4 | 68.9 |
| Flan-T5-XXL | 66.0 | 36.0 | 63.3 | 64.4 | 70.7 |
| LLaMA-2-13B | 50.2 | 37.7 | 59.3 | 50.9 | 54.9 |
| OPT-13B | 52.9 | 28.4 | 45.0 | 45.1 | 43.2 |
| Vicuna-13B-16k | 34.3 | 29.4 | 34.9 | 40.0 | 30.1 |
| WizardLM-13B | 24.1 | 38.0 | 62.6 | 50.9 | 59.8 |
| BLOOM-7B | 50.6 | 24.1 | 47.2 | 42.8 | 40.7 |
| Falcon-7B-Instruct | 51.3 | 25.0 | 52.9 | 46.3 | 44.2 |
| Incite-7B-Base | 50.1 | 36.2 | 47.0 | 46.6 | 40.9 |
| Incite-7B-Instruct | 54.9 | 35.6 | 52.9 | 54.5 | 45.1 |
| LLaMA-2-7B | 50.2 | 33.7 | 55.9 | 47.7 | 47.7 |
| MPT-7B-8k-Instruct | 54.3 | 25.9 | 48.9 | 42.1 | 44.3 |
| OPT-6.7B | 52.4 | 23.1 | 46.3 | 48.9 | 42.2 |
| Vicuna-7B-16k | 3.9 | 14.0 | 35.6 | 28.1 | 14.0 |
| BLOOM-3B | 47.4 | 20.6 | 45.0 | 45.0 | 36.4 |
| Flan-T5-XL | 56.8 | 31.7 | 52.1 | 51.4 | 67.4 |
| Incite-3B-Instruct | 51.1 | 26.9 | 47.4 | 49.6 | 40.2 |
| OPT-2.7B | 53.7 | 22.2 | 46.0 | 44.4 | 39.8 |

Table 55: Average performance for each LLM over the different LEGALBENCH categories. The first block of rows corresponds to large commercial models, the second block corresponds to models in the 11B-13B range, the third block corresponds to models in the 6B-7B range, and the final block corresponds to models in the 2B-3B range. The columns correspond to (in order): issue-spotting, rule-recall, rule-conclusion, interpretation, and rhetorical-understanding. For each class of models (large, 13B, 7B, and 3B), the best performing model in each category of reasoning is underlined.

| LLM | Correctness | Analysis |
|---|---|---|
| GPT-4 | 82.2 | 79.7 |
| GPT-3.5 | 58.5 | 44.2 |
| Claude-v1 | 61.4 | 59.0 |

Table 56: Average performance for the large LLMs on rule-application tasks.

**Variation across families**   Even for LLMs of the same size, we find considerable differences in performance. For instance, we observe significant gaps in performance between Flan-T5-XXL (11B parameters) and Vicuna-13B-16k (13B parameters), across all reasoning categories. This suggests, unsurprisingly, that the choice of pretraining data, regime of instruction-tuning, and architecture play an important role in determining performance, and that certain configurations may be better aligned for LEGALBENCH tasks. Interestingly, we observe that such choices may affect which types of reasoning categories LLMs appear to perform well at. For instance, we observe that WizardLM-13B performs worse than all peers on issue-spotting tasks, best on rule-recall tasks, and nearly matches the performance of the best-performing peer on rule-conclusion tasks. Comparing Incite-7B-Instruct to Incite-7B-Base also provides insight into the effect of instruction-tuning across different categories, at one size point (7B parameters). We observe that instruction-tuning improves performance on four categories (issue-spotting, rule-conclusion, interpretation, and rhetorical-understanding), and worsens performance on rule-recall.

We additionally find that family-specific trends appear to hold across different size points. For instance, the Flan-T5 models outperform all others at both the 3B and 13B scale, while the Vicuna models appear to underperform competitors at both the 7B and 13B scale. We attribute the Vicuna models' low performance to their frequency tendency to generate poorly-formed outputs, which did not map to the expected verbalizer tokens (e.g., blank spaces, random characters, etc.).[34] This could possibly be attributed to the type of data used to fine the model (e.g., user-conversation), although more in-depth experimentation is necessary.

---

[34]In further experimentation, we found that writing prompts using the "### Human:" and "'### Assistant:'" templates did not appear to help.

**The gap between open-source and commercial models** Finally, we find evidence that open-source models are capable of performance that matches or exceeds certain commercial models. For instance, Flan-T5-XXL outperforms GPT-3.5 and Claude-1 on two categories (issue-spotting and rhetorical-understanding), despite the relative gap in parameter count. Notably, the gap between closed and open-source models is largest for the rule-conclusion category. Amongst LEGALBENCH tasks, rule-conclusion tasks most like the other types of multi-step/common-sense reasoning tasks where commercial LLMs have been found to perform well.

## H.3   Comparing GPT models

This section provides a more in-depth study of performance, focusing on the three commercial models (GPT-4, GPT-3.5, and Claude-1). The purpose of this section is to illustrate how LEGALBENCH enables fine-grained analysis of LLM performance. In particular we highlight how LEGALBENCH can provide more rigorous empirical support for anecdotal observations arising out of the legal community's use of these models, and explain performance differences between models.

### H.3.1   Issue-spotting

We first consider average model performance across all issue-spotting tasks. We observe that GPT-4 outperforms GPT-3.5 and Claude-1 (both at $p < 0.001$).[35] In absolute terms, issue tasks present the largest gap in performance between GPT-4 and other closed-API models, with an absolute margin of $20+$ points. GPT-3.5 and Claude-1, in contrast, appear to match each other in performance, separated by an average gap of only 2 points. We additionally find that the open-source models perform poorly here. On 9 tasks, Incite-Base collapses to predicting a single class for all samples.

We note one limitation to our results: because 16/17 of our issue-spotting tasks are drawn from one source (Learned Hands data), average issue performance is skewed by properties of the Learned Hands data distribution (i.e.,,, user-generated questions). For instance, though GPT-3.5 outperforms Claude-1 on 12/16 Learned Hands tasks, Claude-1 outperforms GPT-3.5 on the one non-Learned Hands task (`corporate_lobbying`). Despite the skew, we observe that these tasks appear to vary in difficulty. While GPT-4's balanced-accuracy on `learned_hands_torts` is only 70.6%, on three tasks—`learned_hands_immigration`, `learned_hands_traffic`, and `learned_hands_estate`—it scores $> 95\%$.

### H.3.2   Rule-recall

We first consider average model performance across all rule-recall tasks. While GPT-4 outperforms GPT-3.5 ($p < 0.05$), we surprisingly find that Claude-1 also outperforms GPT-3.5 ($p < 0.05$), and appears almost on par with GPT-4. Moreover, Claude-1 outperforms GPT-4 on three tasks: `rule_qa`, `international_citizenship_questions`, and `nys_judicial_ethics`. This is the only task category where Claude-1 provides performance comparable to GPT-4. Because little is known regarding the architecture and training processes for these models however, it is difficult to explain why this is the case.

Because rules/laws can be analogized to law-specific "facts," rule-recall tasks are similar to general domain LLM tasks designed to measure "hallucination." There, an extensive literature has documented the propensity for LLMs to both generate factually incorrect information, and answer fact-based questions incorrectly [78, 104]. Our results align with the primary findings of that literature. For example, we observe that the small open source models perform considerably worse than the larger models, consistent with the observation that model size plays an important role in fact-retention. Overall, performance on the rule-recall tasks also lend additional empirical support to more anecdotal reports—from the legal community—regarding how LLMs often misstate the law or cases [112].

### H.3.3   Rule-application

Application tasks evaluate whether LLMs can explain how a legal rule applies to a set of facts, and verbalize the necessary inferences. With respect to correctness, we observe that GPT-4 outperforms both GPT-3.5 ($p < 0.01$) and Claude-1 ($p < 0.01$). Across LLMs, we find that variation in performance across tasks is consistent with subjective impressions of task difficulty. For instance, performance on `diversity_jurisdiction_1` (an easy task requiring a model to determine if an amount is greater than \$75k and if the plaintiff and defendant are from different states) is much higher than performance on `successor_liability` (a harder task requiring a model to identify multiple successor liability exceptions in a fact pattern describing a complex transaction).

We observe that LLM generations may be incorrect in many different ways. On the Diversity Tasks, LLMs sometimes perform incorrect arithmetic operations or mathematical comparisons (i.e., stating that \$50,000 is greater than \$75,000). On `telemarketing_sales_rule` in contrast, LLMs will cite to an incorrect portion of the rule. For instance, a generation may explain that certain conduct by a telemarketer runs afoul of the rule

---

[35]Statistical significance is computed using a paired $t$-test over the tasks in the category.

because the telemarketer failed to make a mandatory disclosure (16 CFR § 310.3(a)(1)), but cite to the portion of the rule prohibiting misrepresentations (16 CFR § 310.3(a)(2)). Examples of other types of incorrect generations can be found in Table 57.

| Fact Pattern (Task) | LLM generation | Why is it incorrect? |
| --- | --- | --- |
| Sophia is from Nebraska. Ava is from Washington. Sophia sues Ava for legal malpractice for $34,000. (`diversity_1`) | "No. The parties are not diverse because Sophia and Ava are both from different states. The AiC is too small because it is less than 75k. Because the AiC requirement is not met there is no jurisdiction." **(GPT-3.5)** | While GPT-3.5 correctly identifies that there is no jurisdiction because the AiC is less than $75k (and thus too small), it fails to recognize that the parties (Sophia and Ava) are diverse because they are from different states. |
| The mark "Trim" for nail clippers. (`abercrombie`) | "The mark trim for nail clippers would be considered a suggestive mark. It suggests a characteristic or quality of the product which is that the nail clippers are used to trim ones nails. It implies this purpose and function without directly describing it so trim for nail clippers would be classified as a suggestive trademark." **(Claude-1)** | Claude-1's assertion that the mark "implies this purpose and function without directly describing it" misses the fact that "Trim" directly describes, rather than implies, what nail clippers are used for (i.e., the *trim* nails). Thus, the mark is actually descriptive, and not suggestive. |
| Ana is a lawyer who resides in Texas. While visiting Louisiana, she meets David, who runs a bike shop. She buys a bike from him and uses it to bike back to Texas. Right after she crosses the border, the bike seat explodes, injuring Ana. Ana sues David in Louisiana. (`personal_jurisdiction`) | "Yes. There is personal jurisdiction over David in Louisiana. Explanation. 1. David is domiciled in Louisiana so there is general jurisdiction over him in Louisiana. 2. David has sufficient minimum contacts with Louisiana such that he has availed himself of the privileges of conducting business in Louisiana. He operates a bike shop in Louisiana so he purposefully avails himself of doing business in the state.. 3. Ana's claim arises out of Davids contacts with Louisiana she bought the defective bike from David's bike shop in Louisiana so the claim is directly related to David's business contacts in the state. 4. Requiring David to litigate in Louisiana would not offend traditional notions of fair play." **(Claude-1)** | The LLM states that David is domiciled in Louisiana, when the facts make no mention of this. |

Table 57: Examples of generations for rule-application tasks which are *incorrect*. The last column provides analysis for why the generation is incorrect.

With respect to analysis, we observe that GPT-4 again outperforms both GPT-3.5 ($p < 0.01$) and Claude-1 ($p < 0.01$). Explanations which failed to exhibit analysis can be grouped into several categories. First, some generations will contain just a prediction as to the legal outcome, without an explanation (even when the LLM has been prompted to generate one). The same prompt–applied to other samples in the dataset–will elicit explanations containing analysis. Second, we observe a tendency for LLMs to sometimes generate explanations which merely restate the facts and legal rule, without actually offering an explanation for how the outcome is reached. Examples of such instances are provided in the table below.

### H.3.4 Rule-conclusion

Rule-conclusion evaluates on the same tasks as rule-application, but only requires the LLM to generate a prediction as to the outcome, and not an explanation. We observe that GPT-4 once again outperforms GPT-3.5 ($p < 0.01$) and Claude-1 ($p < 0.01$). Claude-1 and GPT-3.5 appear approximately level on performance.

The rule-conclusion tasks offer a heuristic for characterizing the types of legal inferences LLMs are capable of or struggle with. In particular, several of these tasks organize samples into *slices*, where the samples contained within a slice all represent a similar type of fact pattern, and thus interact with the legal rule in a comparable way. For instance, the `hearsay` task contains a slice corresponding to "non-verbal hearsay." This slice contains fact patterns where an individual communicates something non-verbally (e.g., pointing), thus qualifying their conduct as a "statement" under the hearsay rule. In order to make accurate predictions on this slice, an LLM

| Fact Pattern (task) | LLM generation | What analysis is missing? |
|---|---|---|
| Martha makes special pinwheel hats in her home in Salem, Oregon. She only markets her hats on the west coast (California, Oregon, and Washington). Alex, a citizen of Maine, purchases one of the hats while visiting Salem. When Alex returns to Maine, the pinwheel comes off and seriously cuts Alex. Alex sues Martha for damages in Washington. (`personal_jurisdiction`) | "No. There is not personal jurisdiction." **(Claude-1)** | Claude-1 does not provide any explanation for *why* there is no personal jurisdiction, so the explanation does not contain analysis. |
| On the issue of whether Henry negligently hit Kelly with his car, Kelly's statement, "a witness told me after the crash that Henry was on his cell phone while driving. (`hearsay`) | "Yes, there is hearsay in this situation. The reasoning is as follows:. . 1. The statement in question is an out-of-court statement, as it was made by a witness who is not testifying in court.. 2. The statement is being introduced to prove the truth of the matter asserted, which is that Henry was on his cell phone while driving.. 3. Since the statement is an out-of-court statement being introduced to prove the truth of the matter asserted, it is considered hearsay." **(GPT-4)** | GPT-4 does not explain how the truthfulness of the out-of-court statement (whether or not Henry was on his phone) relates to the purpose of its introduction (proving/disproving that Henry negligently drove). |

Table 58: Examples of generations for rule-application tasks which do not contain analysis. The last column explains why the generation is deficient.

must recognize that (1) the hearsay rule applies to non-verbal communicative conduct, and (2) the non-verbal conduct in these fact patterns is communicative.

Though slices are small—and thus not intended for rigorous statistical analysis—they provide some intuition as to the source of GPT-4's improvement over GPT-3.5, and the overall areas of strength and weakness for both models. On the `hearsay` task for instance (Table 59), the difference between GPT-4 and GPT-3.5 appears primarily attributable to improvements over the slices corresponding to non-verbal hearsay and statements made in court. In looking across slices moreover, it's clear that some are comfortably within the realm of model capabilities (e.g., non-assertive conduct), while others (e.g., not introduced to prove the truth of the matter asserted) still pose a considerable challenge.

Another example is provided by the `abercrombie` task, in which an LLM must determine the relationship between a product and a potential trademark name, by classifying the product-name pair into one of five categories recognized by courts: generic, descriptive, suggestive, arbitrary, and fanciful. Loosely, these categories measure how *distinctive* a product name is for a product, with generic being the least distinctive, and fanciful being the most distinctive. Just as with `hearsay`, comparing LLM performance on each of these categories provides insight into the relative areas of improvement (Table 60). Here, GPT-4's improved overall performance appears most attributable to performance on marks which are suggestive or arbitrary. However, GPT-4 still makes a number of errors for both categories. Interestingly, performance on descriptive marks is consistent between both models.

### H.3.5 Interpretation

On the interpretation tasks, we find that on average GPT-4 outperforms GPT-3.5 ($p < 0.01$), ans GPT-3.5 outperforms Claude-1 ($p < 0.01$). Here, the larger API-models are highly performant on tasks which involve binary classification over short clauses. Averaged across the 38 CUAD tasks (contract clauses), for instance, GPT-4, GPT-3.5, and Claude-1 all have a balanced-accuracy $\geq 88\%$. And on `proa` (statutory clauses), both GPT-4 and GPT-3.5 have a balanced-accuracy $\geq 90\%$. Notably, performance degrades on tasks which contain longer text sequences or involve multi-class classification. On the Supply Chain Disclosure tasks for instance—in which LLMs must classify disclosures which are 1-2 pages in length—the average balanced-accuracy of the large commercial models ranges between 74-75%. And on the MAUD tasks—which require answering multiple choice questions about merger deals—the average balanced-accuracy of GPT-4 drops to 47.8% accuracy.

| Slice | Slice description | GPT-3.5 | GPT-4 |
|---|---|---|---|
| Non-assertive conduct ($n = 19$) | The fact pattern describes conduct which is non-communicative and therefore not hearsay. | 100% | 100% |
| Statement made in-court ($n = 14$) | The fact pattern describes a statement that was made in court and therefore not hearsay. | 57% | 93% |
| Standard hearsay ($n = 29$) | The fact pattern describes traditional hearsay (out-of-court statement introduced to prove the truth of the matter asserted). | 97% | 97% |
| Non-verbal hearsay ($n = 12$) | The fact pattern describes non-verbal communicative conduct that qualifies as hearsay. | 33% | 75% |
| Not introduced to prove truth ($n = 20$) | The fact pattern describes a statement *not* introduced to prove the truth of the matter asserted, which is therefore not hearsay. | 25% | 45% |

Table 59: Comparison between GPT-3.5 and GPT-4 on `hearsay` slices. Accuracy is reported for each slice.

| Mark | Mark description | GPT-3.5 | GPT-4 |
|---|---|---|---|
| Generic ($n = 19$) | The name connotes the basic nature of the product/service. | 94% | 100% |
| Descriptive ($n = 19$) | The name identifies a characteristic or quality of the product/service. | 73% | 72% |
| Suggestive ($n = 20$) | The name suggests, rather than describes, a characteristic of the product/service. | 38% | 70% |
| Arbitrary ($n = 18$) | The name is a real world but has no relation to the product/service. | 41% | 82% |
| Fanciful ($n = 19$) | The name is a made-up word. | 84% | 100% |

Table 60: Comparison between GPT-3.5 and GPT-4 on `abercrombie` categories. Accuracy is reported for each slice.

### H.3.6   Rhetorical-analysis

On average across all rhetorical-understanding tasks, we find that GPT-4 outperforms both GPT-3.5 ($p \leq 0.05$) and Claude-1 ($p \leq 0.05$). We note several results. First, on `definition_extraction`—which requires a LLM to extract the term defined by a sentence taken from a Supreme Court opinion—Incite-Base almost equals GPT-4 in performance (80.6% accuracy to 81.8%). Second, nearly all evaluated models struggle on two tasks requiring LLMs to label the legal "roles" played by either a question or excerpt from an opinion (`function_of_decision_section` and `oral_argument_question_purpose`). Notably, both tasks require the LLM to classify text into one of six or more categories

### H.4   Prompt engineering strategies

Finally, we illustrate—through a series of micro-studies—how LEGALBENCH can be used to explore different aspects of prompt-engineering for LLMs in legal settings. We focus on three questions:

1. Can LLMs rely on their latent knowledge of a rule for rule-conclusion tasks?
2. Does simplifying task descriptions to plain language affect performance?
3. Are LLMs sensitive to the choice of in-context demonstrations?

**Reliance on latent knowledge**   When prompting for general-domain tasks like sentiment or topic classification, prompt-engineers will often rely on the LLM's latent knowledge of the task [5]. In topic classification for instance, a prompt may use the instructions to label whether a news article is about "sports," without offering

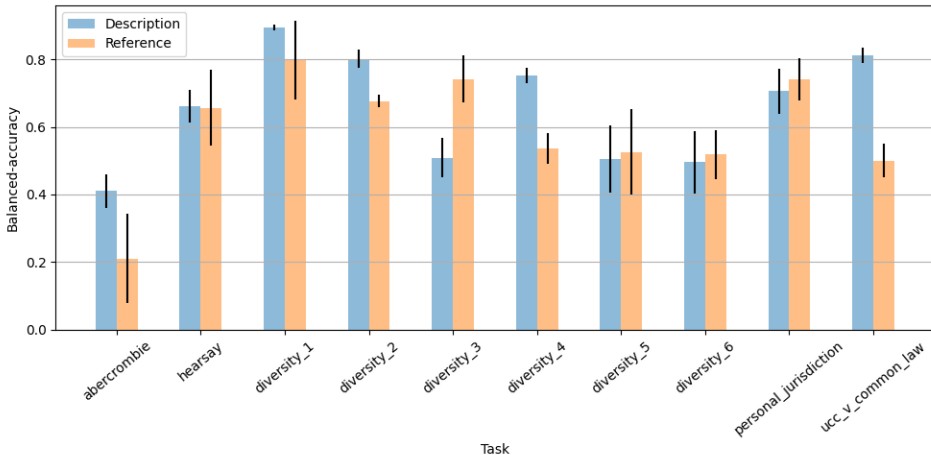

Figure 3: We compare performance of prompts which describe the legal rule to be applied ("description") against prompts which reference the legal rule to be applied ("reference"). Error bars measure standard error, computed using a bootstrap with 1000 resamples.

a detailed description of what "sports" refers to or encompasses. Such a description is not necessary, because general-domain terms like "sports" appear frequently in LLM training corpora, and LLMs can learn from these occurrences what general-domain terms mean. Prompting for legal tasks, however, may require a different strategy. Because legal terms occur less frequently in general domain training corpora, legal prompting may require practitioners to provide additional background information. For example, a general domain LLM may not know what the requirements for diversity jurisdiction are, because diversity jurisdiction is not as commonly discussed in pretraining corpora.

We explore this question through a study of rule-conclusion tasks. For a selection of these tasks, we evaluate GPT-3.5 with two zero-shot prompts: a reference-based prompt and a description-based prompt. In the reference prompt, the task instructions merely state the rule to be applied, i.e., "Determine if the following fact patterns give rise to diversity jurisdiction." In the description-based prompt, the instructions provide an explicit description of the rule, i.e., "Diversity jurisdiction exists when there is (1) complete diversity between plaintiffs and defendants, and (2) the amount-in-controversy (AiC) is greater than $75k." By comparing performance between the reference and description prompt, we can measure whether providing a description of the rule in the prompt provides additional performance boost over the LLM's latent knowledge of the rule.

Figure 3 provides a comparison for the different prompts. Interestingly, we find considerable variation across tasks. On tasks like `abercrombie`, `ucc_v_common_law`, `diversity_2`, and `diversity_4`, description prompts appear to offer significant increase in performance. On the other tasks, performance is approximately the same (or even worse). We identify two possible explanations for diverging results across tasks. First, on certain tasks, subsets of fact-patterns are too challenging for LLMs like GPT-3.5, and description-based prompts do not provide sufficient guidance for LLMs to reason through those fact patterns. Second, legal rules may be described to varying extents within pretraining corpora. Hence, tasks where we observe performance improvements from description-based prompting may correspond to rules which occur less frequently in pretraining data.

**Plain language descriptions of tasks**     Next, we examine the extent to which domain specialization in the language of the prompts affects performance. Like experts in other specialized domains, lawyers have developed their own language (i.e., "legalese"), which forms the basis for most legal writing and communication. It is unclear whether—in interacting with large language models through prompting—lawyers should continue to rely on formalistic legal language, or instead use simpler plain language. While most large language models are "general domain" and thus less specialized to legalese, formalistic legal language is more precise, and may thus induce more accurate behavior from the model.

We explore this question by comparing "plain language" and "technical language" prompts. For a subset of LEGALBENCH tasks, we have access to the formal language provided to law-trained annotators when creating task data. By comparing the performance of a prompt which uses this language—to one which uses a plain-language version—we can measure how the technicality of language affects results.

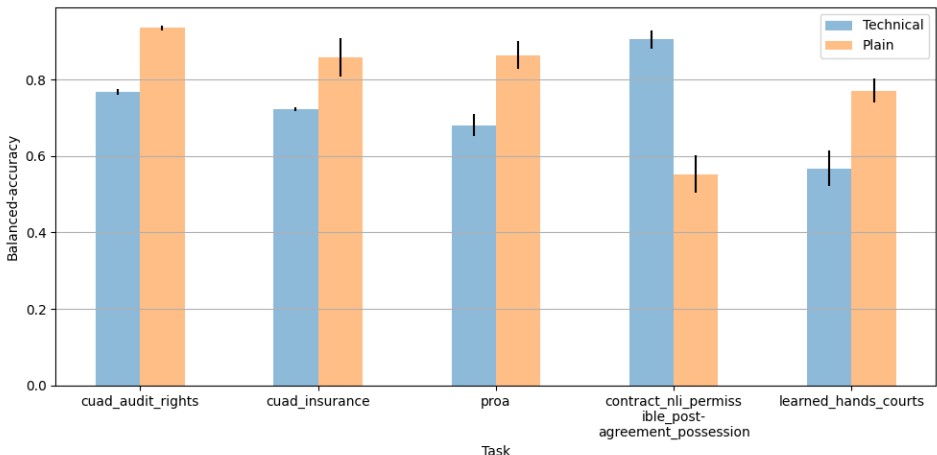

Figure 4: We compare performance of prompts which describe the task in plain language to prompts which describe the task in technical legal language (for GPT-3.5). Error bars measure standard error, computed using a bootstrap with 1000 resamples.

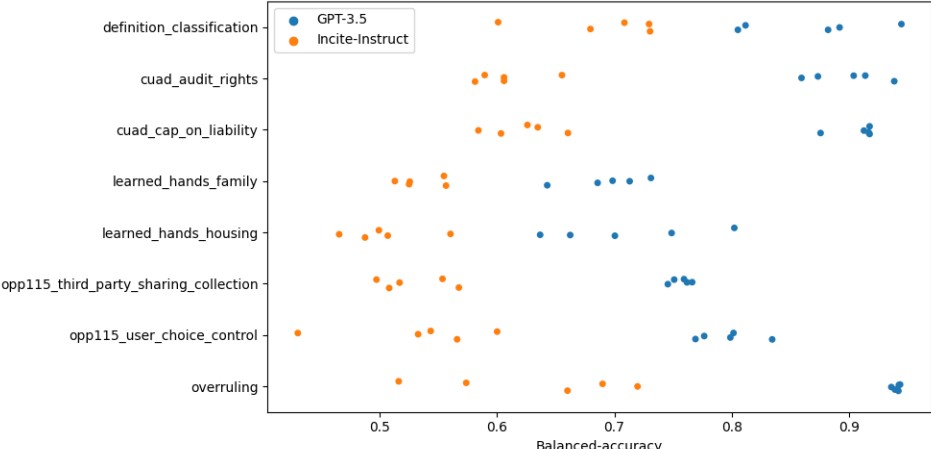

Figure 5: We evaluate GPT-3.5 and Incite-Instruct on five prompts constructed by randomly selecting different samples to use as in-context demonstrations (maintaining class balance in the prompt). In the figure above, each point corresponds to a different prompt.

We conduct preliminary experiments on a sample of five LEGALBENCH tasks (Figure 4).[36] On four of the five tasks, we find that the plain-language prompt significantly outperforms the technical language prompt, by up to 21 points (balanced-accuracy). Interestingly, on `contract_nli_permissible_post-agreement_possession`, we find the opposite phenomenon holds: the plain language prompt is substantially *worse* than the technical prompt.

**Sensitivity to in-context demonstrations**  Finally, we investigate the influence of the in-context demonstrations used in prompts. Prior work in general domain LLMs have observed that few-shot performance is highly sensitive to the choice of demonstrations [56, 123, 141]. We evaluate whether LLMs are similarly sensitive for legal tasks, focusing on a subset of 8 binary classification tasks. For each task we merge the train and evaluation split into a single dataset, and randomly sample four in-context samples to include in the prompt (two from each class), five different times. We evaluated GPT-3.5 and Incite-Instruct-7B with each of the five generated prompts, and plot the the balanced-accuracy of each prompt in Figure 5.

Consistent with findings on general-domain tasks, we observe that LLMs on legal tasks are also highly sensitive to the choice of in-context samples. Notably, this appears to be the case for both GPT-3.5 and Incite-Instruct.

---

[36]Prompts are made available in the LEGALBENCH repository.

Under a permutation test, we find significant differences ($p < 0.01$) between the best and worst performing prompt for Incite-Instruct (on all tasks), and for GPT-3.5 (on all tasks except `opp115_third_party_sharing_collection` and `overruling`).[37] For many tasks, the magnitude of difference is substantial. On `overruling` for instance, the best Incite-Instruct prompt improves upon the worst prompt by over 20 points (balanced-accuracy). Overall, these results suggest that future work is needed to understand how different demonstrations influence performance.

# I  All tables

We provide results for each LLM on each of the tasks. Models are divided into four groups based on type: commercial models, 13B models, 7B models, and 3B models. Model names are abbreviated to the family name to ensure well-formed tables.

| Task | GPT-4 | GPT-3.5 | Claude-1 |
|---|---|---|---|
| abercrombie | 84.2 / 84.2 | 48.4 / 48.4 | 65.2 / 63.1 |
| abercrombie | 84.2 / 84.2 | 48.4 / 48.4 | 65.2 / 63.1 |
| diversity_1 | 96.7 / 96.7 | 86.7 / 86.7 | 86.7 / 86.7 |
| diversity_2 | 100.0 / 100.0 | 53.3 / 50.0 | 73.3 / 73.3 |
| diversity_3 | 96.7 / 96.7 | 83.3 / 66.7 | 53.3 / 53.3 |
| diversity_4 | 93.3 / 93.3 | 70.0 / 66.7 | 43.3 / 43.3 |
| diversity_5 | 76.6 / 76.6 | 66.7 / 66.7 | 36.7 / 36.7 |
| diversity_6 | 80.0 / 80.0 | 6.7 / 6.7 | 53.3 / 53.3 |
| hearsay | 75.5 / 47.9 | 55.3 / 23.4 | 68.1 / 41.5 |
| personal_jurisdiction | 94.0 / 94.0 | 68.0 / 10.0 | 70.0 / 70.0 |
| successor_liability | 19.1 / 19.1 | 15.2 / 15.2 | 38.3 / 38.3 |
| telemarketing_sales_rule | 72.3 / 70.2 | 48.9 / 42.5 | 55.3 / 55.3 |
| ucc_v_common_law | 97.8 / 97.8 | 100.0 / 47.8 | 93.6 / 93.6 |

Table 61: Performance on rule-application tasks for commercial models. We report correctness/analysis.

| Task | GPT-4 | GPT-3.5 | Claude-1 |
|---|---|---|---|
| rule_qa | 72.0 | 46.0 | 77.0 |
| international_citizenship_questions | 59.3 | 52.7 | 60.0 |
| nys_judicial_ethics | 78.3 | 74.3 | 79.1 |
| citation_prediction_classification | 71.3 | 54.6 | 61.1 |
| citation_prediction_open | 15.1 | 3.8 | 11.3 |

Table 62: Commercial models on rule-recall tasks.

---

[37]We conduct the permutation test with 1000 resamples.

| Task | Flan | Llama-2 | OPT | Vicuna | WizardLM |
|---|---|---|---|---|---|
| rule_qa | 0.0 | 22.0 | 8.0 | 14.0 | 20.0 |
| international_citizenship_questions | 52.4 | 50.0 | 18.0 | 21.5 | 49.8 |
| nys_judicial_ethics | 69.1 | 67.6 | 63.3 | 61.3 | 63.8 |
| citation_prediction_classification | 56.5 | 47.2 | 52.8 | 50.0 | 52.8 |
| citation_prediction_open | 1.9 | 1.9 | 0.0 | 0.0 | 3.8 |

Table 63: 13B models on rule-recall tasks.

| Task | Bloom | Falcon | Incite-Base | Incite-Inst. | Llama-2 | MPT | OPT | Vicuna |
|---|---|---|---|---|---|---|---|---|
| rule_qa | 6.0 | 8.0 | 18.0 | 28.5 | 22.0 | 20.0 | 6.0 | 0.0 |
| international_citizenship_questions | 0.0 | 10.2 | 49.5 | 47.0 | 27.2 | 2.4 | 2.4 | 3.0 |
| nys_judicial_ethics | 62.5 | 55.8 | 62.7 | 54.9 | 67.5 | 57.2 | 59.7 | 52.3 |
| citation_prediction_classification | 50.0 | 50.9 | 50.9 | 47.2 | 50.0 | 50.0 | 47.2 | 14.8 |
| citation_prediction_open | 1.9 | 0.0 | 0.0 | 0.0 | 1.9 | 0.0 | 0.0 | 0.0 |

Table 64: 7B models on rule-recall tasks.

| Task | Bloom | Flan | Incite | Opt |
|---|---|---|---|---|
| rule_qa | 0.0 | 0.0 | 14.0 | 2.0 |
| international_citizenship_questions | 0.0 | 50.0 | 18.9 | 3.0 |
| nys_judicial_ethics | 54.0 | 58.5 | 49.9 | 56.7 |
| citation_prediction_classification | 49.1 | 50.0 | 51.9 | 49.1 |
| citation_prediction_open | 0.0 | 0.0 | 0.0 | 0.0 |

Table 65: 3B models on rule-recall tasks.

| Task | GPT-4 | GPT-3.5 | Claude-1 |
|---|---|---|---|
| corporate_lobbying | 81.7 | 59.1 | 75.8 |
| learned_hands_benefits | 87.9 | 62.1 | 66.7 |
| learned_hands_business | 81.6 | 58.6 | 47.7 |
| learned_hands_consumer | 76.2 | 59.3 | 58.1 |
| learned_hands_courts | 52.6 | 54.2 | 46.9 |
| learned_hands_crime | 81.0 | 62.4 | 59.0 |
| learned_hands_divorce | 84.0 | 59.3 | 53.3 |
| learned_hands_domestic_violence | 83.9 | 60.9 | 62.6 |
| learned_hands_education | 91.1 | 57.1 | 55.4 |
| learned_hands_employment | 69.9 | 67.7 | 49.3 |
| learned_hands_estates | 96.6 | 59.0 | 74.7 |
| learned_hands_family | 86.2 | 57.1 | 50.9 |
| learned_hands_health | 87.2 | 65.0 | 49.6 |
| learned_hands_housing | 85.0 | 63.9 | 58.7 |
| learned_hands_immigration | 98.5 | 79.9 | 73.1 |
| learned_hands_torts | 70.6 | 60.0 | 53.2 |
| learned_hands_traffic | 95.3 | 49.8 | 52.3 |

Table 66: Commercial models on issue-spotting tasks.

| Task | Flan | Llama-2 | OPT | Vicuna | WizardLM |
|---|---|---|---|---|---|
| corporate_lobbying | 55.9 | 55.9 | 50.9 | 50.3 | 50.2 |
| learned_hands_benefits | 68.2 | 50.0 | 50.0 | 4.5 | 0.0 |
| learned_hands_business | 61.5 | 50.6 | 48.9 | 1.1 | 46.6 |
| learned_hands_consumer | 72.8 | 50.0 | 45.0 | 0.0 | 44.6 |
| learned_hands_courts | 58.9 | 49.5 | 50.0 | 50.0 | 0.0 |
| learned_hands_crime | 83.0 | 50.1 | 51.5 | 48.4 | 26.7 |
| learned_hands_divorce | 57.3 | 49.3 | 50.0 | 49.3 | 0.0 |
| learned_hands_domestic_violence | 68.4 | 48.9 | 50.0 | 0.0 | 0.0 |
| learned_hands_education | 89.3 | 50.0 | 46.4 | 50.0 | 26.8 |
| learned_hands_employment | 74.2 | 49.2 | 49.9 | 23.1 | 0.0 |
| learned_hands_estates | 67.4 | 50.0 | 68.5 | 49.4 | 38.8 |
| learned_hands_family | 5.4 | 50.1 | 63.6 | 49.3 | 0.0 |
| learned_hands_health | 66.8 | 49.6 | 63.7 | 50.0 | 38.1 |
| learned_hands_housing | 68.7 | 49.8 | 49.5 | 47.8 | 0.0 |
| learned_hands_immigration | 79.9 | 51.5 | 53.0 | 50.0 | 53.0 |
| learned_hands_torts | 60.6 | 50.0 | 50.2 | 49.8 | 46.1 |
| learned_hands_traffic | 84.2 | 49.8 | 58.6 | 10.8 | 38.7 |

Table 67: 13B models on issue-spotting tasks.

| Task | Bloom | Falcon | Incite-Base | Incite-Inst. | Llama-2 | MPT | OPT | Vicuna |
|---|---|---|---|---|---|---|---|---|
| corporate_ lobbying | 40.3 | 44.8 | 50.1 | 49.7 | 50.0 | 49.6 | 50.9 | 45.5 |
| learned_ hands_ benefits | 48.5 | 51.5 | 50.0 | 51.5 | 51.5 | 56.1 | 50.0 | 0.0 |
| learned_ hands_ business | 55.7 | 60.3 | 50.0 | 55.7 | 50.0 | 50.0 | 59.8 | 0.0 |
| learned_ hands_ consumer | 45.9 | 45.0 | 49.7 | 48.0 | 50.0 | 46.6 | 48.9 | 7.5 |
| learned_ hands_ courts | 53.6 | 50.5 | 50.5 | 48.4 | 49.5 | 57.8 | 49.5 | 0.0 |
| learned_ hands_ crime | 50.3 | 51.6 | 50.0 | 54.1 | 51.0 | 53.9 | 51.7 | 0.0 |
| learned_ hands_ divorce | 48.0 | 50.0 | 50.0 | 62.0 | 46.7 | 58.0 | 49.3 | 0.0 |
| learned_ hands_ domestic_ violence | 48.9 | 48.9 | 51.7 | 59.8 | 51.1 | 49.4 | 50.6 | 0.0 |
| learned_ hands_ education | 53.6 | 62.5 | 50.0 | 60.7 | 50.0 | 53.6 | 48.2 | 0.0 |
| learned_ hands_ employment | 50.6 | 49.7 | 49.9 | 54.8 | 49.4 | 51.1 | 49.7 | 0.0 |
| learned_ hands_ estates | 51.7 | 51.1 | 50.0 | 46.1 | 50.6 | 62.4 | 55.6 | 0.0 |
| learned_ hands_ family | 55.0 | 57.3 | 49.6 | 60.8 | 50.2 | 56.0 | 53.3 | 0.0 |
| learned_ hands_ health | 49.1 | 52.7 | 50.0 | 59.3 | 52.2 | 57.1 | 53.5 | 0.0 |
| learned_ hands_ housing | 49.2 | 49.6 | 49.8 | 47.1 | 50.5 | 49.3 | 50.1 | 0.0 |
| learned_ hands_ immigration | 53.7 | 47.0 | 50.0 | 64.9 | 50.0 | 67.9 | 62.7 | 0.0 |
| learned_ hands_ torts | 51.2 | 48.8 | 50.0 | 56.0 | 50.0 | 48.4 | 50.0 | 0.0 |
| learned_ hands_ traffic | 54.7 | 50.0 | 49.8 | 54.3 | 50.2 | 56.5 | 57.2 | 13.3 |

Table 68: 7B models on issue-spotting tasks.

| Task | Bloom | Flan | Incite | Opt |
|---|---|---|---|---|
| corporate_lobbying | 26.2 | 51.9 | 47.6 | 51.0 |
| learned_hands_benefits | 43.9 | 43.9 | 47.0 | 53.0 |
| learned_hands_business | 51.1 | 49.4 | 47.7 | 44.8 |
| learned_hands_consumer | 39.7 | 70.8 | 50.2 | 38.1 |
| learned_hands_courts | 56.2 | 46.9 | 47.4 | 60.4 |
| learned_hands_crime | 49.7 | 60.6 | 51.9 | 51.7 |
| learned_hands_divorce | 58.0 | 50.0 | 61.3 | 56.0 |
| learned_hands_domestic_violence | 36.2 | 49.4 | 52.9 | 51.7 |
| learned_hands_education | 50.0 | 67.9 | 51.8 | 46.4 |
| learned_hands_employment | 50.7 | 46.6 | 48.0 | 50.1 |
| learned_hands_estates | 42.7 | 66.3 | 52.8 | 62.4 |
| learned_hands_family | 52.1 | 47.6 | 59.0 | 64.6 |
| learned_hands_health | 46.9 | 58.4 | 53.1 | 52.2 |
| learned_hands_housing | 51.2 | 51.7 | 47.1 | 51.8 |
| learned_hands_immigration | 59.0 | 73.1 | 53.7 | 56.7 |
| learned_hands_torts | 44.4 | 66.2 | 51.2 | 56.7 |
| learned_hands_traffic | 46.8 | 65.3 | 46.4 | 65.3 |

Table 69: 3B models on issue-spotting tasks.

| Task | GPT-4 | GPT-3.5 | Claude-1 |
|---|---|---|---|
| abercrombie | 85.3 | 63.2 | 66.3 |
| diversity_1 | 100.0 | 88.4 | 83.4 |
| diversity_2 | 99.8 | 87.3 | 92.9 |
| diversity_3 | 97.0 | 89.4 | 84.3 |
| diversity_4 | 100.0 | 90.1 | 97.9 |
| diversity_5 | 93.2 | 92.6 | 81.0 |
| diversity_6 | 90.5 | 77.3 | 56.4 |
| hearsay | 83.8 | 69.2 | 76.4 |
| personal_jurisdiction | 91.4 | 63.3 | 81.9 |
| successor_liability | 57.1 | 52.3 | 72.7 |
| telemarketing_sales_rule | 82.4 | 63.1 | 71.9 |
| ucc_v_common_law | 98.8 | 100.0 | 88.8 |

Table 70: Commercial models on rule-conclusion tasks.

| Task | Flan | Llama-2 | OPT | Vicuna | WizardLM |
|---|---|---|---|---|---|
| abercrombie | 42.1 | 40.0 | 0.0 | 22.1 | 44.2 |
| diversity_1 | 76.0 | 50.0 | 55.0 | 25.5 | 60.5 |
| diversity_2 | 59.4 | 62.1 | 53.9 | 48.4 | 57.1 |
| diversity_3 | 78.6 | 65.8 | 50.0 | 52.6 | 77.8 |
| diversity_4 | 53.2 | 68.3 | 49.6 | 55.4 | 82.5 |
| diversity_5 | 53.5 | 57.7 | 52.4 | 52.7 | 53.2 |
| diversity_6 | 50.0 | 50.0 | 48.1 | 50.0 | 54.3 |
| hearsay | 64.0 | 56.5 | 52.3 | 48.7 | 66.3 |
| personal_jurisdiction | 62.6 | 57.9 | 51.3 | 0.0 | 64.0 |
| successor_liability | 52.7 | 51.2 | 26.4 | 0.0 | 39.1 |
| telemarketing_sales_rule | 69.8 | 74.3 | 50.0 | 63.1 | 72.7 |
| ucc_v_common_law | 98.1 | 77.9 | 52.5 | 0.0 | 79.8 |

Table 71: 13B models on rule-conclusion tasks.

| Task | Bloom | Falcon | Incite-Base | Incite-Inst. | Llama-2 | MPT | OPT | Vicuna |
|---|---|---|---|---|---|---|---|---|
| abercrombie | 17.9 | 24.2 | 27.4 | 34.7 | 32.6 | 34.7 | 17.9 | 2.1 |
| diversity_1 | 47.6 | 51.7 | 56.8 | 61.5 | 73.4 | 54.9 | 58.0 | 55.3 |
| diversity_2 | 50.0 | 56.3 | 50.5 | 65.7 | 50.0 | 50.0 | 49.8 | 49.1 |
| diversity_3 | 51.9 | 53.6 | 50.0 | 50.8 | 62.8 | 50.0 | 49.9 | 49.3 |
| diversity_4 | 58.3 | 68.7 | 50.0 | 67.9 | 72.9 | 50.0 | 49.7 | 49.7 |
| diversity_5 | 57.9 | 50.0 | 50.4 | 49.4 | 51.3 | 50.0 | 50.0 | 46.8 |
| diversity_6 | 50.0 | 50.0 | 50.0 | 44.7 | 50.0 | 50.0 | 49.7 | 50.0 |
| hearsay | 51.7 | 53.7 | 50.0 | 73.2 | 64.1 | 61.7 | 36.4 | 30.7 |
| personal_jurisdiction | 51.8 | 54.6 | 40.0 | 46.1 | 52.6 | 43.8 | 50.6 | 50.0 |
| successor_liability | 21.7 | 39.1 | 37.2 | 32.6 | 43.4 | 35.7 | 38.8 | 0.0 |
| telemarketing_sales_rule | 57.4 | 55.3 | 51.8 | 58.6 | 61.3 | 55.8 | 54.5 | 43.6 |
| ucc_v_common_law | 50.0 | 50.0 | 50.0 | 50.0 | 56.6 | 50.0 | 50.0 | 0.0 |

Table 72: 7B models on rule-conclusion tasks.

| Task | Bloom | Flan | Incite | Opt |
|---|---|---|---|---|
| abercrombie | 20.0 | 31.6 | 25.3 | 26.3 |
| diversity_1 | 50.0 | 50.0 | 50.3 | 51.5 |
| diversity_2 | 50.0 | 50.0 | 49.3 | 52.6 |
| diversity_3 | 50.0 | 50.0 | 46.2 | 50.0 |
| diversity_4 | 50.0 | 50.0 | 62.5 | 51.4 |
| diversity_5 | 51.8 | 50.0 | 52.8 | 50.0 |
| diversity_6 | 50.5 | 50.0 | 51.5 | 50.0 |
| hearsay | 51.2 | 57.5 | 57.0 | 49.7 |
| personal_jurisdiction | 50.0 | 51.1 | 46.1 | 50.0 |
| successor_liability | 21.7 | 44.6 | 24.8 | 14.4 |
| telemarketing_sales_rule | 44.5 | 51.4 | 53.5 | 55.9 |
| ucc_v_common_law | 50.0 | 88.8 | 50.0 | 50.0 |

Table 73: 3B models on rule-conclusion tasks.

| Task | GPT-4 | GPT-3.5 | Claude-1 |
|---|---|---|---|
| canada_tax_court_outcomes | 98.9 | 80.0 | 76.9 |
| definition_classification | 96.6 | 80.2 | 87.9 |
| definition_extraction | 81.8 | 85.0 | 82.7 |
| function_of_decision_section | 43.3 | 35.2 | 37.6 |
| legal_reasoning_causality | 84.5 | 72.1 | 66.9 |
| oral_argument_question_purpose | 37.4 | 28.4 | 35.1 |
| overruling | 95.2 | 88.9 | 95.4 |
| scalr | 77.9 | 58.8 | 64.7 |
| textualism_tool_dictionaries | 93.9 | 65.1 | 71.2 |
| textualism_tool_plain | 84.7 | 73.2 | 70.2 |

Table 74: Commercial models on rhetorical-understanding tasks.

| Task | Flan | Llama-2 | OPT | Vicuna | WizardLM |
|---|---|---|---|---|---|
| canada_tax_court_outcomes | 71.7 | 34.4 | 1.8 | 2.9 | 75.1 |
| definition_classification | 80.8 | 51.2 | 57.4 | 50.0 | 81.5 |
| definition_extraction | 80.5 | 85.0 | 80.1 | 62.4 | 82.0 |
| function_of_decision_section | 34.6 | 18.0 | 20.8 | 14.2 | 13.2 |
| legal_reasoning_causality | 78.6 | 52.8 | 52.3 | 48.4 | 46.2 |
| oral_argument_question_purpose | 24.5 | 20.0 | 15.0 | 16.1 | 29.3 |
| overruling | 94.2 | 92.3 | 78.3 | 52.7 | 89.5 |
| scalr | 66.5 | 56.7 | 20.2 | 5.0 | 45.6 |
| textualism_tool_dictionaries | 93.9 | 66.5 | 54.9 | 5.6 | 61.1 |
| textualism_tool_plain | 81.5 | 72.6 | 51.7 | 43.6 | 74.6 |

Table 75: 13B models on rhetorical-understanding tasks.

| Task | Bloom | Falcon | Incite-Base | Incite-Inst. | Llama-2 | MPT | OPT | Vicuna |
|---|---|---|---|---|---|---|---|---|
| canada_tax_court_outcomes | 0.0 | 0.0 | 0.0 | 5.2 | 35.4 | 0.0 | 28.5 | 0.0 |
| definition_classification | 54.4 | 80.2 | 50.1 | 65.4 | 50.0 | 57.5 | 58.3 | 42.5 |
| definition_extraction | 77.4 | 77.7 | 80.6 | 73.4 | 84.1 | 80.9 | 73.5 | 4.2 |
| function_of_decision_section | 14.2 | 2.3 | 10.2 | 10.6 | 16.7 | 13.9 | 22.3 | 0.0 |
| legal_reasoning_causality | 47.4 | 59.4 | 57.6 | 56.0 | 53.2 | 55.5 | 50.0 | 38.7 |
| oral_argument_question_purpose | 13.7 | 15.5 | 20.2 | 27.0 | 14.5 | 17.2 | 14.4 | 1.2 |
| overruling | 82.3 | 67.9 | 50.3 | 75.2 | 92.2 | 72.4 | 53.3 | 28.7 |
| scalr | 18.6 | 20.7 | 22.0 | 23.0 | 33.7 | 25.7 | 19.3 | 0.0 |
| textualism_tool_dictionaries | 48.5 | 60.9 | 55.7 | 59.6 | 47.4 | 59.5 | 51.0 | 19.8 |
| textualism_tool_plain | 50.0 | 57.2 | 61.6 | 55.9 | 50.0 | 60.5 | 51.0 | 4.6 |

Table 76: 7B models on rhetorical-understanding tasks.

| Task | Bloom | Flan | Incite | Opt |
|---|---|---|---|---|
| canada_tax_court_outcomes | 16.0 | 62.3 | 11.7 | 0.3 |
| definition_classification | 51.0 | 78.5 | 51.3 | 57.2 |
| definition_extraction | 59.1 | 77.6 | 59.4 | 70.6 |
| function_of_decision_section | 9.9 | 34.8 | 26.7 | 14.1 |
| legal_reasoning_causality | 50.1 | 67.5 | 49.5 | 55.7 |
| oral_argument_question_purpose | 5.3 | 19.9 | 21.5 | 14.3 |
| overruling | 63.4 | 93.6 | 54.4 | 54.8 |
| scalr | 17.7 | 64.5 | 21.5 | 20.0 |
| textualism_tool_dictionaries | 39.7 | 92.9 | 55.6 | 54.8 |
| textualism_tool_plain | 51.5 | 82.2 | 50.7 | 55.9 |

Table 77: 3B models on rhetorical-understanding tasks.

Table 78: Commercial models on interpretation tasks.

| Task | GPT-4 | GPT-3.5 | Claude-1 |
|---|---|---|---|
| consumer_contracts_qa | 93.6 | 85.9 | 90.3 |
| contract_nli_confidentiality_of_agreement | 96.3 | 96.3 | 92.7 |
| contract_nli_explicit_identification | 82.4 | 81.1 | 65.0 |
| contract_nli_inclusion_of_verbally_conveyed_information | 90.7 | 83.0 | 83.4 |
| contract_nli_limited_use | 86.6 | 85.4 | 80.8 |
| contract_nli_no_licensing | 92.5 | 76.7 | 79.2 |
| contract_nli_notice_on_compelled_disclosure | 97.2 | 97.2 | 96.5 |
| contract_nli_permissible_acquirement_of_similar_information | 96.1 | 96.6 | 93.8 |
| contract_nli_permissible_copy | 80.4 | 77.7 | 72.8 |
| contract_nli_permissible_development_of_similar_information | 98.5 | 99.3 | 96.3 |
| contract_nli_permissible_post-agreement_possession | 94.6 | 89.3 | 92.2 |
| contract_nli_return_of_confidential_information | 95.6 | 92.5 | 89.4 |
| contract_nli_sharing_with_employees | 94.6 | 94.8 | 95.9 |
| contract_nli_sharing_with_third-parties | 93.3 | 75.0 | 86.6 |
| contract_nli_survival_of_obligations | 94.0 | 74.5 | 78.3 |
| contract_qa | 96.2 | 93.6 | 98.7 |
| cuad_affiliate_license-licensee | 90.9 | 90.9 | 85.9 |
| cuad_affiliate_license-licensor | 92.0 | 95.5 | 89.8 |
| cuad_anti-assignment | 91.4 | 89.1 | 92.4 |
| cuad_audit_rights | 97.9 | 89.5 | 92.7 |
| cuad_cap_on_liability | 95.6 | 94.1 | 92.9 |
| cuad_change_of_control | 88.9 | 89.7 | 89.2 |
| cuad_competitive_restriction_exception | 84.1 | 80.0 | 71.4 |
| cuad_covenant_not_to_sue | 95.8 | 88.0 | 89.3 |
| cuad_effective_date | 92.8 | 75.0 | 74.2 |
| cuad_exclusivity | 92.9 | 89.0 | 87.3 |
| cuad_expiration_date | 82.0 | 87.0 | 78.8 |
| cuad_governing_law | 99.3 | 98.3 | 98.7 |
| cuad_insurance | 99.2 | 95.3 | 94.9 |
| cuad_ip_ownership_assignment | 91.7 | 91.0 | 89.2 |
| cuad_irrevocable_or_perpetual_license | 97.5 | 95.4 | 88.6 |
| cuad_joint_ip_ownership | 94.3 | 91.1 | 85.4 |
| cuad_license_grant | 94.0 | 90.3 | 91.0 |
| cuad_liquidated_damages | 96.4 | 86.4 | 90.9 |
| cuad_minimum_commitment | 89.1 | 86.1 | 88.6 |
| cuad_most_favored_nation | 96.9 | 95.3 | 96.9 |
| cuad_no-solicit_of_customers | 100.0 | 98.8 | 92.9 |
| cuad_no-solicit_of_employees | 100.0 | 97.9 | 96.5 |
| cuad_non-compete | 93.0 | 91.0 | 90.5 |
| cuad_non-disparagement | 97.0 | 95.0 | 87.0 |

| Task | GPT-4 | GPT-3.5 | Claude-1 |
|---|---|---|---|
| cuad_non-transferable_license | 90.2 | 82.1 | 87.6 |
| cuad_notice_period_to_terminate_renewal | 95.9 | 97.7 | 96.4 |
| cuad_post-termination_services | 94.6 | 89.0 | 77.8 |
| cuad_price_restrictions | 95.7 | 87.0 | 89.1 |
| cuad_renewal_term | 96.1 | 95.9 | 95.6 |
| cuad_revenue-profit_sharing | 95.3 | 91.2 | 88.9 |
| cuad_rofr-rofo-rofn | 88.6 | 81.9 | 86.2 |
| cuad_source_code_escrow | 96.6 | 91.5 | 94.1 |
| cuad_termination_for_convenience | 96.7 | 94.2 | 95.8 |
| cuad_third_party_beneficiary | 89.7 | 83.8 | 82.4 |
| cuad_uncapped_liability | 85.4 | 70.4 | 74.1 |
| cuad_unlimited-all-you-can-eat-license | 93.8 | 93.8 | 87.5 |
| cuad_volume_restriction | 80.7 | 68.6 | 80.1 |
| cuad_warranty_duration | 77.8 | 81.2 | 78.1 |
| insurance_policy_interpretation | 69.6 | 55.0 | 64.0 |
| jcrew_blocker | 100.0 | 88.9 | 55.6 |
| maud_ability_to_consummate_concept_is_subject_to_mae_carveouts | 50.0 | 50.0 | 31.5 |
| maud_financial_point_of_view_is_the_sole_consideration | 50.0 | 38.8 | 50.0 |
| maud_accuracy_of_fundamental_target_rws_bringdown_standard | 29.3 | 33.3 | 10.4 |
| maud_accuracy_of_target_general_rw_bringdown_timing_answer | 63.6 | 51.0 | 46.9 |
| maud_accuracy_of_target_capitalization_rw_(outstanding_shares)_bringdown_standard_answer | 20.7 | 16.2 | 13.4 |
| maud_additional_matching_rights_period_for_modifications_(cor) | 57.4 | 43.3 | 24.5 |
| maud_application_of_buyer_consent_requirement_(negative_interim_covenant) | 63.7 | 68.8 | 35.3 |
| maud_buyer_consent_requirement_(ordinary_course) | 50.0 | 60.8 | 28.9 |
| maud_change_in_law__subject_to_disproportionate_impact_modifier | 53.0 | 48.3 | 65.2 |
| maud_changes_in_gaap_or_other_accounting_principles__subject_to_disproportionate_impact_modifier | 51.7 | 47.4 | 53.8 |
| maud_cor_permitted_in_response_to_intervening_event | 50.0 | 52.5 | 50.1 |
| maud_cor_permitted_with_board_fiduciary_determination_only | 21.4 | 50.0 | 50.6 |
| maud_cor_standard_(intervening_event) | 0.5 | 36.5 | 0.0 |
| maud_cor_standard_(superior_offer) | 40.5 | 45.5 | 0.0 |
| maud_definition_contains_knowledge_requirement_-_answer | 25.0 | 33.7 | 20.7 |
| maud_definition_includes_asset_deals | 33.3 | 30.5 | 0.5 |
| maud_definition_includes_stock_deals | 33.3 | 37.5 | 27.4 |
| maud_fiduciary_exception__board_determination_standard | 40.1 | 27.5 | 0.0 |
| maud_fiduciary_exception_board_determination_trigger_(no_shop) | 50.0 | 48.8 | 50.0 |
| maud_fls_(mae)_standard | 25.0 | 44.6 | 22.5 |
| maud_general_economic_and_financial_conditions_subject_to_disproportionate_impact_modifier | 54.2 | 56.0 | 53.6 |
| maud_includes_consistent_with_past_practice | 54.2 | 55.3 | 84.2 |

| Task | GPT-4 | GPT-3.5 | Claude-1 |
|---|---|---|---|
| maud_initial_matching_rights_period_(cor) | 15.4 | 31.9 | 20.7 |
| maud_initial_matching_rights_period_(ftr) | 49.4 | 32.8 | 14.0 |
| maud_intervening_event_-_required_to_occur_after_signing_-_answer | 51.9 | 51.4 | 12.5 |
| maud_knowledge_definition | 51.1 | 49.1 | 38.5 |
| maud_liability_standard_for_no-shop_breach_by_target_non-do_representatives | 44.2 | 51.9 | 49.4 |
| maud_ordinary_course_efforts_standard | 91.1 | 70.3 | 53.6 |
| maud_pandemic_or_other_public_health_event__subject_to_disproportionate_impact_modifier | 48.7 | 50.0 | 51.3 |
| maud_pandemic_or_other_public_health_event_specific_reference_to_pandemic-related_governmental_responses_or_measures | 79.5 | 70.9 | 60.8 |
| maud_relational_language_(mae)_applies_to | 57.9 | 47.2 | 26.0 |
| maud_specific_performance | 51.5 | 90.6 | 73.7 |
| maud_tail_period_length | 68.1 | 39.5 | 60.4 |
| maud_type_of_consideration | 99.5 | 82.7 | 75.1 |
| opp115_data_retention | 67.0 | 70.5 | 55.7 |
| opp115_data_security | 87.5 | 84.2 | 55.6 |
| opp115_do_not_track | 99.1 | 93.6 | 90.0 |
| opp115_first_party_collection_use | 76.7 | 80.6 | 63.0 |
| opp115_international_and_specific_audiences | 92.3 | 82.6 | 79.4 |
| opp115_policy_change | 91.9 | 89.3 | 83.8 |
| opp115_third_party_sharing_collection | 80.1 | 77.0 | 71.0 |
| opp115_user_access,_edit_and_deletion | 90.2 | 87.7 | 79.3 |
| opp115_user_choice_control | 82.9 | 79.3 | 71.3 |
| privacy_policy_entailment | 85.5 | 78.8 | 89.6 |
| privacy_policy_qa | 71.3 | 65.5 | 63.0 |
| proa | 99.0 | 90.6 | 88.5 |
| ssla_company_defendants | 65.0 | 65.3 | 16.5 |
| ssla_individual_defendants | 29.6 | 25.8 | 11.1 |
| ssla_plaintiff | 92.0 | 86.5 | 86.7 |
| sara_entailment | 86.8 | 68.4 | 67.6 |
| sara_numeric | 8.3 | 4.2 | 6.2 |
| supply_chain_disclosure_best_practice_accountability | 71.5 | 69.5 | 74.6 |
| supply_chain_disclosure_best_practice_audits | 74.4 | 76.6 | 75.5 |
| supply_chain_disclosure_best_practice_certification | 76.6 | 77.7 | 77.4 |
| supply_chain_disclosure_best_practice_training | 83.3 | 87.1 | 85.3 |
| supply_chain_disclosure_best_practice_verification | 68.3 | 59.4 | 64.3 |
| supply_chain_disclosure_disclosed_accountability | 77.0 | 80.4 | 75.5 |
| supply_chain_disclosure_disclosed_audits | 81.6 | 83.7 | 80.0 |
| supply_chain_disclosure_disclosed_certification | 71.2 | 67.3 | 75.8 |
| supply_chain_disclosure_disclosed_training | 89.1 | 83.0 | 75.6 |

| Task | GPT-4 | GPT-3.5 | Claude-1 |
|---|---|---|---|
| supply_chain_disclosure_disclosed_verification | 56.6 | 62.0 | 67.6 |
| unfair_tos | 9.1 | 13.7 | 5.5 |

Table 79: 13B models on interpretation tasks.

| Task | Flan | LLaMA-2 | OPT | Vicuna | WizardLM |
|---|---|---|---|---|---|
| consumer_contracts_qa | 92.6 | 68.1 | 24.8 | 37.9 | 67.8 |
| contract_nli_confidentiality_of_agreement | 85.4 | 50.0 | 54.9 | 48.8 | 76.8 |
| contract_nli_explicit_identification | 81.4 | 49.4 | 51.6 | 50.0 | 67.0 |
| contract_nli_inclusion_of_verbally_conveyed_ information | 56.2 | 50.0 | 53.0 | 50.0 | 58.2 |
| contract_nli_limited_use | 71.1 | 44.5 | 60.6 | 49.5 | 57.1 |
| contract_nli_no_licensing | 54.2 | 53.3 | 47.0 | 51.2 | 58.4 |
| contract_nli_notice_on_compelled_disclosure | 90.8 | 52.1 | 61.3 | 55.6 | 73.2 |
| contract_nli_permissible_acquirement_of_ similar_information | 89.3 | 50.0 | 45.5 | 52.2 | 61.2 |
| contract_nli_permissible_copy | 84.3 | 47.8 | 47.8 | 50.0 | 56.2 |
| contract_nli_permissible_development_of_ similar_information | 98.5 | 50.0 | 55.1 | 54.4 | 86.0 |
| contract_nli_permissible_post-agreement_ possession | 93.4 | 48.8 | 44.0 | 50.0 | 52.5 |
| contract_nli_return_of_confidential_information | 89.4 | 50.0 | 44.3 | 51.3 | 66.2 |
| contract_nli_sharing_with_employees | 92.2 | 48.2 | 70.5 | 50.0 | 65.7 |
| contract_nli_sharing_with_third-parties | 86.7 | 49.5 | 50.9 | 50.5 | 64.4 |
| contract_nli_survival_of_obligations | 74.6 | 50.0 | 44.8 | 50.6 | 59.6 |
| contract_qa | 96.3 | 82.7 | 56.8 | 73.1 | 35.9 |
| cuad_affiliate_license-licensee | 83.8 | 58.1 | 49.0 | 50.0 | 65.7 |
| cuad_affiliate_license-licensor | 90.9 | 72.7 | 53.4 | 50.0 | 54.5 |
| cuad_anti-assignment | 85.0 | 52.2 | 48.1 | 50.2 | 76.8 |
| cuad_audit_rights | 87.1 | 51.6 | 71.7 | 50.7 | 58.6 |
| cuad_cap_on_liability | 81.5 | 74.0 | 0.6 | 50.0 | 57.5 |
| cuad_change_of_control | 75.5 | 56.2 | 52.9 | 49.0 | 74.8 |
| cuad_competitive_restriction_exception | 81.8 | 51.8 | 38.2 | 50.0 | 50.0 |
| cuad_covenant_not_to_sue | 86.0 | 66.6 | 49.4 | 50.0 | 56.5 |
| cuad_effective_date | 92.8 | 50.0 | 50.4 | 48.7 | 58.1 |
| cuad_exclusivity | 84.0 | 86.1 | 61.7 | 49.6 | 58.7 |
| cuad_expiration_date | 60.5 | 51.6 | 53.1 | 51.0 | 73.2 |
| cuad_governing_law | 99.5 | 90.6 | 71.7 | 59.5 | 54.3 |
| cuad_insurance | 90.2 | 55.5 | 66.3 | 55.0 | 55.7 |
| cuad_ip_ownership_assignment | 89.8 | 74.0 | 49.8 | 49.7 | 59.9 |
| cuad_irrevocable_or_perpetual_license | 95.7 | 85.4 | 67.9 | 50.0 | 73.9 |
| cuad_joint_ip_ownership | 76.0 | 53.6 | 50.5 | 50.0 | 60.4 |

Table 79 – continued from previous page

| Task | Flan | LLaMA-2 | OPT | Vicuna | WizardLM |
|------|------|---------|-----|--------|----------|
| cuad_license_grant | 92.8 | 62.2 | 50.8 | 49.9 | 68.7 |
| cuad_liquidated_damages | 73.6 | 50.9 | 48.6 | 49.5 | 71.8 |
| cuad_minimum_commitment | 62.3 | 54.5 | 51.6 | 49.9 | 47.0 |
| cuad_most_favored_nation | 75.0 | 56.2 | 43.8 | 50.0 | 57.8 |
| cuad_no-solicit_of_customers | 97.6 | 57.1 | 56.0 | 50.0 | 85.7 |
| cuad_no-solicit_of_employees | 95.8 | 97.9 | 69.0 | 50.0 | 75.4 |
| cuad_non-compete | 91.6 | 52.7 | 46.8 | 49.3 | 55.2 |
| cuad_non-disparagement | 83.0 | 73.0 | 67.0 | 49.0 | 74.0 |
| cuad_non-transferable_license | 69.6 | 55.0 | 54.2 | 49.4 | 75.6 |
| cuad_notice_period_to_terminate_renewal | 92.8 | 50.0 | 54.5 | 50.0 | 83.3 |
| cuad_post-termination_services | 88.7 | 50.7 | 51.4 | 49.6 | 62.5 |
| cuad_price_restrictions | 76.1 | 71.7 | 50.0 | 47.8 | 50.0 |
| cuad_renewal_term | 89.1 | 50.5 | 50.3 | 57.5 | 83.9 |
| cuad_revenue-profit_sharing | 72.1 | 58.4 | 54.5 | 49.7 | 52.7 |
| cuad_rofr-rofo-rofn | 65.1 | 50.4 | 54.1 | 50.0 | 59.6 |
| cuad_source_code_escrow | 66.1 | 58.5 | 67.8 | 50.0 | 52.5 |
| cuad_termination_for_convenience | 91.4 | 50.9 | 49.8 | 55.8 | 84.2 |
| cuad_third_party_beneficiary | 85.3 | 54.4 | 63.2 | 47.1 | 69.1 |
| cuad_uncapped_liability | 50.3 | 55.4 | 46.6 | 51.0 | 74.5 |
| cuad_unlimited-all-you-can-eat-license | 83.3 | 52.1 | 64.6 | 47.9 | 54.2 |
| cuad_volume_restriction | 55.6 | 50.0 | 55.3 | 50.0 | 54.0 |
| cuad_warranty_duration | 74.7 | 57.5 | 55.9 | 50.6 | 63.1 |
| insurance_policy_interpretation | 44.8 | 46.6 | 38.0 | 13.2 | 51.9 |
| jcrew_blocker | 86.7 | 66.7 | 51.1 | 50.0 | 8.9 |
| maud_ability_to_consummate_concept_is_ subject_to_mae_carveouts | 50.0 | 47.3 | 51.8 | 50.0 | 8.2 |
| maud_financial_point_of_view_is_the_sole_ consideration | 53.6 | 51.0 | 50.0 | 53.1 | 49.5 |
| maud_accuracy_of_fundamental_target_rws_ bringdown_standard | 12.5 | 33.3 | 33.3 | 31.7 | 33.3 |
| maud_accuracy_of_target_general_rw_ bringdown_timing_answer | 46.6 | 50.0 | 49.1 | 49.8 | 50.0 |
| maud_accuracy_of_target_capitalization_rw_ (outstanding_shares)_bringdown_standard_ answer | 6.1 | 25.8 | 26.9 | 20.9 | 24.8 |
| maud_additional_matching_rights_period_for_ modifications_(cor) | 0.0 | 25.0 | 19.8 | 18.4 | 12.9 |
| maud_application_of_buyer_consent_ requirement_(negative_interim_covenant) | 63.4 | 47.8 | 45.0 | 3.1 | 49.4 |
| maud_buyer_consent_requirement_(ordinary_ course) | 34.4 | 56.5 | 44.4 | 45.1 | 50.0 |
| maud_change_in_law__subject_to_ disproportionate_impact_modifier | 50.0 | 55.9 | 46.6 | 21.3 | 0.0 |

| Task | Flan | LLaMA-2 | OPT | Vicuna | WizardLM |
|---|---|---|---|---|---|
| maud_changes_in_gaap_or_other_accounting_ principles__subject_to_disproportionate_impact_ modifier | 50.0 | 57.5 | 47.1 | 20.6 | 0.0 |
| maud_cor_permitted_in_response_to_ intervening_event | 50.0 | 57.6 | 58.8 | 26.8 | 47.0 |
| maud_cor_permitted_with_board_fiduciary_ determination_only | 50.0 | 50.0 | 51.2 | 46.7 | 42.0 |
| maud_cor_standard_(intervening_event) | 0.0 | 24.0 | 16.7 | 10.0 | 23.3 |
| maud_cor_standard_(superior_offer) | 11.9 | 16.8 | 3.0 | 0.0 | 26.9 |
| maud_definition_contains_knowledge_ requirement_-_answer | 0.0 | 28.9 | 25.0 | 24.5 | 24.1 |
| maud_definition_includes_asset_deals | 2.8 | 35.4 | 32.7 | 0.9 | 34.2 |
| maud_definition_includes_stock_deals | 4.6 | 31.8 | 24.7 | 30.1 | 17.3 |
| maud_fiduciary_exception__board_ determination_standard | 1.2 | 6.5 | 12.5 | 0.5 | 14.1 |
| maud_fiduciary_exception_board_determination_ trigger_(no_shop) | 50.0 | 59.2 | 44.7 | 8.8 | 48.8 |
| maud_fls_(mae)_standard | 17.1 | 5.0 | 24.5 | 25.0 | 4.6 |
| maud_general_economic_and_financial_ conditions_subject_to_disproportionate_impact_ modifier | 50.0 | 53.6 | 52.4 | 6.0 | 0.0 |
| maud_includes_consistent_with_past_practice | 62.3 | 50.0 | 52.6 | 53.2 | 61.0 |
| maud_initial_matching_rights_period_(cor) | 0.0 | 8.3 | 20.8 | 0.9 | 14.2 |
| maud_initial_matching_rights_period_(ftr) | 0.0 | 13.3 | 23.2 | 19.1 | 18.8 |
| maud_intervening_event_-_required_to_occur_ after_signing_-_answer | 39.2 | 46.2 | 48.9 | 47.1 | 47.1 |
| maud_knowledge_definition | 50.6 | 46.0 | 48.4 | 0.0 | 51.2 |
| maud_liability_standard_for_no-shop_breach_ by_target_non-do_representatives | 48.7 | 48.7 | 49.4 | 0.6 | 50.0 |
| maud_ordinary_course_efforts_standard | 81.1 | 57.8 | 33.7 | 0.0 | 67.3 |
| maud_pandemic_or_other_public_health_event_ _subject_to_disproportionate_impact_modifier | 48.1 | 52.2 | 46.2 | 7.6 | 31.0 |
| maud_pandemic_or_other_public_health_ event_specific_reference_to_pandemic-related_ governmental_responses_or_measures | 50.0 | 50.0 | 51.9 | 50.7 | 37.4 |
| maud_relational_language_(mae)_applies_to | 51.0 | 44.2 | 1.4 | 51.7 | 5.1 |
| maud_specific_performance | 94.9 | 52.1 | 50.0 | 0.0 | 56.7 |
| maud_tail_period_length | 25.8 | 51.5 | 52.5 | 13.5 | 34.6 |
| maud_type_of_consideration | 73.6 | 39.1 | 30.2 | 27.8 | 27.2 |
| opp115_data_retention | 55.7 | 51.1 | 45.5 | 50.0 | 63.6 |
| opp115_data_security | 75.1 | 49.8 | 51.2 | 55.5 | 59.4 |
| opp115_do_not_track | 79.1 | 50.0 | 42.7 | 51.8 | 88.2 |
| opp115_first_party_collection_use | 75.0 | 67.0 | 68.5 | 52.3 | 55.2 |
| opp115_international_and_specific_audiences | 80.1 | 59.5 | 18.2 | 50.4 | 66.2 |
| opp115_policy_change | 70.5 | 64.9 | 60.9 | 52.8 | 56.5 |

| Task | Flan | LLaMA-2 | OPT | Vicuna | WizardLM |
|------|------|---------|-----|--------|----------|
| opp115_third_party_sharing_collection | 71.4 | 54.9 | 58.8 | 52.4 | 60.9 |
| opp115_user_access,_edit_and_deletion | 75.4 | 54.3 | 60.9 | 49.1 | 59.4 |
| opp115_user_choice_control | 80.9 | 53.7 | 50.0 | 47.5 | 58.0 |
| privacy_policy_entailment | 58.9 | 56.2 | 50.1 | 0.6 | 65.9 |
| privacy_policy_qa | 52.5 | 50.5 | 50.9 | 0.0 | 55.6 |
| proa | 94.8 | 76.0 | 52.1 | 50.0 | 80.1 |
| ssla_company_defendants | 34.4 | 63.4 | 56.6 | 3.1 | 2.8 |
| ssla_individual_defendants | 21.9 | 21.6 | 16.2 | 0.0 | 0.0 |
| ssla_plaintiff | 88.8 | 26.4 | 31.6 | 0.0 | 0.0 |
| sara_entailment | 35.3 | 58.1 | 48.9 | 15.4 | 50.0 |
| sara_numeric | 1.0 | 0.0 | 0.0 | 0.0 | 0.0 |
| supply_chain_disclosure_best_practice_accountability | 73.6 | 49.4 | 48.8 | 67.2 | 52.1 |
| supply_chain_disclosure_best_practice_audits | 69.2 | 49.2 | 66.6 | 46.5 | 65.7 |
| supply_chain_disclosure_best_practice_certification | 69.9 | 51.6 | 56.1 | 67.2 | 64.8 |
| supply_chain_disclosure_best_practice_training | 81.0 | 49.7 | 49.4 | 71.5 | 64.9 |
| supply_chain_disclosure_best_practice_verification | 70.5 | 49.1 | 50.8 | 51.6 | 52.5 |
| supply_chain_disclosure_disclosed_accountability | 80.7 | 49.4 | 45.8 | 59.1 | 58.8 |
| supply_chain_disclosure_disclosed_audits | 83.3 | 49.3 | 48.9 | 64.1 | 64.7 |
| supply_chain_disclosure_disclosed_certification | 68.7 | 49.4 | 53.8 | 61.6 | 59.3 |
| supply_chain_disclosure_disclosed_training | 87.0 | 49.2 | 48.0 | 58.1 | 62.8 |
| supply_chain_disclosure_disclosed_verification | 64.3 | 49.3 | 48.6 | 58.0 | 50.4 |
| unfair_tos | 10.0 | 12.8 | 10.0 | 8.3 | 13.8 |

Table 80: 7B models on interpretation tasks.

| Task | BLOOM | Falcon | Incite-Base | Incite-Inst. | LLaMA-2 | MPT | OPT | Vicuna |
|------|-------|--------|-------------|--------------|---------|-----|-----|--------|
| consumer_contracts_qa | 0.6 | 57.9 | 42.4 | 49.0 | 63.5 | 40.5 | 14.7 | 34.0 |
| contract_nli_confidentiality_of_agreement | 53.7 | 64.6 | 65.9 | 59.8 | 50.0 | 52.4 | 57.3 | 45.1 |
| contract_nli_explicit_identification | 49.4 | 66.1 | 61.6 | 68.9 | 50.0 | 50.0 | 59.9 | 34.8 |

| Task | BLOOM | Falcon | Incite-Base | Incite-Inst. | LLaMA-2 | MPT | OPT | Vicuna |
|---|---|---|---|---|---|---|---|---|
| contract_ nli_ inclusion_ of_ verbally_ conveyed_ information | 50.0 | 67.5 | 66.7 | 60.9 | 50.0 | 49.3 | 55.5 | 33.1 |
| contract_ nli_ limited_ use | 56.2 | 46.0 | 59.0 | 66.3 | 51.1 | 61.8 | 61.7 | 24.8 |
| contract_ nli_no_ licensing | 49.4 | 44.6 | 51.9 | 56.3 | 48.2 | 45.9 | 49.5 | 15.9 |
| contract_ nli_ notice_ on_ compelled_ disclosure | 51.4 | 51.4 | 64.1 | 62.7 | 50.7 | 65.5 | 68.3 | 27.5 |
| contract_ nli_ permissible_ acquirement_ of_ similar_ information | 49.4 | 31.5 | 26.4 | 47.8 | 50.0 | 53.4 | 50.0 | 39.9 |
| contract_ nli_ permissible_ copy | 49.9 | 36.8 | 46.4 | 57.6 | 58.9 | 55.0 | 49.6 | 33.2 |
| contract_ nli_ permissible_ development_ of_ similar_ information | 49.3 | 43.4 | 59.6 | 44.1 | 50.0 | 54.4 | 53.7 | 41.9 |
| contract_ nli_ permissible_ post- agreement_ possession | 48.2 | 43.1 | 53.2 | 55.6 | 50.0 | 55.3 | 44.5 | 2.4 |
| contract_ nli_ return_of_ confidential_ information | 50.0 | 57.7 | 50.2 | 70.0 | 50.0 | 56.5 | 75.6 | 5.9 |
| contract_ nli_ sharing_ with_ employees | 55.6 | 61.6 | 48.2 | 70.3 | 50.0 | 48.1 | 58.9 | 6.1 |

| Task | BLOOM | Falcon | Incite-Base | Incite-Inst. | LLaMA-2 | MPT | OPT | Vicuna |
|---|---|---|---|---|---|---|---|---|
| contract_ nli_ sharing_ with_third-parties | 49.5 | 48.1 | 39.3 | 53.7 | 50.0 | 48.4 | 48.7 | 11.9 |
| contract_ nli_ survival_ of_ obligations | 49.5 | 55.1 | 41.2 | 43.2 | 50.0 | 41.7 | 49.3 | 1.8 |
| contract_ qa | 14.8 | 7.7 | 11.4 | 87.7 | 31.9 | 9.0 | 39.0 | 50.3 |
| cuad_ affiliate_ license-licensee | 53.0 | 50.5 | 68.7 | 71.7 | 50.0 | 69.2 | 66.2 | 34.3 |
| cuad_ affiliate_ license-licensor | 45.5 | 70.5 | 64.8 | 85.2 | 50.0 | 76.1 | 52.3 | 38.6 |
| cuad_anti-assignment | 47.7 | 33.2 | 60.4 | 76.2 | 58.8 | 56.5 | 55.1 | 37.8 |
| cuad_ audit_ rights | 67.4 | 59.7 | 80.0 | 71.5 | 50.7 | 52.5 | 72.8 | 28.2 |
| cuad_ cap_on_ liability | 41.8 | 42.1 | 40.4 | 57.4 | 50.2 | 70.3 | 37.0 | 16.8 |
| cuad_ change_ of_control | 55.0 | 49.8 | 57.0 | 67.8 | 50.0 | 56.0 | 55.5 | 39.2 |
| cuad_ competitive_ restriction_ exception | 37.3 | 30.9 | 50.0 | 47.7 | 48.6 | 46.8 | 36.4 | 16.4 |
| cuad_ covenant_ not_to_ sue | 47.7 | 48.1 | 67.9 | 78.2 | 57.5 | 71.8 | 50.0 | 0.0 |
| cuad_ effective_ date | 56.8 | 60.2 | 53.0 | 46.6 | 50.0 | 44.5 | 65.7 | 6.4 |
| cuad_ exclusivity | 62.9 | 57.1 | 66.4 | 65.4 | 63.4 | 56.7 | 60.0 | 14.4 |
| cuad_ expiration_ date | 75.9 | 62.7 | 55.5 | 67.6 | 50.1 | 52.9 | 85.0 | 10.5 |
| cuad_ governing_ law | 79.3 | 77.3 | 28.5 | 68.2 | 55.8 | 66.3 | 73.1 | 17.6 |
| cuad_ insurance | 83.2 | 66.8 | 65.4 | 72.0 | 50.1 | 72.7 | 77.2 | 39.1 |

| Task | BLOOM | Falcon | Incite-Base | Incite-Inst. | LLaMA-2 | MPT | OPT | Vicuna |
|---|---|---|---|---|---|---|---|---|
| cuad_ip_ ownership_ assignment | 53.0 | 65.8 | 61.1 | 73.8 | 51.6 | 60.4 | 70.8 | 14.4 |
| cuad_ irrevocable_ or_ perpetual_ license | 57.5 | 58.6 | 72.1 | 83.2 | 52.9 | 59.6 | 80.0 | 46.1 |
| cuad_ joint_ip_ ownership | 55.2 | 58.3 | 60.9 | 74.5 | 50.0 | 55.2 | 72.4 | 30.2 |
| cuad_ license_ grant | 62.7 | 60.3 | 65.5 | 77.0 | 63.0 | 39.0 | 67.9 | 20.8 |
| cuad_ liquidated_ damages | 54.1 | 65.0 | 65.0 | 70.5 | 50.0 | 51.4 | 57.7 | 36.4 |
| cuad_ minimum_ commitment | 51.4 | 58.4 | 55.3 | 53.8 | 49.9 | 55.3 | 59.2 | 31.3 |
| cuad_ most_ favored_ nation | 51.6 | 48.4 | 60.9 | 59.4 | 51.6 | 57.8 | 48.4 | 23.4 |
| cuad_no-solicit_of_ customers | 47.6 | 38.1 | 61.9 | 77.4 | 50.0 | 51.2 | 27.4 | 22.6 |
| cuad_no-solicit_of_ employees | 39.4 | 35.9 | 54.9 | 80.3 | 69.0 | 70.4 | 42.3 | 19.7 |
| cuad_non-compete | 32.8 | 42.5 | 55.2 | 67.4 | 63.3 | 52.3 | 29.0 | 42.3 |
| cuad_non-disparagement | 41.0 | 50.0 | 64.0 | 70.0 | 64.0 | 51.0 | 47.0 | 34.0 |
| cuad_non-transferable_ license | 67.2 | 65.9 | 56.5 | 78.0 | 50.0 | 48.5 | 68.5 | 27.9 |
| cuad_ notice_ period_to_ terminate_ renewal | 50.0 | 58.6 | 50.5 | 76.6 | 50.0 | 50.9 | 81.5 | 35.6 |
| cuad_post-termination_ services | 48.1 | 45.7 | 60.0 | 57.2 | 50.0 | 64.7 | 57.7 | 20.0 |
| cuad_ price_ restrictions | 58.7 | 43.5 | 45.7 | 58.7 | 50.0 | 54.3 | 52.2 | 41.3 |
| cuad_ renewal_ term | 50.3 | 57.3 | 45.1 | 75.6 | 50.0 | 53.1 | 82.1 | 35.2 |

| Task | BLOOM | Falcon | Incite-Base | Incite-Inst. | LLaMA-2 | MPT | OPT | Vicuna |
|------|-------|--------|-------------|--------------|---------|-----|-----|--------|
| cuad_ revenue-profit_ sharing | 54.3 | 57.9 | 50.6 | 65.2 | 52.3 | 50.0 | 62.4 | 11.1 |
| cuad_rofr-rofo-rofn | 54.6 | 43.3 | 55.9 | 57.0 | 50.0 | 59.9 | 53.0 | 20.6 |
| cuad_ source_ code_ escrow | 75.4 | 59.3 | 62.7 | 65.3 | 52.5 | 51.7 | 79.7 | 24.6 |
| cuad_ termination_ for_ convenience | 70.5 | 67.7 | 47.9 | 83.7 | 50.0 | 49.8 | 64.4 | 42.8 |
| cuad_ third_ party_ beneficiary | 70.6 | 50.0 | 69.1 | 79.4 | 50.0 | 67.6 | 70.6 | 29.4 |
| cuad_ uncapped_ liability | 46.9 | 53.4 | 53.4 | 61.2 | 51.0 | 77.9 | 36.4 | 33.0 |
| cuad_ unlimited-all-you-can-eat-license | 72.9 | 62.5 | 75.0 | 79.2 | 62.5 | 60.4 | 72.9 | 41.7 |
| cuad_ volume_ restriction | 52.8 | 47.5 | 55.0 | 54.0 | 50.9 | 52.5 | 63.0 | 42.2 |
| cuad_ warranty_ duration | 67.5 | 59.4 | 57.8 | 64.1 | 51.9 | 54.4 | 65.3 | 5.3 |
| insurance_ policy_ interpretation | 34.5 | 42.8 | 35.4 | 36.9 | 43.2 | 34.9 | 33.7 | 29.1 |
| jcrew_ blocker | 45.6 | 46.7 | 47.8 | 51.1 | 58.9 | 57.8 | 56.7 | 11.1 |
| maud_ ability_to_ consummate_ concept_ is_ subject_ to_mae_ carveouts | 42.3 | 48.2 | 30.9 | 50.0 | 50.0 | 0.9 | 50.5 | 45.5 |
| maud_ financial_ point_of_ view_is_ the_sole_ consideration | 48.5 | 56.6 | 45.9 | 43.9 | 48.5 | 4.1 | 63.3 | 50.0 |

| Task | BLOOM | Falcon | Incite-Base | Incite-Inst. | LLaMA-2 | MPT | OPT | Vicuna |
|---|---|---|---|---|---|---|---|---|
| maud_ accuracy_ of_ fundamental_ target_ rws_ bringdown_ standard | 33.3 | 34.5 | 31.7 | 35.9 | 34.4 | 31.6 | 37.5 | 33.3 |
| maud_ accuracy_ of_target_ general_ rw_ bringdown_ timing_ answer | 50.0 | 45.8 | 53.7 | 48.3 | 51.7 | 50.8 | 53.9 | 50.0 |
| maud_ accuracy_ of_target_ capitalization_ rw_ (outstanding_ shares)_ bringdown_ standard_ answer | 30.7 | 30.2 | 25.1 | 20.2 | 22.1 | 22.3 | 24.7 | 18.1 |
| maud_ additional_ matching_ rights_ period_ for_ modifications_ (cor) | 19.5 | 18.3 | 21.1 | 16.3 | 22.0 | 8.4 | 20.3 | 20.0 |
| maud_ application_ of_buyer_ consent_ requirement_ (negative_ interim_ covenant) | 47.8 | 40.0 | 50.0 | 55.6 | 50.9 | 50.0 | 47.5 | 43.1 |
| maud_ buyer_ consent_ requirement_ (ordinary_ course) | 50.6 | 54.1 | 49.7 | 50.0 | 50.0 | 7.4 | 57.5 | 50.0 |
| maud_ change_ in_law__ subject_ to_ disproportionate_ impact_ modifier | 36.4 | 45.8 | 11.2 | 49.4 | 50.6 | 12.2 | 50.0 | 50.0 |

| Task | BLOOM | Falcon | Incite-Base | Incite-Inst. | LLaMA-2 | MPT | OPT | Vicuna |
|---|---|---|---|---|---|---|---|---|
| maud_ changes_ in_gaap_ or_other_ accounting_ principles_ _subject_ to_ disproportionate_ impact_ modifier | 35.9 | 48.4 | 10.0 | 53.3 | 50.6 | 7.1 | 50.0 | 50.0 |
| maud_ cor_ permitted_ in_ response_ to_ intervening_ event | 47.6 | 62.5 | 47.5 | 49.4 | 58.6 | 7.3 | 65.6 | 50.7 |
| maud_ cor_ permitted_ with_ board_ fiduciary_ determination_ only | 45.8 | 42.1 | 37.2 | 48.8 | 50.0 | 13.5 | 49.4 | 49.4 |
| maud_ cor_ standard_ (intervening_ event) | 26.7 | 9.3 | 10.9 | 16.7 | 21.6 | 13.8 | 16.7 | 16.7 |
| maud_ cor_ standard_ (superior_ offer) | 16.8 | 7.8 | 7.0 | 15.8 | 17.8 | 9.7 | 11.5 | 6.4 |
| maud_ definition_ contains_ knowledge_ requirement_ -_answer | 25.3 | 26.9 | 20.8 | 23.1 | 22.8 | 28.0 | 25.3 | 25.0 |
| maud_ definition_ includes_ asset_ deals | 23.9 | 29.4 | 18.2 | 28.9 | 32.8 | 6.8 | 35.1 | 28.6 |
| maud_ definition_ includes_ stock_ deals | 27.7 | 16.4 | 5.0 | 24.5 | 18.4 | 7.6 | 17.2 | 22.6 |

| Task | BLOOM | Falcon | Incite-Base | Incite-Inst. | LLaMA-2 | MPT | OPT | Vicuna |
|---|---|---|---|---|---|---|---|---|
| maud_fiduciary_exception__board_determination_standard | 4.3 | 9.6 | 6.9 | 12.7 | 14.6 | 9.9 | 13.7 | 1.0 |
| maud_fiduciary_exception_board_determination_trigger_(no_shop) | 48.8 | 44.7 | 43.3 | 49.3 | 55.8 | 20.1 | 57.0 | 42.1 |
| maud_fls_(mae)_standard | 38.3 | 16.0 | 13.0 | 28.6 | 1.8 | 1.8 | 24.6 | 0.0 |
| maud_general_economic_and_financial_conditions_subject_to_disproportionate_impact_modifier | 54.8 | 62.5 | 38.1 | 48.8 | 50.6 | 14.3 | 50.0 | 50.0 |
| maud_includes_consistent_with_past_practice | 50.0 | 48.2 | 61.6 | 49.6 | 52.5 | 5.4 | 52.8 | 50.0 |
| maud_initial_matching_rights_period_(cor) | 20.7 | 11.2 | 21.5 | 23.4 | 27.2 | 10.8 | 21.0 | 18.9 |
| maud_initial_matching_rights_period_(ftr) | 20.6 | 14.5 | 16.4 | 11.1 | 22.8 | 7.7 | 20.0 | 16.9 |
| maud_intervening_event_-_required_to_occur_after_signing_-_answer | 50.2 | 51.5 | 43.4 | 50.0 | 42.6 | 43.5 | 47.5 | 50.0 |
| maud_knowledge_definition | 47.1 | 46.8 | 40.0 | 46.7 | 51.8 | 49.3 | 50.7 | 45.5 |

| Task | BLOOM | Falcon | Incite-Base | Incite-Inst. | LLaMA-2 | MPT | OPT | Vicuna |
|---|---|---|---|---|---|---|---|---|
| maud_ liability_ standard_ for_no-shop_ breach_ by_target_ non-do_ representatives | 59.6 | 50.0 | 50.0 | 53.8 | 50.0 | 46.8 | 58.3 | 50.0 |
| maud_ ordinary_ course_ efforts_ standard | 34.2 | 32.8 | 41.9 | 32.9 | 77.4 | 32.7 | 34.2 | 33.8 |
| maud_ pandemic_ or_other_ public_ health_ event__ subject_ to_ disproportionate_ impact_ modifier | 48.2 | 10.8 | 49.4 | 48.9 | 51.3 | 8.5 | 47.5 | 49.3 |
| maud_ pandemic_ or_other_ public_ health_ event_ specific_ reference_ to_ pandemic-related_ governmental_ responses_ or_ measures | 41.7 | 51.9 | 46.0 | 48.1 | 50.0 | 4.2 | 50.8 | 50.0 |
| maud_ relational_ language_ (mae)_ applies_to | 37.0 | 47.1 | 18.8 | 50.0 | 7.1 | 16.5 | 28.3 | 46.4 |
| maud_ specific_ performance | 60.4 | 50.0 | 49.2 | 43.5 | 58.5 | 0.6 | 39.4 | 50.0 |
| maud_ tail_ period_ length | 6.9 | 32.0 | 25.3 | 31.4 | 63.9 | 14.2 | 8.6 | 3.1 |
| maud_ type_of_ consideration | 25.0 | 28.5 | 27.1 | 26.4 | 25.3 | 39.3 | 24.3 | 25.0 |
| opp115_ data_ retention | 48.9 | 37.5 | 46.6 | 50.0 | 50.0 | 50.0 | 51.1 | 42.0 |

| Task | BLOOM | Falcon | Incite-Base | Incite-Inst. | LLaMA-2 | MPT | OPT | Vicuna |
|---|---|---|---|---|---|---|---|---|
| opp115_ data_ security | 53.4 | 60.3 | 53.8 | 63.6 | 50.2 | 49.8 | 63.7 | 45.9 |
| opp115_ do_not_ track | 43.6 | 34.5 | 47.3 | 69.1 | 50.0 | 49.1 | 31.8 | 45.5 |
| opp115_ first_ party_ collection_ use | 63.9 | 59.3 | 69.5 | 69.9 | 61.3 | 59.2 | 70.4 | 46.4 |
| opp115_ international_ and_ specific_ audiences | 58.1 | 60.3 | 50.1 | 64.9 | 51.1 | 52.7 | 57.5 | 37.4 |
| opp115_ policy_ change | 68.2 | 66.2 | 55.6 | 55.9 | 50.0 | 50.0 | 74.5 | 44.0 |
| opp115_ third_ party_ sharing_ collection | 53.8 | 57.7 | 50.9 | 68.4 | 50.4 | 50.3 | 54.1 | 36.5 |
| opp115_ user_ access,_ edit_and_ deletion | 60.2 | 51.5 | 56.1 | 64.5 | 54.7 | 50.8 | 59.0 | 41.7 |
| opp115_ user_ choice_ control | 63.6 | 40.0 | 46.5 | 64.5 | 50.2 | 48.6 | 47.4 | 44.2 |
| privacy_ policy_ entailment | 51.4 | 49.3 | 56.8 | 58.1 | 57.7 | 64.1 | 52.8 | 26.6 |
| privacy_ policy_qa | 50.0 | 49.7 | 52.0 | 56.3 | 50.2 | 50.0 | 50.4 | 0.1 |
| proa | 52.1 | 56.2 | 50.0 | 71.6 | 52.1 | 58.3 | 51.0 | 47.9 |
| ssla_ company_ defendants | 35.5 | 44.9 | 54.8 | 54.1 | 60.7 | 51.0 | 50.9 | 0.0 |
| ssla_ individual_ defendants | 14.3 | 17.4 | 20.8 | 19.6 | 23.1 | 20.8 | 13.9 | 0.0 |
| ssla_ plaintiff | 26.1 | 9.3 | 52.3 | 64.4 | 60.5 | 76.0 | 59.8 | 0.0 |
| sara_ entailment | 50.4 | 50.0 | 50.0 | 51.1 | 50.0 | 50.0 | 50.0 | 0.0 |
| sara_ numeric | 2.1 | 1.0 | 2.1 | 0.0 | 0.0 | 2.1 | 1.0 | 0.0 |

| Task | BLOOM | Falcon | Incite-Base | Incite-Inst. | LLaMA-2 | MPT | OPT | Vicuna |
|---|---|---|---|---|---|---|---|---|
| supply_ chain_ disclosure_ best_ practice_ accountability | 6.2 | 55.8 | 50.0 | 58.4 | 50.4 | 56.8 | 32.7 | 31.5 |
| supply_ chain_ disclosure_ best_ practice_ audits | 1.5 | 71.6 | 55.9 | 63.0 | 55.8 | 42.2 | 31.1 | 8.7 |
| supply_ chain_ disclosure_ best_ practice_ certification | 2.8 | 67.1 | 52.1 | 57.6 | 52.3 | 40.8 | 51.5 | 11.3 |
| supply_ chain_ disclosure_ best_ practice_ training | 6.6 | 52.3 | 50.9 | 55.4 | 49.3 | 56.2 | 50.5 | 30.7 |
| supply_ chain_ disclosure_ best_ practice_ verification | 4.5 | 55.8 | 49.8 | 54.3 | 49.1 | 36.0 | 38.1 | 13.9 |
| supply_ chain_ disclosure_ disclosed_ accountability | 2.1 | 48.2 | 49.7 | 48.7 | 49.0 | 24.6 | 20.4 | 15.7 |
| supply_ chain_ disclosure_ disclosed_ audits | 3.6 | 50.2 | 48.7 | 49.6 | 51.5 | 52.0 | 49.7 | 16.3 |
| supply_ chain_ disclosure_ disclosed_ certification | 3.2 | 50.5 | 52.0 | 52.2 | 54.9 | 45.8 | 32.3 | 20.1 |
| supply_ chain_ disclosure_ disclosed_ training | 5.6 | 48.6 | 49.1 | 49.2 | 49.3 | 44.0 | 39.5 | 10.1 |
| supply_ chain_ disclosure_ disclosed_ verification | 5.3 | 48.9 | 49.6 | 47.6 | 48.6 | 48.5 | 49.3 | 23.1 |
| unfair_tos | 8.3 | 12.3 | 12.7 | 8.7 | 10.6 | 11.0 | 10.5 | 0.0 |

Table 81: 3B models on interpretation tasks.

| Task | BLOOM | Flan | Incite | Opt |
|------|-------|------|--------|-----|
| consumer_contracts_qa | 0.8 | 93.2 | 46.1 | 31.7 |
| contract_nli_confidentiality_of_agreement | 57.3 | 86.6 | 56.1 | 46.3 |
| contract_nli_explicit_identification | 57.2 | 81.2 | 63.6 | 53.3 |
| contract_nli_inclusion_of_verbally_conveyed_information | 50.1 | 82.2 | 67.8 | 60.9 |
| contract_nli_limited_use | 63.1 | 72.8 | 60.1 | 63.0 |
| contract_nli_no_licensing | 50.0 | 65.1 | 43.5 | 38.5 |
| contract_nli_notice_on_compelled_disclosure | 57.0 | 66.2 | 65.5 | 71.8 |
| contract_nli_permissible_acquirement_of_similar_information | 59.6 | 69.1 | 46.6 | 54.5 |
| contract_nli_permissible_copy | 51.3 | 83.7 | 55.3 | 56.2 |
| contract_nli_permissible_development_of_similar_information | 60.3 | 75.0 | 49.3 | 59.6 |
| contract_nli_permissible_post-agreement_possession | 43.8 | 66.3 | 43.4 | 42.4 |
| contract_nli_return_of_confidential_information | 51.6 | 82.3 | 61.3 | 51.9 |
| contract_nli_sharing_with_employees | 53.3 | 69.7 | 47.1 | 52.8 |
| contract_nli_sharing_with_third-parties | 50.0 | 80.5 | 51.4 | 49.6 |
| contract_nli_survival_of_obligations | 50.0 | 72.9 | 49.4 | 44.1 |
| contract_qa | 35.9 | 0.0 | 82.9 | 44.6 |
| cuad_affiliate_license-licensee | 59.6 | 66.7 | 60.1 | 68.7 |
| cuad_affiliate_license-licensor | 48.9 | 77.3 | 71.6 | 50.0 |
| cuad_anti-assignment | 50.0 | 74.7 | 51.5 | 33.7 |
| cuad_audit_rights | 50.0 | 56.3 | 56.2 | 64.6 |
| cuad_cap_on_liability | 49.2 | 50.9 | 49.4 | 26.8 |
| cuad_change_of_control | 51.4 | 59.6 | 60.8 | 47.4 |
| cuad_competitive_restriction_exception | 46.4 | 73.2 | 45.5 | 40.5 |
| cuad_covenant_not_to_sue | 50.0 | 59.1 | 49.4 | 44.8 |
| cuad_effective_date | 51.7 | 92.4 | 47.0 | 65.7 |
| cuad_exclusivity | 64.6 | 55.2 | 53.9 | 60.9 |
| cuad_expiration_date | 50.2 | 67.7 | 52.1 | 69.3 |
| cuad_governing_law | 56.8 | 84.4 | 50.5 | 74.8 |
| cuad_insurance | 53.5 | 53.9 | 57.5 | 70.3 |
| cuad_ip_ownership_assignment | 49.0 | 57.6 | 57.3 | 65.1 |
| cuad_irrevocable_or_perpetual_license | 51.8 | 76.1 | 53.6 | 71.1 |
| cuad_joint_ip_ownership | 48.4 | 65.1 | 64.6 | 68.8 |
| cuad_license_grant | 50.1 | 69.1 | 51.4 | 69.2 |
| cuad_liquidated_damages | 50.0 | 52.3 | 57.3 | 34.5 |
| cuad_minimum_commitment | 52.7 | 54.1 | 50.6 | 57.0 |
| cuad_most_favored_nation | 50.0 | 57.8 | 56.2 | 43.8 |
| cuad_no-solicit_of_customers | 52.4 | 47.6 | 59.5 | 32.1 |
| cuad_no-solicit_of_employees | 48.6 | 63.4 | 66.9 | 42.3 |
| cuad_non-compete | 48.0 | 63.1 | 56.1 | 30.3 |
| cuad_non-disparagement | 49.0 | 57.0 | 60.0 | 44.0 |

| Task | BLOOM | Flan | Incite | Opt |
|---|---|---|---|---|
| cuad_non-transferable_license | 56.8 | 58.7 | 50.9 | 62.0 |
| cuad_notice_period_to_terminate_renewal | 59.5 | 61.3 | 54.5 | 68.5 |
| cuad_post-termination_services | 50.0 | 64.0 | 50.7 | 55.9 |
| cuad_price_restrictions | 56.5 | 50.0 | 58.7 | 47.8 |
| cuad_renewal_term | 50.3 | 70.7 | 59.8 | 73.3 |
| cuad_revenue-profit_sharing | 54.1 | 60.9 | 52.5 | 64.1 |
| cuad_rofr-rofo-rofn | 50.3 | 49.4 | 51.6 | 55.4 |
| cuad_source_code_escrow | 55.9 | 42.4 | 70.3 | 56.8 |
| cuad_termination_for_convenience | 50.0 | 64.4 | 58.1 | 38.4 |
| cuad_third_party_beneficiary | 58.8 | 79.4 | 50.0 | 64.7 |
| cuad_uncapped_liability | 50.0 | 53.4 | 49.7 | 28.9 |
| cuad_unlimited-all-you-can-eat-license | 66.7 | 81.2 | 54.2 | 70.8 |
| cuad_volume_restriction | 50.3 | 49.4 | 57.5 | 59.6 |
| cuad_warranty_duration | 53.1 | 61.9 | 51.6 | 54.4 |
| insurance_policy_interpretation | 32.2 | 38.8 | 37.1 | 32.9 |
| jcrew_blocker | 55.6 | 56.7 | 48.9 | 53.3 |
| maud_ability_to_consummate_concept_is_subject_to_mae_carveouts | 47.3 | 13.4 | 50.0 | 49.1 |
| maud_financial_point_of_view_is_the_sole_consideration | 53.1 | 49.0 | 50.0 | 43.4 |
| maud_accuracy_of_fundamental_target_rws_bringdown_standard | 32.6 | 0.0 | 32.6 | 34.8 |
| maud_accuracy_of_target_general_rw_bringdown_timing_answer | 50.0 | 0.0 | 48.2 | 53.2 |
| maud_accuracy_of_target_capitalization_rw_(outstanding_shares)_bringdown_standard_answer | 26.1 | 0.0 | 23.4 | 28.2 |
| maud_additional_matching_rights_period_for_modifications_(cor) | 20.0 | 0.0 | 18.8 | 20.3 |
| maud_application_of_buyer_consent_requirement_(negative_interim_covenant) | 49.1 | 62.5 | 50.3 | 50.0 |
| maud_buyer_consent_requirement_(ordinary_course) | 49.4 | 50.0 | 50.0 | 49.7 |
| maud_change_in_law__subject_to_disproportionate_impact_modifier | 44.3 | 6.3 | 44.0 | 51.1 |
| maud_changes_in_gaap_or_other_accounting_principles__subject_to_disproportionate_impact_modifier | 41.9 | 5.9 | 51.0 | 47.9 |
| maud_cor_permitted_in_response_to_intervening_event | 51.2 | 48.8 | 71.7 | 50.6 |
| maud_cor_permitted_with_board_fiduciary_determination_only | 49.4 | 36.3 | 50.0 | 45.9 |
| maud_cor_standard_(intervening_event) | 16.7 | 1.0 | 16.7 | 16.7 |
| maud_cor_standard_(superior_offer) | 10.8 | 4.1 | 8.8 | 5.4 |
| maud_definition_contains_knowledge_requirement_-_answer | 24.7 | 0.0 | 26.2 | 25.0 |
| maud_definition_includes_asset_deals | 34.4 | 0.0 | 22.5 | 33.3 |
| maud_definition_includes_stock_deals | 40.4 | 0.0 | 9.9 | 0.0 |
| maud_fiduciary_exception__board_determination_standard | 6.3 | 0.5 | 12.9 | 13.3 |
| maud_fiduciary_exception_board_determination_trigger_(no_shop) | 56.7 | 50.0 | 50.5 | 50.0 |
| maud_fls_(mae)_standard | 23.9 | 10.5 | 24.2 | 1.2 |

| Task | BLOOM | Flan | Incite | Opt |
|------|-------|------|--------|-----|
| maud_general_economic_and_financial_conditions_subject_to_disproportionate_impact_modifier | 50.0 | 3.0 | 66.1 | 53.0 |
| maud_includes_consistent_with_past_practice | 50.4 | 89.7 | 50.0 | 43.3 |
| maud_initial_matching_rights_period_(cor) | 13.5 | 0.0 | 18.0 | 17.0 |
| maud_initial_matching_rights_period_(ftr) | 18.9 | 0.0 | 19.6 | 19.4 |
| maud_intervening_event_-_required_to_occur_after_signing_-_answer | 48.5 | 3.3 | 50.0 | 50.0 |
| maud_knowledge_definition | 46.0 | 50.0 | 47.5 | 49.4 |
| maud_liability_standard_for_no-shop_breach_by_target_non-do_representatives | 50.0 | 54.5 | 50.0 | 50.0 |
| maud_ordinary_course_efforts_standard | 33.3 | 58.3 | 32.1 | 33.3 |
| maud_pandemic_or_other_public_health_event__subject_to_disproportionate_impact_modifier | 2.6 | 62.3 | 60.4 | 47.4 |
| maud_pandemic_or_other_public_health_event_specific_reference_to_pandemic-related_governmental_responses_or_measures | 50.0 | 59.9 | 50.0 | 50.0 |
| maud_relational_language_(mae)_applies_to | 32.6 | 0.0 | 50.0 | 49.3 |
| maud_specific_performance | 50.0 | 89.3 | 48.0 | 50.0 |
| maud_tail_period_length | 1.5 | 0.0 | 35.6 | 25.0 |
| maud_type_of_consideration | 24.4 | 2.1 | 27.2 | 25.2 |
| opp115_data_retention | 45.5 | 52.3 | 64.8 | 39.8 |
| opp115_data_security | 60.7 | 69.7 | 55.9 | 53.6 |
| opp115_do_not_track | 45.5 | 55.5 | 42.7 | 37.3 |
| opp115_first_party_collection_use | 60.9 | 52.3 | 69.5 | 56.1 |
| opp115_international_and_specific_audiences | 59.9 | 70.5 | 51.8 | 57.0 |
| opp115_policy_change | 52.1 | 60.4 | 61.2 | 57.4 |
| opp115_third_party_sharing_collection | 55.6 | 63.3 | 64.8 | 47.0 |
| opp115_user_access,_edit_and_deletion | 49.2 | 82.3 | 66.4 | 53.5 |
| opp115_user_choice_control | 48.5 | 58.5 | 59.0 | 47.5 |
| privacy_policy_entailment | 50.0 | 53.6 | 66.1 | 54.1 |
| privacy_policy_qa | 50.3 | 61.0 | 55.1 | 50.5 |
| proa | 54.1 | 80.1 | 64.5 | 53.1 |
| ssla_company_defendants | 36.2 | 13.1 | 51.1 | 47.0 |
| ssla_individual_defendants | 11.9 | 20.0 | 20.5 | 17.3 |
| ssla_plaintiff | 40.1 | 86.4 | 36.8 | 42.8 |
| sara_entailment | 48.5 | 0.0 | 50.7 | 48.9 |
| sara_numeric | 3.1 | 1.0 | 0.0 | 1.0 |
| supply_chain_disclosure_best_practice_accountability | 43.0 | 74.5 | 55.1 | 37.4 |
| supply_chain_disclosure_best_practice_audits | 42.7 | 74.3 | 61.0 | 54.6 |
| supply_chain_disclosure_best_practice_certification | 46.8 | 77.4 | 52.0 | 46.6 |
| supply_chain_disclosure_best_practice_training | 44.9 | 83.4 | 64.2 | 41.0 |
| supply_chain_disclosure_best_practice_verification | 47.7 | 54.9 | 53.1 | 21.8 |
| supply_chain_disclosure_disclosed_accountability | 46.9 | 77.7 | 60.6 | 14.6 |

Table 81 – continued from previous page

| Task | BLOOM | Flan | Incite | Opt |
|---|---|---|---|---|
| supply_chain_disclosure_disclosed_audits | 46.5 | 73.0 | 57.3 | 2.6 |
| supply_chain_disclosure_disclosed_certification | 45.3 | 77.3 | 52.0 | 12.6 |
| supply_chain_disclosure_disclosed_training | 47.9 | 87.2 | 58.8 | 16.9 |
| supply_chain_disclosure_disclosed_verification | 44.1 | 66.1 | 55.3 | 0.2 |
| unfair_tos | 9.0 | 8.2 | 4.0 | 15.3 |

