[147] Sean Welleck, Jiacheng Liu, Ronan Le Bras, Hannaneh Hajishirzi, Yejin Choi, and Kyunghyun Cho. Naturalproofs: Mathematical theorem proving in natural language. *arXiv preprint arXiv:2104.01112*, 2021.

[148] Hannes Westermann, Jaromir Savelka, and Karim Benyekhlef. Llmediator: Gpt-4 assisted online dispute resolution. *arXiv preprint arXiv:2307.16732*, 2023.

[149] Wikipedia. Irac. https://en.wikipedia.org/wiki/IRAC.

[150] Shomir Wilson, Florian Schaub, Aswarth Abhilash Dara, Frederick Liu, Sushain Cherivirala, Pedro Giovanni Leon, Mads Schaarup Andersen, Sebastian Zimmeck, Kanthashree Mysore Sathyendra, N Cameron Russell, et al. The creation and analysis of a website privacy policy corpus. In *Proceedings of the 54th Annual Meeting of the Association for Computational Linguistics (Volume 1: Long Papers)*, pages 1330–1340, 2016.

[151] Lawrence Wrightsman. *Oral arguments before the Supreme Court: An empirical approach*. Oxford University Press, 2008.

[152] Chaojun Xiao, Haoxi Zhong, Zhipeng Guo, Cunchao Tu, Zhiyuan Liu, Maosong Sun, Yansong Feng, Xianpei Han, Zhen Hu, Heng Wang, et al. Cail2018: A large-scale legal dataset for judgment prediction. *arXiv preprint arXiv:1807.02478*, 2018.

[153] Can Xu, Qingfeng Sun, Kai Zheng, Xiubo Geng, Pu Zhao, Jiazhan Feng, Chongyang Tao, and Daxin Jiang. Wizardlm: Empowering large language models to follow complex instructions, 2023.

[154] Fangyi Yu, Lee Quartey, and Frank Schilder. Legal prompting: Teaching a language model to think like a lawyer. *arXiv preprint arXiv:2212.01326*, 2022.

[155] Fangyi Yu, Lee Quartey, and Frank Schilder. Exploring the effectiveness of prompt engineering for legal reasoning tasks. In *Findings of the Association for Computational Linguistics: ACL 2023*, pages 13582–13596, 2023.

[156] Diego Zambrano, Neel Guha, Austin Peters, and Jeffrey Xia. Private enforcement in the states. *University of Pennsylvania Law Review, forthcoming*, 2023.

[157] Susan Zhang, Stephen Roller, Naman Goyal, Mikel Artetxe, Moya Chen, Shuohui Chen, Christopher Dewan, Mona Diab, Xian Li, Xi Victoria Lin, et al. Opt: Open pre-trained transformer language models. *arXiv preprint arXiv:2205.01068*, 2022.

[158] Lucia Zheng, Neel Guha, Brandon R Anderson, Peter Henderson, and Daniel E Ho. When Does Pretraining Help? Assessing Self-Supervised Learning for Law and the CaseHOLD Dataset of 53,000+ Legal Holdings. In *Proceedings of the Eighteenth International Conference on Artificial Intelligence and Law*, pages 159–168, 2021.

[159] Haoxi Zhong, Zhipeng Guo, Cunchao Tu, Chaojun Xiao, Zhiyuan Liu, and Maosong Sun. Legal judgment prediction via topological learning. In *Proceedings of the 2018 conference on empirical methods in natural language processing*, pages 3540–3549, 2018.

[160] Caleb Ziems, William Held, Omar Shaikh, Jiao Chen, Zhehao Zhang, and Diyi Yang. Can large language models transform computational social science? *arXiv preprint arXiv:2305.03514*, 2023.

[161] Lee B. Ziffer. The robots are coming: Ai large language models and the legal profession. *American Bar Association*, 2023.

[162] Sebastian Zimmeck, Peter Story, Daniel Smullen, Abhilasha Ravichander, Ziqi Wang, Joel R Reidenberg, N Cameron Russell, and Norman Sadeh. Maps: Scaling privacy compliance analysis to a million apps. *Proc. Priv. Enhancing Tech.*, 2019:66, 2019.

## Acknowledgments and Disclosure of Funding

Full acknowledgements and disclosures will be included as part of the camera-ready version.

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

—like forms of document review [62, 142] or case summarization [120, 121, 90, 69]—and developed benchmarks to assess the performance of current state-of-the-art techniques. Here, research has focused on the aspects of legal text which are often challenging for NLP methods, like the length of documents or the presence of jargon [21, 111, 87, 41, 79]. A second line of work has focused on developing tasks to evaluate forms of inferential reasoning common to law [21]. This includes, for instance, tasks which require a model to identify the best supporting statement for an argument [80, 158], or perform statutory reasoning [65]. Other work has focused on creating datasets for pretraining models [60, 129], non-English/multilingual tasks [95, 96, 66, 152, 52, 94, 70, 20, 23, 103, 18], legal judgement prediction [89, 38, 17, 159], legal role labeling [85], and different forms of retrieval [68].

Importantly, the majority of previous benchmarking efforts have focused on language models which learn by supervised training or finetuning (e.g., BERT variants [45]), and researchers have consequently studied questions related to the role of domain specific datasets [158, 19, 20]. More recently, researchers have begun to ask whether *large* language models (LLMs) like GPT-3/4 can perform legal reasoning [73, 154, 67, 12, 29, 31, 155], citing to evidence of these models' capacity to perform sophisticated reasoning tasks in domains like math or programming [145, 24]. Unlike BERT-based models, LLMs are evaluated on their ability to learn tasks *in-context*, primarily through prompting. While a few works have experimented with LLMs on existing benchmarks [16, 12], most evaluations focus on standardized tests or other exam equivalents [71, 93, 30]. Studies have explored the role of prompt-engineering [155, 154, 75], potential applications [93, 29, 148, 117, 116], questions regarding human-LLM interaction [31, 63], and comparisons to older finetuned-models [91].

LEGALBENCH builds on prior work in several ways. First, LEGALBENCH enhances opportunities to study legal reasoning in LLMs, by making available 162 evaluation tasks. LEGALBENCH systematizes and standardizes these tasks for LLM evaluation, specifying potential prompts, in-context demonstrations, and metrics. Second, LEGALBENCH presents a framework for organizing and comparing tasks, allowing researchers to identify trends in performance across groupings of tasks. This enables researchers, for instance, to distinguish between task types for which current LLMs are highly performant, and task types for which further work is needed.

A notable consequence of focusing on few-shot LLMs is that LEGALBENCH can contribute a much more diverse set of legal reasoning tasks. Traditional NLP methods require a large training set and a smaller evaluation set. The cost of legal annotations means that constructing benchmarks has required extraordinary financial investment [62, 120] or a "natural" source of existing annotations [158, 21]. Because the few-shot prompting regime requires only a few labeled demonstrations, creating large training sets isn't necessary, and the effort they otherwise would have consumed can be allocated towards developing new tasks.

## F.2    Connections to other LLM benchmarking efforts

We highlight connections to two broader research efforts. First, we draw inspiration from existing efforts within NLP and machine learning to define fine-grained measures of performance, which allow researchers to discuss model capabilities with precision and specificity. Examples include the diagnostic set of the GLUE Benchmark [141], the "reasoning patterns" studied in [100], the task organization used in HELM [80], and the BigBench effort [123]. Fine-grained measurements are valuable because they allow researchers to identify how particular modifications to model architectures or training regimes affect performance. They hold particular value for the field of legal NLP, in which researchers continue to debate how best to specialize language models to the domain [60, 158, 56]

We additionally draw inspiration from other large-scale collaborative efforts in AI, including the BigBench project [123], and studies in medicine [28]. In particular, we believe that LEGALBENCH illustrates a new model of open-source and interdisciplinary collaboration between the legal and AI communities. To the extent that LLMs gain adoption for legal tasks, legal professionals will be primarily charged with supervising them and selecting application use-cases. Involving the legal community in the design and construction of evaluation tasks allows for the construction of benchmarks which are more responsive to their interests and information needs.

# G Task overview