# OpenReview forum: "LegalBench: A Collaboratively Built Benchmark for Measuring Legal Reasoning in Large Language Models"
_NeurIPS.cc/2023/Track/Datasets_and_Benchmarks — NeurIPS 2023 Datasets and Benchmarks Poster_

### Official Review · Reviewer_tmL2 · 2023-07-16
**Reveiw of LegalBench, a benchmark for evaluating LLMs for the legal domain**

**Rating:** 6
**Confidence:** 5
**Correctness:** The claims, dataset creation process …

**Strengths:**

1. LegalBench benchmark covers various types of reasoning that are required by legal professionals in day to day working. And the benchmark is created with the help of legal professionals so it covers the practical aspects of the legal profession.
2. The tasks are well organized (there is a typology of tasks) into various types of reasoning, this can be useful from the perspective of future research.

**Additional Feedback:**

Please see the points mentioned above.

**Clarity:**

The paper is well written and well organized. The paper motivates the benchmark well.

**Documentation:**

Authors describe details of the dataset and benchmark in detail in the paper. However,  authors do not share the code for reproducing the experiments. Also authors do not discuss about the license associated with the dataset. Also, authors should create a website/github for the benchmark.

**Ethics:**

The fact that LLMs are having such high performance on many of the tasks is indicative of the fact that tasks are relatively easy. Authors should discuss the consequences of this from societal aspects. Although authors claim that the idea is not to replace humans but nevertheless, if LLMs are performing so well on many of the tasks then many of such tasks could be automated, what will be the impact of such automation on society? Also, since LLMs are not perfect, what happens when LLMs go wrong?

**Limitations:**

1. The benchmark is created with the sole focus of testing LLMs, however, if in the future a new class of models come into being then these tasks may not serve a good purpose. The benchmark should be generic and there should be a clear separation between the tasks and systems that are used to solve those. The focus of the tasks should be promoting development of legal-tech and this can be independent of what techniques/models are used to achieve this goal.
2. The benchmark is mainly restricted to US legal system and it is not clear if a system/LLM is performing well on LegalBench, would it generalize well to legal systems of other countries as well? Authors should discuss about this and possibly evaluate on benchmarks for other legal domains (e.g., EU, India, China, etc.). Moreover, the benchmark is restricted to English language only. Authors briefly mention about this in the paper.
3. Currently, LegalBench involves tasks that are not subjective and result in an unambiguous outcome. However, in the practice of law, such cases are straight forward to solve. For the benchmark to be reflective of real life conditions it should also involve tasks which may involve subjective outcome. It would be nice to evaluate how LLMs perform over these. Moreover, it will serve a good benchmark for future LLMs.
4. Authors are proposing various tasks and some of these can have negative societal impact and some may not have much impact. Possibly, authors should discuss in detail which of the tasks are more prone to ethical considerations and what could be done to mitigate the impact.

**Opportunities For Improvement:**

Please see the limitations point 1, 2 and 3.

Authors show the results of various models on various tasks but it would be nice to see in depth error-analysis of LLMs on some of the tasks. It would be good to investigate if there are some patterns to mistakes made by LLMs and what could be done to mitigate these mistakes.

**Relation To Prior Work:**

Authors should also include brief descriptions about legal benchmarks of other legal systems and countries like China, India, etc. For example:

1. Chaojun Xiao, Haoxi Zhong, Zhipeng Guo, Cunchao Tu, Zhiyuan Liu, Maosong Sun, Yansong Feng, Xianpei Han, Zhen Hu, Heng Wang, et al. 2018. Cail2018: A large-scale legal dataset for judgment prediction.
2. Bingfeng Luo, Yansong Feng, Jianbo Xu, Xiang Zhang, and Dongyan Zhao. 2017. Learning to Predict Charges for Criminal Cases with Legal Basis, EMNLP
3. Vijit Malik, Rishabh Sanjay, Shubham Kumar Nigam, Kripabandhu Ghosh, Shouvik Kumar Guha, Arnab Bhattacharya, and Ashutosh Modi. 2021. ILDC for CJPE: Indian Legal Documents Corpus for Court Judgment Prediction and Explanation, ACL.
4. Arnav Kapoor, Mudit Dhawan, Anmol Goel, Arjun T H, Akshala Bhatnagar, Vibhu Agrawal, Amul Agrawal, Arnab Bhattacharya, Ponnurangam Kumaraguru, and Ashutosh Modi, 2022. HLDC: Hindi Legal Documents Corpus, ACL Findings
5. Abhinav Joshi, Akshat Sharma, Sai Kiran Tanikella, and Ashutosh Modi, 2023, U-CREAT: Unsupervised Case Retrieval using Events extrAcTion, ACL 2023.
6. Vijit Malik, Rishabh Sanjay, Shouvik Kumar Guha, Angshuman Hazarika, Shubham Nigam, Arnab Bhattacharya, and Ashutosh Modi. 2022. Semantic Segmentation of Legal Documents via Rhetorical Roles.

**Summary And Contributions:**

This paper proposes a new benchmark (LegalBench) for evaluating LLMs for the legal domain. LegalBench consists of 162 different tasks that evaluate various types of legal reasoning. The benchmark is created with the help of legal professionals. Authors introduce a typology of various types of reasoning used for creating the tasks.

---

> ### Author Response · Authors · 2023-08-21
> **Response to tmL2**
>
> We thank tmL2 for their review. We are glad to hear they appreciated the organization of the benchmark and the manner in which it was developed.
>
> **In-depth analysis of errors.** We appreciate the reviewer highlighting the value of more in-depth error analysis, and the role this can play in surfacing systematic errors. In response, we have conducted a detailed error analysis of LLMs on several tasks. These are discussed briefly in  §5.3, and more comprehensively in Appendix J.
>
> **Focus on LLMs.** We agree with the reviewer that an ideal benchmark should be method agnostic so as to allow for a diverse class of models/methods, and that the broader goal of legal benchmarks should be to enable the development of legal tech.
>
> We decided to anchor LegalBench in LLMs because this class of models are currently the subject of large discussion within the legal and AI communities. At the same time, we emphasize that nothing in LegalBench precludes the evaluation of newer, yet-to-be-developed models, that operate in a few or zero-shot setting. LegalBench tasks have been contributed by the legal community because the tasks correspond to real-world applications, or implicate interesting types of legal reasoning. This will be true of the tasks regardless of whether LLMs are being used to perform them, or future-classes of models. We are comforted by the previous works published in the Datasets and Benchmarks track, which have similarly focused on large language models or foundation models, such as the FETA benchmark (2022) or the RAFT benchmark (2021).
>
> **Focus on US.** We thank the reviewer for noting the US-centric nature of the benchmark. Whether LLMs that perform well on tasks primarily corresponding to one legal system work well on tasks from other legal systems is an important question. We have updated the draft to emphasize the focus on the US legal system (see §1, and Limitations), and the need to answer questions pertaining to cross-jurisdiction generalization.
>
> **Subjective tasks.** We appreciate the reviewer noting the importance of legal tasks which entail subjectivity and ambiguity. In particular, we appreciate how LegalBench does not capture these tasks, and have updated our Limitations section to address this.
>
> At the same time, we emphasize that our selection of tasks is aimed to capture (at least in part) legal services of great importance to the community. We note in §1 that the difficulty in accessing legal services and extraordinary workload pressure on legal professionals means that understanding whether LLMs can improve performance or efficiency in performing unobjective/straight-forward tasks still has significant value. Appendix I provides a discussion of the significance and value for each task in the context of legal practice. Many of these correspond to real-life tasks that legal professionals must conduct.
>
> In addition, we emphasize that the inherent subjectivity of alternative tasks proves particular challenges for evaluation, given the absence of a concrete ground truth label. Most importantly, in order to accurately capture subjectivity, an adequate assessment would require a large number of expert evaluators to provide ground truth labels. This can be prohibitively costly. For instance, in a small qualitative study (https://www.juliannyarko.com/papers/expert_annotations.pdf), captured the subjectivity among senior lawyers in annotating a small set of only five contracts. They estimate that their efforts cost about $220,000. At the moment, it appears infeasible for academics to expand similar efforts at scale.
>
> **Social impact.** We acknowledge, as the reviewer has highlighted, the potential variation across tasks with regards to social impact. We have updated the Social Impact section in the Appendix to address this, and additionally refer to our response to the ethics review for additional details.
>
> **Documentation.** We thank the reviewer for bringing attention to certain aspects of the documentation.
>
> - Information about the licenses for each task can be found in Table 4.
> - A website for the benchmark can be found here: https://hazyresearch.stanford.edu/legalbench/.
> - Code for experiments and managing the benchmark can be found here: https://github.com/HazyResearch/legalbench/.
>
> **Prior work.** We thank the reviewer for identifying the missing references. We have added these and more to both the main-body literature review, and the expanded literature review in the Appendix. As a reference point, we note that the updated draft has nearly twice the number of references as the original submission.
>
> **Ethics discussion.** We thank the reviewer for noting these concerns, and agree that they constitute important issues for the use of LLMs in the legal domain. As these overlap with questions raised by the Ethics Review, we defer the reviewer to our discussion there.

---

### Official Review · Reviewer_XJRt · 2023-07-17
**Review for paper 413**

**Rating:** 7
**Confidence:** 4
**Clarity:** Yes, the paper is very clear and easy…

**Strengths:**

- The benchmark consists of 162 tasks, which is comprehensive and professional.
- The framework for legal reasoning is very clear, reasonable, and motivative.
- The supplementary material provides lots of details for this benchmark, which is very solid.


**Additional Feedback:**

None.

**Correctness:**

The evaluation methods and experiment design are appropriate and performed correctly.

**Documentation:**

The documentation is good.

**Ethics:**

I do not see any ethical concerns.


**Limitations:**

The limitations and and potential negative societal impact of their work are well addressed.

**Opportunities For Improvement:**

- It is better to evaluate more LLMs with different numbers of parameters (eg, LLaMA and BLOOM). It is interesting to investigate whether the model size will influence the results. Besides, if some legal domain specific models can be applied, that would be even better.
- I think comparing the performance between zero-shot and few-shot settings is also interesting.


**Relation To Prior Work:**

The related work is well discussed.

**Summary And Contributions:**

This paper proposes a new benchmark LegalBench for legal reasoning. The authors define a typology for organizing and describing legal tasks in terms of the reasoning types they require. Then, we collect data for legalBench that is a comprehensive benchmark with 162 tasks covering six different types of legal reasoning. The tasks are designed and hand-crafted by legal professionals, making the evaluation reasonable and convincing. Different types of LLMs are evaluated on this benchmark. The paper is clear and easy to understand.

---

> ### Author Response · Authors · 2023-08-21
> **Response to XJRt**
>
> We thank XJRt for their review. We are glad to hear they appreciated the comprehensive nature of the benchmark, the framework for organization, and the details provided in the Appendix.
>
> **Evaluation on additional models.** We appreciate the reviewer highlighting the benefits of additional evaluation. Our updated draft contains evaluation results on 20 models (5x as many as the original submission), including models from the LLaMa-2 family and BLOOM family.
>
> We note that as of August 1, 2023, we were unable to find English language legal LLMs that we can evaluate on.
>
> **Few-shot v. zero-shot.** We are grateful to the reviewer noting few-shot v. zero-shot, and we intend to add results on this question for the final draft. In initial experiments, we have observed significant variation in the gap between few-shot and zero-shot prompts depending on the reasoning category. For instance, zero-shot performance appears closest to few-shot performance on interpretation tasks, and farthest on issue-spotting tasks.

---

### Official Review · Reviewer_BYVU · 2023-07-21
**A few shot legal NLP benchmark that is neatly organized and well-documented**

**Rating:** 8
**Confidence:** 3
**Correctness:** Yes.
**Clarity:** Very well written.

**Strengths:**

- Timely contribution in the era of LLM.
- Neat organization of tasks from the view point of lawyers.
- GPT-4 and GPT-3.5 experimental results that gives the information of current status of AI on legal reasoning tasks.

**Additional Feedback:**

There is no.

**Documentation:**

Well documented.

**Ethics:**

.

**Limitations:**

.

**Opportunities For Improvement:**

- It would be useful to provide additional tags to individual tasks according to standard NLP terminology (e.g. common-sense, logical resoning, entailment etc) if possible. It will help NLP reseachers without legal knowledge to understand individual tasks.

- "162" tasks seems to be a little bit exaggerated as more than 100 tasks are generated by re-formatting previous tasks (38 tasks from CUAD, 14 tasks from Contract NLI tasks, 34 tasks from MAUD tasks, 9 tasks from OPP-115 tasks, etc..).
Why not use the number that can be calculated from Table 4?

- Although 7B LLM is frequently used in the literature, LLM with 13B scales are also popular and accessible with decent GPUs. It will be great to include the result from 13B models.

**Relation To Prior Work:**

Some of previous works on legal datasets seem to be missing.
To name a few,
A heterogeneous graph-based approach for automatic legal statute identification from indian legal documents, AAAI, 2022
2023,
Legal case document summarization: Extractive and abstractive methods and their evaluation. AACL, 2022
A Multi-Task Benchmark for Korean Legal Language Understanding and Judgement Prediction, NeurIPS, 2022
CAIL2018: A Large-Scale Legal Dataset for Judgment Prediction, 2018
Proceedings of the Eigth International Competition on Legal Information Extraction/Entailment, (COLIEE 2021)
etc..

**Summary And Contributions:**

The authors propose a legal NLP benchmark specialized for few-shot learning evaluation.
The benchmark consists of 162 tasks categorized neatly according to their legal reasoning type based on IRAC framework.
The authors also provide the performance of four representative LLMs, GPT-4, GPT-3.5, Incite-7B-Instruct, Incite-7B-Base.

---

> ### Author Response · Authors · 2023-08-21
> **Response to BYVU**
>
> We thank BYVU for their review. We are glad to hear they appreciated the organization of tasks, and the timeliness of the contribution.
>
> **Additional tags.** We have added tags for each task according to standard NLP terminology. In the updated draft, tags can be found in Table 7 (page 30).
>
> **How we count tasks.** We appreciate the reviewer highlighting the manner in which tasks are counted, and the potential confusion that could arise. Table 5 counts the number of corpora that LegalBench tasks are drawn from (e.g., CUAD, LearnedHands, Supply Chain Disclosures, SSLA documents). A single corpora may yield multiple “tasks,” where each task has (1) a different prediction goal, and (2) relies on a different subset of documents from the corpora. We have updated our draft to clarify this, and highlighted the number of corpora that LegalBench tasks are drawn from (L54).
>
> We opted to count the number of tasks instead of the number of corpora because we believe that analyzing performance on the level of tasks provides a finer-grained view of performance. Task-based evaluation allows us to study how domain-significant variations in prompting for instance (§5.3), affect performance. It also allows us to compare models granularly, and identify where one LLM lags behind another. Finally, tasks can be better aligned with legal expert understandings of difficulty. For instance, lawyers assess difficulty in contract clause classification depending on the type of clause.
>
> **Evaluation on additional models.** We thank the reviewer for noting the lack of evaluation on larger models. In response, we have updated our draft with evaluations of four 13B LLMs, amongst other models.
>
> **Prior work.** We are grateful to the reviewer for identifying the missing references. We have added these and more to both the main-body literature review, and the expanded literature review in the Appendix. As a reference point, we note that the updated draft has nearly twice the number of references as the original submission.

---

### Official Review · Reviewer_cUyJ · 2023-07-21
**A great resource that warrants more comprehensive evaluation**

**Rating:** 8
**Confidence:** 4
**Correctness:** The process of dataset construction a…

**Strengths:**

* The proposed benchmark undoubtedly serves as a unique and highly valuable resource for the research community.
* The collaboration with legal experts is a significant advantage. It enabled precise analysis of the types of legal reasoning required in the real world, hence assessing models on concrete tasks that will add substantial value in practice.
* The level of detail included in the documentation is remarkable.

**Additional Feedback:**

Overall, a great resource that I am sure will benefit the research community.

**Clarity:**

The paper is excellently written and properly structured, making it both easy to follow and enjoyable to read.

**Documentation:**

The documentation accompanying the benchmark is extraordinarily thorough, with 114 pages detailing each task, its context, construction process, and significance.

**Ethics:**

I do not have any major ethical concerns to report.

**Limitations:**

* Several limitations are acknowledged in a section dedicated to this topic. One particularly pertinent and constraining limitation is that the benchmark is exclusively in English - despite other legal benchmarks like LEXTREME (Niklaus et al., 2023) offering multilingual features (though they are less comprehensive) - and tends to focus on American law, particularly contract law. These factors must be borne in mind for future work before drawing general conclusions about the performance of LLMs "in the legal field."
* An additional limitation that I would suggest is the inherent difficulty in accurately evaluating some of the tasks presented in this benchmark, particularly the seven open-generation tasks. In their assessment, the authors enlist legal experts to evaluate rule-application tasks, but this might not be feasible or affordable for others wishing to work with LegalBench. The research community would likely benefit from a more rigorous study of suitable metrics for each type of task or from guidelines on manual evaluation.

**Opportunities For Improvement:**

* It is regrettable that the authors did not conduct a more exhaustive evaluation of current LLMs, specifically in terms of diversity by considering some major foundational models (LLaMa, Falcon, OPT, MPT, BLOOM) and a few instruction-tuned derivatives (Vicuna, WizardLM, etc), but also in terms of model sizes -- a comparison between 3B, 7B, 13B would have been informative. Given the large number of tasks included in the benchmark and the computational resources required for such evaluations, it is understandable that a limited number of LLMs were considered. However, readers could argue that focusing only on two open-source LLMs does not provide a sufficiently representative picture of the current performance of publicly accessible models.
* It is also disappointing that the authors chose not to incorporate legal tasks involving long documents, due to the current constraint of limited context windows inherent to LLMs, as the latter is increasingly being circumvented by techniques such as dynamic NTK-aware scaling of RoPE for models exploiting rotary positional embeddings.

**Relation To Prior Work:**

Previous work is accurately described in the second section (although a little short). Besides, authors substantiate many claims throughout the paper with relevant references.

**Summary And Contributions:**

* The authors have constructed a benchmark comprising 162 tasks, spanning across six types of legal reasoning derived from a common framework employed in the legal community.
* The tasks are sourced from both existing datasets and new contributions from legal professionals, aiming at evaluating practically useful skills.
* Empirical results based on four LLMs underscore the performance of current models on legal reasoning.

---

> ### Author Response · Authors · 2023-08-21
> **Response to cUyJ**
>
> We thank cUyJ for their review. We are glad to hear they appreciated the unique value of the benchmark, the distinctive nature of interdisciplinary collaboration that produced it, and the level of detail in documentation.
>
> **More exhaustive evaluation**. We appreciate the reviewer noting the missed opportunity for a more comprehensive evaluation. In response, we have evaluated significantly more LLMs (5x the number in the original submission), including models from major families (LLaMa, Falcon, OPT, MPT, Bloom), instruction tuned-derivatives (Vicuna, WizardLM), and differently sized models (3B, 7B, 13B). While not exhaustive, we believe that–per the reviewer’s noted concern–these models do provide a more comprehensive picture of performance. In particular, they show:
>
> - How performance improves with scale.
> - Where the gaps between open and closed-source are smallest and largest.
> - How variations in family can significantly affect performance (e.g., chat-tuning vs Flan-T5 style-tuning).
>
> We additionally point the reader towards new experiments conducted on prompting (§5.3). In particular, these highlight important nuances with the use of open-source models. For instance, we find that open-source models are particularly sensitive to the choice of in-context samples, and that practitioners should be cognizant of the level of technical jargon used in prompts.
>
> **Long document tasks**. We acknowledge, as pointed out by the reviewer, the absence of long-document tasks in LegalBench, particularly given recent progress with regards to improving LLM context-windows. We note that the “call” for tasks to be included in this round of LegalBench was put out in September 2022. At that time, the state-of-the-art model (GPT-3) had a context window of 4k tokens. Hence, we chose to limit our tasks to smaller context windows.
>
> However, we note that the authors behind LegalBench have also begun to curate a collection of long-document tasks (currently numbered ~25). We look forward to releasing these, either as an addition to LegalBench or as a separate benchmark.
>
> **Limitation to English and skew towards American law**. We thank the reviewer for highlighting these important limitations of the benchmark, particularly with regards to linguistic features and country-of-origin. In response, we have revised certain statements in the draft to deemphasize broader conclusions about legal performance.
>
> **Evaluating open-generation tasks**. We appreciate the reviewer noting the inherent difficulty in evaluating the seven open-generation tasks, as future researchers may not be able to work with or hire lawyers to evaluate. Per the reviewer’s recommendation, we have offered guidelines for how non-legal experts may evaluate the generations for these tasks. These guidelines offer detailed descriptions of what generations should or should not contain. Please see the global review for a link to the guidelines.

---

### Official Review · Reviewer_ZLU8 · 2023-07-24
**LegalBench: legal reasoning benchmark review**

**Rating:** 5
**Confidence:** 4

**Strengths:**

Recent progress in LLMs has not previously led to public datasets and benchmarks providing broad evaluation of LLMs' legal reasoning capabilities. The present work makes progress towards categorizing legal reasoning tasks and benchmarking LLMs on such tasks.  This work is done as a collaboration between ML and legal experts, and appears to be amongst the first broad-based collaborative efforts of its kind.  This is a novel work towards an important research direction.

**Additional Feedback:**

I will be glad to discuss/further revisit the above.

**Clarity:**

 Apart from the clarifications suggested elsewhere in this review, I think the paper is clear.

**Correctness:**

Subject to the discussion of evaluation limitations above, the results in the paper seem to be sound. However, items 1. and 2. below should be addressed to ensure reproducibility of the results.

**Documentation:**

The NeurIPS Datasets and Benchmarks call for paper requires the following supplementary materials, which seems to be incomplete or missing in the current submission.

1. Reproducibility:  The codebase on Dropbox (utils.py and demo.ipynb) is aimed to generate prompts in plain text, but no code is available for inference/evaluation. Also Appendix E.2 Composition mentions that labels in the dataset include open-ended generation. I could not find ground truth answers for tasks that require generation (or any human expert corrections of answers generated by the models).  I reviewed the Dropbox folders for the ucc_v_common_law task featured in Table 1 and a few other tasks requiring generated answers.

2. Long-term preservation/hosting/maintenance: The dataset is shared via Dropbox URL link. According to Appendix E.5 the authors intend to create a repository as part of the LegalBench website. However, it is unclear where Dropbox or another repository will be used. Appendix E.7 says that the authors will make older versions of the dataset available upon email request, but there appear to be no other maintenance details. (Note many repositories, like zenodo, make older versions automatically available). One particular question is whether the dataset will be publicly available to everyone subject to the terms of the relevant licenses alone, or users will need to expressly sign any terms and conditions, submit any applications or take any other special steps in order to gain access to the dataset.

3. The required author statement that they bear all responsibility in case of violation of rights, etc. appears to be missing.

4. A persistent identifier: As there appears to be no permanent repository for the dataset, no DOI or other permanent identifier is available either in the current submission.

**Limitations:**

The limitations described in the previous section under a. and d. are very briefly mentioned in the paper, and perhaps would benefit from further elaboration, while the limitation described in b. is not explained.


**Opportunities For Improvement:**

I suggest the following two main items for improvement.

a. The paper does not develop a quantitative metric for evaluating the generation tasks (see also item 1 under Reproducibility below regarding the apparent lack of labels/annotations for the generated tasks). One way to do this would be to consider a graph or sequence of references/entities, such as the "bike->good->UCC->contract" reasoning chain shown in Table 1. The benchmark would evaluate the generated text on the retrieval of these references as well as their number and ordering. Cf. the evaluation of mathematical reasoning in natural language in Welleck et al, NaturalProof: Mathematical Theorem Proving in Natural Language, NeurIPS Datasets 2021.  Another potential approach would be to evaluate whether the issue, rule, application and conclusion (IRAC) reasoning prompts would improve the accuracy to responses to yes/no questions (see references by Yu et al in the next paragraph).

b. The issue, rule, application and conclusion (IRAC) paradigm described in Section 3.1 is normally applied to a given fact pattern in a multi-hop fashion: issue->rule->application->conclusion.  (Moreover, a fact pattern may contain multiple issues, and a practicing lawyer or law student would need to spot and analyse each of them.) In the present paper, the discussion in Section 3.2 and the Dropbox materials  suggest that the issue spotting, rule recall, application and conclusion tasks do not pertain to a single fact pattern. Rather it suggests that one fact pattern is given for the issue spotting, a different fact pattern is given for the rule recall, etc. Therefore, it seems impossible to use the dataset to train/evaluate a model that performs the IRAC reasoning in a multihop fashion for each issue it identifies in a given fact pattern; as mentioned this is a usual setting in legal practice or law exam.  More realistic tasks were described in reference [35] assessing GPT4 performance on the essay section of the multistate bar exam. Related work in the context of Japanese Bar exam answered yes/no answers using prompts based on IRAC and other legal reasoning paradigms:
*Yu et al, Exploring the Effectiveness of Prompt Engineering for Legal Reasoning Tasks,  In Findings of the Association for Computational Linguistics: ACL 2023; and
*Yu et al, Legal Prompting: Teaching a Language Model to Think Like a Lawyer, https://arxiv.org/abs/2212.01326.

Below I outline a few additional missed opportunities that are less significant than the ones above.

c. The tasks adapted from the CUAD [32], MAUD [69] and ContractNLI [38] datasets should permit comparison of the fine-tuned models evaluated in those references with the LLMs evaluated in the present paper. This opportunity appears to be missed since the metrics are different. For example, in the CUAD and MUAD papers use the area under the precision-recall curve while the present paper uses class-balanced accuracy (see also item 1 under Reproducibility below re unavailability of code for inference/evaluation).

d. Since the context windows of most LLMs are constrained to a few pages of text, LegalBench does not include tasks involving long documents.  However, the CUAD, MUAD and ContractNLI datasets used in LegalBench actually include original long-form contracts from EDGAR, and each of those previous papers provides a bespoke solution to address the limited context windows of the relevant transformer models. (MUAD provides manually extracted clauses and actually benchmarks models on the separate extraction task, while CUAD and ContractNLI use some versions of a sliding window.) Accordingly instead of copying into LegalBench short excerpts from those long document datasets, it would be advantageous to maintain the integrity of the original data and use the segmentation baselines associated with each of the original datasets to address the maximum context window limit. This approach would permit not only using LegalBench to compare the LLMs benchmarked in the present paper with the models evaluated in the the CUAD, MUAD and ContractNLI papers (see the previous paragraph) but also to evaluate new segmentation baselines on the legal reasoning tasks. LegalBench does not appear to map its datapoints to the original long-form CUAD/MUAD/ContractNLI docs. If the long-form original are preserved and/or such maps are provided, then the issue extraction task from long text could be added in the future without changing the dataset. Cf. the COLIEE legal information extraction and entailment competition where the entailment task may depend on a separate retrieval task.

**Relation To Prior Work:**

As suggested in items a. and b. above, I think it is important to develop a quantitative metric for evaluating reasoning in generated text.  A more robust literature overview relating to quantitative evaluation of legal and other reasoning in natural text, and the previous datasets and benchmarks on reasoning in natural language should provide some guidance. The references provided in item b. above could serve as a starting point.

As noted in item d. above, it is standard to include the original long documents in legal NLP datasets, and use some kind of manual or automatic segmentation benchmark to address the maximum context window limitations. See CUAD, MUAD and ContactNLI references described above. Other references relevant to overcoming the maximum context window limit in those legal datasets are the following:
*A model evaluated on CUAD developed by a subset of the authors of the present LegalBench paper -- Li et al, Don't Use a Cannon to Kill a Fly: An Efficient Cascading Pipeline for Long Documents; and
*A dataset for segmenting the MAUD agreements -- Rao et al. MarkupMnA: Markup-Based Segmentation of M&A Agreements

**Summary And Contributions:**

The paper categorizes legal tasks based on several reasoning paradigms from the legal domain. It introduces a dataset and benchmark to evaluate the performance of large language models (LLMs) prompted by natural language on legal reasoning tasks. This work appears to be the first benchmark aimed explicitly at assessing US legal reasoning capabilities of such LLMs (although significant share of the tasks have been constructed from tasks categorized as information extraction, reading comprehension and inference in previously released legal datasets.)

---

> ### Author Response · Authors · 2023-08-21
> **Response to ZLU8**
>
> We thank ZLU8 for their review. We are glad to hear they appreciated the unique collaboration which formed the basis for this work, the novelty of the work, and the importance of this research direction.
>
> **Quantitatively evaluating generation tasks (a)**. We appreciate the reviewer highlighting the evaluation challenges posed by the generation tasks, and how many researchers using this benchmark may not have access to lawyers who can perform manual grading for these tasks. In response, we have decided to implement the reviewer’s suggestion to implement a more structured format for evaluation. For each task, we have created a “gradebook,” which provides for each sample a description of what a correct LLM generation should encompass, as an ordered set of inferences. For example: “(1) the contract is for a bike, (2) a bike is a tangible good (3) tangible goods are covered by the UCC, (4) the contract is covered by the UCC.” Per your recommendation, this is inspired by NaturalProof ([147] in the updated draft). Gradebooks are accessible here: https://hazyresearch.stanford.edu/legalbench/getting-started/.
>
> **Multi-hop IRAC (b)**. We thank the reviewer for pointing out differences in how we employ IRAC in contrast to how it has been utilized in previous contributions. In response, we have included a discussion that emphasizes those differences. In particular, we highlight how the references cited (e.g., Katz et al, 2023) evaluate open-ended LLM generations under the IRAC structure. In that work, lawyers manually evaluated whether responses adhered to IRAC, and graded whether each element of IRAC was found in the generation, and correctly performed.
>
> In contrast, we use IRAC as an organizing principle for grouping tasks. The tasks we study implicate only one aspect of IRAC (e.g., only issue-spotting). As a result, evaluating performance in the manner of Katz et al, would be inappropriate, because these tasks do not implicate other prongs of IRAC.
>
> We have updated our draft to clarify that our goal is to assess performance at each step of IRAC, and not across the chain (L260). We additionally note that the purpose of this grouping is to draw higher level conclusions about where LLMs perform well or struggle, using terminology that is familiar to the legal community. For instance, this grouping allows us to identify that Claude-1 can match GPT-4’s performance on rule-recall, but performs worse on issue-spotting.
>
> Daniel Martin Katz, Michael James Bommarito, Shang Gao, and Pablo Arredondo. Gpt-4
> passes the bar exam. Available at SSRN 4389233, 2023.
>
> **Reporting performance metrics from CUAD, MAUD, and ContractNLI**. We appreciate the reviewer highlighting the value of enabling a more direct comparison to baselines evaluated in prior work (e.g., for CUAD, MAUD, ContractNLI). In response, we have released predictions for all LLMs for all tasks. They are available here: https://drive.google.com/file/d/1MwmfhKCOS7F1atJdLt_ldbautauWAaM4/view?usp=sharing.
>
> We note the caveat, however, that it is not possible to directly compare results on LegalBench for tasks derived from these datasets to the performance reported in the original papers because the datasets presented in those papers have been significantly modified in order to be incorporated into LegalBench. For instance, the CUAD and ContractNLI classification tasks operate on subsamples of the original datasets, with different class-balances. Additionally, the MAUD paper only reports aggregate category performance, while LegalBench has created tasks out of certain individual questions.
>
> **Linking segmented clauses to original documents**. We acknowledge, as pointed out by the reviewer, the additional value that would derive from linking clauses in LegalBench’s derived version of the long-document tasks to their original documents. Accordingly, per their recommendation, we have released mappings for CUAD and ContractNLI which associate clauses with the original documents, and are working on generating mappings for MAUD. The mappings are available as a column on each of these tasks, accessible on the Huggingface data.
>
> **Documentation**. We are grateful to the reviewer for identifying gaps in the documentation for the files submitted. We highlight where these elements can now be found:
>
> - Code for evaluation is available on the LegalBench Github repository.
> - Ground truth answers and guidelines for grading are available on the LegalBench website (see global review).
> - Inference was performed using the open-source Manifest library, which is available here: https://github.com/HazyResearch/legalbench/.
> - The dataset is publicly available to everyone subject to the terms of the task-specific licenses. It is hosted on Huggingface. Instructions for accessing older-versions of the benchmark are available on the website.
> - The updated draft has the required author statement regarding bearing responsibility of rights.

---

### Author Response · Authors · 2023-08-21
**Global review**

We thank all reviewers for their feedback. We are glad to hear that reviewers recognized the first-of-its-kind collaborative process by which LegalBench was constructed [ZLU8, cUyJ, tmL2], the organization of LegalBench tasks [ZLU8, cUyJ, BYVU, XJRt, tmL2], the breadth and number of tasks [ZLU8, cUyJ, XJRt], and the extensiveness of documentation [cUyJ, XJRt].

We have made a number of changes in response to reviewer comments, and are happy to report updates to the project since submission. We have uploaded a revised version of the submission, and summarize improvements below.

- We have published LegalBench (https://hazyresearch.stanford.edu/legalbench/). The data can be downloaded via HuggingFace (https://huggingface.co/datasets/nguha/legalbench). The Github repository stores both prompts and code for loading/evaluating tasks (https://github.com/HazyResearch/legalbench/).

- We now report a significant number of additional experiments on LegalBench. The full results are available in Appendix J. We highlight three specific improvements:
    - We now report results from 20 different LLMs (up from 4 in the original submission), spanning 11 different families.
    - We show how LegalBench can be used to perform an in-depth comparison of LLMs, with special attention to specific error-modes and performance tradeoffs.
    - We perform a study of prompt-engineering on LegalBench tasks, and identify trends relating to the importance of latent legal knowledge, plain language versions of instructions, and in-context demonstrations.
- We have expanded the related works section (§2) to engage with a broader section of the literature. We have additionally included an extended discussion of related work in Appendix F, given the space restrictions on the main-body discussion.
- We have provided answer-guides for grading generation tasks, and have regraded generations to ensure consistency with the grading guides. Information about manual grading can be found here: https://hazyresearch.stanford.edu/legalbench/getting-started/

We appreciate several reviewers noting  that the benchmark is largely limited to the US setting. We acknowledge this limitation, and are excited to expand the benchmark to other jurisdictions in future work. At the same time, we would like to emphasize that our efforts to generate a benchmark for the US alone involved substantial effort. The role of legal experts in curating and developing LegalBench is essential to its legitimacy. Involving legal experts ensures that the benchmark captures tasks which are meaningful to the legal community. It also provides AI researchers with the confidence that tasks contained in the benchmark correspond to interesting or useful capabilities.

Following this method for benchmark construction–an aspect of this work recognized by a number of reviewers as novel and valuable–was a significant endeavor. Because of the lack of significant interdisciplinary collaboration between law and AI, building this benchmark required substantial coordination and outreach to recruit legal experts, and provide them with the necessary background on LLM evaluation. These legal experts then devoted substantial time to hand labeling data, generating manual explanations, and iterating on task design.

In light of this, it appears difficult for any single team of researchers to compile a benchmark that simultaneously allows for the assessment of LLMs across different legal systems. To that end, we view our efforts as a first, self-contained step towards a collection of more comprehensive benchmarks. We are confident that legal scholars with expertise in other jurisdictions will help us to achieve the goal of ultimately forming a more comprehensive, globally representative benchmark. Consistent with that goal, we highlight in the paper our continued solicitation of different task contributions, and that we hope to expand to additional jurisdictions as part of future work.

Once again, we are extremely grateful to the reviewers for their thoughtful and detailed comments, and look forward to clarifying remaining questions or concerns.

---

### Decision · Program_Chairs · 2023-09-22

**Decision:**

Accept (Poster)

**Comment:**

This paper reports on a collaboratively assembled benchmark for assessing legal reasoning. The authors include both CS/AI folks and the stake holders from legal profession--making it more likely that the benchmark may find acceptance in the legal community.

The reviews are mixed--with three reviewers pretty strongly advocating it, and two viewing the paper and the contribution as borderline.

Given the sensitivity of the benchmark, an ethics review was also done (and the authors responded to the concerns raised).

Speaking for myself (AC), one question that I had was this: The benchmark seems to focus mostly on reasoning by pattern matching (rule application, rule recall, rule conclusion etc). To what extent is legal reasoning really glorified pattern matching, as against what logical reasoning involving long inference chains? I ask because as the authors may have noticed, there is considerable divergence of opinion as to whether LLMs are actually able to do the latter type of deep reasoning. I think the paper and the benchmark would be strengthened if it discusses the issue.